# A randomized, double-blind, placebo-controlled trial of niclosamide nanohybrid for the treatment of patients with mild to moderate COVID-19

Jung Ho Kim [1], Sungmin Kym[2], Shin-Woo Kim[3], Dae Won Park[4], Ki Tae Kwon[5], Jun-Won Seo [6], Seungjin Yu[7,8], Goeun Choi[7,8], Sanoj Rejinold N[8], Jin-Ho Choy [8,9] ✉, Geun-woo Jin[10] & Jun Yong Choi [1] ✉

Effective and reliable treatments for SARS-CoV-2 infections are a key part of global COVID-19 management. Based on vitro studies, niclosamide has been considered as a potential drug candidate for SARS-CoV-2, but its clinical development has been limited due to poor solubility and bioavailability. Here we report results from a randomized, double-blind, placebo-controlled clinical trial involving 300 patients (Clinical Trial Registration Number: KCT0007307) that assessed the efficacy and safety of the niclosamide nanohybrid CP-COV03 at two different doses. Oral CP-COV03 was well tolerated, with no serious adverse events reported in any treatment group. The primary endpoints demonstrated that CP-COV03 significantly alleviated all 12 FDA-recommended COVID-19 symptoms, with symptom improvement sustained for more than 48 h. Additionally, CP-COV03 reduced SARS-CoV-2 viral load by 56.7% within 16 h of the initial dose compared to baseline. Secondary endpoints, including time to sustained symptom resolution, time to return to usual health, and reduction in hospitalization risk, also showed favorable results in the CP-COV03 group compared to placebo. These findings indicate that CP-COV03 is a safe and effective therapeutic option for the treatment of mild to moderate COVID-19 and represents a promising advancement in the repurposing of niclosamide through nanohybrid engineering.

The coronavirus disease 2019 (COVID-19) pandemic, caused by severe acute respiratory syndrome coronavirus 2 (SARS-CoV-2), has led to significant global health concerns[1]. Despite widespread vaccination, emerging variants such as Delta and Omicron continue to challenge pandemic control, highlighting the need for improved therapeutics[2–4].

To date, several oral antivirals; nirmatrelvir + ritonavir (Paxlovid®), molnupiravir (Lagevrio), and ensitrelvir (Xocova) have been developed[5–8]. However, limitations such as drug–drug interactions (ritonavir)[9], potential teratogenicity (molnupiravir)[10], and reduced

clinical efficacy in vaccinated individuals underscore the need for alternative options. Furthermore, molnupiravir has not significantly reduced COVID-19-related hospitalizations or deaths among high-risk vaccinated adults[11].

In this context, Niclosamide[12], an FDA-approved antiparasitic drug, has shown broad-spectrum antiviral activity[13], including against SARS-CoV-2 and its variants (supplementary Table S1[14–22]), and benefits from a long-standing safety profile[23,24] and low cost. However, its clinical use has been hindered by poor solubility and

bioavailability[25–30]. A prior randomized trial in patients with mild to moderate COVID-19[31] using pristine niclosamide did not significantly shorten symptom duration, likely due to these limitations

To overcome this, we developed CP-COV03, a nanohybrid formulation of niclosamide with magnesium oxide and hydroxypropyl methylcellulose (NIC-MgO-HPMC), which significantly enhances niclosamide's bioavailability (Supplementary note-1, pages 32-43). In a Syrian hamster model, CP-COV03 effectively reduced SARS-CoV-2 replication and lung pathology compared to controls[32].

Based on these findings, CP-COV03 progressed into clinical evaluation. This study presents the clinical trial design, methodology, and findings on the safety and efficacy of CP-COV03 in patients with mild to moderate COVID-19.

## Results

### Patients
Between May 11 and November 28, 2022, a total of 317 patients were screened, out of which 300 were enrolled and randomized equally into three groups: the low dose (300 mg), the high dose (450 mg), and the placebo. After excluding seven patients who withdrew before treatment, 293 received the study drug, and 291 completed the trial. All participants were hospitalized for at least six days. The safety and ITT (intention-to-treat) analyses included the 293 treated patients, all of whom began treatment within five days of symptom onset (Fig. 1). Additional analysis sets included modified ITT-1 ((mITT-1) (264 patients treated within three days of onset)), mITT-2 (253 patients with valid baseline PCR), and the per-protocol set (PPS), which included 227 patients who had no major protocol deviations and completed the full follow-up.

Major protocol deviations were defined in the protocol (study protocol in the supplementary file pages 2–17) and reviewed during the blind data meeting (the blind meeting was held on January 31, 2023, and the data were unblinded on February 22, 2023). Breakdowns of protocol deviations leading to exclusion from the PPS are presented in supplementary Table S2. The baseline characteristics of the study participants are shown in Table 1.

### Efficacy
The primary efficacy endpoint was the median time required for sustained improvement ($\geq 48$ h) of targeted COVID-19 symptoms by Day 14. In the PPS analysis, the low dose group showed a median of 9.0 days (95% CI, 7.00–10.00), significantly shorter than the placebo group's 13.0 days (95% CI, 10.50–ND (Not Determined); $P = 0.0083$). Similar trends were observed in the mITT-1 and mITT-2 populations, with statistically significant differences between the low dose and the placebo groups ($P = 0.024$ and $P = 0.0275$, respectively). While the ITT group showed a shorter symptom duration for the low dose group (10.0 vs. 12.25 days), the difference was not statistically significant. Notably, the benefit of early administration was pronounced among high-risk individuals (aged $\geq 60$, with comorbidities, or immunocompromised), where the low dose group showed a median of 7.5 days to symptom improvement compared to 12.5 days in the placebo group ($P = 0.017$).

### Viral load, pharmacokinetic parameters, and their association
Viral load data (Fig. 2A) revealed that both the low and the high dose groups achieved significant reductions compared to the placebo as early as 16 h after administration (Day 2), with 56.7% and 55.2% reductions, respectively, versus just 4.1% in the placebo group. This represented a ~13.5-fold greater reduction in viral load in the CP-COV03-treated groups. These changes were significantly associated with the pharmacokinetics of niclosamide (Fig. 2B), in which the low dose group had a Cmax of 285.25 ng/mL and an AUCt of 10,562.09 ng·h/mL, while the high dose group had a higher Cmax of 389.90 ng/mL and AUCt of 12,876.29 ng·h/mL. Although the high dose group, which showed higher drug exposure, demonstrated a stronger negative correlation with viral load reduction compared to the low dose group, the time–concentration curves were similar for both doses. This suggests that the antiviral efficacy of CP-COV03 may be more time-dependent than concentration-dependent. Correlation analysis confirmed a significant inverse relationship between drug exposure and viral load (r = –0.330 for the low dose group and r = –0.482 for the high dose group; Fig. 3A, B).

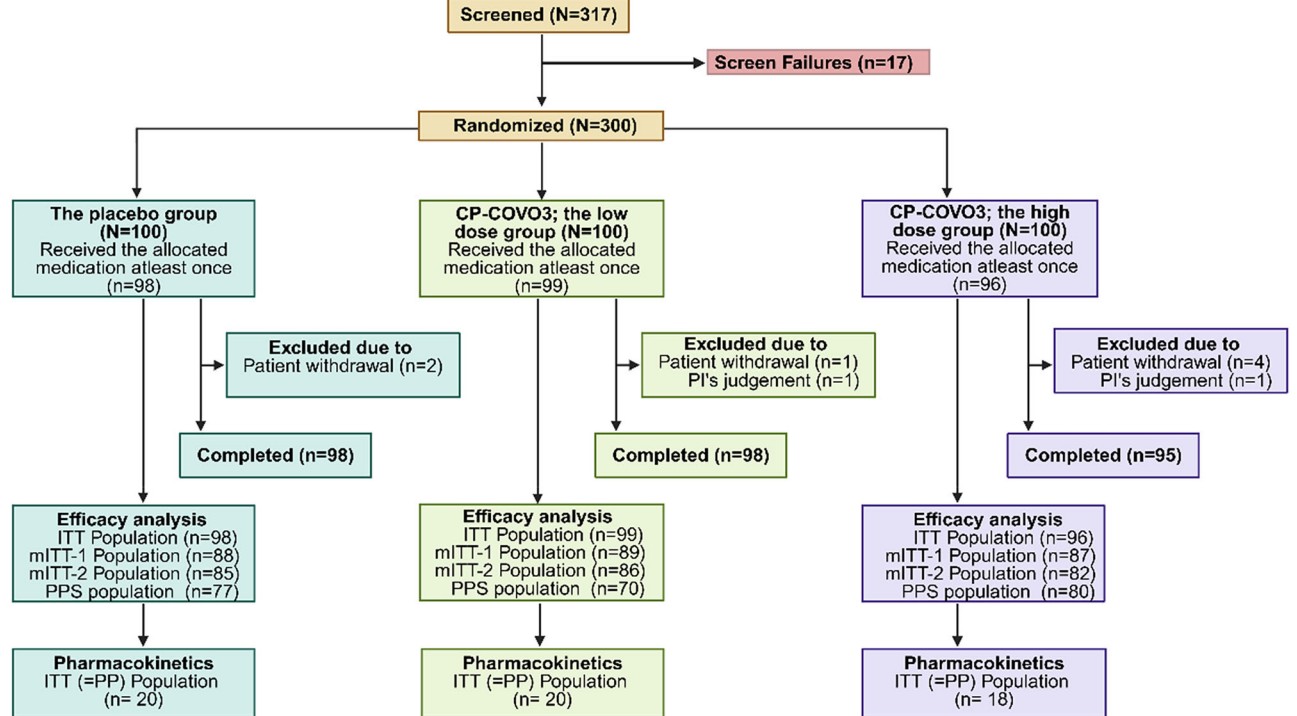

**Fig. 1 | CONSORT diagram for clinical trial.** Shown is the study flow chart for randomized, double-blind, placebo-controlled trial of multiple doses of CP-COV03 in mild or moderate COVID-19 (Created in BioRender. REJINOLD, S. (2025) https://BioRender.com/ wlivq3e).

**Table 1 | Baseline characteristics (intention-to-treat population)**

| | The placebo group (n = 98) | The low dose group (n = 99) | The high dose group (n = 96) |
|---|---|---|---|
| **Demographics** | | | |
| Age (y), mean (SD) | 43.79 (12.94) | 42.18 (13.50) | 41.89 (12.32) |
| Age groups | | | |
| 19–29 yrs., n (%) | 17 (17.4) | 24 (24.2) | 20 (20.8) |
| 30–39 yrs., n (%) | 21 (21.4) | 16 (16.2) | 21 (21.9) |
| 40–49 yrs., n (%) | 25 (25.5) | 25 (25.3) | 29 (30.2) |
| 50–59 yrs., n (%) | 21 (21.4) | 25 (25.3) | 16 (16.7) |
| ≥60 yrs., n (%) | 14 (14.3) | 9 (9.1) | 10 (10.4) |
| Sex | | | |
| Male, n (%) | 62 (63.3) | 70 (70.7) | 61 (63.5) |
| Female, n (%) | 36 (36.7) | 29 (29.3) | 35 (36.5) |
| Height, cm, mean (SD) | 168.81 (8.57) | 169.29 (8.39) | 169.77 (7.79) |
| Weight, kg, mean (SD) | 70.92 (14.05) | 70.04 (13.16) | 69.79 (12.99) |
| **Comorbidities** | | | |
| Hypertension, n (%) | 11 (11.0) | 7 (7.1) | 4 (4.2) |
| Diabetes mellitus, n (%) | 2 (2.0) | 5 (5.1) | 2 (2.1) |
| Hyperlipidemia, n (%) | 4 (4.1) | 11 (11.1) | 5 (5.2) |
| **COVID-19 symptoms** | | | |
| Fever, n (%) | 9 (9.2) | 12 (12.1) | 7 (7.3) |
| Chill, n (%) | 87 (88.8) | 82 (82.8) | 82 (85.4) |
| Muscle ache, n (%) | 86 (87.8) | 88 (88.9) | 85 (88.5) |
| Headache, n (%) | 69 (70.4) | 70 (70.7) | 63 (65.6) |
| Fatigue, n (%) | 80 (81.6) | 84 (84.8) | 73 (76) |
| Cough, n (%) | 63 (64.3) | 66 (66.7) | 57 (59.4) |
| Sore throat, n (%) | 66 (67.3) | 70 (70.7) | 64 (66.7) |
| Stuffy or runny nose, n (%) | 70 (71.4) | 75 (75.8) | 68 (70.8) |
| Difficulty of breathing, n (%) | 21 (21.4) | 23 (23.2) | 24 (25) |
| Nausea | 21 (21.4) | 27 (27.3) | 37 (38.5) |
| Vomiting | 6 (6.1) | 7 (7.1) | 6 (6.3) |
| Diarrhea | 18 (18.4) | 22 (22.2) | 20 (20.8) |
| **Mean total symptom score (±SD)** | 11.39 (± 4.93) | 11.96 (± 5.54) | 11.58 (± 5.33) |
| **COVID-19 severity** | | | |
| Mild, n (%) | 84 (85.7) | 86 (86.9) | 86 (89.6) |
| Moderate, n (%) | 14 (14.3) | 13 (13.1) | 10 (10.4) |

COVID-19: coronavirus disease 2019, SD: standard deviation.

### Safety

Adverse events were mild and similar across all groups (Table 2). No serious adverse reactions were reported. While the number of adverse events was higher in the high dose group (47 events among 33 patients) compared to the low dose and the placebo groups, there were no significant differences in severity or clinical laboratory parameters. Importantly, no adverse interactions were noted despite the concurrent use of medications for chronic conditions (supplementary Table S3).

### Rescue medicine and severity progression

Use of rescue medications (acetaminophen, ibuprofen, antidiarrheals) was permitted and did not differ significantly among groups (Tables 43–46 in Supplementary note-2). Similarly, there was no observed difference in progression to severe COVID-19, which aligns with the generally low severity profile of the Omicron variant

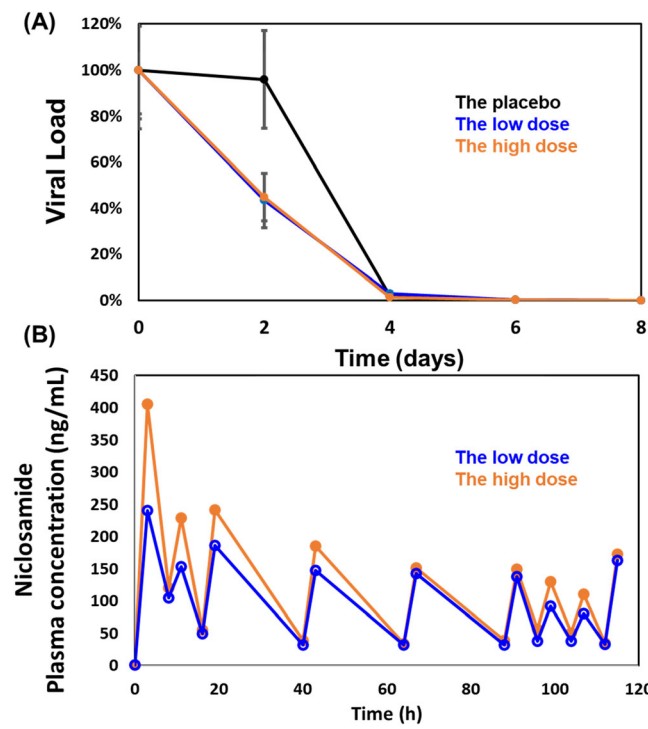

**Fig. 2 | Viral load and pharmacokinetic analyses. A** The adjusted mean change in viral load from baseline of Severe Acute Respiratory Syndrome Coronavirus 2 (SARS-CoV-2) monitored starting from Day 0. Any data point that is more than 3 times the IQR above the third quartile or below the first quartile is an outlier ($n = 77$ for the placebo group, $n = 70$ for the low dose group, $n = 80$ for the high dose group). Error bars represent standard error (S.E.). **B** Pharmacokinetic profiles of niclosamide from clinical trial ($n = 20$ for the placebo group, $n = 20$ for the low dose group, $n = 18$ for the high dose group). Error bars represent standard error (S.E.).

circulating during the trial period. Rescue medication was administered for ethical reasons to protect clinical trial participants. The number of administrations was recorded, and it was shown that there were no differences in the usage of rescue medication between treatment groups.

Additionally, there were no significant differences between the groups in the proportion of participants experiencing severe COVID-19 progression from Day 1 to Day 28. This result aligns with the characteristic low rate of severe progression associated with the Omicron variant, which was the predominant variant during this clinical trial (Table 47 in Supplementary note-2)[33].

### Discussion

In this study, conducted to support emergency use authorization (EUA), the low-dose group demonstrated a statistically significant reduction in both viral load and the number of days (reported as median values in half-day intervals) required for overall improvement in the severity scores of the 12 targeted COVID-19 symptoms, compared to the placebo group.

Although the high dose group had a higher AUCt than the low dose one (12,876.29 vs. 10,562.09 ng·h/mL), both dose groups showed similar pharmacokinetic profiles and achieved steady-state levels, leading to comparable reductions in viral load. This suggests that CP-COV03's efficacy is time-dependent rather than concentration-dependent[34–36], consistent with niclosamide's mechanism of inducing autophagy. The low dose group appears to have reached the pharmacodynamic ceiling needed for sustained antiviral action.

Analysis of the primary endpoint in the PPS population showed that the low dose group achieved sustained symptom improvement in

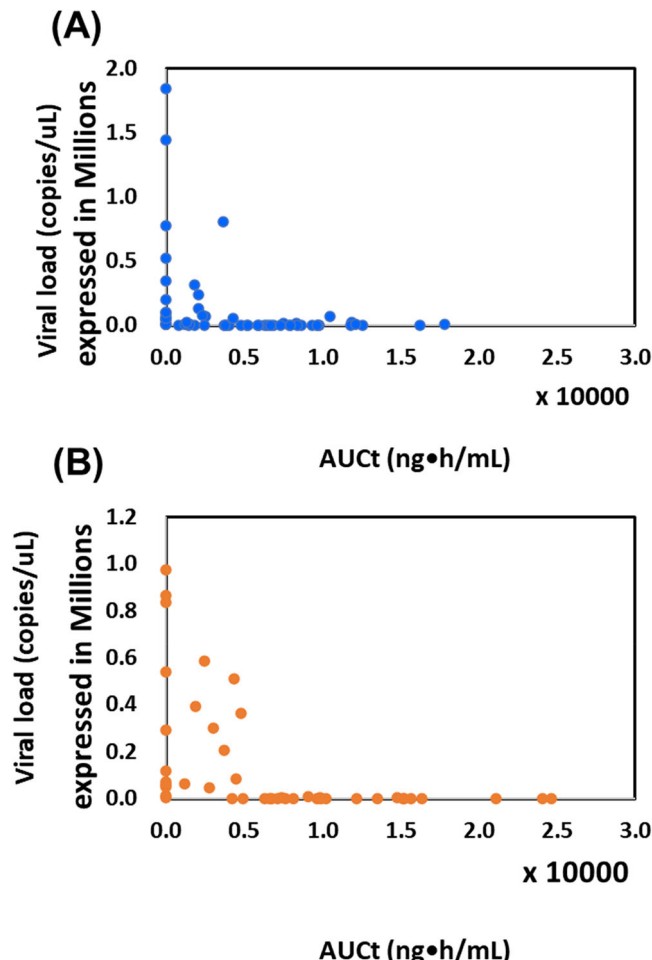

**Fig. 3 | Correlation between CP-COV03 pharmacokinetic parameters and the viral load of Severe Acute Respiratory Syndrome Coronavirus 2 (SARS-CoV-2). A** the low dose group; **B** the high dose group (n = 15 for the low dose, n = 11 for the high dose groups).

a median of 9.0 days (95% CI, 7.00–10.00), significantly shorter than the 13.0 days required by the placebo group (95% CI, 10.50–ND; $P = 0.0083$). In the secondary endpoint, the low dose group showed generally faster improvement in individual symptoms, with sore throat, headache, and fatigue resolving significantly earlier than in the placebo group (Sore throat: p = 0.0168, the placebo group 5.23 [95% CI, 4.53-5.93] / the low dose group 3.99 [95% CI, 3.26-4.72]; Headache: p = 0.0285, the placebo group 5.48 [95% CI, 4.72-6.24] / the low dose group 4.25 [95% CI, 3.46-5.04]; Fatigue: p = 0.0116, the placebo group 5.62 [95% CI, 4.81-6.42] / the low dose group 4.10 [95% CI, 3.24-4.95]) (Table 6 in Supplementary note-2).

The mean number of days required for each of the 12 targeted COVID-19 symptoms to improve and be sustained for more than 48 h was further analyzed across the ITT, PPS, and mITT groups and is provided in supplementary Table S4. In the ITT population, the median time to resolution of all 12 symptoms was 10.0 days (95% CI, 8.50-12.50) for the low dose group and 12.25 days (95% CI, 10.50-ND) for the placebo group, though not statistically significant. However, in the additional analysis of the mITT-1 population—comprising participants who received the study drug within 3 days of symptom onset—the median time to improvement of targeted symptoms was significantly shorter in the low dose group at 9.0 days (95% CI, 7.50–10.50) compared to 12.5 days (95% CI, 10.50–ND) in the placebo group ($P = 0.024$). Furthermore, in the high-risk subgroup within the mITT-1 population (aged ≥60 years, or with obesity, chronic conditions such as diabetes

or hypertension, immunocompromised status, or long-term immunosuppressant use), the low dose group showed a median improvement time of 7.5 days (95% CI, 7.00–9.00) versus 12.5 days (95% CI, 8.00–ND) for the placebo group, also reaching statistical significance ($P = 0.017$).

These findings surpass the reported efficacy of Paxlovid®, which shortened symptom resolution time by 3 days in high-risk patients[37,38], and ensitrelvir, which reduced the time to improvement in five symptoms by 1 day in the general population[8,39].

However, the high dose group did not show a statistically significant improvement in symptoms (PPS analysis: 12.25 days [95% CI, 9.50-ND] for the high dose group vs. 13.0 days [95% CI, 10.50-ND] for the placebo one). A possible explanation for this is the higher amount of magnesium oxide (MgO) in the investigational product administered to the high dose group compared to the low dose group. This higher MgO content may have caused gastrointestinal symptoms, thereby confounding participant-reported symptom evaluations. The daily MgO intake in the high dose was 945 mg, higher than the 630 mg in the low dose and exceeding the recommended daily intake of 800 mg[40]. Excessive MgO intake is associated with gastrointestinal symptoms such as diarrhea, nausea, and vomiting, which overlap with the targeted COVID-19 symptoms in this trial[41].

In an ad-hoc analysis excluding the influence of MgO, the median time for improvement of three representative COVID-19 symptoms (fever, headache, and sore throat) was 4.0 days (95% CI, 3.5-4.5), 4.5 days (95% CI, 4.0-5.0), and 6.0 days (95% CI, 4.0-7.5) in the low dose, the high dose, and the placebo groups, respectively. Both the low and the high dose groups demonstrated statistically significant reductions in the time required for symptom improvement compared to the placebo one ($P = 0.011$ and $P = 0.044$ for the low and the high doses, respectively), suggesting that MgO confounded the results for the high dose by affecting gastrointestinal symptoms.

The higher proportion of certain COVID-19 symptoms at baseline in the high dose group may have contributed to the lack of statistically significant improvement in symptoms observed in this group. Factors that could influence clinical outcomes, such as age and disease severity, were stratified to maintain balance between groups, but it was not feasible to stratify for all 12 symptoms to ensure group balance. As a result, while there were no statistically significant differences in the number of patients showing symptoms like fever, cough, sore throat, headache, muscle ache, chill, stuffy or runny nose, fatigue, difficulty of breathing, vomiting or diarrhea at baseline between groups, significantly more patients with nausea were included in the high dose (Chi-squared test, the placebo group vs the low dose one, p-value = 0.0092).

In this trial, a key indicator of CP-COV03's antiviral efficacy—its ability to reduce viral load—was confirmed, as both dose groups showed statistically significant reductions compared to the placebo one. Notably, the low dose group achieved an approximate 13.8-fold reduction in viral load just 16 h after administration, while the high dose group showed a similar 13.5-fold reduction. For comparison, Paxlovid® achieved a ~10-fold reduction by Day 5 in the EPIC-HR and EPIC-SR trials, and ensitrelvir showed reductions by Day 4. The early and substantial viral load decline observed with CP-COV03 represents one of the most rapid and effective responses reported among current COVID-19 therapies. This swift reduction may not only alleviate acute symptoms but also help limit tissue damage and reduce the risk of persistent symptoms associated with long COVID[42–45].

In this study, the primary endpoint—defined as the median number of days required for targeted COVID-19 symptoms to improve and be sustained for more than 48 h by Day 14—warrants further consideration given the nature of COVID-19. Many of the symptoms assessed were found to persist beyond the 14-day period, a pattern that aligns with observations reported in other studies[46,47].

**Table 2 | Summary of adverse events (Safety analysis set)**

| | The placebo group (n = 98) | The low dose group (n = 99) | The high dose group (n = 96) |
|---|---|---|---|
| Patients with any TEAE, n (%) | 26 (26.5) | 21 (21.2) | 33 (34.4) |
| Patients with any serious TEAE or death, n | 0 | 0 | 0 |
| Patients with TEAEs leading to treatment discontinuation, n | 0 | 0 | 0 |
| Patients with sequelae of TEAE, n | 0 | 0 | 0 |
| Cases of any TEAE, n | 37 | 32 | 47 |
| TEAEs occurring in ≥2% in either group | | | |
| Cardiomegaly, n | 5 | 2 | 7 |
| Abdominal pain upper, n | 0 | 0 | 2 |
| Dyspepsia, n | 0 | 2 | 0 |
| Pneumonia, n | 8 | 9 | 7 |
| Blood glucose increased, n | 5 | 1 | 3 |
| White blood cell count decreased, n | 2 | 1 | 1 |
| Eosinophil count increased, n | 0 | 0 | 2 |
| Lipase increased, n | 1 | 2 | 2 |
| Gamma-glutamyl transferase increased, n | 2 | 0 | 1 |
| Ageusia, n | 0 | 1 | 2 |
| Dysmenorrhea, n | 0 | 0 | 2 |
| Urticaria, n | 1 | 1 | 3 |
| Cases of any treatment-related AE, n | 19 | 12 | 22 |
| Treatment-related AEs occurring in ≥2% in either group | | | |
| Blood glucose increased, n | 5 | 1 | 3 |
| White blood cell count decreased, n | 2 | 1 | 1 |
| Eosinophil count increased, n | 0 | 0 | 2 |
| Lipase increased, n | 1 | 2 | 2 |
| Urticaria, n | 1 | 1 | 3 |

AE: adverse event, TEAE: Treatment emergent adverse event.

For instance, one study reported that 137 out of 290 COVID-19 patients (47.2%) continued to experience symptoms even one month after infection[48]. Similarly, in the pivotal EPIC-HR trial of Paxlovid®, which assessed symptom resolution through Day 28, only 43.9% of patients in the treatment group achieved full symptom alleviation by that time point[49]. In the present study, when assessing symptom improvement up to Day 14, 43.7% of participants in the ITT population −comprising 46 in the placebo group, 39 in the low dose, and 43 in the high dose−did not reach sustained improvement of all 12 targeted symptoms for at least 48 h. These findings are consistent with prior research and further support the observation that COVID-19 symptoms can persist well beyond the acute phase of infection.

Additionally, in the Cox proportional hazards regression model, the date of sustained symptom improvement was censored as the last recorded symptom assessment. Given the prolonged and variable nature of COVID-19 symptoms, limiting the evaluation period to 14 days carries a risk of misclassification. For example, participants whose symptoms improved by Day 14 but relapsed afterward would still be counted as having achieved sustained improvement, while those whose symptoms improved after Day 14 would be incorrectly classified as not having improved. This analytical limitation underscores the need for extended follow-up to more accurately capture treatment outcomes.

The clinical trial protocol, including the definition of the primary endpoint, was developed in consultation with the MFDS during the planning phase. Following precedents set by protocols for other acute respiratory infections such as influenza, the study aimed to assess whether CP-COV03 could facilitate rapid symptom improvement within a 14-day period. However, given the increasingly recognized persistence of COVID-19 symptoms beyond the acute phase, this 14-day timeframe presents a risk of underestimating treatment efficacy. To address this limitation, an ad-hoc analysis was conducted to evaluate symptom improvement through Day 28.

The ad-hoc analysis demonstrated that, in the ITT population, the low dose group experienced symptom improvement 3 days earlier than the placebo group (median: 10.50 days [95% CI, 9.00–18.00] vs. 13.50 days [95% CI, 11.00–17.00]; $P = 0.9305$). In the PPS population, the difference was even more pronounced, with the low dose group showing a 5.75-day shorter time to symptom improvement compared to the placebo (median: 9.25 days [95% CI, 7.50–12.50] vs. 15.00 days [95% CI, 11.50–19.00]; $P = 0.0275$). These findings indicate that extending the evaluation period to Day 28 reinforces the initial results observed at Day 14, confirming that the low dose of CP-COV03 consistently accelerates symptom resolution relative to the placebo (Table 51 in Supplementary note-2).

In light of these results, the planned Phase 3 study will address the limitations of the current design by extending the follow-up period to 28 days to more accurately capture sustained symptom improvement.

CP-COV03, which increases the bioavailability of niclosamide, demonstrated efficacy against COVID-19 symptoms without serious safety issues and showed dose-dependent pharmacokinetics that correlated with viral load reduction. The low dose CP-COV03 significantly shortened the number of days required to improve the 12 symptoms of COVID-19 and sustain the effect for 48 h. These findings suggest the potential of CP-COV03 as an oral treatment option for COVID-19.

## Methods
### Trial design and randomization
This study was conducted in accordance with all relevant ethical regulations governing research involving human participants. The clinical trial protocol was reviewed and approved by the Institutional Review Boards (IRBs) of the following hospitals and institutions in South Korea: Bestian Hospital, Gimpo Woori Hospital, Korea University Ansan Hospital, Chungnam National University Sejong Hospital, Kyungpook National University Hospital, Keimyung University Dongsan Hospital, Hyundae General Hospital, Kyungpook National University Chilgok Hospital, Chosun University Hospital.

Written informed consent was obtained from all participants prior to their enrollment in the study. The trial was conducted in compliance with the principles outlined in the Declaration of Helsinki and adhered to Good Clinical Practice (GCP) guidelines.

The study was a randomized, double-blind, the placebo-controlled trial of multiple doses of CP-COV03 in 300 adults with mild or moderate COVID-19 (CRIS Registration number: KCT0007307). Screening tests, such as physical examination, chest imaging, electrocardiogram, and clinical laboratory tests, were performed to determine eligibility for non-hospitalized, symptomatic patients with a confirmed diagnosis of COVID-19 who voluntarily agreed to participate in this study by Day -1. Eligible patients were enrolled and assigned in the ratio of 1:1:1 to the low dose group, the high dose group, and the placebo (has the same size, color, weight, and appearance as the CP-COV03 capsule). The enrolled participants were mild to moderate COVID-19 patients who did not require hospitalization, but they were hospitalized based on clinical severity criteria for close observation and sample collection by trained nurses for the pharmacokinetics study and viral load measurements and were treated with the study drugs from Day 1 to Day 6 (supplementary Table S5).

Also, during the study period, individuals infected with COVID-19 in Korea were required to quarantine at home. To minimize unnecessary outings and movement in accordance with this guideline, the enrolled participants were hospitalized. The participants were allocated to take 9 capsules three times a day from Day 1 evening to Day 6 noon, for a total of 5 days. The study drug was taken approximately 8 h apart, typically before breakfast, 2 - 4 h after lunch, and 2 - 4 h after dinner, preferably on an empty stomach. The low dose group received 6 capsules of CP-COV03 50 mg and 3 capsules of the placebo, and the high dose group received 9 capsules of CP-COV03 50 mg.

The safety and efficacy assessments were conducted following a predefined protocol. Investigators interviewed all the participants in person or on the phone every day during hospitalization. After hospital discharge on Day 6, participants visited the study site on Days 8, 14, and 28 for the efficacy and safety assessments; all participants were not discharged on day 6 regardless of their symptoms. According to the protocol, the hospital discharge date could be extended based on symptoms or in line with the Korea Disease Control and Prevention Agency (KDCA) guidelines for COVID-19.

## Patients

The eligibility criteria of this trial are as follows: 1) an adult aged ≥ 19 years who voluntarily decided to participate and gave written (electronic) consent to abide by the precautions of the trial; 2) those who have confirmed SARS-CoV-2 infection within 3 days, and symptom onset within 5 days prior to randomization, with at least two symptoms rated 2 points or higher on the COVID-19 symptom scale at the time of randomization; 3) those who confirmed with COVID-19 infection through reverse transcription–polymerase chain reaction (PCR) test or expert rapid antigen test within three days from the date of randomization; 4) those with mild or moderate COVID-19 based on the US National Institutes of Health severity categories at the time of screening and randomization. Those who received other antiviral drugs or neutralizing antibody treatments for COVID-19 within 28 days of screening were excluded. Complete information of the eligibility criteria is described in the trial protocols (Supplementary information).

The participants were randomized in a 1:1:1 ratio into the low dose group, the high dose group, and the placebo group. Randomization was stratified based on age, severity, and consent for pharmacokinetic (PK) blood sampling. Additional PK blood sampling was conducted for 60 participants who consented to PK sampling (the low dose group: the high dose group: the placebo group = 20:20:20). PK sampling participants were not stratified by age or severity. An independent statistical team, separate from this clinical trial, generated the randomization table, and the investigational drugs were packaged according to this table. Participants were assigned to each group through the IWRS (Interactive Web Response System). The randomization method used was the block randomization method, with a block size of 6.

As of May 11, 2022, the start of participant enrollment, the vaccination rate for adults in Korea was 96.5%, so most participants likely received two or more doses.

## Trial oversight

The clinical trial was approved on May 2, 2022, by the Ministry of Food and Drug Safety (MFDS) of South Korea and by the institutional review board (IRB) at each trial site (IRB approval date for the first study site: May 9, 2022) before the start of recruitment (recruitment began on May 10, 2022). The trial was conducted in accordance with the Declaration of Helsinki and Good Clinical Practice (GCP) guidelines. The clinical trial plan was guided by the Korean MFDS in February 2022, because of the seriousness of the Omicron outbreak in Korea, for EUA. This clinical trial was designed by referencing the clinical trial protocols of Paxlovid® (NCT04960202) and ensitrelvir (jRCT2031210350) utilizing preliminary information such as the drug

administration days, efficacy analysis group, and end points. In January 2022, the sponsor submitted a phase 2 clinical trial plan to explore the safety and efficacy of CP-COV03 with a target of 80 participants (the placebo, 40; treatment, 40). However, the Korean MFDS recommended that the number of clinical trial participants should be increased to obtain confirmatory clinical results. Hence, the study population of the trial was increased to 300 participants (Supplementary methods, page 30). This number corresponds to the number of Asians who participated in the Paxlovid® clinical trial, which the Korean MFDS considered when determining its effectiveness for Koreans, before granting EUA for Paxlovid® in Korea on December 27, 2021. Additionally, it represents the top 30% of participants in phase 3 clinical trials conducted in Korea.

## Efficacy

The primary objective of the study was to assess the efficacy of CP-COV03 as compared with the placebo measured by the number of days required for targeted COVID-19 symptoms to improve and maintain for more than 48 h until Day 14. The time to sustained improvement (in days) was reported as a median value, with half-day intervals recorded based on symptom assessments conducted every morning and evening. Targeted COVID-19 symptoms included fever (38.0°C or above), cough, sore throat, headache, muscle aches, chills or shivering, stuffy or runny nose, tiredness or low energy, difficulty breathing or shortness of breath, nausea, vomiting, and diarrhea. In accordance with the FDA guideline ("Assessing COVID-19-related symptoms in outpatient adult and adolescent subjects in clinical trials of drugs and biological products for COVID-19 prevention or treatment"), symptoms were evaluated on a 4-point scale: 0 = absent, 1 = mild, 2 = moderate, and 3 = severe. Improvement was considered to have occurred in one of the following three cases: 1) if symptoms observed at baseline by 2 points or more have improved to 1 point or less; 2) if a symptom observed as 1 point at baseline has improved to 0 points; 3) symptoms that were not observed at the baseline but newly occurred during the clinical trial improved to 0 again. In other words, If COVID-19 symptoms improve and persist for at least 48 h but any symptom recurs and then improves again, the improvement date is recorded as the last date on which symptoms improved and persisted for 48 h. In case of recurrence, the sustained improvement time is counted from the start of treatment, not from the time of symptom recurrence. An example of improvement time evaluation is provided in Fig. S1. For participants whose symptoms did not improve until Day 14 or who were not satisfied with maintaining symptoms for more than 48 h after the symptom improvement until Day 14.5 (evening of Day 14), the number of days required for symptom improvement was calculated as 13, the maximum value that could appear in the analysis, and classified as censored.

The key secondary endpoint was change in SARS-CoV-2 viral load which was measured through quantitative PCR on Day 2, Day 4, Day 6, and Day 8. The pharmacokinetics of CP-COV03 at a steady state after repeated administration were assessed and the correlation between the blood concentration of CP-COV03 and the SARS-CoV-2 viral load was also evaluated. Blood samples were collected from 60 participants. It involved sampling two times on Day 1 (before and 3 h after evening administration), four times on Days 2 and 6 (before and 3 h after morning and noon administration), two times on Days 3 and 4 (before and 3 h after noon administration), and four times on Day 5 (before and 3 h after noon and evening administration). The other secondary endpoints and their results are presented in Table 1–50 excluding Tables 39–42 (viral load results) in the Supplementary note-2.

## Safety

The safety endpoints were safety profile of CP-COV03 compared with the placebo measured by incidence of adverse events, vital signs,

 

laboratory tests, thoracic (chest) imaging tests, and electro-cardiogram. The safety data were provided through Day 28 for each treatment group within safety analysis population, which included all participants who received at least one dose of the study drug or the placebo.

## Statistical analysis

The statistical analysis plan (version 2.0) was approved on February 21, 2023 by Hyundai Bioscience, Co. Ltd, Seoul, Republic of Korea, prior to the unblinding and analysis. From May 11, 2022, to November 28, 2022, 300 participants underwent randomization. The planned analysis groups for the efficacy evaluation of this clinical trial were ITT and PPS. All participants who received at least one dose of the study drug were included in the ITT, and the PPS comprised participants in the ITT population, except for those with a protocol violation that could affect the assessment of antiviral activity.

For additional analysis, the mITT-1 population included participants who took the study drug within 3 days of symptom onset. The mITT-2 population excluded participants from the mITT-1 population whose baseline PCR was negative or missing.

The primary endpoint for efficacy was assessed in the ITT, mITT-1, mITT-2, and PPS and the comparison between treatment groups was analyzed using a Cox Proportional Hazards Regression Model with treatment group as a factor and stratification factors (age, severity) as covariates. The hazard ratio for the treatment group, along with the corresponding 95% confidence interval and p-value, was presented. To ensure accurate measurement of the efficacy of the study drug, additional analyses for primary efficacy were conducted using mITT-1 and mITT-2, which involved censoring concomitant medications that could affect symptom assessment (concomitant medication administration day - 0.5 day). This censoring method, together with the Cox proportional hazards regression model, was also employed in an ad-hoc analysis designed to assess representative COVID-19 symptoms—such as fever, headache, and sore throat—while specifically excluding the gastrointestinal symptoms due to the high dose of MgO.

To analyse the key secondary endpoints, the changes in SARS-CoV-2 viral load, the SARS-CoV-2 viral load changes at each time point (Day 2, Day 4, Day 6, and Day 8) for each treatment group were presented. The comparison between treatment groups was analysed using an Analysis of covariance (ANCOVA) model with the treatment group as a factor, and baseline values and stratification factors (age, severity) as covariates.

The safety analysis was conducted on all participants who had received at least one dose of the study drug (SAS, Safety Analysis Set). Adverse reactions were standardized and analyzed by System Organ Class (SOC) and Preferred Term (PT), with differences between groups assessed using the Chi-square or Fisher's exact test. Significance was set at $p < 0.05$, and SAS Ver. 9.4 was used for analysis. More detailed statistical analysis methods are provided in the Supplementary Information (pages 30-31).

## Viral load analysis

Global Clinical Central Lab (GCCL), a certified Good Clinical Laboratory Practice (GCLP) facility, provided nasal swabs to each clinical hospital and collected nasal swabs from participants. To eliminate potential variability related to self-swabbing, trained nurses collected nasal swab samples from the participants in the afternoon on the scheduled days for viral load analysis. The nasal swab samples were stored under -20 °C or below condition before sent to GC Labs for sample processing and analysis. The samples were fully thawed under refrigerated conditions for analysis. For sample preparation, 5 μL of Internal Control A (STANDARD M nCoV Real-Time Detection kit, SD Biosensor, INC., Cat. No. M-NCOV-01) was dispensed into each well of the sample plate (Dxseq Viral Nucleic Acid Isolation Kit, DXOME CO., LTD., Cat. No. MVP-VIK01096). The samples were gently vortexed, and 200 μL of

each sample was dispensed into the wells of the sample plate. Nucleic acids were extracted using the KingFisherTM Flex Purification System (Thermo Fisher Scientific). To prepare the calibration curve, Reference RNA (Twist Synthetic SARS-CoV-2 RNA Control 2, TWIST BIOSCIENCE, Cat. No. 102024) was thawed and kept on ice. A mixture of 7.5 μL of Reference RNA and 52.2 μL of nuclease-free water (Thermo Fisher Scientific, Cat. No. R0581) was prepared to a final volume of 59.7 μL, yielding a final concentration of $1.0 \times 10^5$ copies/μL. This was further diluted to prepare calibration curve samples ($1.0 \times 10^5$, $1.0 \times 10^4$, $1.0 \times 10^3$, $1.0 \times 10^2$, $1.0 \times 10^1$, $5.0 \times 10^0$ copies/μL). On ice, 14 μL of 2019-nCoV Reaction Solution (STANDARD M nCoV Real-Time Detection kit, SD Biosensor, INC., Cat. No. M-NCOV-01) and 6 μL of Rtase Mix (STANDARD M nCoV Real-Time Detection kit, SD Biosensor, INC., Cat. No. M-NCOV-01) were mixed to prepare the PCR mixture. The prepared PCR mixture was then dispensed into a 96-well PCR plate, and Real-time PCR was performed using the CFX96TM Real-Time PCR Detection System (Bio-Rad Laboratories, Inc.). The viral load (copy number) of each sample was calculated based on $C_t$ values derived from a standard curve plotting $C_t$ values against log10. $C_t$ values and copy numbers were calculated using the average of duplicate measurements, and the average copy numbers were converted to logarithmic values (log10).

## Reporting summary

Further information on research design is available in the Nature Portfolio Reporting Summary linked to this article.

## Data availability

The datasets generated and analyzed during the current study are not publicly available due to clinical data privacy restrictions and ethical considerations involving patient confidentiality. Access to these clinical data may be granted for non-commercial academic research purposes upon reasonable request and is subjected to approval by the study sponsor, Hyundai Bioscience. Requests should be directed to the corresponding authors, Prof. Jin-Ho Choy (e-mail: jhchoy@dankook.ac.kr) and Prof. Jun Yong Choi (e-mail: seran@yuhs.ac). Requests will be evaluated within two weeks, and the data will be available for one month after approval. The source code used for the data analysis and modeling in this study is publicly available at GitHub: https://github.com/jhchoy1/CPCOV03-CODE.git. All other data supporting the findings of this study, including processed results, are available in the Supplementary Information.

## Code availability

The code related to the study are available at: https://github.com/jhchoy1/CPCOV03-CODE.git.

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

## Acknowledgements

We thank all the patients and medical staff at the study sites who participated in this clinical trial. This work was funded by Hyundai Bioscience Co., Ltd. and JHC was supported by the Research Grant 2024 of the National Academy of Sciences, Republic of Korea.

## Author contributions

J.H.K., S.K., S.W.K., D.W.P., K.T.W., J.W.S., were involved in the clinical trial analyses supervised by J.Y.C. along with G.W.J. and J.H.C. The material parts involved synthesis, characterization and analyses were done by S.Y., G.C. and N.S.R. supervised by J.H.C. All authors contributed to the interpretation of the results and have given approval to the final version of the manuscript.

## Competing interests

The authors declare no competing interests.

## Additional information

¹Department of Internal Medicine, Yonsei University College of Medicine, Seoul 03722, Republic of Korea. ²Division of Infectious Diseases, Department of Internal Medicine, Chungnam National University Sejong Hospital, Chungnam National University School of Medicine, Sejong 30099, Republic of Korea. ³Department of Internal Medicine, School of Medicine, Kyungpook National University, Daegu 41944, Republic of Korea. ⁴Division of Infectious Diseases, Department of Internal Medicine, Korea University College of Medicine, Korea University Ansan Hospital, 123, Jeokgeum-ro, Danwon-gu, Ansan-si, Gyeonggi-do 15355, Republic of Korea. ⁵Division of Infectious Diseases, Department of Internal Medicine, Kyungpook National University Chilgok Hospital, School of Medicine, Kyungpook National University, Daegu 41404, Republic of Korea. ⁶Division of Infectious Disease, Department of Internal Medicine, College of Medicine, Chosun University, 365, Pilmun-daero, Dong-gu, Gwangju Metropolitan City 61453, Republic of Korea. ⁷Department of Nanobiomedical Science, Dankook University, Cheonan 31116, Republic of Korea. ⁸Intelligent Nanohybrid Materials Laboratory (INML), Department of Chemistry, College of Science and Technology, Dankook University, Cheonan 31116, Republic of Korea. ⁹Division of Natural Sciences, The National Academy of Sciences, Seoul 06579, Republic of Korea. ¹⁰R&D Center, Hyundai Bioscience Co., LTD, Seoul 07790, Republic of Korea. ✉e-mail: jhchoy@dankook.ac.kr; seran@yuhs.ac

