## [Peer Review file · Nature Communications]

A randomized, double-blind, placebo-controlled trial of niclosamide nano hybrid for the treatment of patients with mild to moderate COVID-19

Corresponding Author: Professor JIN-HO CHOY

Version 0:

Reviewer comments:

Reviewer #1

(Remarks to the Author)

This is a moderately sized randomised controlled trial of an evaluation of an anti-viral treatment for mild COVID-19. It focuses on a surrogate endpoint of viral load reduction and global symptom improvement. It is limited by some important risks of bias and some unclear reporting.

Major comments:

1. The details of the outcome measurement definitions are not clear enough in terms of the precise definition of measurement of e.g. symptoms, the follow-up time point, and population within which it was planned to be assessed. Linked to this, which outcome measurements were pre-specified or planned as the primary efficacy outcome are not stated clearly. There are concerns as to whether the outcome reported have been selected from many analyses in the way the methods and results are currently presented. Which ones are pre-specified? Present the results for these clearly with treatment effect estimates and confidence intervals for the key assessments.
2. The randomisation method is not clearly stated in terms of sequence generation, allocation concealment and blocks, noting that there is perfect 1:1:1 randomisation across the 3 groups.
3. The discussion on why the symptom reduction difference is not seen in the higher dose group is unclear. It is hypothesised that it might be related to GI symptom duration but was there evidence for this? It didn't seem like it from the data available. Did the trial plan to evaluate all symptoms? Any sensitivity analysis of just e.g. fever, cough, shortness of breath?
4. The manuscript is much too long. For example, the introduction is 7 paragraphs. Not necessary to discuss drug companies by name and the final 2 paragraphs overstate the scope somewhat and start introducing results. The multiple populations described for different analyses are confusing - could stick to e.g. just the primary and secondary efficacy and safety outcomes in the main manuscript and present the results for one main analysis each of time to symptom improvement, viral load reduction, and adverse events. For example, the correlation analysis is not needed for the main manuscript.
5. The funding and potential conflicts of interest in the trial are not clear from the material. Who funded the trial? presumably the drug manufacturer? Were the authors linked to the company?

Minor comments:

1. Please identify this study as a randomised controlled trial in the title
2. The abstract mistakenly suggests this trial was in hospitalised patients but I believe the patients were outpatients who were admitted to hospital for the explicit purpose of the trial
3. The abstract does not state the viral load and symptom improvement results in a clear quantitative measure with confidence intervals - the 56% viral load reduction is unclear on the scale and whether this is relative to a baseline measurement and the symptom improvement numeric result is not stated

4. Was a sample size/power calculation done?

5. In order to judge the clinical utility of the treatment it is presumably required to evaluate for risk of hospitalisation or will treatment approval be given for symptom/viral load reduction alone? Why were the patients hospitalised for the trial? It also seems like an enormous number of tablets were needed for the treatment - 6 capsules three times a day for 5 days = 90 tablets for lower dose and 135 tablets for the higher dose - presumably this will be a major problem for treatment delivery?

6. No need for "childbearing potential" row in tables

7. Lots of results presented in figure captions - should be in main text instead?

Reviewer #2

(Remarks to the Author)

Thanks for sharing this interesting manuscript describing the results from a clinical trial on mild-to-moderated COVID-19.

Importance of the paper:

It has long been speculated if salicylamides such as niclosamide and nitazoxanide have the ability to provide significant levels of efficacy on clinical readouts in studies in the treatment of viral diseases such as influenza and COVID-19. Results so far show no significant effects on antiviral and clinical readouts in studies on influenza and nitazoxanide although positive trends was observed for some primary and secondary readouts in influenza and COVID-19 studies for nitazoxanide. Thus leaving the question on whether this class of compounds are useful in treatment of viral infections unanswered. It has been speculated that the probable cause of the poor outcomes is the low exposure level caused by suboptimal drug property profiles. For niclosamide poor solubility and low bioavailability is cited as major contributors.

The manuscript describes the study of CP-COV03, a newly developed magnesium oxide HPMC based nanohybrid formulation that significantly increases the plasma exposure of niclosamide. It is nicely demonstrated in the manuscript that the increased exposure leads to a significant decrease of the viral load as well as a significant improvement in symptom clearance at 300 mg TID dose level measured as a composite score of indicators. Importantly, there is no increase in adverse events observed upon multi-day dosing using the new formulation.

Comments:

- The authors comment on emergence of new SARS-CoV-2 variants, this is also mentioned as a potential, unlikely, source of variable efficacy impacting the study outcomes. It would be beneficial for the paper if the authors could comment on this and provide a reference of the SARS-CoV-2 variant efficacy for niclosamide or add own data in the supplementary section.

- The authors describe the limitations of other antiviral treatments for SARS-CoV-2 infection. I would recommend that this paragraph is reviewed and rewritten to address a number of points e.g. the major limitation relation for the nirmatrelvir is that it is co-dosed with the CYP inhibitor ritonavir, (nirmatrelvir + ritonavir provides the drug Paxlovid) to limited metabolic clearance, the use of ritonavir limits its use in certain patient populations. For molnupiravir there is a perceived risk for teratogenicity based on the molecular mechanism of action etc.

Please recheck the background information and update the description of limitations associate with the three antivirals mentioned.

- Please check the manuscript and ensure that the distinction between use of nirmatrelvir and the drug Paxlovid is clear (in clinical studies the drug is used not nirmatrelvir on its own)

- The authors comment on the niclosamide has a well-documented safety profile. With the new nanohybrid formulation the exposure increases significantly without any significant observations - this is a key point. Are there any reference values from previous studies that could be added to the introduction.

- The lack of improvement of the primary readout at the 450 mg dose is notable and difficult to explain considering the overlapping niclosamide exposure levels. There is and unsubstantiated mentioning of magnesium oxide contributing to this effect. I would like to see a more in-depth discussion and evidence on this topic with reference to previous studies confirming that a 50 % increase in the amount of magnesium oxide could contribute significantly. Does the placebo control contain magnesium oxide as well, which I would expect, if so the reasoning is not valid as all groups would receive the same amount of magnesium oxide. Would it be possible to correlate the effects (primary and secondary) to actual exposure levels in individual patients in the combined 300 and 450 mg study groups to gain better clarity on the link between efficacy and plasma exposure?

- There is no additional increase in the viral clearance upon dose increase increasing the dose from 300 to 450 mg the PK profiles seem to be the explanation for these result. Are there any additional data from previous human PK studies indicating the lack of a dose dependent increase of drug plasma exposure. There is also a slight reduction in exposure over time – what is known about induction of drug clearance over time?

- Please comment on the lack of balance in base-line characteristics for some of the parameters (diarrhea, nausea) seen in Table 1.

- Please add error bars to the plasma sample to understand if there is a significant difference in exposures between the two dose groups.

Reviewer #3

(Remarks to the Author)

Thank you for the opportunity to review this potentially important paper. In summary, I found this report difficult to review in the absence of a standard checklist which outlines the required items to be covered in a randomized control trial report. For example, I did not find a statistical analysis plan, the various populations analyst or somewhat confusing and it is not clear if these were pre specified. Trial processes are unclear. Including the pharmacology and development of the study drug adds complexity, and may be better put in a separate publication. Some suggestions to consider for improving this report include:

The title should be non-declamatory and preferably state the study design.

The introduction in the abstract needs reworking. For example, there is an implicit assumption that broad spectrum antiviral drugs are warranted because of the 'Surging long coronavirus disease 2019 cases.' We have no convincing evidence as yet that treating acute viral illnesses with antiviral drugs will diminish persisting symptoms.

Line 44: what broad-spectrum efficacy is niclosamide known for? Certainly, the evidence for this drug outside its parasitic indication is controversial. Has a systematic search been done for trials of niclosamide as a treatment for COVID-19?

Line 48 'trials' should be singular

Lines 51 to 52: 56% of what?

lines 57 and 58: precise estimates are available.

Line 80 Molnupiravir has not been found to be 'ineffective' for vaccinated individuals. Molnupiravir reduced time taken for recovery in vaccinated individuals. However, it did not reduce an already low hospital admission rate in a vaccinated population in the United Kingdom.

Lines 90 to 91: Suggest provide a brief summary of the data in the animal models which suggest antiviral activity.

Line 106: Best for the authors not to apply positive valued judgments to their own work.

Lines 1/21 to 124. Was there a statistical analysis plan that was approved before the analysis? Was the protocol registered and approved before the study began recruitment?

More detail is needed on the methods: for example, were there any contraindications to the drug that resulted in potential participants being ineligible? What was the sampling regimen and how a samples analyzed. What was the primary outcome measure? Was there a power calculation? What was the definition of sustained improvement? I noticed that a lot of this comes later, following the journal's tradition of putting the methods at the end of the discussion, but I've found it difficult to interpret the results without understanding what the primary outcome really was, and what populations the results were referring to. For example, excluding people who took medication who could influence outcome is a very unusual thing to do in a trial, especially if this was done post hoc. I could not find the list of medications that resulted in exclusion from the main analysis. Was the sub group analysis in the MIT population pre specified? Line 145: what is the ITT-1 population Line 157 what is the mITT2 population? Line 207: what is the PPS analysis

Line 160: How was the viral load determined, and when were participants swabbed, who took the swabs, and how were they analysed?

Line 201 it is unclear what a clinically designed study is: suggest do not use the abbreviation here of EUA (and other abbreviations), unless previously spelled out in full.

Under methods, we are told that enrolled participants were hospitalized: did this apply also to those who had mild illness? Why were they hospitalized? Were participants all hospitalised for trial purposes? If so, how many declined to participate because they did not wish to be hospitalised, especially since the majority had mild symptoms? Were all patients discharged on day six, regardless of their symptoms? What proportion completed the three additional follow up visits to the facilities? Even although the journal puts methods at the end of the report, explaining what the different study populations was in the report itself would be really helpful.

I'm sure I missed it but how many facilities contributed to the study and what was their nature?

Lines 342 to 347. This appears to be a retrospective justification for the sample size, rather than a prospective estimation, and the relevance of some of these considerations is questionable. This justification is inadequate. Was there a sample size calculation based on the pre specified primary outcome?

It is not clear what the scale is that is being referred to on lines 356 to 57. Is this a four point ordinal scale? I eventually found this in the appendix.

Line 379 to 8: the authors return to sample size here and although they say 300 participants is expected to be sufficient to evaluate the efficacy and safety of the drug, there is no actual rationale for this expectation. Stating that this was sufficient for other trials, which may or may not have been done in a similar population using the same outcome measure is not sufficient explanation for the chosen sample size.

Is this the first study of niclosamide for viral illnesses including COVID-19? We know that this is the first trial of this formulation, but what about of other formulations? Was a systematic search done? What were the previous findings

The consort diagram suggests that only 17 people failed screening: Of the 300 randomised, we have exactly 100 in each group. Is this exact division the result of good fortune or stratification?

Were the patients help seeking? How was the trial brought to their attention?

Study groups were not well balanced at baseline for all characteristics of note. For example those receiving 450 milligrams had more frequent nausea and diarrhea. This needs to be scrutinized and fully discussed.

There is a lot of repetition including of 265 -269

Line 298 The placebo should be described here: e.g. TID? Matched for colour and taste?

I found the description of eligibility check and randomisation a bit confusing. We are told, “ those who developed at least one of the COVID-19 symptoms within five days from the date of randomization and who had at least two or more symptoms of 2 points or more among the symptom scores determined on the day of randomization; 3) those who confirmed with COVID-19 infection through reverse transcription–polymerase chain reaction (PCR) test or expert rapid antigen test within three days from the date of randomization...” Does this mean that potential participants could have been asymptomatic when they were randomized? Could people have been randomized without a positive test of some sort for COVID-19?

Lines 383: “In the ITT-1 population, the symptom assessment data of participants, who were administered concomitant medications that may influence symptom evaluation, were censored at the day before the administration of the concomitant medication (concomitant medication administration day - 0.5 day) mITT, which involved censoring concomitant medications that could 391 affect symptom assessment.

. Was this pre specified? What worthy relevant concomitant medicines, was the list pre specified, and why was this done in the first place? The study was randomized and so additional medication use should have been balanced between groups. Excluding those who took additional drugs such as and anti inflammatories would introduce are likely bias. Were equal numbers excluded in all groups?

The per protocol set (PPS) population included participants in the safety population who had no important protocol deviations leading to exclusion from the per protocol population and had completed the day 28 follow-up visit

Suppl. 177. “Day 14.5, the number of days required for symptom improvement was calculated as 13, i.e., the maximum value that could appear in the analysis, and classified as censored”. This is irregular. Trials have found that up to 25% of people are still symptomatic at 28 days. Other trials have found that it takes a median of 16 days for Participants to feel recovered. How many were censored at this point?

The mIT-1 and mITT2 populations are virtually the same size and not that different from the ITT population in terms of size. Unless these populations were pre specified in a statistical analysis plan signed off before the end of recruitment and data log, I would leave these analysis for other outputs.

Version 1:

Reviewer comments:

Reviewer #1

(Remarks to the Author)

The authors have made considerable efforts to respond to concerns raised by reviewers. I feel they have addressed my concerns in the revisions.

(Remarks on code availability)

Reviewer #2

(Remarks to the Author)

Thanks for providing the revisions, questions asked have been answered.

Only one comment, the statement on line 70 regarding ensitrelvir having similar challenges as described for nirmatrelvir are not correct the PK challenges observed for nirmatrelvir does not apply for ensitrelvir. Also the reference 12 cite discusses risk for mutation not the exposure and safety concerns highlighted for nirmatrelvir and ensitrelvir respectively. Suggest removing the comment and reference.

(Remarks on code availability)

Reviewer #3

(Remarks to the Author)

Many thanks for asking me to review this updated version of the manuscript. It is much improved.

However, I am still concerned about the analysis of the primary outcome. As I understand it, this is a time to event analysis. The 'event' is improvement in targeted COVID-19 symptoms. If any one symptom did not reduce sufficiently and remain reduced for 48 hours, does that mean the endpoint was not met? What happened if symptoms reduced for 48 hours, and then rebounded, which is fairly typical in many cases of COVID-19?

In one analysis, the median time to recovery in a placebo group was 13 days. However, symptoms were only ascertained up to 14 days. This suggests that around half of the patients had not met the end point at the time of censoring.

I would like to see a table of the median time to events for all the symptoms, and then for any symptom.

The proportion of patients not meeting the end point on day 14, in other words patients who were still symptomatic when symptom data stopped being collected, should be made explicit in the text.

The fact that so many participants symptom data was censored at day 14 needs to be covered in the discussion.

Chris Butler

(Remarks on code availability)

Version 2:

Reviewer comments:

Reviewer #3

(Remarks to the Author)

Thank you for asking me to consider the responses of the authors to previous questions about this paper.

1. Responses to the query about whether all symptoms needed to be affected or whether the primary outcome would be made if a single symptom was affected, could still be made clearer. The response under paragraph 3B, it would be worth inserting ".... any of the COVID-19 symptoms we ascertained persisted ...".

2. I notice that Table 4 does not make it clear what the figures refer to: if these are the median number of days, then the term 'median'; should be there. The numbers of participants which each symptom ascertained are not present in the table. In line 1:30, the word median should be included.

Line 326' add in "for medical reasons". Presumably, not every single participant stayed in hospital for the full six days? The numbers who did complete the six day stay should be mentioned.

I remain critically concerned about the choice of primary outcome and reporting of secondary outcomes. The protocol makes it clear that symptoms were ascertained until day 28. However, the primary outcome truncates these symptoms at day 14. This is a potential weakness and should be discussed. Whatever the wisdom of this decision, from what I can tell, there is a considerable amount of information on recovery that is not reported. All secondary outcomes should be reported. Differences in recovery over the full 28 day are critically important. As secondary outcomes and as these data have been collected according to the protocol, these and any differences between study groups should be reported in this publication. Where are the findings for example, rescue medication, Secondary endpoints 9 and 10?

(Remarks on code availability)

Version 3:

Reviewer comments:

Reviewer #4

(Remarks to the Author)

Level ----- Section / Page ----- Quote ----- Comment

1:CRITICAL ----- Line 267 ----- "A possible explanation for this is the presence of magnesium oxide (MgO) in the 450 mg formulation, which may have caused gastrointestinal symptoms, thereby confounding participant-reported symptom evaluations." ----- As written, this seems to imply that the placebo did not match well at least one of the of the active groups. If true, this would be critical issue.

1:CRITICAL ----- Line 282 ----- "The higher proportion of certain COVID-19 symptoms at baseline in the 450 mg treatment group may have contributed to the lack of statistically significant improvement in symptoms observed in this group" ----- This reads like a failure of the randomisation system to ensure proper balance or the authors to consider all the pertinent issues ahead of randomisation. Or bad luck.

2:MAJOR ----- Line 274 ----- "In an ad-hoc analysis excluding the influence of MgO, th" ----- How does this work? Did I miss this in the Methods?

2:MAJOR ----- Throughout ----- n/a ----- The version I received is a mess of highlighting and red text with no explanation. What on Earth is going on? I haven't see this document before. Did I miss something?

2:MAJOR ----- Throughout ----- e.g. "CP-COV03 300 mg dose, 450 mg dose, and placebo groups" ----- The groups are misnamed. The Placebo group has placebo every time but one of the other groups has placebo for much of their tablets. This should be clearer. Or call it "No CP-COV3" and don't reference the blinding every moment. Or Higher, Lower and None.

3:CLARITY ----- Figures 2 and 3 ----- n/a ----- These could be better presented

3:CLARITY ----- Line 129 ----- "the primary outcome of the study was to assess the efficacy of CP-COV03 compared to placebo, measured by the median number of days" ----- No mention of median in the Methods? And is quarter and half days in the summary methods misleading when no one is able to have a half-day or quarter-day?

3:CLARITY ----- Line 368 ----- "patients who did not require hospitalization, but they were hospitalized for close observation" -
---- So, they did require hospitalisation?

3:CLARITY ----- Line 371 ----- n/a ----- Starts talking about D1 without defining it. From disease onset, from symptoms, from hospitalisation, from screening, from randomisation: which?

3:CLARITY ----- Line 371 ----- "The participants took 9 capsules three times a day from Day 1 evening to Day 6 noon, for a total of 5 days" ----- Did they take or were they allocated to take? It's the latter, isn't it?

3:CLARITY ----- Line 433 ----- "symptoms to improve and maintain for more than 48 hours until Day 14" ----- I find this difficult to whether it's time to the start of that 48 hours or the end of the 48 hours for the measurement. And what if symptoms improve on Day 13: does measurement continue? How does this work for censoring when there's no 48 hours to follow?

3:CLARITY ----- Line 454 ----- n/a ----- Why didn't bloods continue to Day 14?

3:CLARITY ----- Line 472 ----- "The statistical analysis plan (version 2.0) was approved" ----- Approved by who?

3:CLARITY ----- Line 478 ----- " , except for those with a protocol violation that could affect the assessment of antiviral activity." ----- Expand what is meant by protocol violation. Not just eligibility? Could the violation be more common on arm than that others?

3:CLARITY ----- Line 481 ----- "This clinical trial was designed by referencing the clinical trial protocols of Paxlovid® and ensitrelvir utilizing preliminary information such as the drug administration days, efficacy analysis group, and end points" ----- Give trial registration number

3:CLARITY ----- Line 483 ----- n/a ----- What's the time unit here? Days would be imprecise, perhaps, when only going to 14 days whereas hours would be spurious precision and open to possible measurement bias.

3:CLARITY ----- Table 2 ----- n/a ----- Can it be a TEAE if there was no treatment? (Placebo) Does these also include problems from Covid-19?

3:CLARITY ----- Throughout ----- n/a ----- What was the role of Rescue medication on outcome measure recording? Was this a protocol violation?

4:NOTE ----- e.g. Lines 175 & 235 ----- n/a ----- Bits of Results belong in Discussion and the Discussion has quite a lot of repetition of the Results

4:NOTE ----- Line 131 ----- “the median number of days required for the improvement of targeted COVID-19 symptoms was 9.0 days (95% CI, 7.00-10.00), 12.25 days (95% CI, 9.50-NR), and 13.0 days (95% CI, 10.50-NR) in the 300 mg dose, 450 mg dose, and placebo groups, respectively.” ----- The lower dose is doing better. Odd?

4:NOTE ----- Line 131 ----- “the median number of days required for the improvement of targeted COVID-19 symptoms was 9.0 days (95% CI, 7.00-10.00), 12.25 days (95% CI, 9.50-NR), and 13.0 days (95% CI, 10.50-NR) in the 300 mg dose, 450 mg dose, and placebo groups, respectively.” ----- What does NR mean and how is this helpful? Can't it be estimated by HR method?

4:NOTE ----- Line 82 ----- “Although niclosamide was well tolerated, but ended in failure, since there was no significant difference in the oropharyngeal clearance of SARS-CoV-2 between the placebo and niclosamide groups³¹.” ----- Not a fair definition of “failure” – the trial recruited and gave an answer. Just because it wasn't the answer they wanted, doesn't make the trial a failure.

4:NOTE ----- Throughout ----- n/a ----- Almost certainly an issue of the journal system but there's a lot of files here and it's really difficult to know what's in what: the file names don't match the cross-referencing.

5:TRIVIAL ----- Figure 1 ----- n/a ----- The groups are presented in a different order to in the text. Keep it the same. (Figure 1's order is better)

5:TRIVIAL ----- Intro ----- n/a ----- Feels too long

5:TRIVIAL ----- Line 357 ----- n/a ----- The Methods section is difficult to find

5:TRIVIAL ----- Line 377 ----- “The safety and efficacy assessments were conducted according to the study schedule.” ----- As opposed to what?

5:TRIVIAL ----- Title ----- n/a ----- No “and” needed

(Remarks on code availability)

They say it will be made available. I haven't looked at the code (not checked if it's actually there) as I had other priorities for the researchers to address.

Version 4:

Reviewer comments:

Reviewer #4

(Remarks to the Author)

I appreciated the clarifications from the authors. It was interesting to know that some of the answers had already been in the manuscript, but had got lost in the complex wording and presentation. I have gone through and offer some further comments.

The imbalance in individual symptoms could be made reassuring by making clear in the Methods that the severity score accounts across all of the symptoms and by putting severity (overall) above individual symptoms in Table 1.

The Results call out Table 47 of the Supplement. There's more than 50 tables. I'm not going to read this in detail: it's impossible for this reviewer. Most tables in the Supplement are not referenced from the paper.

I did not engage much with the Discussion last time as I had too many questions to be sure I would understand it. At 1,600 words, it's way too long and seems to repeat a lot of the methods and results.

Giving “not determined” as an upper bound for median time is unhelpful. This can be estimated through modelling, but it is always better to read up from a pre-specified time on the x-axis (which will always given bounds to the estimate) than across from the y-axis (which can give the same problem the researchers have here).

The edits have made it mostly tidier and easier to read but there are still points of inconsistency to make me unsure about my understanding:

In labelling of allocated groups every time

Trial vs study (particularly in “completed the ...”)

Outcome (prognosis) vs outcome measure (how the prognosis was measured) vs endpoint (or event – the key (bad) thing being looked before in the outcome measure)

A reminder is needed in the Results and Discussion that the smallest unit of time in the time-to-event analyses is a half-day.

Colours of groups in Figure 3 could usefully match the allocated colour in Figure 2

The use of middle-endian date formats (month first) is anti-scientific but I think this is a problem with the journal rather than the authors. Similarly, putting the Methods after the Results feels like an excuse to finesse what happened: again, a “feature” of the journal, rather than a problem by the authors, I suspect.

The use of “dropout” sounds careless. Talk about early cessation of participation. Similar, “completed the study” is complex, because I think they mean the participant completed the maximum amount of time they needed to contribute which isn’t directly related to the time required by the researchers to complete the study overall.

(Remarks on code availability)

The images or other third party material in this Peer Review File are included in the article’s Creative Commons license, unless indicated otherwise in a credit line to the material. If material is not included in the article’s Creative Commons license and your intended use is not permitted by statutory regulation or exceeds the permitted use, you will need to obtain permission directly from the copyright holder.

Authors point by point responses to Reviewer's comments

Authors point by point responses to Reviewer #1:

This a moderately sized randomized controlled trial of an evaluation of an anti-viral treatment for mild COVID-19. It focuses on a surrogate endpoint of viral load reduction and global symptom improvement. It is limited by some important risks of bias and some unclear reporting.

Authors point by point responses to Major comments:

1. The details of the outcome measurement definitions are not clear enough in terms of the precise definition of measurement of **a)** e.g. symptoms, the follow-up time point, and population within which it was planned to be assessed. **b)** Linked to this, which outcome measurements were **pre-specified or planned as the primary efficacy outcome** are not stated clearly. There are concerns as to whether the outcome reported have been selected from many analyses in the way the methods and results are currently presented. Which ones are pre-specified? **c)** Present the results for these clearly with treatment effect estimates and confidence intervals for the key assessments.

Response by the authors:

1a) Thank you for your feedback. We acknowledge that the outcome measurement definitions has been already clarified in the revised manuscript. Additional information on **definition of symptom measurement, the exact follow-up time points, and the specific population** in which these assessments are planned are detailed as follows;

Definition of symptom measurement, the exact follow-up time points:

Efficacy

The primary objective of the study was to assess the efficacy of CP-COV03 as compared with placebo measured by the number of days required for targeted COVID-19 symptoms to improve and maintain for more than 48 hours until Day 14. Targeted COVID-19 symptoms included fever (38.0°C or above), cough, sore throat, headache, muscle aches, chills or shivering, stuffy or runny nose, tiredness or low energy, difficulty breathing or shortness of breath, nausea, vomiting, and diarrhea. In accordance with the FDA guideline ("Assessing COVID-19-related symptoms in outpatient adult and adolescent subjects in clinical trials of drugs and biological products for COVID-19 prevention or treatment"), symptoms were evaluated on a 4-point scale: 0 = absent, 1 = mild, 2 = moderate, and 3 = severe. Improvement was considered to have occurred in one of the following three cases: 1) if symptoms observed at baseline by 2 points or more have improved to 1 point or less; 2) if a symptom observed as 1 point at baseline has improved to 0 points; 3) symptoms that were not observed at the baseline but newly occurred during the clinical trial improved to 0 again. For participants whose symptoms did not improve until Day 14 or who were not satisfied with maintaining symptoms for more than 48 hours after the symptom improvement until Day 14.5 (evening of Day 14), the number of days required for symptom improvement was calculated as 13, the maximum value that could appear in the analysis, and classified as censored. (Refer pages 15-16, lines 361-380)

The specific population:

The primary outcome of the study was to assess the efficacy of CP-COV03 compared to placebo, measured by the number of days required for targeted COVID-19 symptoms to improve and be maintained for more than 48 hours until Day 14. In the PPS group, the part of planned analysis, which included participants enrolled within 5 days of COVID-19 symptom onset and with no major protocol deviations, the median number of days required for the improvement of targeted COVID-19 symptoms was 9.0 days (95% CI, 7.00-10.00), 12.25 days (95% CI, 9.50-NR), and 13.0 days (95% CI, 10.50-NR) in the 300 mg dose, 450 mg dose, and placebo groups, respectively. A statistically significant difference ($P = 0.0083$) was observed in the 300 mg dose group compared to the placebo group. In another planned analysis group, the ITT population, the median number of days required for all 12 COVID-19 symptoms to show sustained improvement for more than 48 hours was 10.0 (95% CI, 8.50-12.50), 12.5 (95% CI, 10.50-NR), and 12.25 days (95% CI, 10.50-NR) in the 300 mg, 450 mg, and placebo groups, respectively. Although the median time in the 300 mg dose group was shorter than the placebo group, the difference was not statistically significant. In the additional analysis group, the mITT-1 population, which included participants who received the study drug within 3 days of symptom onset, the median number of days required for targeted COVID-19 symptoms to improve was 9.0 (95% CI, 7.50-10.50), 12.5 (95% CI, 10.00-NR), and 12.5 days (95% CI, 10.50-NR) in the 300 mg dose, 450 mg dose, and placebo groups, respectively. The 300 mg dose group showed a statistically significant reduction in the number of days required for symptom improvement compared to the placebo group ($P=0.024$). This was also confirmed in the mITT-2 population, where the median number of days required for targeted COVID-19 symptoms to improve was 9.0 (95% CI, 7.50-10.50), 12.5 (95% CI, 10.00-NR), and 12.5 (95% CI, 10.50-NR) days in the 300 mg dose, 450 mg dose, and placebo groups, respectively, with a statistically significant difference ($P=0.0275$) observed between the 300 mg dose group and the placebo group. This result was consistent in the subgroup analysis targeting the high-risk group (age 60 and above, patients with obesity, chronic conditions such as diabetes or hypertension, immunocompromised individuals, or those on long-term immune-suppressant therapy). In this high-risk group, the time required for symptom improvement was 7.5 days (95% CI, 7.00-9.00) in those receiving the 300 mg dose, which was significantly shorter than the 12.5 days (95% CI, 8.00-NR) required for the placebo group ($P=0.017$). (Refer pages 5-7, lines 124-157)

1b) Additionally, specific outcome measurements that were pre-specified or planned as the primary efficacy outcome have been stated clearly as follows;

Please refer to the response to the referees comment 1a) as described above. (Refer pages 5-7, lines 124-157)

In addition, the planned and additional analysis results were separately described as below;

Analysis of the primary outcome—the time required for the improvement of targeted COVID-19 symptoms—revealed that in the planned analysis of the PPS group, the placebo group required 13.0 days (95% CI, 10.50-NR), while the 300 mg CP-COV03 group required 9.0 days (95% CI, 7.00-10.00), a statistically significant difference ($P=0.0083$). In the ITT population, the median time for the resolution of all 12 COVID-19 symptoms was 10.0 days (95% CI, 8.50-12.50) for the

300 mg group, compared to 12.25 days (95% CI, 10.50-NR) for the placebo one; however, this difference was not statistically significant.

In the additional analysis of the mITT-1 population, which included participants who received the study drug within 3 days of symptom onset, the median time to improvement of targeted symptoms was 9.0 days (95% CI, 7.50-10.50) in the 300 mg dose group, compared to 12.5 days (95% CI, 10.50-NR) in the placebo one, with the difference being statistically significant (P=0.024). (Refer page 9, lines 218-225)

1c) Present the results for these clearly with treatment effect estimates and confidence intervals for the key assessments;

As clearly described in the Efficacy and discussion parts, confidence intervals are given in all key assessments as below;

Efficacy:

The primary outcome of the study was to assess the efficacy of CP-COV03 compared to placebo, measured by the number of days required for targeted COVID-19 symptoms to improve and be maintained for more than 48 hours until Day 14. In the PPS group, the part of planned analysis, which included participants enrolled within 5 days of COVID-19 symptom onset and with no major protocol deviations, the median number of days required for the improvement of targeted COVID-19 symptoms was 9.0 days (95% CI, 7.00-10.00), 12.25 days (95% CI, 9.50-NR), and 13.0 days (95% CI, 10.50-NR) in the 300 mg dose, 450 mg dose, and placebo groups, respectively. A statistically significant difference (P = 0.0083) was observed in the 300 mg dose group compared to the placebo group. In another planned analysis group, the ITT population, the median number of days required for all 12 COVID-19 symptoms to show sustained improvement for more than 48 hours was 10.0 (95% CI, 8.50-12.50), 12.5 (95% CI, 10.50-NR), and 12.25 days (95% CI, 10.50-NR) in the 300 mg, 450 mg, and placebo groups, respectively. Although the median time in the 300 mg dose group was shorter than the placebo group, the difference was not statistically significant. In the additional analysis group, the mITT-1 population, which included participants who received the study drug within 3 days of symptom onset, the median number of days required for targeted COVID-19 symptoms to improve was 9.0 (95% CI, 7.50-10.50), 12.5 (95% CI, 10.00-NR), and 12.5 days (95% CI, 10.50-NR) in the 300 mg dose, 450 mg dose, and placebo groups, respectively. The 300 mg dose group showed a statistically significant reduction in the number of days required for symptom improvement compared to the placebo group (P=0.024). This was also confirmed in the mITT-2 population, where the median number of days required for targeted COVID-19 symptoms to improve was 9.0 (95% CI, 7.50-10.50), 12.5 (95% CI, 10.00-NR), and 12.5 (95% CI, 10.50-NR) days in the 300 mg dose, 450 mg dose, and placebo groups, respectively, with a statistically significant difference (P=0.0275) observed between the 300 mg dose group and the placebo group. This result was consistent in the subgroup analysis targeting the high-risk group (age 60 and above, patients with obesity, chronic conditions such as diabetes or hypertension, immunocompromised individuals, or those on long-term immune-suppressant therapy). In this high-risk group, the time required for symptom improvement was 7.5 days (95% CI, 7.00-9.00) in those receiving the 300 mg dose, which was significantly shorter than the 12.5 days (95% CI, 8.00-NR) required for the placebo group (P=0.017). (Refer pages 5-7, lines 124-157)

Discussion:

Analysis of the primary outcome—the time required for the improvement of targeted COVID-19 symptoms—revealed that in the planned analysis of the PPS group, the placebo group required 13.0 days (95% CI, 10.50-NR), while the 300 mg CP-COV03 group required 9.0 days (95% CI, 7.00-10.00), a statistically significant difference ($P=0.0083$). In the ITT population, the median time for the resolution of all 12 COVID-19 symptoms was 10.0 days (95% CI, 8.50-12.50) for the 300 mg group, compared to 12.25 days (95% CI, 10.50-NR) for the placebo one; however, this difference was not statistically significant.

In the additional analysis of the mITT-1 population, which included participants who received the study drug within 3 days of symptom onset, the median time to improvement of targeted symptoms was 9.0 days (95% CI, 7.50-10.50) in the 300 mg dose group, compared to 12.5 days (95% CI, 10.50-NR) in the placebo one, with the difference being statistically significant ($P=0.024$).

In the mITT-1 population, the subgroup analysis of the high-risk group (participants aged 60 years and above, or those with obesity, chronic conditions such as diabetes or hypertension, immunocompromised individuals, or long-term users of immunosuppressants) showed a significantly shorter time to symptom improvement in the 300 mg CP-COV03 group than the placebo one. Specifically, the median time for symptom improvement was 7.5 days (95% CI, 7.00-9.00) in the 300 mg group, compared to 12.5 days (95% CI, 8.00-NR) in the placebo one ($P=0.017$). These findings surpass the reported efficacy of Paxlovid®, which reduced symptom resolution time by 3 days in high-risk patients^{28,29}, and ensitrelvir, which reduced the time to improvement in five symptoms by 1 day in the general population.^{8,30}

However, the 450 mg dose group did not show a statistically significant improvement in symptoms (PPS analysis: 12.25 days [95% CI, 9.50-NR] for the 450 mg dose group vs. 13.0 days [95% CI, 10.50-NR] for the placebo one). A possible explanation for this is the presence of magnesium oxide (MgO) in the 450 mg formulation, which may have caused gastrointestinal symptoms, thereby confounding participant-reported symptom evaluations. The daily MgO intake in the 450 mg group was 945 mg, higher than the 630 mg in the 300 mg group and exceeding the recommended daily intake of 800 mg. Excessive MgO intake is associated with gastrointestinal symptoms such as diarrhea, nausea, and vomiting, which overlap with the targeted COVID-19 symptoms in this trial.³¹

In an ad-hoc analysis excluding the influence of MgO, the median time for improvement of three representative COVID-19 symptoms (fever, headache, and sore throat) was 4.0 days (95% CI, 3.5-4.5), 4.5 days (95% CI, 4.0-5.0), and 6.0 days (95% CI, 4.0-7.5) in the 300 mg, 450 mg, and placebo groups, respectively. Both the 300 mg and 450 mg dose groups demonstrated statistically significant reductions in the time required for symptom improvement compared to the placebo one ($P=0.011$ and $P=0.044$ for the 300 mg and 450 mg doses, respectively), suggesting that MgO confounded the results for the 450 mg group by affecting gastrointestinal symptoms. (Refer page 9-11, lines 218-258)

2. The randomization method is not clearly stated in terms of sequence generation, allocation concealment and blocks, noting that there is perfect 1:1:1 randomisation across the 3 groups.

Response by the authors:

The randomization method has been clearly stated in the revised manuscript as follows;

The participants were randomized in a 1:1:1 ratio into 300mg dose group, 450mg dose group, and the placebo group. Randomization was stratified based on age, severity, and consent for pharmacokinetic (PK) blood sampling. Additional PK blood sampling was conducted for 60 participants who consented to PK sampling (300mg dose group: 450mg dose group: placebo = 20:20:20). PK sampling participants were not stratified by age or severity. An independent statistical team, separate from this clinical trial, generated the randomization table, and the investigational drugs were packaged according to this table. Participants were assigned to each group through the IWRS (Interactive Web Response System). The randomization method used was the block randomization method, with a block size of 6. (Refer page 14, lines 329-338)

3. The discussion on why the symptom reduction difference is not seen in the higher dose group is unclear. **a)** It is hypothesised that it might be related to GI symptom duration but was there evidence for this? It didn't seem like it from the data available. **b)** Did the trial plan to evaluate all symptoms? Any sensitivity analysis of just e.g. fever, cough, shortness of breath?

Response by the authors:

3a) In the discussion part, we calculated the amount of MgO included in both the 300mg and 450mg doses and added an explanation referencing the gastrointestinal symptoms associated with the daily intake of MgO (945mg) in the 450mg CP-COV03 group. Additionally, through further analysis, we demonstrated that symptom improvement, excluding gastrointestinal symptoms caused by excess MgO intake, was statistically significant not only in the 300mg group but also in the 450mg group as below;

However, the 450 mg dose group did not show a statistically significant improvement in symptoms (PPS analysis: 12.25 days [95% CI, 9.50-NR] for the 450 mg dose group vs. 13.0 days [95% CI, 10.50-NR] for the placebo one). A possible explanation for this is the presence of magnesium oxide (MgO) in the 450 mg formulation, which may have caused gastrointestinal symptoms, thereby confounding participant-reported symptom evaluations. The daily MgO intake in the 450 mg group was 945 mg, higher than the 630 mg in the 300 mg group and exceeding the recommended daily intake of 800 mg. Excessive MgO intake is associated with gastrointestinal symptoms such as diarrhea, nausea, and vomiting, which overlap with the targeted COVID-19 symptoms in this trial³¹.

In an ad-hoc analysis excluding the influence of MgO, the median time for improvement of three representative COVID-19 symptoms (fever, headache, and sore throat) was 4.0 days (95% CI, 3.5-4.5), 4.5 days (95% CI, 4.0-5.0), and 6.0 days (95% CI, 4.0-7.5) in the 300 mg, 450 mg, and placebo groups, respectively. Both the 300 mg and 450 mg dose groups demonstrated statistically significant reductions in the time required for symptom improvement compared to the placebo one (P=0.011 and P=0.044 for the 300 mg and 450 mg doses, respectively), suggesting that MgO

confounded the results for the 450 mg group by affecting gastrointestinal symptoms. (Refer pages 10-11, lines 241-258)

3b) As mentioned in the response to comment **1a**, in the efficacy section, we measured the time required for overall improvement of the 12 COVID-19 symptoms outlined in the FDA guidelines. The definition of improvement is also provided in the efficacy section. A sensitivity analysis was not conducted separately (Refer pages 15-16, lines 361-380)

4. The manuscript is much too long. **a)** For example, the introduction is 7 paragraphs. Not necessary to discuss drug companies by name and the final 2 paragraphs overstate the scope somewhat and start introducing results. The multiple populations described for different analyses are confusing - could stick to e.g. **b)** just the primary and secondary efficacy and safety outcomes in the main manuscript and present the results for one main analysis each of time to symptom improvement, viral load reduction, and adverse events. **c)** For example, the correlation analysis is not needed for the main manuscript.

Response by the authors:

Thank you for your valuable feedback. According to the reviewer's comment, we have revised the manuscript to make it more concise. As suggested, minor details such as the drug company name have been removed, and the results and discussion parts have been reorganized to focus primarily on the primary, secondary efficacy, and safety outcomes.

4a) Introduction: We have totally reformulated introduction, and reduced from 688 to 531 words, to get a more concise overview. Unnecessary mentions of specific drug companies have also been removed; (Refer Pages 3-5, lines 57-107).

4b) Results and discussion: In the results section (efficacy, viral load, pharmacokinetic parameters, and their association, safety) and the discussion part, we have revised the manuscript to focus on the primary, secondary efficacy, and safety outcomes, as suggested by the reviewer, making it more concise; (Refer pages 5-8, lines 124-197).

4c) Correlation Analysis: To further interpret and support the primary outcome results from multiple perspectives, we have included the mITT results in the manuscript, particularly reflecting outcomes from participants who received the drug within 3 days of symptom onset.

5. The funding and potential conflicts of interest in the trial are not clear from the material. Who funded the trial? presumably the drug manufacturer? Were the authors linked to the company?

Response by the authors:

The detailed information on funding and potential conflicts of interest in the trial have been provided as follows: (**Refer Page 26, lines 635-640**). Authors declare there are no conflict of interests.

Acknowledgements

We thank all the patients and medical staffs at the study sites who participated in this clinical trial. This work was funded by Hyundai Bioscience Co., Ltd and JHC is grateful to the National Academy of Sciences, Republic of Korea, that supported this research in the form of the 2024 Research Participation Grant on the International Academic Organization Research Project.

Authors' response to the Minor comments from Reviewer-1:

1. Please identify this study as a randomized controlled trial in the title.

Response by the authors:

The title was revised from

“Niclosamide based nano hybrid system for the treatment of mild to moderate COVID-19 patients: Rapid viral load reduction and symptom alleviation”

to

“A randomized, double-blind and placebo-controlled trial of niclosamide nano hybrid for the treatment of patients with mild to moderate COVID-19”. (**Refer page 1, lines 1-2**)

2. The abstract mistakenly suggests this trial was in hospitalised patients but I believe the patients were outpatients who were admitted to hospital for the explicit purpose of the trial.

Response by the authors: The mild to moderate COVID-19 patients in this clinical trial did not require hospitalization for medical reasons. However, they were hospitalized to ensure that trained nurses could accurately collect samples and closely observe the patients. The original abstract referred to "hospitalized patients," which may have caused some misunderstanding, but the abstract has been appropriately revised.

Abstract

Coronavirus disease 2019 (COVID-19), caused by severe acute respiratory syndrome coronavirus 2 (SARS-CoV-2), has precipitated profound global social and economic disruptions and continues to drive high rates of infection. Amid this ongoing crisis, there is a persistent and urgent demand

for an effective and reliable treatment for COVID-19. As part of the efforts to develop the treatment, niclosamide has been considered as a potential drug candidate for SARS-CoV-2 according to the in vitro studies. However, no clinical trials were successful due to its intrinsic properties like low solubility and bioavailability. In the present study, these limitations have been successfully addressed by developing a new niclosamide nano hybrid (CP-COV03), demonstrating significant antiviral activity in clinical trial for COVID-19; the findings from a randomized, double-blind and placebo-controlled clinical trial involving 300 patients (Clinical Trial Registration Number: KCT0007307) that assessed the efficacy and safety of CP-COV03 have been reported. The results highlight that the present nano hybrid drug significantly alleviated 12 FDA-recommended COVID-19 symptoms with sustained improvement (symptom improvement maintained for more than 48 hours) and reduced viral load by 56.7% compared to baseline within 16 hours of the initial dose. Importantly, no serious adverse events were reported across all treatment groups during the trial. (Refer page 2, lines 38-56)

The enrolled participants were mild to moderate COVID-19 patients who did not require hospitalization, but they were hospitalized for close observation and sample collection by trained nurses for the pharmacokinetics study and viral load measurements and were treated with the study drugs from Day 1 to Day 6 (Table S3). (Refer page 12, lines 298-302)

3. The abstract does not state the viral load and symptom improvement results in a clear quantitative measure with confidence intervals - the 56% viral load reduction is unclear on the scale and whether this is relative to a baseline measurement and the symptom improvement numeric result is not stated

Response by the authors: As recommended by the reviewer, the abstract has been revised to ensure the results are presented more clearly and quantitatively (56.7% compared to baseline) in the Abstract. (Refer page 2, lines 38-56). Regarding the symptom improvement results, please refer to Response for 1c, stating the numeric results including their confidence interval in efficacy part (Refer pages 5-7, lines 124-157), and in discussion part (pages 9-11, lines 218-258).

4. Was a sample size/power calculation done?

Response by the authors:

The method we used to calculate the sample size/power is as follows;

At the time of planning this clinical trial, data on the progression to severe COVID-19 infection was available, but there was no reference data on symptom improvement. Therefore, we internally calculated the sample size for the clinical trial by referring to the symptom improvement effect of oseltamivir in influenza infection.

For reference, Paxlovid's sample size calculation was also based on the clinical trial (BLAZE-1) of the antibody treatment Bamlanivimab (Pfizer, EPIC-HR protocol, protocol number: C4671005).

According to Table 2 from previous oseltamivir (Tamiflu) studies (Treanor, John J., et al. "Efficacy and safety of the oral neuraminidase inhibitor oseltamivir in treating acute influenza: a randomized controlled trial." JAMA 283.8 (2000): 1016-1024), the duration of symptoms in the placebo group

(129 participants) was a mean of 103.3 hours, with a 95% confidence interval of 92.6–118.7, while in the treatment group (121 participants), the mean was 69.9 hours, with a 95% confidence interval of 60.0–87.9. Based on these results, the standard deviations were calculated as approximately 75.62 hours for the placebo group and 78.29 hours for the treatment group.

The effect size (Cohen's *d*) was approximately 0.434. Considering a significance level of 5% and a power of 90%, the required sample size for the clinical trial was calculated to be 92 participants per group. Therefore, setting the sample size for this clinical trial at 100 participants per group is appropriate. **(Refer to supplementary file page 30, statistical analysis).**

5. In order to judge the clinical utility of the treatment it is presumably required to evaluate for risk of hospitalisation or will treatment approval be given for symptom/viral load reduction alone? Why were the patients hospitalised for the trial? It also seems like an enormous number of tablets were needed for the treatment - 6 capsules three times a day for 5 days = 90 tablets for lower dose and 135 tablets for the higher dose - presumably this will be a major problem for treatment delivery?

Response by the authors:

To judge the clinical utility, the symptom alleviations can be evaluated;

In accordance with the FDA guideline ("Assessing COVID-19-related symptoms in outpatient adult and adolescent subjects in clinical trials of drugs and biological products for COVID-19 prevention or treatment"), symptoms were evaluated on a 4-point scale: 0 = absent, 1 = mild, 2 = moderate, and 3 = severe. **(Refer page 15, lines 367-371)**

The purpose of hospitalization of the enrolled participants has been given in the revised manuscript as follows:

The enrolled participants were mild to moderate COVID-19 patients who did not require hospitalization, but they were hospitalized for close observation and sample collection by trained nurses for the pharmacokinetics study and viral load measurements and were treated with the study drugs from Day 1 to Day 6 (Table S3). **(Refer page 12, lines 298-302)**

Further to comment on the Capsule use as Reviewer 1 pointed out;

Administering 90–135 capsules is a considerable amount. We acknowledge that this could pose practical challenges, especially for elderly patients who often struggle with swallowing larger capsules. However, it's important to note that we used small capsules in this study, which helped mitigate these issues, and there were no reported difficulties with administration due to capsule size. Nonetheless, we recognize the need for more practical dosing options in future formulations to improve ease of use and adherence. (we used capsule #4 for CP-COV03)

Size	Length(mm)								Outside Diameter(mm)				Volume(ml)				Weight(mg)	
	Gelatin Capsules				HPMC Capsules				Gelatin Capsules		HPMC Capsules		Gelatin Capsules		HPMC Capsules		Gelatin Capsules	HPMC Capsules
	Cap	Body	Open L. (Empty)	Closed L. (Filled)	Cap	Body	Open L. (Empty)	Closed L. (Filled)	Cap	Body	Cap	Body	Cap	Body	Cap	Body		
#000	12.9	22.2	28.1	26.1	12.9	22.2	28.1	26.1	9.91	9.55	9.91	9.55	0.19	1.37	0.19	1.37	163	160
#00EL	12.9	22.2	27.5	25.3	13.0	22.2	27.5	25.3	8.52	8.16	8.53	8.18	0.15	1.04	0.15	1.04	140	132
#00	11.8	20.2	25.5	23.3	11.7	20.2	25.5	23.3	8.52	8.16	8.53	8.18	0.15	0.95	0.15	0.91	125	123
#0EL	12.0	20.7	25.6	23.6	12.0	20.7	25.6	23.6	7.65	7.33	7.65	7.33	0.11	0.79	0.11	0.79	110	106
#0	11.1	18.5	23.4	21.4	10.7	18.4	23.6	21.7	7.64	7.33	7.64	7.34	0.11	0.68	0.11	0.68	97	95
#1EL	10.5	17.7	22.4	20.4	10.5	17.7	22.4	20.4	6.91	6.63	6.91	6.63	0.08	0.54	0.08	0.54	81	79
#1	9.7	16.5	20.9	19.1	9.8	16.6	21.3	19.4	6.91	6.63	6.91	6.63	0.08	0.47	0.08	0.50	77	75
#1SL	9.78	13.95	18.6	16.7	9.78	13.95	18.6	16.7	6.91	6.63	6.91	6.63	0.08	0.40	0.08	0.40	69	67
#2EL	9.7	16.7	21.0	19.3	9.7	16.7	21.0	19.3	6.35	6.09	6.35	6.09	0.07	0.41	0.07	0.41	68	65
#2	9.2	15.3	19.2	17.6	8.9	15.3	19.6	18.0	6.35	6.07	6.35	6.07	0.06	0.37	0.06	0.37	64	62
#2SL	8.9	13.3	17.65	16.0	8.9	13.3	17.65	16.0	6.35	6.07	6.35	6.07	0.06	0.31	0.06	0.31	60	58
#3	8.2	13.5	17.2	15.7	8.1	13.6	17.7	15.9	5.83	5.57	5.82	5.56	0.05	0.27	0.05	0.30	50	48
#4	7.4	12.2	15.6	14.2	7.2	12.2	16.0	14.3	5.32	5.05	5.31	5.05	0.03	0.20	0.03	0.21	40	38
#5	6.2	9.3	12.8	11.1	6.2	9.3	12.8	11.1	4.91	4.68	4.91	4.68	0.02	0.13	0.02	0.13	28	26
Tolerance	±0.4	±0.4	±0.5	±0.4	±0.4	±0.4	±0.5	±0.4	±0.06	±0.06	±0.06	±0.06	±3%	±3%	±3%	±3%	±10%	±11%

6. No need for "childbearing potential" row in tables

Response by the authors:

We completely agree with your opinion. And it has been removed from the Table 1 in main text.

7. Lots of results presented in figure captions

Response by the authors:

According to the referee's comment, we revised all the figure captions (1, 2, and 3) to make them more concise as follows;

- 1) **Fig. 1. CONSORT diagram for clinical trial.** Shown is the study flow chart for randomized, double-blind, placebo-controlled trial of multiple doses of CP-COV03 in mild or moderate COVID-19.
- 2) **Fig. 2. Viral load and pharmacokinetic analyses.** (A) The adjusted mean change in viral load from baseline of Severe Acute Respiratory Syndrome Coronavirus 2 (SARS-CoV-2) monitored starting from Day 0. Any data point that is more than 3 times the IQR above the third quartile or below the first quartile is an outlier (n = 77 for placebo, n = 70 for 300 mg, n = 80 for 450 mg). Error bars represent standard error (S.E.). (B) Pharmacokinetic profiles of niclosamide from clinical trial (n = 20 for placebo, n = 20 for 300 mg, n = 18 for 450 mg). Error bars represent standard error (S.E.).
- 3) **Fig. 3. Correlation between CP-COV03 pharmacokinetic parameters and the viral load of Severe Acute Respiratory Syndrome Coronavirus 2 (SARS-CoV-2).** (A) CP-COV03 300 mg dose group; (B) CP-COV03 450 mg dose group (n = 15 for 300 mg, n = 11 for 450 mg).

Authors point by point responses to Reviewer #2:

Comments:

1. The authors comment on emergence of new SARS-CoV-2 variants, this is also mentioned as a potential, unlikely, source of variable efficacy impacting the study outcomes. It would be beneficial for the paper if the authors could comment on this and provide a reference of the SARS-CoV-2 variant efficacy for niclosamide or add own data in the supplementary section.

Response by the authors:

We have provided information in the main text with reference (14) and provided the information as supplementary Table S1.

Niclosamide has demonstrated efficacy against SARS-CoV-2 and its various variants¹⁴, as well as other viruses, as highlighted in Table S1¹⁵⁻²³ (Refer page 3, line 73-74)

Table S1. Broad spectrum antiviral activity of niclosamide against various viral families

Family	Virus species	References
Coronaviridae	MERS-CoV, SARS-CoV, SARS-CoV-2, Alpha (B. 1.1. 7), Beta (B. 1.351) and Delta variant (B. 1.617.2)	Gassen et. al., Nat. Commun. (2019) 10: 5770; Wu et. al., Antimicrob. Agents Chemother. (2004) 48: 2693; Jeon et. al, Antimicrob Agents Chemother. (2020) 64(7):e00819; Weis et.al., PLoS one. (2021) 16: e0260958.
Adenoviridae	Human adenovirus	Marrugal-Lorenzo et. al., (2019) Sci Rep 9: 17
Orthomyxoviridae	Influenza virus (IV)	Jurgeit et. al., PLoS Pathog. (2012) 8(10):e1002976; Mazzon et. al., Viruses (2019) 11:176
Picornaviridae	Human rhinovirus (HRV), Coxsackieviruses (CV)	Jurgeit et. al., PLoS Pathog. (2012) 8(10):e1002976
Herpesviridae	Herpes virus (HSV), Epstein-Barr virus (EBV), Kaposi's sarcoma-associated herpesvirus (KSHV)	Jurgeit et. al., PLoS Pathog. (2012) 8(10):e1002976; Anderson et. al., (2019) Viruses 11:964; Huang et. al., (2017) Antiviral Res. 38: 68–78
Pneumoviridae	Respiratory syncytial virus (RSV)	Niyomdechana N et. al., Virus Res. (2021) 295:198277
Flaviviridae	Zika virus (ZIKV), Dengue virus (DENV), West Nile virus (WNV), Yellow fever virus (YFV), Japanese	Simeonov et. al., (2016) Nat. Med. 22: 1101–7; Li et. al., Cell Res. (2017) 27: 1046–64; Edwards et. al., J. Med. Chem. (2011) 54: 8670–80

	encephalitis virus (JEV),		
	Hepatitis C virus (HCV)		
Togaviridae	Chikungunya virus (CHIKV),	Sindbis virus (SINV),	Wang et. al., Antiviral Res. (2016) 135:81-90; Mazzon et. al., Viruses (2019) 11(10):176
	Semliki forest virus (SFV),	Ross river virus (RRV)	

2. The authors describe the limitations of other antiviral treatments for SARS-CoV-2 infection. I would recommend that this paragraph is reviewed and rewritten to address a number of points e.g. the major limitation relation for the nirmatrelvir is that it is co-dosed with the CYP inhibitor ritonavir, (nirmatrelvir + ritonavir provides the drug Paxlovid) to limited metabolic clearance, the use of ritonavir limits its use in certain patient populations. For molnupiravir there is a perceived risk for teratogenicity based on the molecular mechanism of action etc. Please recheck the background information and update the description of limitations associate with the three antivirals mentioned.

Response by the authors:

We have included the major limitation of other antiviral treatments for SARS-CoV-2 infection in the revised manuscript as follows;

A significant limitation of nirmatrelvir is that it must be co-administered with ritonavir⁹, a CYP inhibitor that acts as a pharmacokinetic booster (nirmatrelvir + ritonavir is marketed as Paxlovid[®]), but ritonavir restricts its use in some patient populations. Molnupiravir, on the other hand, carries a potential risk of teratogenicity due to its mechanism of inducing replication errors¹⁰. Furthermore, molnupiravir has not significantly reduced COVID-19-related hospitalizations or deaths among high-risk vaccinated adults¹¹.

(Refer page 3, lines 64-70)

3. Please check the manuscript and ensure that the distinction between use of nirmatrelvir and the drug P (Refer is clear (in clinical studies the drug is used not nirmatrelvir on its own)

Response by the authors:

We have rectified these mistakes and uniformed them throughout the manuscript;

To date, oral anti-SARS-CoV-2 drugs such as nirmatrelvir + ritonavir (Paxlovid[®]), molnupiravir (Lagevrio), and ensitrelvir (Xocova) have been developed⁵⁻⁸. However, concerns remain regarding their safety, efficacy, and accessibility. A significant limitation of nirmatrelvir is that it must be co-administered with ritonavir⁹, a CYP inhibitor that acts as a pharmacokinetic booster, but ritonavir restricts its use in some patient populations. (Refer page 3, lines 62-67)

4. The authors comment on the niclosamide has a well-documented safety profile. With the new nano-hybrid formulation the exposure increases significantly without any significant observations

- this is a key point. Are there any reference values from previous studies that could be added to the introduction.

Response by the authors:

Thank you for accurately pointing out the key point regarding safety profile of niclosamide in the introduction part. As per the referee's suggestion, we found references that mention the 'well-documented safety profile' of niclosamide and added the two main references in the introduction section to demonstrate the safety profile for niclosamide as follows:

24. Tao, H. et al. Niclosamide ethanolamine-induced mild mitochondrial uncoupling improves diabetic symptoms in mice. *Nat. Med.* 20, 1263-1269(2014).
<https://doi.org/10.1038/nm.3699>

25. Tam, John, et al. "Host-targeted niclosamide inhibits *C. difficile* virulence and prevents disease in mice without disrupting the gut microbiota. *Nat. Commun.* 9, 5233 (2018)
<https://doi.org/10.1038/s41467-018-07705-w>

5. a) The lack of improvement of the primary readout at the 450 mg dose is notable and difficult to explain considering the overlapping niclosamide exposure levels. There is and unsubstantiated mentioning of magnesium oxide contributing to this effect. I would like to see a more in-depth discussion and evidence on this topic with reference to previous studies confirming that a 50 % increase in the amount of magnesium oxide could contribute significantly. Does the placebo control contain magnesium oxide as well, which I would expect, if so the reasoning is not valid as all groups would receive -the same amount of magnesium oxide. b) Would it be possible to correlate the effects (primary and secondary) to actual exposure levels in individual patients in the combined 300 and 450 mg study groups to gain better clarity on the link between efficacy and plasma exposure?

Response by the authors:

5a) Thank you for your detailed feedback. The following contents can address these concerns questioned:

1. **In-Depth Discussion:** We have provided a more comprehensive discussion regarding the lack of improvement at the 450 mg dose. For reference, MgO was not included in the placebo. Please consider the following manuscript content in light of this.

Discussion

In this study, designed to support emergency use authorization (EUA), the 300 mg dose of CP-COV03 demonstrated a statistically significant reduction in both viral load and time required to alleviate the 12 targeted COVID-19 symptoms compared to the placebo group.

Although the 450 mg dose group showed a higher AUC_t than the 300 mg one (10,562.09 ng·h/mL for the 300 mg dose and 12,876.29 ng·h/mL for the 450 mg dose), the pharmacokinetic profiles of both doses were similar, reaching steady-state concentrations at comparable levels. This was reflected in the similar viral load reduction patterns observed in both dose groups. These findings suggest that CP-COV03's antiviral efficacy is time-dependent (sustained exposure over time; Figure 2B) rather than concentration-dependent 34-36. This is attributed to the mechanism of action of niclosamide, the active ingredient in CP-COV03, which induces autophagy, resulting in antiviral effects. The plasma concentration necessary to sustain autophagy induction appears to have been achieved with the 300 mg dose indicating that it reaches a pharmacodynamic ceiling.

Analysis of the primary outcome—the time required for the improvement of targeted COVID-19 symptoms—revealed that in the planned analysis of the PPS group, the placebo group required 13.0 days (95% CI, 10.50-NR), while the 300 mg CP-COV03 group required 9.0 days (95% CI, 7.00-10.00), a statistically significant difference (P=0.0083). In the ITT population, the median time for the resolution of all 12 COVID-19 symptoms was 10.0 days (95% CI, 8.50-12.50) for the 300 mg group, compared to 12.25 days (95% CI, 10.50-NR) for the placebo one; however, this difference was not statistically significant.

In the additional analysis of the mITT-1 population, which included participants who received the study drug within 3 days of symptom onset, the median time to improvement of targeted symptoms was 9.0 days (95% CI, 7.50-10.50) in the 300 mg dose group, compared to 12.5 days (95% CI, 10.50-NR) in the placebo one, with the difference being statistically significant (P=0.024).

In the mITT-1 population, the subgroup analysis of the high-risk group (participants aged 60 years and above, or those with obesity, chronic conditions such as diabetes or hypertension, immunocompromised individuals, or long-term users of immunosuppressants) showed a significantly shorter time to symptom improvement in the 300 mg CP-COV03 group than the placebo one. Specifically, the median time for symptom improvement was 7.5 days (95% CI, 7.00-9.00) in the 300 mg group, compared to 12.5 days (95% CI, 8.00-NR) in the placebo one (P=0.017).

These findings surpass the reported efficacy of Paxlovid®, which reduced symptom resolution time by 3 days in high-risk patients^{37,38}, and ensitrelvir, which reduced the time to improvement in five symptoms by 1 day in the general population. 8,39

However, the 450 mg dose group did not show a statistically significant improvement in symptoms (PPS analysis: 12.25 days [95% CI, 9.50-NR] for the 450 mg dose group vs. 13.0 days [95% CI, 10.50-NR] for the placebo one). A possible explanation for this is the presence of magnesium oxide (MgO) in the 450 mg formulation, which may have caused gastrointestinal symptoms, thereby confounding participant-reported symptom evaluations. The daily MgO intake in the 450 mg group was 945 mg, higher than the 630 mg in the 300 mg group and exceeding the

recommended daily intake of 800 mg⁴⁰. Excessive MgO intake is associated with gastrointestinal symptoms such as diarrhea, nausea, and vomiting, which overlap with the targeted COVID-19 symptoms in this trial⁴¹.

In an ad-hoc analysis excluding the influence of MgO, the median time for improvement of three representative COVID-19 symptoms (fever, headache, and sore throat) was 4.0 days (95% CI, 3.5-4.5), 4.5 days (95% CI, 4.0-5.0), and 6.0 days (95% CI, 4.0-7.5) in the 300 mg, 450 mg, and placebo groups, respectively. Both the 300 mg and 450 mg dose groups demonstrated statistically significant reductions in the time required for symptom improvement compared to the placebo one (P=0.011 and P=0.044 for the 300 mg and 450 mg doses, respectively), suggesting that MgO confounded the results for the 450 mg group by affecting gastrointestinal symptoms.

The higher proportion of certain COVID-19 symptoms at baseline in the 450 mg treatment group may have contributed to the lack of statistically significant improvement in symptoms observed in this group. Factors that could influence clinical outcomes, such as age and disease severity, were stratified to maintain balance between groups, but it was not feasible to stratify for all 12 symptoms to ensure group balance. As a result, while there were no statistically significant differences in the number of patients showing symptoms like fever, cough, sore throat, headache, muscle ache, chill, stuffy or runny nose, fatigue, difficulty of breathing, vomiting or diarrhea at baseline between groups, significantly more patients with nausea were included in the 450 mg group (Chi-squared test, placebo vs 300mg, p-value = 0.0092). (Refer pages 8-11, lines 200-268)

5b) To investigate the link between exposure and efficacy, we examined the association between AUC_t and individual viral loads. As a result, a significant negative association was observed between the two, with correlation coefficients of -0.330 (P=0.0101) and -0.482 (P=0.0009), respectively. Since the 450 mg group, which showed higher drug exposure, also demonstrated a stronger negative correlation with viral load reduction compared to the 300 mg group, it can be concluded that the 450 mg group experienced a more effective reduction in viral load.

This point has been reflected in the manuscript as follows;

Since the 450 mg group, which showed higher drug exposure, also demonstrated a stronger negative correlation with viral load reduction compared to the 300 mg group, it can be concluded that the 450 mg group experienced a more effective reduction in viral load. (Refer pages 8, lines 184-187)

6. a) There is no additional increase in the viral clearance upon dose increase increasing the dose from 300 to 450 mg the PK profiles seem to be the explanation for these result. b) Are there any additional data from previous human PK studies indicating the lack of a dose dependent increase of drug plasma exposure. c) There is also a slight reduction in exposure over time – what is known about induction of drug clearance over time?

Response by the authors:

6a) We have already given the possible mechanism in the revised manuscript as follows;

“Although the 450 mg dose group showed a higher AUC_t than the 300 mg one (10,562.09 ng·h/mL for the 300 mg dose and 12,876.29 ng·h/mL for the 450 mg dose), the pharmacokinetic profiles of both doses were similar, reaching steady-state concentrations at comparable levels. This was reflected in the similar viral load reduction patterns observed in both dose groups. These findings suggest that CP-COV03’s antiviral efficacy is time-dependent (sustained exposure over time; Figure 2B) rather than concentration-dependent 34-36. This is attributed to the mechanism of action of niclosamide, the active ingredient in CP-COV03, which induces autophagy, resulting in antiviral effects. The plasma concentration necessary to sustain autophagy induction appears to have been achieved with the 300 mg dose indicating that it reaches a pharmacodynamic ceiling. (Refer Page 9, lines 207-217)

6b) In previous studies, no information regarding the dose dependency of niclosamide was found. However, as mentioned in the response to **6a**, CP-COV03 showed an increase in exposure with increasing dose (10,562.09 ng·h/mL for the 300 mg dose and 12,876.29 ng·h/mL for the 450 mg dose).

6c) As described in the supplementary file, due to the mucoadhesive property of HPMC, increasing its quantity leads to the prolongation of the duration of niclosamide plasma concentration while reducing C_{max}.

Due to the mucoadhesive property of HPMC, its quantity increases lead to the prolongation of the duration of niclosamide plasma concentration while reducing C_{max}. Based on this finding, we have designed a formulation to be used in clinical trials aimed at maintaining appropriate blood concentrations when administered. (Refer to supplementary file page 35, Results).

7. Please comment on the lack of balance in base-line characteristics for some of the parameters (diarrhea, nausea) seen in Table 1.

Response by the authors:

In designing the study, we stratified by severity (mild/moderate) and age, which are factors that could influence the outcomes of the COVID-19 clinical trial, to ensure balance between study groups. Given that there are 12 target symptoms of COVID-19, it is not feasible to achieve a perfect balance across all symptoms between the study groups. However, the analysis showed that, aside from nausea, there were no statistically significant differences (P<0.05) in the frequency of symptoms between groups. In the case of nausea, it occurred significantly more frequently in the 450 mg dose group, which may have had a negative impact on symptom improvement in this group.

This point has been further addressed in the discussion part as follows.

The higher proportion of certain COVID-19 symptoms at baseline in the 450 mg treatment group may have contributed to the lack of statistically significant improvement in symptoms observed in this group. Factors that could influence clinical outcomes, such as age and disease severity, were stratified to maintain balance between groups, but it was not feasible to stratify for all 12 symptoms to ensure group balance. As a result, while there were no statistically significant differences in the number of patients showing symptoms like fever, cough, sore throat, headache, muscle ache, chill, stuffy or runny nose, fatigue, difficulty of breathing, vomiting or diarrhea at baseline between groups, significantly more patients with nausea were included in the 450 mg group (Chi-squared test, placebo vs 300mg, p-value = 0.0092). (Refer page 11, lines 259-268)

Additionally, we identified error in the number of patients with symptoms at baseline in the original table. The original table reported the number of patients who exhibited symptoms over the entire period, rather than just at baseline. We have corrected this error as follows. We apologize for the confusion.

Table 1. Baseline characteristics (intention-to-treat population)

	Placebo (n=98)	CP-COV03 30mg dose (n=99)	CP-COV03 450 mg dose (n=96)
Demographics			
Age (y), mean (SD)	43.79 (12.94)	42.18 (13.50)	41.89 (12.32)
Age groups			
19–29 yrs., n (%)	17 (17.4)	24 (24.2)	20 (20.8)
30–39 yrs., n (%)	21 (21.4)	16 (16.2)	21 (21.9)
40–49 yrs., n (%)	25 (25.5)	25 (25.3)	29 (30.2)
50–59 yrs., n (%)	21 (21.4)	25 (25.3)	16 (16.7)
≥60 yrs., n (%)	14 (14.3)	9 (9.1)	10 (10.4)
Sex			
Male, n (%)	62 (63.3)	70 (70.7)	61 (63.5)
Female, n (%)	36 (36.7)	29 (29.3)	35 (36.5)
Height, cm, mean (SD)	168.81 (8.57)	169.29 (8.39)	169.77 (7.79)
Weight, kg, mean (SD)	70.92 (14.05)	70.04 (13.16)	69.79 (12.99)
Comorbidities			
Hypertension, n (%)	11 (11.0)	7 (7.1)	4 (4.2)
Diabetes mellitus, n (%)	2 (2.0)	5 (5.1)	2 (2.1)
Hyperlipidemia, n (%)	4 (4.1)	11 (11.1)	5 (5.2)
COVID-19 symptoms			
Fever, n (%)	9 (9.2)	12 (12.1)	7 (7.3)
Chill, n (%)	87 (88.8)	82 (82.8)	82 (85.4)
Muscle ache, n (%)	86 (87.8)	88 (88.9)	85 (88.5)
Headache, n (%)	69 (70.4)	70 (70.7)	63 (65.6)
Fatigue, n (%)	80 (81.6)	84 (84.8)	73 (76)
Cough, n (%)	63 (64.3)	66 (66.7)	57 (59.4)

Sore throat, n (%)	66 (67.3)	70 (70.7)	64 (66.7)
Stuffy or runny nose, n (%)	70 (71.4)	75 (75.8)	68 (70.8)
Difficulty of breathing, n (%)	21 (21.4)	23 (23.2)	24 (25)
Nausea	21 (21.4)	27 (27.3)	37 (38.5)
Vomiting	6 (6.1)	7 (7.1)	6 (6.3)
Diarrhea	18 (18.4)	22 (22.2)	20 (20.8)
COVID-19 severity			
Mild, n (%)	84 (85.7)	86 (86.9)	86 (89.6)
Moderate, n (%)	14 (14.3)	13 (13.1)	10 (10.4)

COVID-19: coronavirus disease 2019

8. Please add error bars to the plasma sample to understand if there is a significant difference in exposures between the two dose groups.

Response by the authors: We have fixed the error bars in the figure 2 as follows;

Fig. 2. Viral load and pharmacokinetic analyses. (A) The adjusted mean change in viral load from baseline of Severe Acute Respiratory Syndrome Coronavirus 2 (SARS-CoV-2) monitored starting from Day 0. Any data point that is more than 3 times the IQR above the third quartile or below the first quartile is an outlier (n = 77 for placebo, n = 70 for 300 mg, n = 80 for 450 mg). Error bars represent standard error (S.E.). (B) Pharmacokinetic profiles of niclosamide from clinical trial (n = 20 for placebo, n = 20 for 300 mg, n = 18 for 450 mg). Error bars represent standard error (S.E.).

Authors point by point responses to Reviewer #3:

1. The title should be non-declamatory and preferably state the study design.

Response by the authors: The title was revised from

“Niclosamide based nanohybrid system for the treatment of mild to moderate COVID-19 patients: Rapid viral load reduction and symptom alleviation”

to

“A randomized, double-blind and placebo-controlled trial of niclosamide nanohybrid for the treatment of patients with mild to moderate COVID-19”. (Refer page 1, lines 1-2)

2. The introduction in the abstract needs reworking. For example, there is an implicit assumption that broad spectrum antiviral drugs are warranted because of the ‘Surging long coronavirus disease 2019 cases.’ We have no convincing evidence as yet that treating acute viral illnesses with antiviral drugs will diminish persisting symptoms.

Response by the authors:

As recommended by the reviewer, the abstract has been revised to ensure the results are presented more clearly and quantitatively as follows:

Abstract

Coronavirus disease 2019 (COVID-19), caused by severe acute respiratory syndrome coronavirus 2 (SARS-CoV-2), has precipitated profound global social and economic disruptions and continues to drive high rates of infection. Amid this ongoing crisis, there is a persistent and urgent demand for an effective and reliable treatment for COVID-19. As part of the efforts to develop the treatment, niclosamide has been considered as a potential drug candidate for SARS-CoV-2 according to the in vitro studies. However, no clinical trials were successful due to its intrinsic properties like low solubility and bioavailability. In the present study, these limitations have been successfully addressed by developing a new niclosamide nanohybrid (CP-COV03), demonstrating significant antiviral activity in clinical trial for COVID-19; the findings from a randomized, double-blind and placebo-controlled clinical trial involving 300 patients (Clinical Trial Registration Number: KCT0007307) that assessed the efficacy and safety of CP-COV03 have been reported. The results highlight that the present nanohybrid drug significantly alleviated 12 FDA-recommended COVID-19 symptoms with sustained improvement (symptom improvement maintained for more than 48 hours) and reduced viral load by 56.7% compared to baseline within 16 hours of the initial dose. Importantly, no serious adverse events were reported across all treatment groups during the trial. (Refer page 2, lines 38-56)

3. Line 44: what broad-spectrum efficacy is niclosamide known for? Certainly, the evidence for this drug outside its parasitic indication is controversial. Has a systematic search been done for trials of niclosamide as a treatment for COVID-19?

Response by the authors:

As mentioned in Table S1, there are studies that have demonstrated the broad-spectrum antiviral activity of niclosamide through in vitro experiments. Additionally, we found clinical trials of niclosamide (without enhanced bioavailability) for COVID-19 and referenced them in the introduction part as follows:

A randomized, placebo-controlled clinical trial for pristine niclosamide was previously conducted in patients with mild to moderate COVID-19³². In this trial, 73 participants were enrolled, with 36 randomized to the niclosamide group and 37 to the placebo group. The niclosamide group received 2g of niclosamide orally once daily for 7 days, while the placebo group received an identically labeled placebo with the same dosing schedule. Although niclosamide was well tolerated, but ended in failure, since there was no significant difference in the oropharyngeal clearance of SARS-CoV-2 between the placebo and niclosamide groups³². (Refer pages 3-4, lines 78-87)

4. Line 48 'trials' should be singular

Response by the authors:

(This was corrected from Trials->trial)

5. Lines 51 to 52: 56% of what?

Response by the authors:

This was corrected as 56.7% compared to baseline)

6. lines 57 and 58: precise estimates are available.

Response by the authors:

It has been corrected as follows:

76 millions affected and 7 millions of deaths reported (Refer pages 3, lines 58)

- <https://data.who.int/dashboards/covid19/cases>
- <https://data.who.int/dashboards/covid19/deaths>

7. Line 80 Molnupiravir has not been found to be ‘ineffective’ for vaccinated individuals. Molnupiravir reduced time taken for recovery in vaccinated individuals. However, it did not reduce an already low hospital admission rate in a vaccinated population in the United Kingdom.

Response by the authors:

According to the referee’s comment, we changed the sentence in line 80 “while molnupiravir has been found ineffective for vaccinated individuals and may” to “Furthermore, molnupiravir has not significantly reduced COVID-19-related hospitalizations or deaths among high-risk vaccinated adults¹¹” (Refer page 3, lines 68-70) by citing the following reference;

Butler, C.C., et al. Molnupiravir plus usual care versus usual care alone as early treatment for adults with COVID-19 at increased risk of adverse outcomes (PANORAMIC): an open-label, platform-adaptive randomised controlled trial. *The Lancet* 401.10373 (2023): 281-293.

[https://doi.org/10.1016/S0140-6736\(22\)02597-1](https://doi.org/10.1016/S0140-6736(22)02597-1)

8. Lines 90 to 91: Suggest provide a brief summary of the data in the animal models which suggest antiviral activity.

Response by the authors:

This information has been included in the revised manuscript as follows;

A study demonstrated CP-COV03's efficacy against COVID-19 in a Syrian hamster model, highlighting the significance of innovative solutions in combating SARS-CoV-2. Compared to the control, the CP-COV03-treated group showed a reduced rate of SARS-CoV-2 replication in lung tissue. Additionally, necropsy revealed reduced total lung lesions in CP-COV03-treated hamsters. Furthermore, histological analysis showed significantly lower lung injury scores in the CP-COV03-treated group than the control group³³. (Refer page 4, lines 93-100)

9. Line 106: Best for the authors not to apply positive valued judgments to their own work.

Response by the authors:

Thank you for your comment. Accordingly, we have removed the following sentences in line 106;

This study is of great interest due to the ongoing need for effective COVID-19 treatments, especially given the limitations of current antivirals and the persistent threat of new variants along with surging long-COVID complications. CP-COV03 represents a promising new therapeutic option with broad-spectrum antiviral potential, addressing significant gaps in COVID-19 treatment strategies.

10. Lines 1/21 to 124. Was there a statistical analysis plan that was approved before the analysis?

Response by the authors:

Yes.

Between May 11 and November 28, 2022, 317 patients were screened for inclusion, of which 300 were enrolled in the trial (Refer page 5, lines 110-111)

11. Was the protocol registered and approved before the study began recruitment?

Response by the authors:

Yes, the protocol (version 5.0) was approved by the MFDS (Ministry of Food and Drug Safety, Korea FDA) on May 2, 2022, and by the IRB of the first study site on May 9, 2022. Recruitment of study participants began on May 10, 2022.

This information can be found in the manuscript as follows;

Trial oversight

The clinical trial was approved on May 2, 2022, by the Ministry of Food and Drug Safety (MFDS) of South Korea and by the institutional review board (IRB) at each trial site (IRB approval date for the first study site: May 9, 2022) before the start of recruitment (recruitment began on May 10, 2022). (Refer page 114, lines 341-345)

12. More detail is needed on the methods: **a)** for example, were there any contraindications to the drug that resulted in potential participants being ineligible? **b)** What was the sampling regimen and how a samples analyzed. What was the primary outcome measure? **c)** Was there a power calculation? **d)** What was the definition of sustained improvement? I noticed that a lot of this comes later, following the journal's tradition of putting the methods at the end of the discussion, but I've found it difficult to interpret the results without understanding what the primary outcome really was, and what populations the results were referring to. For example, excluding people who took medication who could influence outcome is a very unusual thing to do in a trial, especially if this was done post hoc.

Response by the authors:

Thank you for your careful evaluation on these aspects related to the method. Accordingly, corrections have been made as follows;

12a) for example, were there any contraindications to the drug that resulted in potential participants being ineligible?

In this clinical trial, there were no potential participants who were excluded during screening due to contraindications to the drug.

12b) What was the sampling regimen and how a samples analyzed. What was the primary outcome measure?

If "sample" refers to the analysis population, the population and analysis method are defined in the revised manuscript as follows;

Patients

Between May 11 and November 28, 2022, 317 patients were screened for inclusion, of which 300 were enrolled in the trial. They were randomly assigned to the CP-COV03 300 mg dose, 450 mg dose, and placebo groups in a 1:1:1 ratio. The study drug was administered to 293 patients, excluding seven who dropped out before administration, and 291 patients completed the trial. Safety and intent-to-treat (ITT) analyses were performed on the 293 patients (who were treated within 5 days of symptom onset), while the modified ITT-1 (mITT-1) population included 264 of the 293 participants (who were treated within 3 days of symptom onset). The mITT-2 population, consisting of 253 participants, excluded those from the mITT-1 population whose baseline PCR results were negative (below LLOQ) or who had missing PCR results. A per-protocol population set (PPS) analysis was conducted on 227 patients who had no important protocol deviations leading to exclusion from the per protocol population and had completed the day 28 follow-up visit and were treated within 5 days of symptom onset (Fig. 1). The baseline characteristics of the study participants are shown in Table 1. (Refer page 5, lines 109-123)

Efficacy

The primary objective of the study was to assess the efficacy of CP-COV03 as compared with placebo measured by the number of days required for targeted COVID-19 symptoms to improve and maintain for more than 48 hours until Day 14. Targeted COVID-19 symptoms included fever (38.0°C or above), cough, sore throat, headache, muscle aches, chills or shivering, stuffy or runny nose, tiredness or low energy, difficulty breathing or shortness of breath, nausea, vomiting, and diarrhea. In accordance with the FDA guideline ("Assessing COVID-19-related symptoms in outpatient adult and adolescent subjects in clinical trials of drugs and biological products for COVID-19 prevention or treatment"), symptoms were evaluated on a 4-point scale: 0 = absent, 1 = mild, 2 = moderate, and 3 = severe. Improvement was considered to have occurred in one of the following three cases: 1) if symptoms observed at baseline by 2 points or more have improved to 1 point or less; 2) if a symptom observed as 1 point at baseline has improved to 0 points; 3) symptoms that were not observed at the baseline but newly occurred during the clinical trial improved to 0 again. For participants whose symptoms did not improve until Day 14 or who were not satisfied with maintaining symptoms for more than 48 hours after the symptom improvement until Day 14.5 (evening of Day 14), the number of days required for symptom improvement was calculated as 13, the maximum value that could appear in the analysis, and classified as censored. (Refer page 15-16, lines 361-380)

If "sample" refers to PCR samples, the sampling and analysis method is outlined as follows;

Viral load analysis

Global Clinical Central Lab (GCCL), a certified Good Clinical Laboratory Practice (GCLP) facility, provided nasal swabs to each clinical hospital and collected nasal swabs from participants. To eliminate potential variability related to self-swabbing, trained nurses collected nasal swab samples from the participants in the afternoon on the scheduled days for viral load analysis. The nasal swab samples were stored under -20°C or below condition before sent to GC Labs for sample processing and analysis. The samples were fully thawed under refrigerated conditions for analysis. For sample preparation, $5\ \mu\text{L}$ of Internal Control A (STANDARD M nCoV Real-Time Detection kit, SD Biosensor, INC., Cat. No. M-NCOV-01) was dispensed into each well of the sample plate (Dxseq Viral Nucleic Acid Isolation Kit, DXOME CO., LTD., Cat. No. MVP-VIK01096). The samples were gently vortexed, and $200\ \mu\text{L}$ of each sample was dispensed into the wells of the sample plate. Nucleic acids were extracted using the KingFisher™ Flex Purification System (Thermo Fisher Scientific). To prepare the calibration curve, Reference RNA (Twist Synthetic SARS-CoV-2 RNA Control 2, TWIST BIOSCIENCE, Cat. No. 102024) was thawed and kept on ice. A mixture of $7.5\ \mu\text{L}$ of Reference RNA and $52.2\ \mu\text{L}$ of nuclease-free water (Thermo Fisher Scientific, Cat. No. R0581) was prepared to a final volume of $59.7\ \mu\text{L}$, yielding a final concentration of 1.0×10^5 copies/ μL . This was further diluted to prepare calibration curve samples (1.0×10^5 , 1.0×10^4 , 1.0×10^3 , 1.0×10^2 , 1.0×10^1 , 5.0×10^0 copies/ μL). On ice, $14\ \mu\text{L}$ of 2019-nCoV Reaction Solution (STANDARD M nCoV Real-Time Detection kit, SD Biosensor, INC., Cat. No. M-NCOV-01) and $6\ \mu\text{L}$ of Rtase Mix (STANDARD M nCoV Real-Time Detection kit, SD Biosensor, INC., Cat. No. M-NCOV-01) were mixed to prepare the PCR mixture. The prepared PCR mixture was then dispensed into a 96-well PCR plate, and Real-time PCR was performed using the CFX96™ Real-Time PCR Detection System (Bio-Rad Laboratories, Inc.). The viral load (copy number) of each sample was calculated based on Ct values derived from a standard curve plotting Ct values against \log_{10} . Ct values and copy numbers were calculated using the average of duplicate measurements, and the average copy numbers were converted to logarithmic values (\log_{10}). (Refer page 18-19, lines 432-461)

The primary outcome was the measurement of 12 COVID-19 symptoms, which is described in the manuscript as follows;

Efficacy

The primary objective of the study was to assess the efficacy of CP-COV03 as compared with placebo measured by the number of days required for targeted COVID-19 symptoms to improve and maintain for more than 48 hours until Day 14. Targeted COVID-19 symptoms included fever (38.0°C or above), cough, sore throat, headache, muscle aches, chills or shivering, stuffy or runny nose, tiredness or low energy, difficulty breathing or shortness of breath, nausea, vomiting, and diarrhea. In accordance with the FDA guideline ("Assessing COVID-19-related symptoms in outpatient adult and adolescent subjects in clinical trials of drugs and biological products for COVID-19 prevention or treatment"), symptoms were evaluated on a 4-point scale: 0 = absent, 1 = mild, 2 = moderate, and 3 = severe. Improvement was considered to have occurred in one of the following three cases: 1) if symptoms observed at baseline by 2 points or more have improved to 1 point or less; 2) if a symptom observed as 1 point at baseline has improved to 0 points; 3)

symptoms that were not observed at the baseline but newly occurred during the clinical trial improved to 0 again. For participants whose symptoms did not improve until Day 14 or who were not satisfied with maintaining symptoms for more than 48 hours after the symptom improvement until Day 14.5 (evening of Day 14), the number of days required for symptom improvement was calculated as 13, the maximum value that could appear in the analysis, and classified as censored. (Refer page 15-16, lines 361-380)

12c) Was there a power calculation?

The power calculation is provided in the supplementary file as follows;

At the time of planning this clinical trial, data on the progression to severe COVID-19 infection was available, but there was no reference data on symptom improvement. Therefore, we internally calculated the sample size for the clinical trial by referring to the symptom improvement effect of oseltamivir in influenza infection.

For reference, Paxlovid's sample size calculation was also based on the clinical trial (BLAZE-1) of the antibody treatment Bamlanivimab (Pfizer, EPIC-HR protocol, protocol number: C4671005).

According to Table 2 from previous oseltamivir (Tamiflu) studies (Treanor, John J., et al. "Efficacy and safety of the oral neuraminidase inhibitor oseltamivir in treating acute influenza: a randomized controlled trial." JAMA 283.8 (2000): 1016-1024), the duration of symptoms in the placebo group (129 participants) was a mean of 103.3 hours, with a 95% confidence interval of 92.6–118.7, while in the treatment group (121 participants), the mean was 69.9 hours, with a 95% confidence interval of 60.0–87.9. Based on these results, the standard deviations were calculated as approximately 75.62 hours for the placebo group and 78.29 hours for the treatment group.

The effect size (Cohen's d) was approximately 0.434. Considering a significance level of 5% and a power of 90%, the required sample size for the clinical trial was calculated to be 92 participants per group. Therefore, setting the sample size for this clinical trial at 100 participants per group is appropriate. (Refer to supplementary file page 30, statistical analysis).

12d) What was the definition of sustained improvement?

Sustained improvement refers to symptom improvement maintained for more than 48 hours. The definition of the primary outcome (definition of improvement) and the analysis groups can be found in the revised manuscript as follows;

Coronavirus disease 2019 (COVID-19), caused by severe acute respiratory syndrome coronavirus 2 (SARS-CoV-2), has precipitated profound global social and economic disruptions and continues to drive high rates of infection. Amid this ongoing crisis, there is a persistent and urgent demand for an effective and reliable treatment for COVID-19. As part of the efforts to develop the treatment, niclosamide has been considered as a potential drug candidate for SARS-CoV-2 according to the in vitro studies. However, no clinical trials were successful due to its intrinsic properties like low solubility and bioavailability. In the present study, these limitations have been successfully

addressed by developing a new niclosamide nano hybrid (CP-COV03), demonstrating significant antiviral activity in clinical trial for COVID-19; the findings from a randomized, double-blind and placebo-controlled clinical trial involving 300 patients (Clinical Trial Registration Number: KCT0007307) that assessed the efficacy and safety of CP-COV03 have been reported. The results highlight that the present nano hybrid drug significantly alleviated 12 FDA-recommended COVID-19 symptoms with sustained improvement (symptom improvement maintained for more than 48 hours) and reduced viral load by 56.7% compared to baseline within 16 hours of the initial dose. Importantly, no serious adverse events were reported across all treatment groups during the trial. (Refer page 2, lines 39-56)

The primary objective of the study was to assess the efficacy of CP-COV03 as compared with placebo measured by the number of days required for targeted COVID-19 symptoms to improve and maintain for more than 48 hours until Day 14. Targeted COVID-19 symptoms included fever (38.0°C or above), cough, sore throat, headache, muscle aches, chills or shivering, stuffy or runny nose, tiredness or low energy, difficulty breathing or shortness of breath, nausea, vomiting, and diarrhea. In accordance with the FDA guideline ("Assessing COVID-19-related symptoms in outpatient adult and adolescent subjects in clinical trials of drugs and biological products for COVID-19 prevention or treatment"), symptoms were evaluated on a 4-point scale: 0 = absent, 1 = mild, 2 = moderate, and 3 = severe. Improvement was considered to have occurred in one of the following three cases: 1) if symptoms observed at baseline by 2 points or more have improved to 1 point or less; 2) if a symptom observed as 1 point at baseline has improved to 0 points; 3) symptoms that were not observed at the baseline but newly occurred during the clinical trial improved to 0 again. For participants whose symptoms did not improve until Day 14 or who were not satisfied with maintaining symptoms for more than 48 hours after the symptom improvement until Day 14.5 (evening of Day 14), the number of days required for symptom improvement was calculated as 13, the maximum value that could appear in the analysis, and classified as censored. (Refer page 15-16, lines 361-380)

12e) I could not find the list of medications that resulted in exclusion from the main analysis. Was the subgroup analysis in the MIT population pre specified? Line 145: what is the ITT-1 population Line 157 what is the mITT2 population? Line 207: what is the PPS analysis

12e) The medications that potentially affect the study's targeted COVID-19 symptoms including gastrointestinal symptoms (e.g., nausea, vomiting, diarrhea), respiratory symptoms (e.g., runny/stuffy nose), and pain (e.g., muscle ache). The list of medications are as follows:

- drugs under ATC code A02B (peptic ulcer and gastro-oesophageal reflux disease): cimetidine, esomeprazole, famotidine, lansoprazole, nizatidine, pantoprazole, rabeprazole, and rebamipide.
- drugs under ATC code R01BA (sympathomimetics): pseudoephedrine.
- drugs under ATC code R06A (antihistamines for systemic use): azelastine, bepotastine, cetirizine, chlorpheniramine, fexofenadine, and levocetirizine.
- drugs under ATC code N02 (analgesics): aspirin, propacetamol, and tramadol.

The planned analysis included PPS and ITT, while other analyses, including mITT-1, were additional analyses. The planned analysis and additional analysis are described separately as follows;

The primary outcome of the study was to assess the efficacy of CP-COV03 compared to placebo, measured by the number of days required for targeted COVID-19 symptoms to improve and be maintained for more than 48 hours until Day 14. In the PPS group, the part of planned analysis, which included participants enrolled within 5 days of COVID-19 symptom onset and with no major protocol deviations, the median number of days required for the improvement of targeted COVID-19 symptoms was 9.0 days (95% CI, 7.00-10.00), 12.25 days (95% CI, 9.50-NR), and 13.0 days (95% CI, 10.50-NR) in the 300 mg dose, 450 mg dose, and placebo groups, respectively. A statistically significant difference ($P = 0.0083$) was observed in the 300 mg dose group compared to the placebo group. In another planned analysis group, the ITT population, the median number of days required for all 12 COVID-19 symptoms to show sustained improvement for more than 48 hours was 10.0 (95% CI, 8.50-12.50), 12.5 (95% CI, 10.50-NR), and 12.25 days (95% CI, 10.50-NR) in the 300 mg, 450 mg, and placebo groups, respectively. Although the median time in the 300 mg dose group was shorter than the placebo group, the difference was not statistically significant. In the additional analysis group, the mITT-1 population, which included participants who received the study drug within 3 days of symptom onset, the median number of days required for targeted COVID-19 symptoms to improve was 9.0 (95% CI, 7.50-10.50), 12.5 (95% CI, 10.00-NR), and 12.5 days (95% CI, 10.50-NR) in the 300 mg dose, 450 mg dose, and placebo groups, respectively. The 300 mg dose group showed a statistically significant reduction in the number of days required for symptom improvement compared to the placebo group ($P=0.024$). This was also confirmed in the mITT-2 population, where the median number of days required for targeted COVID-19 symptoms to improve was 9.0 (95% CI, 7.50-10.50), 12.5 (95% CI, 10.00-NR), and 12.5 (95% CI, 10.50-NR) days in the 300 mg dose, 450 mg dose, and placebo groups, respectively, with a statistically significant difference ($P=0.0275$) observed between the 300 mg dose group and the placebo group. This result was consistent in the subgroup analysis targeting the high-risk group (age 60 and above, patients with obesity, chronic conditions such as diabetes or hypertension, immunocompromised individuals, or those on long-term immune-suppressant therapy). In this high-risk group, the time required for symptom improvement was 7.5 days (95% CI, 7.00-9.00) in those receiving the 300 mg dose, which was significantly shorter than the 12.5 days (95% CI, 8.00-NR) required for the placebo group ($P=0.017$). (Refer page 6-7, lines 124-157)

13. Line 160: How was the viral load determined, and when were participants swabbed, who took the swabs, and how were they analyzed?

Response by the authors:

All the information on the viral load determination, participant's swab plan, and the methodology, etc, have been described as follows;

Viral load analysis

Global Clinical Central Lab (GCCL), a certified Good Clinical Laboratory Practice (GCLP) facility, provided nasal swabs to each clinical hospital and collected nasal swabs from participants. To eliminate potential variability related to self-swabbing, trained nurses collected nasal swab samples from the participants in the afternoon on the scheduled days for viral load analysis. The nasal swab samples were stored under -20 °C or below condition before sent to GC Labs for sample processing and analysis. The samples were fully thawed under refrigerated conditions for analysis. For sample preparation, 5 µL of Internal Control A (STANDARD M nCoV Real-Time Detection kit, SD Biosensor, INC., Cat. No. M-NCOV-01) was dispensed into each well of the sample plate (Dxseq Viral Nucleic Acid Isolation Kit, DXOME CO., LTD., Cat. No. MVP-VIK01096). The samples were gently vortexed, and 200 µL of each sample was dispensed into the wells of the sample plate. Nucleic acids were extracted using the KingFisher™ Flex Purification System (Thermo Fisher Scientific). To prepare the calibration curve, Reference RNA (Twist Synthetic SARS-CoV-2 RNA Control 2, TWIST BIOSCIENCE, Cat. No. 102024) was thawed and kept on ice. A mixture of 7.5 µL of Reference RNA and 52.2 µL of nuclease-free water (Thermo Fisher Scientific, Cat. No. R0581) was prepared to a final volume of 59.7 µL, yielding a final concentration of 1.0×10^5 copies/µL. This was further diluted to prepare calibration curve samples (1.0×10^5 , 1.0×10^4 , 1.0×10^3 , 1.0×10^2 , 1.0×10^1 , 5.0×10^0 copies/µL). On ice, 14 µL of 2019-nCoV Reaction Solution (STANDARD M nCoV Real-Time Detection kit, SD Biosensor, INC., Cat. No. M-NCOV-01) and 6 µL of Rtase Mix (STANDARD M nCoV Real-Time Detection kit, SD Biosensor, INC., Cat. No. M-NCOV-01) were mixed to prepare the PCR mixture. The prepared PCR mixture was then dispensed into a 96-well PCR plate, and Real-time PCR was performed using the CFX96™ Real-Time PCR Detection System (Bio-Rad Laboratories, Inc.). The viral load (copy number) of each sample was calculated based on Ct values derived from a standard curve plotting Ct values against log₁₀. Ct values and copy numbers were calculated using the average of duplicate measurements, and the average copy numbers were converted to logarithmic values (log₁₀). (Refer Pages 18-19, lines 432-461)

14. Line 201 it is unclear what a clinically designed study is: suggest do not use the abbreviation here of EUA (and other abbreviations), unless previously spelled out in full.

Response by the authors:

Thank you for your careful suggestion. Accordingly, we have corrected it as follows;

In this clinically designed study aimed at facilitating emergency use authorization (EUA) (Refer page 8 , line 203)

15. Under methods, we are told that enrolled participants were hospitalized: **a)** did this apply also to those who had mild illness? Why were they hospitalized? Were participants all hospitalised for trial purposes? **b)** If so, how many declined to participate because they did not wish to be hospitalised, especially since the majority had mild symptoms? **c)** Were all patients discharged on day six, regardless of their symptoms? **d)** What proportion completed the three additional follow

up visits to the facilities? e) Even although the journal puts methods at the end of the report, explaining what the different study populations was in the report itself would be really helpful.

Response by the authors:

15a) The reason why the clinical participants were hospitalized, despite having mild to moderate COVID-19, is explained as follows;

The enrolled participants were mild to moderate COVID-19 patients who did not require hospitalization, but they were hospitalized for close observation and sample collection by trained nurses for the pharmacokinetics study and viral load measurements and were treated with the study drugs from Day 1 to Day 6 (Table S3). (Refer Page 12, lines 298-302)

15b)

Three of the seventeen screening failures and six of the nine dropouts declined to participate in the study, withdrawing their consent prior to randomization. Although the screening failure and dropout documentation did not specify the reasons for the consent withdrawals, it is suspected that most were likely related to hospitalization. (Refer supplementary file page 28, participants section).

15c) For reference, not all participants were discharged on day 6 regardless of their symptoms. According to the protocol, the hospital discharge date could be extended based on symptoms or in accordance with the Korea Disease Control and Prevention Agency (KDCA) guidelines for COVID-19. (Refer supplementary file page 28, participants section).

15d) 97.0% (291 out of 300) of participants completed the three additional follow-up visits. (Refer supplementary file page 28, participants section).

15e) We have provided explanations of the different study populations in the manuscript as follows.

Patients

Between May 11 and November 28, 2022, 317 patients were screened for inclusion, of which 300 were enrolled in the trial. They were randomly assigned to the CP-COV03 300 mg dose, 450 mg dose, and placebo groups in a 1:1:1 ratio. The study drug was administered to 293 patients, excluding seven who dropped out before administration, and 291 patients completed the trial. Safety and intent-to-treat (ITT) analyses were performed on the 293 patients (who were treated within 5 days of symptom onset), while the modified ITT-1 (mITT-1) population included 264 of the 293 participants (who were treated within 3 days of symptom onset). The mITT-2 population, consisting of 253 participants, excluded those from the mITT-1 population whose baseline PCR results were negative (below LLOQ) or who had missing PCR results. A per-protocol population set (PPS) analysis was conducted on 227 patients who had no important protocol deviations leading to exclusion from the per protocol population and had completed the day 28 follow-up visit and were treated within 5 days of symptom onset (Fig. 1). The baseline characteristics of the study participants are shown in Table 1. (Refer Page 5, lines 109-123)

Efficacy

The primary outcome of the study was to assess the efficacy of CP-COV03 compared to placebo, measured by the number of days required for targeted COVID-19 symptoms to improve and be maintained for more than 48 hours until Day 14. In the PPS group, the part of planned analysis, which included participants enrolled within 5 days of COVID-19 symptom onset and with no major protocol deviations, the median number of days required for the improvement of targeted COVID-19 symptoms was 9.0 days (95% CI, 7.00-10.00), 12.25 days (95% CI, 9.50-NR), and 13.0 days (95% CI, 10.50-NR) in the 300 mg dose, 450 mg dose, and placebo groups, respectively. A statistically significant difference ($P = 0.0083$) was observed in the 300 mg dose group compared to the placebo group. In another planned analysis group, the ITT population, the median number of days required for all 12 COVID-19 symptoms to show sustained improvement for more than 48 hours was 10.0 (95% CI, 8.50-12.50), 12.5 (95% CI, 10.50-NR), and 12.25 days (95% CI, 10.50-NR) in the 300 mg, 450 mg, and placebo groups, respectively. Although the median time in the 300 mg dose group was shorter than the placebo group, the difference was not statistically significant. In the additional analysis group, the mITT-1 population, which included participants who received the study drug within 3 days of symptom onset, the median number of days required for targeted COVID-19 symptoms to improve was 9.0 (95% CI, 7.50-10.50), 12.5 (95% CI, 10.00-NR), and 12.5 days (95% CI, 10.50-NR) in the 300 mg dose, 450 mg dose, and placebo groups, respectively. The 300 mg dose group showed a statistically significant reduction in the number of days required for symptom improvement compared to the placebo group ($P=0.024$). This was also confirmed in the mITT-2 population, where the median number of days required for targeted COVID-19 symptoms to improve was 9.0 (95% CI, 7.50-10.50), 12.5 (95% CI, 10.00-NR), and 12.5 (95% CI, 10.50-NR) days in the 300 mg dose, 450 mg dose, and placebo groups, respectively, with a statistically significant difference ($P=0.0275$) observed between the 300 mg dose group and the placebo group. This result was consistent in the subgroup analysis targeting the high-risk group (age 60 and above, patients with obesity, chronic conditions such as diabetes or hypertension, immunocompromised individuals, or those on long-term immune-suppressant therapy). In this high-risk group, the time required for symptom improvement was 7.5 days (95% CI, 7.00-9.00) in those receiving the 300 mg dose, which was significantly shorter than the 12.5 days (95% CI, 8.00-NR) required for the placebo group ($P=0.017$). (Refer Page 6-7, lines 124-157)

16. I'm sure I missed it but how many facilities contributed to the study and what was their nature?

Response by the authors:

As shown in the table below, 9 hospitals contributed to this clinical trial, and the number of patients recruited by each hospital is as follows: This information was provided as Supplementary Table S3 as follows (Refer supplementary file page 45).

Table S3. Hospitals involved in this study

Bestian Hospital	• Screened: 147• Randomized: 135• Completed: 133
Gimpo Woori Hospital	• Screened: 106• Randomized: 104• Completed: 101
Korea University Ansan Hospital	• Screened: 3• Randomized: 3• Completed: 3
Chungnam National University Sejong Hospital	• Screened: 3• Randomized: 3• Completed: 3
Kyungpook National University Hospital	• Screened: 2• Randomized: 2• Completed: 2
Keimyung University Dongsan Hospital	• Screened: 6• Randomized: 6• Completed: 6
Hyundai General Hospital	• Screened: 34• Randomized: 31• Completed: 29
Kyungpook National University Chilgok Hospital	• Screened: 14• Randomized: 14• Completed: 12
Chosun University Hospital	• Screened: 2• Randomized: 2• Completed: 2

17. Lines 342 to 347. This appears to be a retrospective justification for the **a)** sample size, rather than a prospective estimation, and the relevance of some of these considerations is questionable. This justification is inadequate. **b)** Was there a sample size calculation based on the pre specified primary outcome?

Response by the authors:

17a) We have calculated sample size/power as below;

At the time of planning this clinical trial, data on the progression to severe COVID-19 infection was available, but there was no reference data on symptom improvement. Therefore, we internally

calculated the sample size for the clinical trial by referring to the symptom improvement effect of oseltamivir in influenza infection.

For reference, Paxlovid's sample size calculation was also based on the clinical trial (BLAZE-1) of the antibody treatment Bamlanivimab (Pfizer, EPIC-HR protocol, protocol number: C4671005).

According to Table 2 from previous oseltamivir (Tamiflu) studies (Treanor, John J., et al. "Efficacy and safety of the oral neuraminidase inhibitor oseltamivir in treating acute influenza: a randomized controlled trial." JAMA 283.8 (2000): 1016-1024), the duration of symptoms in the placebo group (129 participants) was a mean of 103.3 hours, with a 95% confidence interval of 92.6–118.7, while in the treatment group (121 participants), the mean was 69.9 hours, with a 95% confidence interval of 60.0–87.9. Based on these results, the standard deviations were calculated as approximately 75.62 hours for the placebo group and 78.29 hours for the treatment group.

The effect size (Cohen's d) was approximately 0.434. Considering a significance level of 5% and a power of 90%, the required sample size for the clinical trial was calculated to be 92 participants per group. Therefore, setting the sample size for this clinical trial at 100 participants per group is appropriate. **(Refer to supplementary page 30, statistical analysis).**

17b) Thank you for your feedback. We acknowledge that the outcome measurement definitions has been already clarified in the revised manuscript. Additional information on **definition of symptom measurement, the exact follow-up time points, and the specific population** in which these assessments are planned are detailed as follows;

Definition of symptom measurement, the exact follow-up time points:

The primary objective of the study was to assess the efficacy of CP-COV03 as compared with placebo measured by the number of days required for targeted COVID-19 symptoms to improve and maintain for more than 48 hours until Day 14. Targeted COVID-19 symptoms included fever (38.0°C or above), cough, sore throat, headache, muscle aches, chills or shivering, stuffy or runny nose, tiredness or low energy, difficulty breathing or shortness of breath, nausea, vomiting, and diarrhea. In accordance with the FDA guideline ("Assessing COVID-19-related symptoms in outpatient adult and adolescent subjects in clinical trials of drugs and biological products for COVID-19 prevention or treatment"), symptoms were evaluated on a 4-point scale: 0 = absent, 1 = mild, 2 = moderate, and 3 = severe. Improvement was considered to have occurred in one of the following three cases: 1) if symptoms observed at baseline by 2 points or more have improved to 1 point or less; 2) if a symptom observed as 1 point at baseline has improved to 0 points; 3) symptoms that were not observed at the baseline but newly occurred during the clinical trial improved to 0 again. For participants whose symptoms did not improve until Day 14 or who were not satisfied with maintaining symptoms for more than 48 hours after the symptom improvement until Day 14.5 (evening of Day 14), the number of days required for symptom improvement was calculated as 13, the maximum value that could appear in the analysis, and classified as censored. **(Refer pages 15-16, lines 361-38)**

The specific population:

The primary outcome of the study was to assess the efficacy of CP-COV03 compared to placebo, measured by the number of days required for targeted COVID-19 symptoms to improve and be maintained for more than 48 hours until Day 14. In the PPS group, the part of planned analysis,

which included participants enrolled within 5 days of COVID-19 symptom onset and with no major protocol deviations, the median number of days required for the improvement of targeted COVID-19 symptoms was 9.0 days (95% CI, 7.00-10.00), 12.25 days (95% CI, 9.50-NR), and 13.0 days (95% CI, 10.50-NR) in the 300 mg dose, 450 mg dose, and placebo groups, respectively. A statistically significant difference ($P = 0.0083$) was observed in the 300 mg dose group compared to the placebo group. In another planned analysis group, the ITT population, the median number of days required for all 12 COVID-19 symptoms to show sustained improvement for more than 48 hours was 10.0 (95% CI, 8.50-12.50), 12.5 (95% CI, 10.50-NR), and 12.25 days (95% CI, 10.50-NR) in the 300 mg, 450 mg, and placebo groups, respectively. Although the median time in the 300 mg dose group was shorter than the placebo group, the difference was not statistically significant. In the additional analysis group, the mITT-1 population, which included participants who received the study drug within 3 days of symptom onset, the median number of days required for targeted COVID-19 symptoms to improve was 9.0 (95% CI, 7.50-10.50), 12.5 (95% CI, 10.00-NR), and 12.5 days (95% CI, 10.50-NR) in the 300 mg dose, 450 mg dose, and placebo groups, respectively. The 300 mg dose group showed a statistically significant reduction in the number of days required for symptom improvement compared to the placebo group ($P=0.024$). This was also confirmed in the mITT-2 population, where the median number of days required for targeted COVID-19 symptoms to improve was 9.0 (95% CI, 7.50-10.50), 12.5 (95% CI, 10.00-NR), and 12.5 (95% CI, 10.50-NR) days in the 300 mg dose, 450 mg dose, and placebo groups, respectively, with a statistically significant difference ($P=0.0275$) observed between the 300 mg dose group and the placebo group. This result was consistent in the subgroup analysis targeting the high-risk group (age 60 and above, patients with obesity, chronic conditions such as diabetes or hypertension, immunocompromised individuals, or those on long-term immune-suppressant therapy). In this high-risk group, the time required for symptom improvement was 7.5 days (95% CI, 7.00-9.00) in those receiving the 300 mg dose, which was significantly shorter than the 12.5 days (95% CI, 8.00-NR) required for the placebo group ($P=0.017$). (Refer pages 5-6, lines 124-157)

18. It is not clear what the scale is that is being referred to on lines 356 to 57. Is this a four-point ordinal scale? I eventually found this in the appendix.

Response by the authors:

We have included this in main text as follows;

In accordance with the FDA guideline ("Assessing COVID-19-related symptoms in outpatient adult and adolescent subjects in clinical trials of drugs and biological products for COVID-19 prevention or treatment"), symptoms were evaluated on a 4-point scale: 0 = absent, 1 = mild, 2 = moderate, and 3 = severe. (Refer page 15, lines 367-371)

19. Line 379 to 8: the authors return to sample size here and although they say 300 participants is expected to be sufficient to evaluate the efficacy and safety of the drug, there is no actual rationale for this expectation. Stating that this was sufficient for other trials, which may or may not have been done in a similar population using the same outcome measure is not sufficient explanation for the chosen sample size.

Response by the authors:

Yes, we have calculated sample size/power as below;

At the time of planning this clinical trial, data on the progression to severe COVID-19 infection was available, but there was no reference data on symptom improvement. Therefore, we internally calculated the sample size for the clinical trial by referring to the symptom improvement effect of oseltamivir in influenza infection.

For reference, Paxlovid's sample size calculation was also based on the clinical trial (BLAZE-1) of the antibody treatment Bamlanivimab (Pfizer, EPIC-HR protocol, protocol number: C4671005).

According to Table 2 from previous oseltamivir (Tamiflu) studies (Treanor, John J., et al. "Efficacy and safety of the oral neuraminidase inhibitor oseltamivir in treating acute influenza: a randomized controlled trial." JAMA 283.8 (2000): 1016-1024), the duration of symptoms in the placebo group (129 participants) was a mean of 103.3 hours, with a 95% confidence interval of 92.6–118.7, while in the treatment group (121 participants), the mean was 69.9 hours, with a 95% confidence interval of 60.0–87.9. Based on these results, the standard deviations were calculated as approximately 75.62 hours for the placebo group and 78.29 hours for the treatment group.

The effect size (Cohen's d) was approximately 0.434. Considering a significance level of 5% and a power of 90%, the required sample size for the clinical trial was calculated to be 92 participants per group. Therefore, setting the sample size for this clinical trial at 100 participants per group is appropriate. **(Refer to supplementary file page 30, statistical analysis).**

- 20.** Is this the first study of niclosamide for viral illnesses including COVID-19? We know that this is the first trial of this formulation, but what about of other formulations? Was a systematic search done? What were the previous findings

Response by the authors:

Yes, we conducted a systematic search on niclosamide, and among the results, we found clinical data related to COVID-19. A clinical trial was conducted using 2g of niclosamide, without improving its bioavailability, for 7 days in patients with mild to moderate COVID-19. (Cairns, Dana M., et al. "Efficacy of niclosamide vs placebo in SARS-CoV-2 respiratory viral clearance, viral shedding, and duration of symptoms among patients with mild to moderate COVID-19: a phase 2 randomized clinical trial." JAMA Network Open 5.2 (2022): e2144942-e2144942.) A summary of this clinical trial has been included in the introduction as follows.

A randomized, placebo-controlled clinical trial for pristine niclosamide was previously conducted in patients with mild to moderate COVID-19³². In this trial, 73 participants were enrolled, with 36 randomized to the niclosamide group and 37 to the placebo group. The niclosamide group received 2g of niclosamide orally once daily for 7 days, while the placebo group received an identically labeled placebo with the same dosing schedule. Although niclosamide was well tolerated, but ended in failure, since there was no significant difference in the oropharyngeal clearance of SARS-CoV-2 between the placebo and niclosamide groups³². **(Refer pages 3,4, lines 78-87).**

21. The consort diagram suggests that only 17 people failed screening: Of the 300 randomised, we have exactly 100 in each group. Is this exact division the result of good fortune or stratification?

Response by the authors:

Exactly 100 patients per group were enrolled in the study through IWRS randomization, with recruitment closing immediately after the 300th patient was randomized. During the study period, individuals infected with COVID-19 were required to quarantine in Korea until May 30, 2022, with quarantine being highly recommended from June 1, 2022. To minimize unnecessary outings and movement, the investigators conducted pre-screening consultations over the phone, allowing patients to self-assess their symptoms before visiting the clinical site for screening. As a result, there were few screen failures during the actual screening process.

22. Were the patients help seeking? How was the trial brought to their attention?

Response by the authors:

The trial was brought to the patient's attention through IRB-approved online and offline advertisements, both within and outside the study sites (The following is an offline advertisement approved by the IRB, which was used to promote the clinical trial).

Korean Version

English Version

연구 목적
본 임상시험의 주요 목적은 신종 코로나바이러스 감염증(코로나19) 치료제인 CP-COV03의 임상효능과 안전성을 평가하고 CP-COV03의 임상효능과 안전성을 평가하는 것입니다.

대상자 선정 기준
1) 18세 이상 75세 이하의 성인(연구 참여를 위한 기준)
2) COVID-19 진단을 받은 후 7일 이내로 발열, 기침, 호흡곤란, 근육통, 두통, 인후통, 코막힘, 또는 후각·미각 상실(또는 후각·미각 상실의 의심)을 경험한 자
3) COVID-19 진단을 받은 후 7일 이내로 발열, 기침, 호흡곤란, 근육통, 두통, 인후통, 코막힘, 또는 후각·미각 상실(또는 후각·미각 상실의 의심)을 경험한 자

임상시험 방법
본 임상시험은 1:1 무작위 배정으로 두 가지 치료군으로 나뉘어 시행됩니다. 두 군은 각각 CP-COV03군과 위약군으로 나뉘어 시행됩니다. 두 군은 각각 CP-COV03군과 위약군으로 나뉘어 시행됩니다. 두 군은 각각 CP-COV03군과 위약군으로 나뉘어 시행됩니다.

예측 가능한 부작용
1) 면역계 이상 반응: 알레르기 반응, 아나필락시스 반응, 아나필락시스 반응, 아나필락시스 반응
2) 신경계 이상 반응: 어지러움, 두통, 어지러움, 두통, 어지러움, 두통
3) 순환계 이상 반응: 호흡곤란, 호흡곤란, 호흡곤란, 호흡곤란, 호흡곤란, 호흡곤란
4) 기타: 연구 참여를 위한 기준에 따라 평가될 수 있습니다.

임상시험 실시 기간
- CP-COV03군: 2022년 5월 30일 ~ 2022년 6월 30일
- 위약군: 2022년 5월 30일 ~ 2022년 6월 30일
- 연구 참여를 위한 기준에 따라 평가될 수 있습니다.

주최 기관
GWH 김포유리병원 HYUNDAI BIOSCIENCE

Research goal
In this clinical trial, we aim to explore the safety and efficacy of CP-COV03 in patients with mild to moderate COVID-19.

Eligibility criteria
1) Adults aged 19 years or older (as of the date of written (electronic) consent).
2) Confirmed to be infected with COVID-19 through RT-PCR testing or rapid antigen testing by a professional.

Clinical trial method
1) Participants will be randomly assigned to Treatment Group 1, Treatment Group 2, or the control (placebo) group.
2) During the visit, screening tests will be conducted, including demographic information, clinical laboratory tests, and electrocardiograms, to assess the severity of COVID-19.
3) The entire study will be conducted over approximately 29 days, involving 6 or 7 days of hospitalization and 3 outpatient visits.

Predictable side effects
1) Immune system disorders: allergic reactions, anaphylactic reactions, anaphylactic shock
2) Nervous system disorders: dizziness (drowsiness)
3) Vascular disorders: cyanosis (blue lips)
In addition to the aforementioned side effects, unexpected side effects may occur.

Clinical trial site
- Besian Hospital, 191 Osongsaengmyeong 1-ro, Osong-eup, Heungdeok-gu, Cheongju, Chungcheongbuk-do, South Korea
- Gimpo Woon Hospital, 11 Gamsan-ro, Gimpo, Gyeonggi-do, South Korea
- Sponsor: HyundaiBioScience Co., Ltd., 106 Apogongdan-gil, Apo-eup, Gancheon, Gyeongangbuk-do, South Korea

주최 기관
GWH 김포유리병원 HYUNDAI BIOSCIENCE

- Detailed information on Korean version is summarized as follows
COVID-19 Treatment
Recruitment for Clinical Trial Participants

We are recruiting patients with mild to moderate COVID-19 symptoms for a Phase 2 clinical trial to evaluate the efficacy and safety of Hyundai Bioscience's antiviral drug, CP-COV03, compared to a placebo. This is a randomized, double-blind, placebo-controlled clinical trial.

For Registration
1544-5021

Research Objective

To evaluate the efficacy and safety of CP-COV03, an antiviral drug, in patients with mild to moderate COVID-19 symptoms.

This study will compare the treatment with a placebo and assess the overall safety and effectiveness in improving patient symptoms.

Expected Side Effects

As with any medication, there may be potential side effects.

The most common side effects may include nausea, headache, and digestive discomfort.

If any side effects occur, medical personnel will be available to provide necessary treatment and guidance.

Study Location

Participants will be treated at designated clinical sites, which include several hospitals across the country.

Eligibility Requirements

Individuals diagnosed with mild to moderate COVID-19.

Aged 18 and above.

Able to provide informed consent and participate in all required assessments during the trial.

Inquiries

For more information or to participate in the trial, please contact us at the number provided above.

23. Study groups were not well balanced at baseline for all characteristics of note. For example those receiving 450 milligrams had more frequent nausea and diarrhea. This needs to be scrutinized and fully discussed.

Response by the authors:

In designing the study, we stratified by severity (mild/moderate) and age, which are factors that could influence the outcomes of the COVID-19 clinical trial, to ensure balance between study groups. Given that there are 12 target symptoms of COVID-19, it is not feasible to achieve a perfect balance across all symptoms between the study groups. However, the analysis showed that, aside from nausea, there were no statistically significant differences ($P < 0.05$) in the frequency of symptoms between groups. In the case of nausea, it occurred significantly more frequently in the 450 mg dose group, which may have had a negative impact on symptom improvement in this group.

This point has been further addressed in the discussion part as follows;

The higher proportion of certain COVID-19 symptoms at baseline in the 450 mg treatment group may have contributed to the lack of statistically significant improvement in symptoms observed in this group. Factors that could influence clinical outcomes, such as age and disease severity, were stratified to maintain balance between groups, but it was not feasible to stratify for all 12 symptoms to ensure group balance. As a result, while there were no statistically significant differences in the number of patients showing symptoms like fever, cough, sore throat, headache, muscle ache, chill, stuffy or runny nose, fatigue, difficulty of breathing, vomiting or diarrhea at baseline between groups, significantly more patients with nausea were included in the 450 mg group (Chi-squared test, placebo vs 300mg, p-value = 0.0092). (Refer page 11, lines 259-268)

Additionally, we identified an error in the number of patients with symptoms at baseline in the original table. The original table reported the number of patients who exhibited symptoms over the entire period, rather than just at baseline. We have corrected this error as follows. We apologize for the confusion.

Table 1. Baseline characteristics (intention-to-treat population)

	Placebo (n=98)	CP-COV03 30mg dose (n=99)	CP-COV03 450 mg dose (n=96)
Demographics			
Age (y), mean (SD)	43.79 (12.94)	42.18 (13.50)	41.89 (12.32)
Age groups			
19–29 yrs., n (%)	17 (17.4)	24 (24.2)	20 (20.8)
30–39 yrs., n (%)	21 (21.4)	16 (16.2)	21 (21.9)
40–49 yrs., n (%)	25 (25.5)	25 (25.3)	29 (30.2)
50–59 yrs., n (%)	21 (21.4)	25 (25.3)	16 (16.7)
≥60 yrs., n (%)	14 (14.3)	9 (9.1)	10 (10.4)
Sex			
Male, n (%)	62 (63.3)	70 (70.7)	61 (63.5)
Female, n (%)	36 (36.7)	29 (29.3)	35 (36.5)
Height, cm, mean (SD)	168.81 (8.57)	169.29 (8.39)	169.77 (7.79)
Weight, kg, mean (SD)	70.92 (14.05)	70.04 (13.16)	69.79 (12.99)
Comorbidities			
Hypertension, n (%)	11 (11.0)	7 (7.1)	4 (4.2)
Diabetes mellitus, n (%)	2 (2.0)	5 (5.1)	2 (2.1)
Hyperlipidemia, n (%)	4 (4.1)	11 (11.1)	5 (5.2)
COVID-19 symptoms			
Fever, n (%)	9 (9.2)	12 (12.1)	7 (7.3)
Chill, n (%)	87 (88.8)	82 (82.8)	82 (85.4)
Muscle ache, n (%)	86 (87.8)	88 (88.9)	85 (88.5)
Headache, n (%)	69 (70.4)	70 (70.7)	63 (65.6)
Fatigue, n (%)	80 (81.6)	84 (84.8)	73 (76)

Cough, n (%)	63 (64.3)	66 (66.7)	57 (59.4)
Sore throat, n (%)	66 (67.3)	70 (70.7)	64 (66.7)
Stuffy or runny nose, n (%)	70 (71.4)	75 (75.8)	68 (70.8)
Difficulty of breathing, n (%)	21 (21.4)	23 (23.2)	24 (25)
Nausea	21 (21.4)	27 (27.3)	37 (38.5)
Vomiting	6 (6.1)	7 (7.1)	6 (6.3)
Diarrhea	18 (18.4)	22 (22.2)	20 (20.8)
COVID-19 severity			
Mild, n (%)	84 (85.7)	86 (86.9)	86 (89.6)
Moderate, n (%)	14 (14.3)	13 (13.1)	10 (10.4)

COVID-19: coronavirus disease 2019

24. There is a lot of repetition including of 265 -269

Response by the authors:

Thank you very much for your careful reading and comments. Accordingly, these repetitions have been removed.

25. Line 298 The placebo should be described here: e.g. TID? Matched for colour and taste?

Response by the authors:

It was corrected as follows;

- placebo TID group (has the same size, color, weight, and appearance as the CP-COV03 capsule) (Refer Page 12, lines 297-298)

26. I found the description of eligibility check and randomisation a bit confusing. We are told, “ those who developed at least one of the COVID-19 symptoms within five days from the date of randomization and who had at least two or more symptoms of 2 points or more among the symptom scores determined on the day of randomization; 3) those who confirmed with COVID-19 infection through reverse transcription–polymerase chain reaction (PCR) test or expert rapid antigen test within three days from the date of randomization...” Does this mean that potential participants could have been asymptomatic when they were randomized? Could people have been randomized without a positive test of some sort for COVID-19?

Response by the authors:

We are sorry to make you confused in describing eligibility check and randomization of participants;

The participants could not have been asymptomatic at the time of randomization due to the inclusion criteria, which required them to have at least two symptoms with a score of 2 or higher, assessed on the day of randomization.

No, participants could not be randomized without a positive COVID-19 test, as the inclusion criteria required confirmation of COVID-19 infection via reverse transcription–polymerase chain reaction (PCR) or expert rapid antigen test within three days prior to randomization.

To clearly convey this meaning, we have revised the manuscript as follows;

those who have confirmed SARS-CoV-2 infection within 3 days, and symptom onset within 5 days prior to randomization, with at least two symptoms rated 2 points or higher on the COVID-19 symptom scale at the time of randomization;(Refer Page 13, lines 318-321)

27. Lines 383: “In the ITT-1 population, the symptom assessment data of participants, who were administered concomitant medications that may influence symptom evaluation, were censored at the day before the administration of the concomitant medication (concomitant medication administration day - 0.5 day) mITT, which involved censoring concomitant medications that could affect symptom assessment.

Response by the authors:

In response to the reviewers' comments requesting a clear definition of the analysis groups, we have revised the method section as follows. The specific points raised by the reviewer have been addressed as follows;

The primary outcome for efficacy was assessed in the ITT, mITT-1, mITT-2, and PPS and the comparison between treatment groups was analyzed using a Cox Proportional Hazards Regression Model with treatment group as a factor and stratification factors (age, severity) as covariates. The hazard ratio for the treatment group, along with the corresponding 95% confidence interval and p-value, was presented. To ensure accurate measurement of the efficacy of the study drug, additional analyses for primary efficacy were conducted using mITT-1 and mITT-2, which involved

c
e
n

28. a) Was this pre specified? What worthy relevant concomitant medicines, was the list pre specified, and why was this done in the first place? b) The study was randomized and so additional medication use should have been balanced between groups. Excluding those who took additional drugs such as and anti inflammatories would introduce are likely bias. Were equal numbers excluded in all groups?

g

Response by the authors:

28a) Concomitant medications were pre-specified, and we have revised the manuscript to prioritize the planned analysis over the additional analysis, as described below;

u
c
o
m
i
t
a

Analysis of the primary outcome—the time required for the improvement of targeted COVID-19 symptoms—revealed that in the planned analysis of the PPS group, the placebo group required 13.0 days (95% CI, 10.50-NR), while the 300 mg CP-COV03 group required 9.0 days (95% CI, 7.00-10.00), a statistically significant difference (P=0.0083). In the ITT population, the median time for the resolution of all 12 COVID-19 symptoms was 10.0 days (95% CI, 8.50-12.50) for the 300 mg group, compared to 12.25 days (95% CI, 10.50-NR) for the placebo one; however, this difference was not statistically significant.

In the additional analysis of the mITT-1 population, which included participants who received the study drug within 3 days of symptom onset, the median time to improvement of targeted symptoms was 9.0 days (95% CI, 7.50-10.50) in the 300 mg dose group, compared to 12.5 days (95% CI, 10.50-NR) in the placebo one, with the difference being statistically significant (P=0.024).

In the mITT-1 population, the subgroup analysis of the high-risk group (participants aged 60 years and above, or those with obesity, chronic conditions such as diabetes or hypertension, immunocompromised individuals, or long-term users of immunosuppressants) showed a significantly shorter time to symptom improvement in the 300 mg CP-COV03 group than the placebo one. Specifically, the median time for symptom improvement was 7.5 days (95% CI, 7.00-9.00) in the 300 mg group, compared to 12.5 days (95% CI, 8.00-NR) in the placebo one (P=0.017). (Refer page 9-10, lines 218-237)

28b) The distribution of patients excluded due to concomitant drug use is as follows. When analyzed using the Chi-squared test, the p-value was 0.1042, indicating no significant difference between the groups. Therefore, it cannot be concluded that the distribution of concomitant drug users was uneven between the groups to affect the interpretation of the clinical outcomes.

	number
placebo	9
300mg	14
450mg	5

- 29.** The per protocol set (PPS) population included participants in the safety population who had no important protocol deviations leading to exclusion from the per protocol population and had completed the day 28 follow-up visit

Response by the authors:

This has been corrected in the revised manuscript as follows;

A per-protocol population set (PPS) analysis was conducted on 227 patients who had no important protocol deviations leading to exclusion from the per protocol population and had completed the day 28 follow-up visit and were treated within 5 days of symptom onset (Refer page 5, lines 119-122)

30. Suppl. 177. “Day 14.5, the number of days required for symptom improvement was calculated as 13, i.e., the maximum value that could appear in the analysis, and classified as censored”. This is irregular. Trials have found that up to 25% of people are still symptomatic at 28 days. Other trials have found that it takes a median of 16 days for Participants to feel recovered. How many were censored at this point?

Response by the authors:

The number of participants censored in the ITT population is as follows. According to one study, 137 out of 290 patients, or 47.2%, reported persistent symptoms even one month after infection, so this is not an unusual figure (Gallant, Maxime, et al. "Prevalence of persistent symptoms at least 1 month after SARS-CoV-2 Omicron infection in adults." *Journal of the Association of Medical Microbiology and Infectious Disease Canada* 8.1 (2022): 57-63.).

	number
placebo	46
300mg	39
450mg	43

31. The mITT-1 and mITT2 populations are virtually the same size and not that different from the ITT population in terms of size. Unless these populations were pre specified in a statistical analysis plan signed off before the end of recruitment and data log, I would leave these analysis for other outputs.

Response by the authors:

We have revised the manuscript to reflect the referee's comments. We clarified the definition of the analysis population in the manuscript, prioritizing the planned analysis and describing the additional analysis afterward.

Statistical analysis

The statistical analysis plan (version 2.0) was approved on February 21, 2023, prior to the unblinding and analysis. From May 11, 2022, to November 28, 2022, 300 participants underwent randomization. The planned analysis groups for the efficacy evaluation of this clinical trial were ITT and PPS. All participants who received at least one dose of the study drug were included in the ITT, and the PPS comprised participants in the ITT population, except for those with a protocol violation that could affect the assessment of antiviral activity.

For additional analysis, the mITT-1 population included participants who took the study drug within 3 days of symptom onset. The mITT-2 population excluded participants from the mITT-1 population whose baseline PCR was negative or missing.

The primary outcome for efficacy was assessed in the ITT, mITT-1, mITT-2, and PPS and the comparison between treatment groups was analyzed using a Cox Proportional Hazards Regression Model with treatment group as a factor and stratification factors (age, severity) as covariates. The hazard ratio for the treatment group, along with the corresponding 95% confidence interval and p-

value, was presented. To ensure accurate measurement of the efficacy of the study drug, additional analyses for primary efficacy were conducted using mITT-1 and mITT-2, which involved censoring concomitant medications that could affect symptom assessment. This censoring method was also applied to the additional analysis of representative COVID-19 symptoms such as fever, headache, and sore throat. **(Refer page 16, lines 397-418)**

Analysis of the primary outcome—the time required for the improvement of targeted COVID-19 symptoms—revealed that in the planned analysis of the PPS group, the placebo group required 13.0 days (95% CI, 10.50-NR), while the 300 mg CP-COV03 group required 9.0 days (95% CI, 7.00-10.00), a statistically significant difference ($P=0.0083$). In the ITT population, the median time for the resolution of all 12 COVID-19 symptoms was 10.0 days (95% CI, 8.50-12.50) for the 300 mg group, compared to 12.25 days (95% CI, 10.50-NR) for the placebo one; however, this difference was not statistically significant.

In the additional analysis of the mITT-1 population, which included participants who received the study drug within 3 days of symptom onset, the median time to improvement of targeted symptoms was 9.0 days (95% CI, 7.50-10.50) in the 300 mg dose group, compared to 12.5 days (95% CI, 10.50-NR) in the placebo one, with the difference being statistically significant ($P=0.024$).

In the mITT-1 population, the subgroup analysis of the high-risk group (participants aged 60 years and above, or those with obesity, chronic conditions such as diabetes or hypertension, immunocompromised individuals, or long-term users of immunosuppressants) showed a significantly shorter time to symptom improvement in the 300 mg CP-COV03 group than the placebo one. Specifically, the median time for symptom improvement was 7.5 days (95% CI, 7.00-9.00) in the 300 mg group, compared to 12.5 days (95% CI, 8.00-NR) in the placebo one ($P=0.017$). **(Refer page 9-10, lines 218-237)**

Comment of Reviewer 3

The proportion of patients who did not achieve sustained symptom improvement by Day 14—meaning patients who were still symptomatic or in the middle of improvement at the data cut-off point—should be explicitly stated in the text. The fact that symptom data for a large number of participants was censored also needs to be addressed in the discussion.

However, as you pointed out, the analysis regarding patients in the ITT population who did not achieve sustained symptom improvement was insufficiently discussed. Therefore, we have further revised the main manuscript as outlined below.

In this study, COVID-19 symptoms were observed to persist for a prolonged period, consistent with findings from other studies⁴⁵⁻⁴⁶. For example, one study found that 137 out of 290 COVID-19 patients (47.2%) reported persistent symptoms even one month after infection⁴⁷. In the results of the Paxlovid® pivotal clinical trial (EPIC-HR), which tracked symptoms up to Day 28, only 43.9% of patients in the treatment group reported symptom alleviation by Day 28.⁴⁸

In this study, evaluating of symptom improvement up to Day 14, 43.7% of trial participants (46 in the placebo group, 39 in the 300mg group, and 43 in the 450mg group from the ITT population) did not achieve a 48-hour sustained improvement of all 12 COVID-19 symptoms without further recurrence by the end of Day 14. The sustained improvement date was censored as the last symptom assessed date in the Cox proportional hazards regression model analysis.

As observed in previous studies mentioned earlier,^{47,48} this study similarly found many participants with prolonged COVID-19 symptoms. Unlike other respiratory infectious diseases, this finding suggests that a longer observation period of more than a month may be necessary to fully evaluate symptom resolution in COVID-19. This observation should be reflected in future clinical trials for COVID-19 treatments. In this clinical trial, symptoms were evaluated up to Day 14 based on the study design developed in consultation with the MFDS to investigate treatments for acute COVID-19 infection. However, the follow-up Phase 3 study, which an Investigational New Drug (IND) application has been submitted to the MFDS, will include a longer follow-up period. (page 12-13, lines 292-306)

Additionally, regarding your inquiry about the impact of censored data, we provide the following response:

Censored data in the Cox proportional hazards regression model is used to distinguish between data with incomplete information, where the improvement date is unknown, and data with complete information, where a specific improvement date is available. This distinction is crucial

for accurately estimating probabilities and hazard ratios in the Cox model, enabling it to effectively handle all available data for valid inference.

We hope these clarifications address your concerns. Should you require further details or revised documents, please let us know, and we will send them promptly.

Authors' response to Reviewer's comments

Reviewer Comment 1:

Responses to the query about whether all symptoms needed to be affected or whether the primary outcome would be made if a single symptom was affected, could still be made clearer. The response under paragraph 3B, it would be worth inserting "... any of the COVID-19 symptoms we ascertained persisted ...".

Authors' Response:

Thank you for this insightful suggestion. To address your comment, we have revised the text in paragraph 1B to include the phrase "... any of the COVID-19 symptoms we ascertained persisted ...". This change ensures that our response is clearer and aligns with your feedback. The revised sentence now reads "And any of the COVID-19 symptoms we ascertained persisted for a prolonged period, which were consistent with findings from other studies^{46,47}." (Refer Page 13 lines 307-308). This adjustment clarifies that the primary outcome does not require all symptoms to persist but is based on any symptom among those ascertained.

Reviewer Comment 2:

I notice that Table 4 does not make it clear what the figures refer to: if these are the median number of days, then the term 'median' should be there. The numbers of participants with each symptom ascertained are not present in the table. In line 130, the word median should be included.

Authors' Response: We appreciate your attention to detail. First of all, we would like to make a clarification regarding the statistical analyses made on primary and secondary outcomes. The primary outcomes were analysed based on Cox Proportional Hazards Model and secondary outcomes analysed based on ANCOVA, which is Analysis of Covariance. The ANCOVA and Cox Proportional Hazards Model differ in how they handle means and medians due to their purposes. ANCOVA focuses on comparing **group means** while adjusting for covariates, assuming normally distributed data, and is used for analyzing continuous outcomes. In contrast, the Cox model is used in survival analysis to study time-to-event data. It emphasizes **median survival time** (the time at which 50% of events occur) because survival data is often skewed, and means are less meaningful. While ANCOVA is about group averages, the Cox model is about risks and medians in the context of time. Therefore, we have made appropriate corrections on the captions related to secondary outcome results reported in Table S4.

Additionally, as per the Reviewer's suggestion, all secondary outcomes have been reported in Annexure-I. We have also included the following sentences in the revised manuscript (**Refer Page 10, lines 235-245**)

According to the PPS results of the secondary outcome, "the time (days) taken to sustain symptom improvement for more than 48 hours by Day 14," the time (days) to improvement of individual targeted COVID-19 symptoms in the group treated with the 300 mg dose was generally shorter compared to the placebo group. In particular, the improvement time (days) for sore throat, headache, and fatigue symptoms was significantly shorter compared to the placebo group (Sore throat: p=0.0168, placebo group 5.23 [95% CI, 4.53-5.93] / CP-COV03 300 mg dose group 3.99 [95% CI, 3.26-4.72]; Headache: p=0.0285, placebo group 5.48 [95% CI, 4.72-6.24] / CP-COV03 300 mg dose group 4.25 [95% CI, 3.46-5.04]; Fatigue: p=0.0116, placebo group 5.62 [95% CI, 4.81-6.42] / CP-COV03 300 mg dose group 4.10 [95% CI, 3.24-4.95]) (Annexure-I).

Authors' response to Reviewer's comments

The data presented in Table S4 represent mean values, which have been indicated in the caption as recommended by the Reviewer. Additionally, the table S4 now includes the number of participants for each symptom. The word "**median**" has also been included in line 129 in the revised manuscript.

Table S4. The mean number of days required for each targeted COVID-19 symptom to improve and be maintained for more than 48 hours as analyzed in the (a) ITT and (b) PPS groups.

(a)

Symptoms		Placebo vs 300mg		Placebo vs 450 mg	
		Placebo	300 mg	Placebo	450 mg
Fever	N	34	40	34	34
	LS Mean [95% CI]	5.0555 [4.0721- 6.0390]	4.9028 [3.9963- 5.8093]	5.0688 [3.9424- 6.1951]	5.2548 [4.1284- 6.3811]
Cough	N	94	94	94	95
	LS Mean [95% CI]	6.8921 [6.0420- 7.7421]	7.3154 [6.4653- 8.1654]	6.9225 [6.0637- 7.7814]	7.6030 [6.7487- 8.4573]
Sore throat	N	93	93	93	93
	LS Mean [95% CI]	5.3359 [4.6876- 5.9842]	4.5136 [3.8653- 5.1619]	5.3171 [4.7124- 5.9218]	4.3711 [3.7664- 4.9758]
Headache	N	89	93	89	88
	LS Mean [95% CI]	5.5485 [4.8478- 6.2493]	4.8890 [4.2036- 5.5745]	5.5161 [4.7887- 6.2435]	5.0121 [4.2806- 5.7437]
Muscle aches	N	88	89	88	83
	LS Mean [95% CI]	3.9062 [3.4625- 4.3498]	3.6714 [3.2303- 4.1126]	3.9124 [3.4272- 4.3976]	3.9122 [3.4125- 4.4119]
Chills or shivering	N	75	77	75	69
	LS Mean [95% CI]	3.2944 [2.9016- 3.6872]	3.4925 [3.1048- 3.8801]	3.2876 [2.8544- 3.7208]	3.5062 [3.0544- 3.9580]
Stuffy or runny nose	N	92	98	92	87
	LS Mean [95% CI]	7.6231 [6.7744- 8.4717]	7.6141 [6.7919- 8.4363]	7.6503 [6.8112- 8.4894]	7.7491 [6.8861- 8.6121]
Tiredness or low energy	N	89	92	89	89
	LS Mean [95% CI]	5.7485 [4.9780- 6.5190]	5.0259 [4.2680- 5.7837]	5.7468 [4.9465- 6.5471]	6.1352 [5.3349- 6.9355]

Authors' response to Reviewer's comments

Symptoms		Placebo vs 300mg		Placebo vs 450 mg	
		Placebo	300 mg	Placebo	450 mg
Difficulty breathing or shortness of breath	N	39	38	39	39
	LS Mean [95% CI]	4.5112 [3.4181-5.6043]	5.3964 [4.2890-6.5039]	4.4733 [3.3394-5.6072]	5.4370 [4.3030-6.5709]
Nausea	N	38	59	38	61
	LS Mean [95% CI]	4.8050 [3.7545-5.8555]	5.2442 [4.4015-6.0870]	4.7161 [3.8264-5.6059]	4.2752 [3.5741-4.9763]
Vomiting	N	11	16	11	13
	LS Mean [95% CI]	4.9981 [3.4724-6.5238]	4.6576 [3.3956-5.9195]	4.8465 [2.6869-7.0062]	4.3991 [2.4239-6.3743]
Diarrhea	N	52	75	52	81
	LS Mean [95% CI]	6.5094 [5.7014-7.3174]	6.7935 [6.1213-7.4656]	6.4949 [5.7007-7.2890]	7.2193 [6.5842-7.8545]

* Observed difference is statistically significant at the 5% level.

(b)

Symptoms		Placebo vs 300mg		Placebo vs 450mg	
		Placebo	300mg	Placebo	450 mg
Fever	N				
	LS Mean [95% CI]	5.0193 [4.0239-6.0148]	4.0877 [3.0497-5.1257]	4.9977 [3.6604-6.3350]	5.0599 [3.7486-6.3712]
Cough	N				
	LS Mean [95% CI]	6.9660 [6.0129-7.9192]	6.7315 [5.7298-7.7333]	6.9995 [6.0132-7.9858]	7.3929 [6.4383-8.3474]
Sore throat	N				
	LS Mean [95% CI]	5.2315 [4.5326-5.9304]	3.9941 [3.2645-4.7237]	5.2114 [4.5401-5.8828]	4.1437 [3.4900-4.7974]
Headache	N				
	LS Mean [95% CI]	5.4773 [4.7158-6.2387]	4.2476 [3.4573-5.0378]	5.4934 [4.7059-6.2809]	4.5338 [3.7626-5.3049]
Muscle aches	N				
	LS Mean [95% CI]	3.9680 [3.4910-4.4451]	3.3206 [2.8052-3.8360]	3.9665 [3.4025-4.5305]	3.8729 [3.3168-4.4290]

Authors' response to Reviewer's comments

Symptoms		Placebo vs 300mg		Placebo vs 450mg	
		Placebo	300mg	Placebo	450 mg
Chills or shivering	N				
	LS Mean [95% CI]	3.2403 [2.8996- 3.5810]	3.1174 [2.7552- 3.4797]	3.2301 [2.7644- 3.6958]	3.3724 [2.9142- 3.8306]
Stuffy or runny nose	N				
	LS Mean [95% CI]	7.4439 [6.5178- 8.3700]	6.5159 [5.5633- 7.4685]	7.4576 [6.5322- 8.3830]	7.5766 [6.6512- 8.5020]
Tiredness or low energy	N				
	LS Mean [95% CI]	5.6167 [4.8147- 6.4186]	4.0979 [3.2421- 4.9537]	5.6568 [4.7768- 6.5368]	5.9918 [5.1118- 6.8718]
Difficulty breathing or shortness of breath	N				
	LS Mean [95% CI]	4.4220 [3.1491- 5.6949]	5.4820 [4.1211- 6.8429]	4.3433 [3.0380- 5.6486]	5.4067 [4.1014- 6.7120]
Nausea	N				
	LS Mean [95% CI]	4.3755 [3.3439- 5.4072]	4.2529 [3.3091- 5.1967]	4.3745 [3.4220- 5.3269]	4.0734 [3.3466- 4.8003]
Vomiting	N				
	LS Mean [95% CI]	5.2535 [3.8310- 6.6759]	3.3465 [1.9241- 4.7690]	5.1902 [2.7372- 7.6433]	4.4634 [2.1331- 6.7937]
Diarrhea	N				
	LS Mean [95% CI]	6.4991 [5.6138- 7.3843]	6.1396 [5.3691- 6.9101]	6.4719 [5.5609- 7.3830]	7.0022 [6.2979- 7.7065]

* Observed difference is statistically significant at the 5% level.

Reviewer Comment 3:

Line 326: Add in “for medical reasons”. Presumably, not every single participant stayed in hospital for the full six days? The numbers who did complete the six-day stay should be mentioned.

Authors' Response: 97.0% (291 out of 300) of participants completed the three additional follow-up visits, and all of them were hospitalized for a minimum of six days. To explain the situation in South Korea at the time when the trial was conducted, COVID-19-infected individuals in South Korea were required to quarantine until May 30, 2022, and quarantine was highly recommended from June 1, 2022. To minimize unnecessary outings and movements, investigators conducted pre-screening consultations over the phone, allowing patients to self-assess their symptoms before visiting the clinical site for screening. This approach resulted in a few screen failures during the actual screening process.

Authors' response to Reviewer's comments

Three of the seventeen screening failures and six of the nine dropouts declined to participate in the study, withdrawing their consent prior to randomization. While the screening failure and dropout documentation did not specify the reasons for the consent withdrawals, it is suspected that most were likely related to hospitalization, especially given that the majority had mild symptoms.

It should be noted that not all participants were discharged on day six regardless of their symptoms. According to the protocol, the hospital discharge date could be extended based on symptoms or in compliance with the Korea Disease Control and Prevention Agency (KDCA) guidelines for COVID-19. This is clearly stated in the manuscript (**Refer pages 15-16, lines 378-383**).

To address reviewer 3's comment and clarify these points, the following sentence has been included in the revised manuscript:

“The study drug was administered to 293 patients, excluding seven who dropped out before administration, and 291 patients completed the trial, with all of them hospitalized for at least six days.”
(Refer Page 5, 109-111)

Reviewer Comment 4:

I remain critically concerned about the choice of primary outcome and reporting of secondary outcomes.

a) The protocol makes it clear that symptoms were ascertained until day 28. However, the primary outcome truncates these symptoms at day 14. This is a potential weakness and should be discussed. Whatever the wisdom of this decision, from what I can tell, there is a considerable amount of information on recovery that is not reported. All secondary outcomes should be reported.

b) Differences in recovery over the full 28-day period are critically important. As secondary outcomes and as these data have been collected according to the protocol, these and any differences between study groups should be reported in this publication.

c) Where are the findings, for example, rescue medication, secondary outcomes 9 and 10?

Authors' Response:

We understand your concerns about the choice of primary outcome and appreciate your emphasis on the need to report all secondary outcomes. In response:

a) We have included a detailed discussion in the manuscript to address the rationale for truncating the primary outcome at day 14, acknowledging this as a potential limitation. This discussion highlights the reasoning behind this decision while reflecting on its implications. (Please refer to Annexure-I) attached below.

b) We have reviewed and incorporated the data on all secondary outcomes, including differences in recovery over the full 28-day period. (Please refer to Annexure-II) attached below.

Accordingly, the following sentences have been included in the revised manuscript (**Refer Pages 13-14, lines 304-349**)

Authors' response to Reviewer's comments

In this study, the primary outcome (median number of days required for targeted COVID-19 symptoms to improve and be maintained for more than 48 hours until Day 14) requires additional consideration in light of the characteristics of COVID-19.

And any of the COVID-19 symptoms we ascertained persisted for a prolonged period, which were consistent with findings from other studies^{46,47}.

For example, one study found that 137 out of 290 COVID-19 patients (47.2%) experienced persistent symptoms even one month after infection⁴⁸. Similarly, the pivotal clinical trial for Paxlovid[®] (EPIC-HR), which tracked symptoms up to Day 28, reported that only 43.9% of patients in the treatment group achieved symptom alleviation by Day 28⁴⁹. In this study, when evaluating symptom improvement up to Day 14, 43.7% of trial participants (46 in the placebo group, 39 in the 300 mg group, and 43 in the 450 mg group from the ITT population) did not achieve a 48-hour sustained improvement of all 12 COVID-19 symptoms by Day 14. These findings suggest, consistent with prior studies, that many participants in this study also experienced prolonged COVID-19 symptoms.

Additionally, the sustained improvement date was censored as the last symptom assessment date in the Cox proportional hazards regression model analysis. Given these characteristics of COVID-19 and the applied analytical approach, evaluating symptoms only up to Day 14 poses a risk of incomplete assessment of the primary outcome. For instance, participants whose symptoms improved by Day 14 but relapsed afterward would be classified as having achieved symptom improvement. Conversely, participants whose symptoms had not improved by Day 14 but improved afterward would be classified as not achieving symptom improvement if the evaluation period is restricted to 14 days.

The clinical trial protocol, including the primary outcome, was designed in consultation with the MFDS during the planning stage. Similar to protocols for other respiratory infections like influenza, the study was designed to evaluate whether CP-COV03 could rapidly improve symptoms caused by acute COVID-19 infection within a 14-day timeframe. However, as noted earlier, considering the prolonged nature of symptoms in COVID-19, evaluating symptoms only up to Day 14 poses a risk of incomplete

Authors' response to Reviewer's comments

assessment of the primary outcome. To address this limitation, an ad-hoc analysis was conducted to evaluate symptom improvement up to Day 28.

The ad-hoc analysis results demonstrated that in the pre-determined analysis population, ITT, the time to symptom improvement in the 300 mg dose group was 3 days shorter than in the placebo group ($p=0.9305$, placebo group 13.50 [95% CI, 11.00–17.00] / 300 mg dose group 10.50 [95% CI, 9.00–18.00]). In the other pre-determined analysis population, PPS, the 300 mg dose group exhibited a 5.75-day shorter time to symptom improvement compared to the placebo group ($p=0.0275$, placebo group 15.00 [95% CI, 11.50–19.00] / 300 mg dose group 9.25 [95% CI, 7.50–12.50]) (Refer to Annexure-II). These findings indicate that even when symptom improvement is evaluated up to Day 28, the results align with those derived from the primary outcome evaluation up to Day 14. Specifically, the 300 mg dose group demonstrated faster symptom improvement compared to the placebo group.

Considering these findings, the follow-up Phase 3 study will take these limitations into account and plan to evaluate symptom improvement over an extended period of up to 28 days.

The other secondary outcomes and their results are presented in Annexure-II (Refer Page 19, line 464-465).

28 Days full data related to recovery

Time (days) for sustained improvement of all symptoms (ITT, n=293)	Placebo (n=98)	CP-COV03 300 mg (n=96)	CP-COV03 450 mg (n=99)
Median to Improvement [95% CI]	13.50 [11.00, 17.00]	10.50 [9.00, 18.00]	14.50 [12.50, 18.50]
Difference vs. placebo	-	3.00	-1.00
p-value	-	0.9305	0.2949

Authors' response to Reviewer's comments

Time (days) for sustained improvement of all symptoms (PPS, n=227)	Placebo (n=77)	CP-COV03 300 mg (n=70)	CP-COV03 450 mg (n=80)
Median to Improvement [95% CI]	15.00 [11.50, 19.00]	9.25 [7.50, 12.50]	13.25 [11.50, 18.50]
Difference vs. placebo	-	5.75	1.75
p-value	-	0.0275	0.8310

c) Specifically, regarding the rescue medication, please refer to Annexure-I (outcome-8) attached below: Also we have added findings for secondary outcomes 9 and 10, including rescue medication usage, into the results section and updated corresponding tables and supplementary materials. (Please refer to Annexure-I) attached below.

Additionally, these revisions have been included in the revised manuscript as below: **(Refer Page 9, lines 205-214)**

Rescue medicine and severity progression

In this clinical trial, acetaminophen, ibuprofen, and antidiarrheal medications were provided as rescue medicines. However, no significant differences were observed between the groups regarding the dose and dosing frequency. This indicates that the use of rescue medications did not affect the clinical trial results (Annexure-I).

Additionally, there were no significant differences between the groups in the proportion of participants experiencing severe COVID-19 progression from Day 1 to Day 28. This result aligns with the characteristic low rate of severe progression associated with the Omicron variant, which was the predominant variant during this clinical trial (Annexure-I)³³.

Detailed information can be seen in the attached Annexure-I as follows:

8) The dose and dosing frequency of Acetaminophen, Ibuprofen, and antidiarrheal from Day 1 to Day 28

For the placebo, CP-COV03 300 mg dose, and CP-COV03 450 mg dose group, from Day 1 to Day 28, the dosage and number of administrations for rescue medications including Acetaminophen, Ibuprofen,

Authors' response to Reviewer's comments

and antidiarrheals were detailed. Specifically, for Acetaminophen and Ibuprofen (for combination drugs, measure only Acetaminophen or Ibuprofen doses only), the number of participants administered (daily N) and dosages are presented in Tables 43–44. Descriptive statistics for the number of administrations of antidiarrheals are provided in Tables 45–46.

Additionally, an ANCOVA analysis, considering age and severity as covariates, was conducted on the total dosages of Acetaminophen and Ibuprofen administered during the period. Due to insufficient data, the number of administrations for antidiarrheals was not presented through ANCOVA analysis. The ANCOVA analysis results showed that there was no statistical significance between the placebo and the CP-COV03 groups on the total amounts of Acetaminophen and Ibuprofen used until Day 28, at the 5% significance level.

Acetaminophen, Ibuprofen		Placebo N=98	CP-COV03 300 mg N=99	CP-COV03 450 mg N=96
Day 1	N	19	21	17
	Mean	580.26	647.62	641.18
	SD	134.00	141.84	36.38
	Median	650.00	650.00	650.00
	Min	200.00	350.00	500.00
	Max	650.00	1000.00	650.00
Day 2	N	9	22	8
	Mean	652.78	681.82	650.00
	SD	168.84	102.99	0.00
	Median	650.00	650.00	650.00
	Min	325.00	650.00	650.00
	Max	1000.00	1000.00	650.00
Day 3	N	6	16	6
	Mean	595.83	693.75	650.00
	SD	132.68	119.55	0.00
	Median	650.00	650.00	650.00
	Min	325.00	650.00	650.00
	Max	650.00	1000.00	650.00
Day 4	N	5	3	4
	Mean	725.00	766.67	650.00
	SD	283.95	202.07	0.00
	Median	650.00	650.00	650.00
	Min	325.00	650.00	650.00
	Max	1000.00	1000.00	650.00
Day 5	N	6	2	2

Authors' response to Reviewer's comments

Acetaminophen, Ibuprofen		Placebo N=98	CP-COV03 300 mg N=99	CP-COV03 450 mg N=96
	Mean	654.17	825.00	650.00
	SD	213.55	247.49	0.00
	Median	650.00	825.00	650.00
	Min	325.00	650.00	650.00
	Max	1000.00	1000.00	650.00
Day 6	N	2	0	2
	Mean	662.50	-	650.00
	SD	477.30	-	0.00
	Median	662.50	-	650.00
	Min	325.00	-	650.00
	Max	1000.00	-	650.00
Day 7	N	2	0	1
	Mean	662.50	-	400.00
	SD	477.30	-	.
	Median	662.50	-	400.00
	Min	325.00	-	400.00
	Max	1000.00	-	400.00
Day 8	N	2	1	0
	Mean	662.50	1000.00	-
	SD	477.30	.	-
	Median	662.50	1000.00	-
	Min	325.00	1000.00	-
	Max	1000.00	1000.00	-
Day 9	N	3	0	0
	Mean	641.67	-	-
	SD	339.42	-	-
	Median	600.00	-	-
	Min	325.00	-	-
	Max	1000.00	-	-
Day 10	N	2	1	0
	Mean	662.50	1000.00	-
	SD	477.30	.	-
	Median	662.50	1000.00	-
	Min	325.00	1000.00	-

Authors' response to Reviewer's comments

Acetaminophen, Ibuprofen		Placebo N=98	CP-COV03 300 mg N=99	CP-COV03 450 mg N=96
	Max	1000.00	1000.00	-
Day 11	N	2	0	1
	Mean	662.50	-	300.00
	SD	477.30	-	.
	Median	662.50	-	300.00
	Min	325.00	-	300.00
	Max	1000.00	-	300.00
Day 12	N	1	0	0
	Mean	500.00	-	-
	SD	.	-	-
	Median	500.00	-	-
	Min	500.00	-	-
	Max	500.00	-	-
Day 13	N	1	0	1
	Mean	1000.00	-	300.00
	SD	.	-	.
	Median	1000.00	-	300.00
	Min	1000.00	-	300.00
	Max	1000.00	-	300.00
Day 14	N	0	1	0
	Mean	-	300.00	-
	SD	-	.	-
	Median	-	300.00	-
	Min	-	300.00	-
	Max	-	300.00	-
Day 15	N	0	0	0
	Mean	-	-	-
	SD	-	-	-
	Median	-	-	-
	Min	-	-	-
	Max	-	-	-
Day 16	N	1	0	0
	Mean	500.00	-	-
	SD	.	-	-
	Median	500.00	-	-

Authors' response to Reviewer's comments

Acetaminophen, Ibuprofen		Placebo N=98	CP-COV03 300 mg N=99	CP-COV03 450 mg N=96
	Min	500.00	-	-
	Max	500.00	-	-
Day 17	N	0	0	0
	Mean	-	-	-
	SD	-	-	-
	Median	-	-	-
	Min	-	-	-
	Max	-	-	-
Day 18	N	0	0	0
	Mean	-	-	-
	SD	-	-	-
	Median	-	-	-
	Min	-	-	-
	Max	-	-	-
Day 19	N	0	0	0
	Mean	-	-	-
	SD	-	-	-
	Median	-	-	-
	Min	-	-	-
	Max	-	-	-
Day 20	N	0	0	0
	Mean	-	-	-
	SD	-	-	-
	Median	-	-	-
	Min	-	-	-
	Max	-	-	-
Day 21	N	0	0	0
	Mean	-	-	-
	SD	-	-	-
	Median	-	-	-
	Min	-	-	-
	Max	-	-	-
Day 22	N	0	1	0
	Mean	-	1000.00	-
	SD	-	.	-

Authors' response to Reviewer's comments

Acetaminophen, Ibuprofen		Placebo N=98	CP-COV03 300 mg N=99	CP-COV03 450 mg N=96
	Median	-	1000.00	-
	Min	-	1000.00	-
	Max	-	1000.00	-
Day 23	N	0	0	0
	Mean	-	-	-
	SD	-	-	-
	Median	-	-	-
	Min	-	-	-
	Max	-	-	-
Day 24	N	0	0	0
	Mean	-	-	-
	SD	-	-	-
	Median	-	-	-
	Min	-	-	-
	Max	-	-	-
Day 25	N	0	1	0
	Mean	-	1000.00	-
	SD	-	.	-
	Median	-	1000.00	-
	Min	-	1000.00	-
	Max	-	1000.00	-
Day 26	N	0	1	0
	Mean	-	325.00	-
	SD	-	.	-
	Median	-	325.00	-
	Min	-	325.00	-
	Max	-	325.00	-
Day 27	N	0	0	0
	Mean	-	-	-
	SD	-	-	-
	Median	-	-	-
	Min	-	-	-
	Max	-	-	-
Day 28	N	0	0	0
	Mean	-	-	-

Authors' response to Reviewer's comments

Acetaminophen, Ibuprofen		Placebo N=98	CP-COV03 300 mg N=99	CP-COV03 450 mg N=96
	SD	-	-	-
	Median	-	-	-
	Min	-	-	-
	Max	-	-	-

Acetaminophen, Ibuprofen		Placebo N=77	CP-COV03 300 mg N=70	CP-COV03 450 mg N=80
Day 1	N			
	Mean			
	SD			
	Median			
	Min			
	Max			
Day 2	N			
	Mean			
	SD			
	Median			
	Min			
	Max			
Day 3	N			
	Mean			
	SD			
	Median			
	Min			
	Max			
Day 4	N	0		
	Mean	-		
	SD	-		
	Median	-		
	Min	-		
	Max	-		
Day 5	N			
	Mean			

Authors' response to Reviewer's comments

Acetaminophen, Ibuprofen		Placebo N=77	CP-COV03 300 mg N=70	CP-COV03 450 mg N=80
	SD			
	Median			
	Min			
	Max			
Day 6	N	0	0	
	Mean	-	-	
	SD	-	-	
	Median	-	-	
	Min	-	-	
	Max	-	-	
Day 7	N	0	0	0
	Mean	-	-	-
	SD	-	-	-
	Median	-	-	-
	Min	-	-	-
	Max	-	-	-
Day 8	N	0	0	0
	Mean	-	-	-
	SD	-	-	-
	Median	-	-	-
	Min	-	-	-
	Max	-	-	-
Day 9	N	0	0	0
	Mean	-	-	-
	SD	-	-	-
	Median	-	-	-
	Min	-	-	-
	Max	-	-	-
Day 10	N	0	0	0
	Mean	-	-	-
	SD	-	-	-
	Median	-	-	-
	Min	-	-	-
	Max	-	-	-
Day 11	N	0	0	

Authors' response to Reviewer's comments

Acetaminophen, Ibuprofen		Placebo N=77	CP-COV03 300 mg N=70	CP-COV03 450 mg N=80
	Mean	-	-	
	SD	-	-	
	Median	-	-	
	Min	-	-	
	Max	-	-	
Day 12	N	0	0	0
	Mean	-	-	-
	SD	-	-	-
	Median	-	-	-
	Min	-	-	-
	Max	-	-	-
Day 13	N	0	0	
	Mean	-	-	
	SD	-	-	
	Median	-	-	
	Min	-	-	
	Max	-	-	
Day 14	N	0		0
	Mean	-		-
	SD	-		-
	Median	-		-
	Min	-		-
	Max	-		-
Day 15	N	0	0	0
	Mean	-	-	-
	SD	-	-	-
	Median	-	-	-
	Min	-	-	-
	Max	-	-	-
Day 16	N		0	0
	Mean		-	-
	SD		-	-
	Median		-	-
	Min		-	-
	Max		-	-

Authors' response to Reviewer's comments

Acetaminophen, Ibuprofen		Placebo N=77	CP-COV03 300 mg N=70	CP-COV03 450 mg N=80
Day 17	N	0	0	0
	Mean	-	-	-
	SD	-	-	-
	Median	-	-	-
	Min	-	-	-
	Max	-	-	-
Day 18	N	0	0	0
	Mean	-	-	-
	SD	-	-	-
	Median	-	-	-
	Min	-	-	-
	Max	-	-	-
Day 19	N	0	0	0
	Mean	-	-	-
	SD	-	-	-
	Median	-	-	-
	Min	-	-	-
	Max	-	-	-
Day 20	N	0	0	0
	Mean	-	-	-
	SD	-	-	-
	Median	-	-	-
	Min	-	-	-
	Max	-	-	-
Day 21	N	0	0	0
	Mean	-	-	-
	SD	-	-	-
	Median	-	-	-
	Min	-	-	-
	Max	-	-	-
Day 22	N	0	0	0
	Mean	-	-	-
	SD	-	-	-
	Median	-	-	-
	Min	-	-	-

Authors' response to Reviewer's comments

Acetaminophen, Ibuprofen		Placebo N=77	CP-COV03 300 mg N=70	CP-COV03 450 mg N=80
	Max	-	-	-
Day 23	N	0	0	0
	Mean	-	-	-
	SD	-	-	-
	Median	-	-	-
	Min	-	-	-
	Max	-	-	-
Day 24	N	0	0	0
	Mean	-	-	-
	SD	-	-	-
	Median	-	-	-
	Min	-	-	-
	Max	-	-	-
Day 25	N	0	0	0
	Mean	-	-	-
	SD	-	-	-
	Median	-	-	-
	Min	-	-	-
	Max	-	-	-
Day 26	N	0	0	0
	Mean	-	-	-
	SD	-	-	-
	Median	-	-	-
	Min	-	-	-
	Max	-	-	-
Day 27	N	0	0	0
	Mean	-	-	-
	SD	-	-	-
	Median	-	-	-
	Min	-	-	-
	Max	-	-	-
Day 28	N	0	0	0
	Mean	-	-	-
	SD	-	-	-
	Median	-	-	-

Authors' response to Reviewer's comments

Acetaminophen, Ibuprofen		Placebo N=77	CP-COV03 300 mg N=70	CP-COV03 450 mg N=80
	Min	-	-	-
	Max	-	-	-

Antidiarrheals		Placebo N=98	CP-COV03 300 mg N=99	CP-COV03 450 mg N=96
Day 2	N	0	0	1
	Mean	-	-	1.00
	SD	-	-	0.00
	Median	-	-	1.00
	Min	-	-	1.00
	Max	-	-	1.00
Day 3	N	0	1	0
	Mean	-	1.00	-
	SD	-	0.00	-
	Median	-	1.00	-
	Min	-	1.00	-
	Max	-	1.00	-
Day 4	N	0	3	0
	Mean	-	1.00	-
	SD	-	0.00	-
	Median	-	1.00	-
	Min	-	1.00	-
	Max	-	1.00	-
Day 5	N	0	3	1
	Mean	-	1.00	1.00
	SD	-	0.00	0.00
	Median	-	1.00	1.00
	Min	-	1.00	1.00
	Max	-	1.00	1.00

Antidiarrheals	Placebo N=77	CP-COV03 300 mg	CP-COV03 450 mg
----------------	-----------------	--------------------	--------------------

Authors' response to Reviewer's comments

			N=70	N=80
Day 1-3	N	0	0	0
	Mean	-	-	-
	SD	-	-	-
	Median	-	-	-
	Min	-	-	-
	Max	-	-	-
Day 4	N	0	2	0
	Mean	-	1.00	-
	SD	-	0.00	-
	Median	-	1.00	-
	Min	-	1.00	-
	Max	-	1.00	-
Day 5	N	0	3	0
	Mean	-	1.00	-
	SD	-	0.00	-
	Median	-	1.00	-
	Min	-	1.00	-
	Max	-	1.00	-
Day 6-28	N	0	3	0
	Mean	-	1.00	-
	SD	-	0.00	-
	Median	-	1.00	-
	Min	-	1.00	-
	Max	-	1.00	-

Response related to outcomes 9 and 10 are also given in Annexure-I as follows which can be seen in Annexure-I attached below.

9) Proportion of participants with severe COVID-19 progression from Day 1 to Day 28

Over the study period from Day 1 through Day 28, there were no participants with severe COVID-19 progression.

Table 47. Secondary outcome 9) participants progressed to severe COVID-19 (ITT)

Participants who progressed to severe COVID-19 or not		Placebo N=98 n (%)	CP-COV03 300 mg N=99 n (%)	CP-COV03 450 mg N=96 n (%)
Day 1- Day 28	Total	98(100.0)	99(100.0)	96(100.0)
	Progressed to severity	0(0.0)	0(0.0)	0(0.0)
	Not progressed to severity	98(100.0)	99(100.0)	96(100.0)

10) Pharmacokinetic characteristics of niclosamide and correlation between pharmacokinetic parameters and viral load

Table 48 presents the descriptive statistics of the pharmacokinetic variables. In the CP-COV03 450 mg dose group, participant R03031 was dropped out before dosing, and for participant R03060, pharmacokinetic blood sampling was not conducted due to a consent error related to pharmacokinetic blood collection.

The geometric mean of C_{max} for the CP-COV03 300 mg dose group was 285.25 ng/mL (minimum 139.82, maximum 936.53), while for the CP-COV03 450 mg dose group, it was 389.90 ng/mL (minimum 129.39, maximum 1061.79).

Table 48. Secondary outcome 10) descriptive statistics of pharmacokinetic variables for niclosamide (ITT)

Treatment	Parameters	N	Arithmetic Mean	SD	Geometric Mean	Median	Min	Max
CP-COV03 300 mg (N=20)	C_{max} (ng/mL)	20	317.78	186.42	285.25	246.39	139.82	936.53
	AUC_t (ng·h/mL)	20	11046.21	3451.07	10562.09	9754.47	6479.00	17836.36
CP-COV03 450 mg (N=18)	C_{max} (ng/mL)	18	460.44	275.45	389.90	391.83	129.39	1061.79
	AUC_t (ng·h/mL)	18	13969.54	5880.64	12876.29	12912.31	6809.88	24605.15

Table 49. Mean and confidence interval for niclosamide AUC and Cmax (ITT)

	Geometric Mean	LSMean Ratio (T/R)		ANOVA-CV	
		Point Estimate	90% CI		
AUC _t	CP-COV03 300 mg	10562.09	1.2191	[1.0000, 1.4862]	36.12
	CP-COV03 450 mg	12876.29			
C _{max}	CP-COV03 300 mg	285.25	1.3669	[1.0277, 1.8180]	51.99
	CP-COV03 450 mg	389.90			

- Correlation between niclosamide and viral load

To determine the correlation between niclosamide AUC_t and viral load (RdRp gene) measured by qPCR, results using the CORR procedure (correlation analysis) of the SAS[®] program are presented in Table 50, and scatter plots are shown in Figure 1 (CP-COV03 300 mg dose) and Figure 2 (CP-COV03 450 mg dose). For the correlation analysis, niclosamide AUC_t was calculated for AUC_{day0}, AUC_{Day2}, AUC_{Day4}, and AUC_{Day6}, and correlation analysis was conducted with the viral load figures corresponding to each date.

There was a significant negative correlation between niclosamide AUC_t and viral load in both the CP-COV03 300 mg dose and 450 mg dose groups. Each correlation coefficient (p-value) was -0.3296 (p=0.0101) for the CP-COV03 300 mg dose group and -0.4818 (p=0.0009) for the CP-COV03 450 mg dose group, showing that the negative correlation was more significant in higher doses. In other words, it was found that as the administered dose increased, the viral load decreased more significantly.

Table 50. Secondary outcome 10) correlation between niclosamide and viral load (ITT)

Niclosamide AUC _t vs viral load		Niclosamide AUC _t	viral load
CP-COV03 300 mg	Pearson correlation coefficient	Niclosamide AUC _t	1.0000
		viral load	-0.3296
			1.0000

Niclosamide AUC _t vs viral load		Niclosamide AUC _t	viral load
	Significance probability	Niclosamide AUC _t	-
		viral load	0.0101
CP-COV03 450 mg	Pearson correlation coefficient	Niclosamide AUC _t	1.0000
		viral load	-0.4818
	Significance probability	Niclosamide AUC _t	
		viral load	0.0009

<Figure 1. Scatter plot of niclosamide AUC_t and viral load (qPCR-RdRp gene) for the CP-COV03 300 mg dose group>

<Figure 2. Scatter plot of niclosamide AUC_t and viral load (qPCR-RdRp gene)
for the CP-COV03 450 mg dose group>

** These figures 1 and 2 are already available in the main text as combined one (**Figure 2**)

These additions ensure comprehensive reporting, improving the robustness of our findings and addressing the critical points raised in your review.

We are confident that these revisions address your concerns and enhance the overall quality and clarity of the manuscript. Thank you once again for your valuable feedback, which has been instrumental in strengthening this work.

Annexure-I

Secondary Outcome Results

1) The time (days) taken to sustained improvement of each symptom for more than 48 hours by Day 14

Descriptive statistics for the time taken to sustain symptom improvement for over 48 hours by Day 14 for each COVID-19 symptoms are presented for the placebo, CP-COV03 300 mg dose, and CP-COV03 450 mg dose group in Tables 1–2.

Participants who did not improve or sustain symptom improvement for more than 48 hours by the evening of Day 14 were censored to have the maximum improvement time as 13.00 days (maximum value 13 days = 14.5 days - (Day 1 of initial dose + 0.5 days)).

ANCOVA analysis was conducted for each symptom, considering age and severity as covariates, with results presented in Tables 3-4.

Table 1. Secondary outcome 1) descriptive statistics (ITT)

Symptom		Placebo N=98	CP-COV03 300 mg N=99	CP-COV03 450 mg N=96
Fever	N	34	40	34
	Mean	5.09	4.88	5.24
	SD	2.97	2.88	3.53
	Median	3.50	3.50	3.50
	Min	2.50	2.50	2.50
	Max	13.00	13.00	13.00
Cough	N	94	94	95
	Mean	6.91	7.29	7.61
	SD	4.16	4.27	4.23
	Median	5.50	6.00	6.50
	Min	2.50	1.50	2.50
	Max	13.00	13.00	13.00
Sore throat	N	93	93	93
	Mean	5.32	4.53	4.37
	SD	3.32	2.98	2.49
	Median	4.00	3.50	3.50
	Min	2.50	2.50	2.50
	Max	13.00	13.00	13.00
Headache	N	89	93	88

Authors' response to Reviewer's comments

Symptom		Placebo N=98	CP-COV03 300 mg N=99	CP-COV03 450 mg N=96
	Mean	5.54	4.89	4.98
	SD	3.76	3.10	3.21
	Median	3.50	3.50	3.50
	Min	2.50	2.50	2.50
	Max	13.00	13.00	13.00
Muscle ache	N	88	89	83
	Mean	3.89	3.69	3.94
	SD	2.22	2.01	2.39
	Median	3.00	3.00	3.00
	Min	2.50	1.50	2.50
	Max	13.00	13.00	13.00
Chill	N	75	77	69
	Mean	3.28	3.51	3.51
	SD	1.61	1.94	2.13
	Median	2.50	2.50	2.50
	Min	2.50	2.50	2.50
	Max	12.00	13.00	13.00
Stuffy or runny nose	N	92	98	87
	Mean	7.61	7.62	7.79
	SD	4.23	4.01	3.87
	Median	6.75	6.25	7.00
	Min	2.50	2.50	2.50
	Max	13.00	13.00	13.00
Fatigue	N	89	92	89
	Mean	5.73	5.04	6.15
	SD	3.80	3.72	3.93
	Median	3.50	3.25	4.00
	Min	2.50	2.50	2.50
	Max	13.00	13.00	13.00
Difficulty of breathing	N	39	38	39
	Mean	4.54	5.37	5.37
	SD	2.86	3.97	3.99
	Median	3.50	3.00	3.50
	Min	2.50	2.50	2.50
	Max	13.00	13.00	13.00

Authors' response to Reviewer's comments

Symptom		Placebo N=98	CP-COV03 300 mg N=99	CP-COV03 450 mg N=96
Nausea	N	38	59	61
	Mean	4.76	5.27	4.25
	SD	3.15	3.43	2.46
	Median	3.00	4.00	3.50
	Min	2.50	2.50	2.50
	Max	13.00	13.00	13.00
Vomiting	N	11	16	13
	Mean	4.59	4.94	4.62
	SD	3.69	2.78	2.91
	Median	3.00	4.00	3.50
	Min	2.50	2.50	2.50
	Max	12.00	13.00	13.00
Diarrhea	N	52	75	81
	Mean	6.42	6.85	7.27
	SD	3.22	2.85	2.67
	Median	6.00	6.50	7.50
	Min	2.50	2.50	2.50
	Max	13.00	13.00	13.00

Table 2. Secondary outcome 1) descriptive statistics (PPS)

Symptom		Placebo N=77	CP-COV03 300 mg N=70	CP-COV03 450 mg N=80
Fever	N	25	23	26
	Mean	5.00	4.11	5.06
	SD	2.88	1.80	3.59
	Median	3.50	3.50	3.50
	Min	2.50	2.50	2.50
	Max	13.00	9.00	13.00
Cough	N	74	67	79
	Mean	6.99	6.71	7.41
	SD	4.25	4.09	4.29
	Median	5.25	5.00	6.00
	Min	2.50	2.50	2.50
	Max	13.00	13.00	13.00

Authors' response to Reviewer's comments

Symptom		Placebo N=77	CP-COV03 300 mg N=70	CP-COV03 450 mg N=80
Sore throat	N	73	67	77
	Mean	5.21	4.02	4.15
	SD	3.39	2.62	2.33
	Median	4.00	3.00	3.50
	Min	2.50	2.50	2.50
	Max	13.00	13.00	13.00
Headache	N	70	65	73
	Mean	5.49	4.24	4.54
	SD	3.74	2.59	2.88
	Median	3.50	3.50	3.50
	Min	2.50	2.50	2.50
	Max	13.00	13.00	13.00
Muscle ache	N	70	60	72
	Mean	3.96	3.33	3.88
	SD	2.36	1.53	2.39
	Median	3.00	3.00	3.00
	Min	2.50	2.50	2.50
	Max	13.00	13.00	13.00
Chill	N	61	54	63
	Mean	3.23	3.13	3.37
	SD	1.57	1.04	2.03
	Median	2.50	2.75	2.50
	Min	2.50	2.50	2.50
	Max	12.00	7.00	13.00
Stuffy or runny nose	N	73	69	73
	Mean	7.41	6.55	7.62
	SD	4.35	3.65	3.79
	Median	6.00	5.00	7.00
	Min	2.50	2.50	2.50
	Max	13.00	13.00	13.00
Fatigue	N	74	65	74
	Mean	5.59	4.13	6.06
	SD	3.89	3.05	3.96
	Median	3.50	3.00	4.00
	Min	2.50	2.50	2.50

Authors' response to Reviewer's comments

Symptom		Placebo N=77	CP-COV03 300 mg N=70	CP-COV03 450 mg N=80
	Max	13.00	13.00	13.00
Difficulty of breathing	N	32	28	32
	Mean	4.39	5.52	5.36
	SD	3.00	4.18	4.12
	Median	3.00	3.25	3.00
	Min	2.50	2.50	2.50
	Max	13.00	13.00	13.00
Nausea	N	31	37	53
	Mean	4.34	4.28	4.09
	SD	2.90	2.79	2.48
	Median	3.00	3.00	3.50
	Min	2.50	2.50	2.50
	Max	13.00	13.00	13.00
Vomiting	N	10	10	11
	Mean	4.80	3.80	4.82
	SD	3.82	1.57	3.13
	Median	3.00	3.25	3.50
	Min	2.50	2.50	2.50
	Max	12.00	7.00	13.00
Diarrhea	N	41	54	68
	Mean	6.38	6.23	7.06
	SD	3.31	2.61	2.67
	Median	5.50	6.25	7.00
	Min	2.50	2.50	2.50
	Max	13.00	13.00	13.00

Table 3. Secondary outcome 1) ANCOVA results (ITT)

Symptom	Placebo vs CP-COV03 300 mg			Placebo vs CP-COV03 450 mg		
	LS Mean [95% CI]		p-value	LS Mean [95% CI]		p-value
	Placebo	300 mg		Placebo	450 mg	
Fever	34	40	0.8208	34	34	0.8164
	5.0555 [4.0721, 6.0390]	4.9028 [3.9963, 5.8093]		5.0688 [3.9424, 6.1951]	5.2548 [4.1284, 6.3811]	
Cough	94	94		94	95	

Authors' response to Reviewer's comments

Symptom	Placebo vs CP-COV03 300 mg			Placebo vs CP-COV03 450 mg		
	LS Mean [95% CI]		p-value	LS Mean [95% CI]		p-value
	Placebo	300 mg		Placebo	450 mg	
	6.8921 [6.0420, 7.7421]	7.3154 [6.4653, 8.1654]	0.4883	6.9225 [6.0637, 7.7814]	7.6030 [6.7487, 8.4573]	0.2699
Sore throat	93	93		93	93	
	5.3359 [4.6876, 5.9842]	4.5136 [3.8653, 5.1619]	0.0789	5.3171 [4.7124, 5.9218]	4.3711 [3.7664, 4.9758]	0.0305*
Headache	89	93		89	88	
	5.5485 [4.8478, 6.2493]	4.8890 [4.2036, 5.5745]	0.1863	5.5161 [4.7887, 6.2435]	5.0121 [4.2806, 5.7437]	0.3370
Muscle ache	88	89		88	83	
	3.9062 [3.4625, 4.3498]	3.6714 [3.2303, 4.1126]	0.4605	3.9124 [3.4272, 4.3976]	3.9122 [3.4125, 4.4119]	0.9995
Chill	75	77		75	69	
	3.2944 [2.9016, 3.6872]	3.4925 [3.1048, 3.8801]	0.4799	3.2876 [2.8544, 3.7208]	3.5062 [3.0544, 3.9580]	0.4931
Stuffy or runny nose	92	98		92	87	
	7.6231 [6.7744, 8.4717]	7.6141 [6.7919, 8.4363]	0.9881	7.6503 [6.8112, 8.4894]	7.7491 [6.8861, 8.6121]	0.8719
Fatigue	89	92		89	89	
	5.7485 [4.9780, 6.5190]	5.0259 [4.2680, 5.7837]	0.1888	5.7468 [4.9465, 6.5471]	6.1352 [5.3349, 6.9355]	0.4994
Difficulty of breathing	39	38		39	39	
	4.5112 [3.4181, 5.6043]	5.3964 [4.2890, 6.5039]	0.2610	4.4733 [3.3394, 5.6072]	5.4370 [4.3030, 6.5709]	0.2413
Nausea	38	59		38	61	
	4.8050 [3.7545, 5.8555]	5.2442 [4.4015, 6.0870]	0.5191	4.7161 [3.8264, 5.6059]	4.2752 [3.5741, 4.9763]	0.4431
Vomiting	11	16		11	13	
	4.9981 [3.4724, 6.5238]	4.6576 [3.3956, 5.9195]	0.7269	4.8465 [2.6869, 7.0062]	4.3991 [2.4239, 6.3743]	0.7609
Diarrhea	52	75		52	81	

Authors' response to Reviewer's comments

Symptom	Placebo vs CP-COV03 300 mg			Placebo vs CP-COV03 450 mg		
	LS Mean [95% CI]		p-value	LS Mean [95% CI]		p-value
	Placebo	300 mg		Placebo	450 mg	
	6.5094 [5.7014, 7.3174]	6.7935 [6.1213, 7.4656]	0.5945	6.4949 [5.7007, 7.2890]	7.2193 [6.5842, 7.8545]	0.1627

* Statistically significant difference was observed at the 5% significance level

Table 4. Secondary outcome 1) ANCOVA results (PPS)

Symptom	Placebo vs CP-COV03 300 mg			Placebo vs CP-COV03 450 mg		
	LS Mean [95% CI]		p-value	LS Mean [95% CI]		p-value
	Placebo	300 mg		Placebo	450 mg	
Fever	25	23	0.1993	25	26	0.9470
	5.0193 [4.0239, 6.0148]	4.0877 [3.0497, 5.1257]		4.9977 [3.6604, 6.3350]	5.0599 [3.7486, 6.3712]	
Cough	74	67	0.7381	74	79	0.5723
	6.9660 [6.0129, 7.9192]	6.7315 [5.7298, 7.7333]		6.9995 [6.0132, 7.9858]	7.3929 [6.4383, 8.3474]	
Sore throat	73	67	0.0168*	73	77	0.0259*
	5.2315 [4.5326, 5.9304]	3.9941 [3.2645, 4.7237]		5.2114 [4.5401, 5.8828]	4.1437 [3.4900, 4.7974]	
Headache	70	65	0.0285*	70	73	0.0875
	5.4773 [4.7158, 6.2387]	4.2476 [3.4573, 5.0378]		5.4934 [4.7059, 6.2809]	4.5338 [3.7626, 5.3049]	
Muscle ache	70	60	0.0708	70	72	0.8158
	3.9680 [3.4910, 4.4451]	3.3206 [2.8052, 3.8360]		3.9665 [3.4025, 4.5305]	3.8729 [3.3168, 4.4290]	
Chill	61	54	0.6263	61	63	0.6672
	3.2403 [2.8996, 3.5810]	3.1174 [2.7552, 3.4797]		3.2301 [2.7644, 3.6958]	3.3724 [2.9142, 3.8306]	
Stuffy or runny nose	73	69	0.1699	73	73	0.8579
	7.4439 [6.5178, 8.3700]	6.5159 [5.5633, 7.4685]		7.4576 [6.5322, 8.3830]	7.5766 [6.6512, 8.5020]	
Fatigue	74	65		74	73	

Authors' response to Reviewer's comments

Symptom	Placebo vs CP-COV03 300 mg			Placebo vs CP-COV03 450 mg		
	LS Mean [95% CI]		p-value	LS Mean [95% CI]		p-value
	Placebo	300 mg		Placebo	450 mg	
	5.6167 [4.8147, 6.4186]	4.0979 [3.2421, 4.9537]	0.0116*	5.6568 [4.7768, 6.5368]	5.9918 [5.1118, 6.8718]	0.5960
	32	28		32	32	
Difficulty of breathing	4.4220 [3.1491, 5.6949]	5.4820 [4.1211, 6.8429]	0.2596	4.3433 [3.0380, 5.6486]	5.4067 [4.1014, 6.7120]	0.2596
	31	37		31	53	
Nausea	4.3755 [3.3439, 5.4072]	4.2529 [3.3091, 5.1967]	0.8619	4.3745 [3.4220, 5.3269]	4.0734 [3.3466, 4.8003]	0.6197
	10	10		10	11	
Vomiting	5.2535 [3.8310, 6.6759]	3.3465 [1.9241, 4.7690]	0.0665	5.1902 [2.7372, 7.6433]	4.4634 [2.1331, 6.7937]	0.6678
	41	54		41	68	
Diarrhea	6.4991 [5.6138, 7.3843]	6.1396 [5.3691, 6.9101]	0.5462	6.4719 [5.5609, 7.3830]	7.0022 [6.2979, 7.7065]	0.3672

* Statistically significant difference was observed at the 5% significance level

In the ITT analysis, ANCOVA results indicated a significant difference at the 5% significance level in the time of sustained symptom improvement for more than 48 hours in the sore throat symptom between the placebo and CP-COV03 450 mg dose group ($p=0.0305$). The LSMEAN [95% confidence interval] was 5.32 [4.71, 5.92] for the placebo group and 4.37 [3.77, 4.98] for CP-COV03 450 mg dose group, indicating that symptom improvement for sore throat occurred 0.95 days faster in CP-COV03 450 mg dose group compared to the placebo group.

In the PPS analysis, ANCOVA results at the 5% significance level showed significant differences in sore throat, headache, and fatigue symptoms between the placebo group and CP-COV03 300 mg dose group. CP-COV03 300 mg dose group showed faster symptom improvement by 1.24 days for sore throat, 1.23 days for headache, and 1.52 days for fatigue. The time of sustained symptom improvement for more than 48 hours [95% confidence interval] for significant differences were as follows:

Sore throat ($p=0.0168$), placebo group 5.23 [4.53, 5.93] / CP-COV03 300 mg dose group 3.99 [3.26, 4.72]

Headache ($p=0.0285$), placebo group 5.48 [4.72, 6.24] / CP-COV03 300 mg dose group 4.25 [3.46, 5.04]

Fatigue ($p=0.0116$), placebo group 5.62 [4.81, 6.42] / CP-COV03 300 mg dose group 4.10 [3.24, 4.95]

Additionally, significant difference ($p=0.0259$) was observed in the time of sustained symptom improvement for more than 48 hours in sore throat between the placebo and CP-COV03 450 mg dose

Authors' response to Reviewer's comments

group, with LSMEAN [95% confidence interval] of 5.21 [4.54, 5.88] for the placebo group and 4.14 [3.49, 4.80] for CP-COV03 450 mg dose group, indicating faster symptom improvement by 1.07 days in CP-COV03 450 mg dose group.

2) Time (days) taken improvement of each COVID-19 symptom by Day 14

Descriptive statistics for the time (days) taken improvement of each COVID-19 symptom by Day 14 for the placebo, CP-COV03 300 mg dose, and CP-COV03 450 mg dose group were presented in Tables 5–6 by symptom.

Participants not improved symptoms by the evening of Day 14.5 were censored to have a maximum improvement time as 13.00 days (maximum value 13 days = 14.5 days - (first day of administration + 0.5 day)). ANCOVA analysis was conducted for each symptom, considering age and Severity as covariates, and the results were presented in Tables 7–8.

Table 5. Secondary outcome 2) descriptive statistics (ITT)

Symptom		Placebo N=98	CP-COV03 300 mg N=99	CP-COV03 450 mg N=96
Fever	N	34	40	34
	Mean	3.18	2.93	3.40
	SD	3.22	3.04	3.91
	Median	1.50	1.50	1.50
	Min	0.50	0.50	0.50
	Max	13.00	13.00	13.00
Cough	N	94	94	95
	Mean	5.30	5.80	6.10
	SD	4.76	4.98	4.89
	Median	3.50	4.00	4.50
	Min	0.50	-0.50*	0.50
	Max	13.00	13.00	13.00
Sore throat	N	93	93	93
	Mean	3.45	2.67	2.43
	SD	3.64	3.38	2.71
	Median	2.00	1.50	1.50
	Min	0.50	0.50	0.50
	Max	13.00	13.00	13.00
Headache	N	89	93	88
	Mean	3.73	3.03	3.10

Authors' response to Reviewer's comments

Symptom		Placebo N=98	CP-COV03 300 mg N=99	CP-COV03 450 mg N=96
	SD	4.15	3.48	3.53
	Median	1.50	1.50	1.50
	Min	0.50	0.50	0.50
	Max	13.00	13.00	13.00
Muscle ache	N	88	89	83
	Mean	1.91	1.71	1.96
	SD	2.32	2.12	2.46
	Median	1.00	1.00	1.00
	Min	0.50	-0.50*	0.50
	Max	13.00	13.00	12.50
Chill	N	75	77	69
	Mean	1.28	1.53	1.57
	SD	1.61	2.08	2.37
	Median	0.50	0.50	0.50
	Min	0.50	0.50	0.50
	Max	10.00	13.00	13.00
Stuffy or runny nose	N	92	98	87
	Mean	6.01	6.00	6.17
	SD	4.78	4.56	4.42
	Median	4.75	4.25	5.00
	Min	0.50	0.50	0.50
	Max	13.00	13.00	13.00
Fatigue	N	89	92	89
	Mean	4.02	3.28	4.40
	SD	4.39	4.24	4.39
	Median	1.50	1.25	2.00
	Min	0.50	0.50	0.50
	Max	13.00	13.00	13.00
Difficulty of breathing	N	39	38	39
	Mean	2.59	3.64	3.63
	SD	3.03	4.53	4.52
	Median	1.50	1.00	1.50
	Min	0.50	0.50	0.50
	Max	13.00	13.00	13.00
Nausea	N	38	59	61

Authors' response to Reviewer's comments

Symptom		Placebo N=98	CP-COV03 300 mg N=99	CP-COV03 450 mg N=96
	Mean	2.82	3.41	2.28
	SD	3.30	3.76	2.59
	Median	1.00	2.00	1.50
	Min	0.50	0.50	0.50
	Max	12.50	13.00	13.00
Vomiting	N	11	16	13
	Mean	2.59	3.03	2.73
	SD	3.69	3.08	3.28
	Median	1.00	2.00	1.50
	Min	0.50	0.50	0.50
	Max	10.00	12.50	12.50
Diarrhea	N	52	75	81
	Mean	4.46	5.01	5.41
	SD	3.31	3.20	3.01
	Median	4.00	4.50	5.50
	Min	0.50	0.50	0.50
	Max	13.00	13.00	13.00

* In CP-COV03 300 mg dose group, one participant (S02008) experienced an improvement in cough and muscle ache in the morning of Day 2. However, as the first IP administration did not occur until the noon of Day 2 (resulting in a missed Day 1 night and Day 2 morning dose), the formula for calculating the time (days) required for symptom improvement resulted in a negative number. The calculation was based on the formula: (Day of symptom improvement observation - (First day of administration + 0.5 days)) = 2 (morning of Day 2) - (2 (Day 2 of first administration) + 0.5) = -0.5.

Table 6. Secondary outcome 2) descriptive statistics (PPS)

Symptom		Placebo N=77	CP-COV03 300 mg N=70	CP-COV03 450 mg N=80
Fever	N	25	23	26
	Mean	3.08	2.11	3.21
	SD	3.13	1.80	3.96
	Median	1.50	1.50	1.50
	Min	0.50	0.50	0.50
	Max	13.00	7.00	13.00
Cough	N	74	67	79

Authors' response to Reviewer's comments

Symptom		Placebo N=77	CP-COV03 300 mg N=70	CP-COV03 450 mg N=80
	Mean	5.42	5.09	5.89
	SD	4.89	4.71	4.96
	Median	3.25	3.00	4.00
	Min	0.50	0.50	0.50
	Max	13.00	13.00	13.00
Sore throat	N	73	67	77
	Mean	3.34	2.14	2.19
	SD	3.72	3.05	2.52
	Median	2.00	1.00	1.50
	Min	0.50	0.50	0.50
	Max	13.00	13.00	13.00
Headache	N	70	65	73
	Mean	3.66	2.33	2.65
	SD	4.11	2.92	3.23
	Median	1.50	1.50	1.50
	Min	0.50	0.50	0.50
	Max	13.00	13.00	13.00
Muscle ache	N	70	60	72
	Mean	1.99	1.37	1.90
	SD	2.48	1.75	2.48
	Median	1.00	1.00	1.00
	Min	0.50	0.50	0.50
	Max	13.00	13.00	12.50
Chill	N	61	54	63
	Mean	1.23	1.13	1.43
	SD	1.57	1.04	2.31
	Median	0.50	0.75	0.50
	Min	0.50	0.50	0.50
	Max	10.00	5.00	13.00
Stuffy or runny nose	N	73	69	73
	Mean	5.80	4.72	5.96
	SD	4.89	3.98	4.31
	Median	4.00	3.00	5.00
	Min	0.50	0.50	0.50
	Max	13.00	13.00	13.00

Authors' response to Reviewer's comments

Symptom		Placebo N=77	CP-COV03 300 mg N=70	CP-COV03 450 mg N=80
Fatigue	N	74	65	74
	Mean	3.89	2.28	4.30
	SD	4.48	3.51	4.42
	Median	1.50	1.00	2.00
	Min	0.50	0.50	0.50
	Max	13.00	13.00	13.00
Difficulty of breathing	N	32	28	32
	Mean	2.45	3.82	3.61
	SD	3.20	4.76	4.62
	Median	1.00	1.25	1.00
	Min	0.50	0.50	0.50
	Max	13.00	13.00	13.00
Nausea	N	31	37	53
	Mean	2.35	2.34	2.13
	SD	2.95	2.97	2.63
	Median	1.00	1.00	1.50
	Min	0.50	0.50	0.50
	Max	11.50	13.00	13.00
Vomiting	N	10	10	11
	Mean	2.80	1.80	2.95
	SD	3.82	1.57	3.53
	Median	1.00	1.25	1.50
	Min	0.50	0.50	0.50
	Max	10.00	5.00	12.50
Diarrhea	N	41	54	68
	Mean	4.43	4.33	5.20
	SD	3.43	2.89	3.01
	Median	3.50	4.25	5.00
	Min	0.50	0.50	0.50
	Max	13.00	13.00	13.00

Authors' response to Reviewer's comments

Table 7. Secondary outcome 2) ANCOVA results (ITT)

Symptom	Placebo vs CP-COV03 300 mg			Placebo vs CP-COV03 450 mg		
	LS Mean [95% CI]		p-value	LS Mean [95% CI]		p-value
	Placebo	300 mg		Placebo	450 mg	
Fever	34	40	0.7937	34	34	0.7675
	3.1429 [2.0867, 4.1990]	2.9536 [1.9801, 3.9271]		3.1565 [1.9181, 4.3949]	3.4170 [2.1786, 4.6554]	
Cough	94	94	0.4332	94	95	0.2738
	5.2764 [4.2937, 6.2592]	5.8299 [4.8472, 6.8127]		5.3134 [4.3253, 6.3015]	6.0899 [5.1070, 7.0728]	
Sore throat	93	93	0.1143	93	93	0.0331*
	3.4675 [2.7457, 4.1892]	2.6454 [1.9237, 3.3672]		3.4477 [2.7863, 4.1090]	2.4287 [1.7673, 3.0900]	
Headache	89	93	0.1985	89	88	0.3296
	3.7346 [2.9571, 4.5121]	3.0228 [2.2623, 3.7834]		3.6986 [2.8966, 4.5006]	3.1344 [2.3278, 3.9409]	
Muscle ache	88	89	0.4785	88	83	0.9807
	1.9298 [1.4636, 2.3961]	1.6930 [1.2294, 2.1566]		1.9371 [1.4336, 2.4406]	1.9282 [1.4096, 2.4467]	
Chill	75	77	0.4453	75	69	0.4360
	1.2952 [0.8868, 1.7036]	1.5177 [1.1147, 1.9207]		1.2893 [0.8258, 1.7528]	1.5551[1.071 7, 2.0385]	
Stuffy or runny nose	92	98	0.9735	92	87	0.9048
	6.0169 [5.0557, 6.9781]	5.9943 [5.0630, 6.9257]		6.0461 [5.0935, 6.9988]	6.1294 [5.1495, 7.1092]	
Fatigue	89	92	0.2137	89	89	0.6007
	4.0423 [3.1582, 4.9265]	3.2579 [2.3884, 4.1275]		4.0398 [3.1306, 4.9489]	4.3816 [3.4725, 5.2907]	
Difficulty of breathing	39	38	0.2074	39	39	0.2065
	2.5637 [1.3456, 3.7818]	3.6715 [2.4374, 4.9055]		2.5321 [1.2740, 3.7902]	3.6858 [2.4277, 4.9439]	
Nausea	38	59	0.4801	38	61	0.4497
	2.8590 [1.7240, 3.9941]	3.3789 [2.4683, 4.2895]		2.7657 [1.8324, 3.6990]	2.3099 [1.5745, 3.0453]	

Authors' response to Reviewer's comments

Symptom	Placebo vs CP-COV03 300 mg			Placebo vs CP-COV03 450 mg		
	LS Mean [95% CI]		p-value	LS Mean [95% CI]		p-value
	Placebo	300 mg		Placebo	450 mg	
Vomiting	11	16	0.7760	11	13	0.8349
	3.0223 [1.4395, 4.6050]	2.7347 [1.4255, 4.0438]		2.8424 [0.5548, 5.1300]	2.5179 [0.4257, 4.6101]	
Diarrhea	52	75	0.5048	52	81	0.1435
	4.5566 [3.6849, 5.4283]	4.9407 [4.2156, 5.6659]		4.5380 [3.6802, 5.3959]	5.3583 [4.6722, 6.0444]	

* Statistically significant difference was observed at the 5% significance level

Table 8. Secondary outcome 2) ANCOVA results (PPS)

Symptom	Placebo vs CP-COV03 300 mg			Placebo vs CP-COV03 450 mg		
	LS Mean [95% CI]		p-value	LS Mean [95% CI]		p-value
	Placebo	300 mg		Placebo	450 mg	
Fever	25	23	0.1905	25	26	0.8942
	3.0996 [2.0396, 4.1596]	2.0874 [0.9821, 3.1927]		3.0775 [1.6117, 4.5432]	3.2139 [1.7767, 4.6512]	
Cough	74	67	0.7282	74	79	0.5845
	5.3958 [4.2978, 6.4937]	5.1151 [3.9611, 6.2691]		5.4364 [4.2984, 6.5744]	5.8760 [4.7746, 6.9774]	
Sore throat	73	67	0.0308*	73	77	0.0269*
	3.3687 [2.5825, 4.1549]	2.1132 [1.2925, 2.9340]		3.3485 [2.6147, 4.0824]	2.1890 [1.4746, 2.9035]	
Headache	70	65	0.0353*	70	73	0.0993
	3.6468 [2.8053, 4.4883]	2.3419 [1.4686, 3.2152]		3.6659 [2.7947, 4.5370]	2.6423 [1.7893, 3.4954]	
Muscle ache	70	60	0.0940	70	72	0.8038
	1.9980 [1.4842, 2.5118]	1.3523 [0.7972, 1.9074]		1.9964 [1.4084, 2.5845]	1.8924 [1.3126, 2.4722]	
Chill	61	54	0.6263	61	63	0.5867
	1.2403 [0.8996, 1.5810]	1.1174 [0.7552, 1.4797]		1.2312 [0.7251, 1.7374]	1.4269 [0.9289, 1.9249]	
	73	69		73	73	

Authors' response to Reviewer's comments

Symptom	Placebo vs CP-COV03 300 mg			Placebo vs CP-COV03 450 mg		
	LS Mean [95% CI]		p-value	LS Mean [95% CI]		p-value
	Placebo	300 mg		Placebo	450 mg	
Stuffy or runny nose	5.8291 [4.7960, 6.8623]	4.6953 [3.6325, 5.7580]	0.1331	5.8477 [4.7997, 6.8956]	5.9126 [4.8647, 6.9605]	0.9312
Fatigue	74 3.9163 [2.9905, 4.8421]	65 2.2491 [1.2612, 3.2371]	0.0163*	74 3.9609 [2.9613, 4.9605]	74 4.2283 [3.2287, 5.2279]	0.7094
Difficulty of breathing	32 2.4899 [1.0772, 3.9025]	28 3.7794 [2.2692, 5.2897]	0.2170	32 2.4090 [0.9672, 3.8507]	32 3.6535 [2.2118, 5.0953]	0.2326
Nausea	31 2.3897 [1.3120, 3.4673]	37 2.3087 [1.3228, 3.2946]	0.9124	31 2.3916 [1.3990, 3.3842]	53 2.1106 [1.3531, 2.8680]	0.6566
Vomiting	10 3.2535 [1.8310, 4.6759]	10 1.3465 [-0.0759, 2.7690]	0.0665	10 3.1891 [0.5858, 5.7925]	11 2.6008 [0.1276, 5.0739]	0.7431
Diarrhea	41 4.5656 [3.6295, 5.5017]	54 4.2280 [3.4133, 5.0427]	0.5918	41 4.5324 [3.5436, 5.5212]	68 5.1349 [4.3704, 5.8993]	0.3454

* Statistically significant difference was observed at the 5% significance level

† If the lower limit of the 95% CI for the LS mean is negative, it indicates that the estimate value is close to zero (same hereafter).

In the ITT analysis, the ANCOVA results showed statistically significant difference at the 5% level in the time (days) taken for the improvement of sore throat symptoms between the placebo and CP-COV03 450 mg dose group ($p=0.0331$). The LSMEAN [95% confidence interval] for this was 3.45 [2.79, 4.11] for the placebo group and 2.43 [1.77, 3.09] for CP-COV03 450 mg dose group, indicating that CP-COV03 450 mg dose group improved in sore throat symptoms 1.02 days faster than the placebo group. No significant difference was observed between CP-COV03 300 mg dose and the placebo group. The LSMEAN [95% confidence interval] for the time taken for the improvement of sore throat symptoms for each group was 3.47 [2.75, 4.19] for the placebo group and 2.65 [1.92, 3.37] for CP-COV03 300 mg dose group.

In the PPS analysis, the ANCOVA analysis showed a significant difference at the 5% significance level for symptoms of sore throat, headache, and fatigue between the placebo and CP-COV03 300 mg dose group. CP-COV03 300 mg dose group improved symptoms faster than the placebo group by 1.26 days for sore throat, 1.31 days for headache, and 1.67 days for fatigue. The LSMEAN [95% confidence

Authors' response to Reviewer's comments

interval] for the time taken for symptom improvement after sustaining for more than 48 hours for each symptom showing a significant difference between the placebo and CP-COV03 300 mg dose group were as follows:

Sore throat ($p=0.0308$), placebo group 3.37 [2.58, 4.15] / CP-COV03 300 mg dose group 2.11 [1.29, 2.93]

Headache ($p=0.0353$), placebo group 3.65 [2.81, 4.49] / CP-COV03 300 mg dose group 2.34 [1.47, 3.22]

Fatigue ($p=0.0163$), placebo group 3.92 [2.99, 4.84] / CP-COV03 300 mg dose group 2.25 [1.26, 3.24]

Significant difference at the 5% significance level was showed in the time taken for sore throat symptoms to improve after sustaining for more than 48 hours between the placebo and CP-COV03 450 mg dose group ($p=0.0269$). The LSMEAN [95% confidence interval] for sore throat was 3.35 [2.61, 4.08] for the placebo group and 2.19 [1.47, 2.90] for CP-COV03 450 mg dose group, indicating that CP-COV03 450 mg dose group improved sore throat symptoms 1.16 days faster than the placebo group.

3) Time (days) taken for each COVID-19 symptom score to improve by more than 1 point by Day 14:

Descriptive statistics for the time taken by Day 14 for each COVID-19 symptom score to decrease by more than one point for the placebo, CP-COV03 300 mg dose, and CP-COV03 450 mg dose group are presented by symptom in Tables 9–10.

Participants not improved symptoms by the evening of Day 14.5 were censored to have a maximum improvement time as 13.00 days (maximum value 13 days = 14.5 days - (first day of administration + 0.5 day)).

ANCOVA analyses were conducted for each symptom, using age and severity as covariates, and the results are presented in Tables 11–12.

Table 9. Secondary outcome 3) descriptive statistics (ITT)

Symptom		Placebo N=98	CP-COV03 300 mg N=99	CP-COV03 450 mg N=96
Fever	N	9	12	7
	Mean	2.72	2.00	2.36
	SD	4.18	3.51	4.48
	Median	1.00	1.00	0.50
	Min	0.50	0.50	0.50
	Max	13.00	13.00	12.50
Cough	N	87	82	82
	Mean	4.67	5.20	5.21
	SD	4.70	4.93	4.83
	Median	2.00	3.25	3.00

Authors' response to Reviewer's comments

Symptom		Placebo N=98	CP-COV03 300 mg N=99	CP-COV03 450 mg N=96
	Min	0.50	-0.50*	0.50
	Max	13.00	13.00	13.00
Sore throat	N	86	88	85
	Mean	3.16	2.27	1.75
	SD	3.67	3.12	2.28
	Median	1.50	1.00	0.50
	Min	0.50	0.50	0.50
	Max	13.00	13.00	13.00
Headache	N	69	70	63
	Mean	2.28	1.96	2.21
	SD	3.09	2.57	3.22
	Median	1.00	1.00	0.50
	Min	0.50	0.50	0.50
	Max	13.00	13.00	13.00
Muscle ache	N	80	84	73
	Mean	1.78	1.38	1.46
	SD	2.39	2.03	2.19
	Median	0.50	0.50	0.50
	Min	0.50	-0.50*	0.50
	Max	13.00	13.00	12.50
Chill	N	63	66	57
	Mean	0.94	1.17	1.46
	SD	1.36	2.01	2.54
	Median	0.50	0.50	0.50
	Min	0.50	0.50	0.50
	Max	10.00	13.00	13.00
Stuffy or runny nose	N	66	70	64
	Mean	4.60	4.84	4.73
	SD	4.78	4.45	4.69
	Median	2.00	3.25	2.75
	Min	0.50	0.50	0.50
	Max	13.00	13.00	13.00
Fatigue	N	70	75	68
	Mean	2.69	1.87	2.91
	SD	3.50	3.18	3.86

Authors' response to Reviewer's comments

Symptom		Placebo N=98	CP-COV03 300 mg N=99	CP-COV03 450 mg N=96
	Median	1.00	0.50	1.00
	Min	0.50	0.50	0.50
	Max	13.00	13.00	13.00
Difficulty of breathing	N	21	23	24
	Mean	1.45	1.63	3.40
	SD	1.92	2.72	4.96
	Median	0.50	0.50	0.50
	Min	0.50	0.50	0.50
	Max	7.50	13.00	13.00
Nausea	N	21	27	37
	Mean	1.29	1.02	1.53
	SD	1.60	1.58	2.53
	Median	0.50	0.50	0.50
	Min	0.50	0.50	0.50
	Max	6.50	8.50	13.00
Vomiting	N	6	7	6
	Mean	0.75	1.57	0.92
	SD	0.27	1.69	0.58
	Median	0.75	0.50	0.75
	Min	0.50	0.50	0.50
	Max	1.00	5.00	2.00
Diarrhea	N	18	22	20
	Mean	2.89	3.27	4.33
	SD	3.08	3.11	3.02
	Median	0.75	2.75	4.50
	Min	0.50	0.50	0.50
	Max	9.50	13.00	13.00

* In CP-COV03 300 mg dose group, one participant (S02008) experienced an improvement of more than one point in cough and muscle ache in the morning of Day 2. However, as the first IP administration did not occur until the noon of Day 2 (resulting in a missed Day 1 night and Day 2 morning dose), the formula for calculating the time (days) required for symptom improvement resulted in a negative number. The calculation was based on the formula: (Day of symptom improvement observation - (First day of administration + 0.5 days)) = 2 (morning of Day 2) - (2 (Day 2 of first administration) + 0.5) = -0.5.

Authors' response to Reviewer's comments

Table 10. Secondary outcome 3) descriptive statistics (PPS)

Symptom		Placebo N=77	CP-COV03 300 mg N=70	CP-COV03 450 mg N=80
Fever	N	5	9	7
	Mean	3.10	1.06	2.36
	SD	5.54	0.63	4.48
	Median	0.50	1.00	0.50
	Min	0.50	0.50	0.50
	Max	13.00	2.50	12.50
Cough	N	69	61	72
	Mean	4.89	4.37	5.16
	SD	4.91	4.44	4.93
	Median	2.50	2.50	2.75
	Min	0.50	0.50	0.50
	Max	13.00	13.00	13.00
Sore throat	N	68	66	73
	Mean	3.09	1.86	1.60
	SD	3.77	2.78	1.97
	Median	1.50	0.50	0.50
	Min	0.50	0.50	0.50
	Max	13.00	13.00	13.00
Headache	N	59	52	55
	Mean	2.35	1.87	1.82
	SD	3.16	2.69	2.74
	Median	1.00	1.00	0.50
	Min	0.50	0.50	0.50
	Max	13.00	13.00	13.00
Muscle ache	N	64	57	64
	Mean	1.90	1.15	1.38
	SD	2.58	1.79	2.13
	Median	0.50	0.50	0.50
	Min	0.50	0.50	0.50
	Max	13.00	13.00	12.50
Chill	N	55	48	53
	Mean	0.96	0.84	1.34
	SD	1.44	0.71	2.46
	Median	0.50	0.50	0.50

Authors' response to Reviewer's comments

Symptom		Placebo N=77	CP-COV03 300 mg N=70	CP-COV03 450 mg N=80
	Min	0.50	0.50	0.50
	Max	10.00	4.00	13.00
Stuffy or runny nose	N	54	57	57
	Mean	4.30	4.23	4.82
	SD	4.75	4.03	4.65
	Median	1.75	3.00	3.00
	Min	0.50	0.50	0.50
	Max	13.00	13.00	13.00
Fatigue	N	60	60	60
	Mean	2.69	1.48	3.08
	SD	3.65	2.67	4.02
	Median	1.00	0.50	1.00
	Min	0.50	0.50	0.50
	Max	13.00	13.00	13.00
Difficulty of breathing	N	19	20	22
	Mean	1.53	1.78	3.09
	SD	2.00	2.89	4.70
	Median	0.50	0.50	0.50
	Min	0.50	0.50	0.50
	Max	7.50	13.00	13.00
Nausea	N	17	22	35
	Mean	0.79	0.68	1.51
	SD	0.59	0.39	2.58
	Median	0.50	0.50	0.50
	Min	0.50	0.50	0.50
	Max	2.50	2.00	13.00
Vomiting	N	5	6	5
	Mean	0.80	1.75	0.70
	SD	0.27	1.78	0.27
	Median	1.00	1.00	0.50
	Min	0.50	0.50	0.50
	Max	1.00	5.00	1.00
Diarrhea	N	16	21	20
	Mean	3.19	3.40	4.33
	SD	3.15	3.12	3.02

Authors' response to Reviewer's comments

Symptom		Placebo N=77	CP-COV03 300 mg N=70	CP-COV03 450 mg N=80
	Median	1.75	3.00	4.50
	Min	0.50	0.50	0.50
	Max	9.50	13.00	13.00

Table 11. Secondary outcome 3) ANCOVA results (ITT)

Symptom	Placebo vs CP-COV03 300 mg			Placebo vs CP-COV03 450 mg		
	LS Mean [95% CI]		p-value	LS Mean [95% CI]		p-value
	Placebo	300 mg		Placebo	450 mg	
Fever	9	12	0.6458	9	7	0.6307
	2.8042 [-0.0832, 5.6917]	1.9385 [-0.5435, 4.4205]		3.1307 [-0.3622, 6.6237]	1.8319 [-2.2070, 5.8708]	
Cough	87	82	0.4964	87	82	0.4748
	4.6847 [3.6697, 5.6997]	5.1881 [4.1426, 6.2337]		4.6754 [3.6607, 5.6901]	5.2042 [4.1589, 6.2494]	
Sore throat	86	88	0.0740	86	85	0.0032*
	3.1813 [2.4544, 3.9082]	2.2489 [1.5304, 2.9675]		3.1569 [2.5024, 3.8114]	1.7471 [1.0888, 2.4055]	
Headache	69	70	0.4751	69	63	0.8859
	2.2903 [1.6240, 2.9566]	1.9496 [1.2881, 2.6111]		2.2877 [1.5377, 3.0376]	2.2088 [1.4239, 2.9936]	
Muscle ache	80	84	0.1420	80	73	0.2259
	1.8351 [1.3526, 2.3177]	1.3296 [0.8588, 1.8005]		1.8422 [1.3399, 2.3444]	1.3921 [0.8661, 1.9182]	
Chill	63	66	0.5773	63	57	0.1763
	0.9713 [0.5611, 1.3815]	1.1334 [0.7327, 1.5342]		0.9441 [0.4396, 1.4485]	1.4478 [0.9174, 1.9782]	
Stuffy or runny nose	66	70	0.8003	66	64	0.9994
	4.6199 [3.4890, 5.7508]	4.8227 [3.7248, 5.9205]		4.6651 [3.5032, 5.8269]	4.6657 [3.4857, 5.8458]	
Fatigue	70	75	0.0834	70	68	0.7670
	2.7518 [1.9811, 3.5225]	1.8050 [1.0606, 2.5495]		2.7056 [1.8382, 3.5730]	2.8913 [2.0112, 3.7713]	

Authors' response to Reviewer's comments

Symptom	Placebo vs CP-COV03 300 mg			Placebo vs CP-COV03 450 mg		
	LS Mean [95% CI]		p-value	LS Mean [95% CI]		p-value
	Placebo	300 mg		Placebo	450 mg	
Difficulty of breathing	21	23	0.9998	21	24	0.0740
	1.5455 [0.6318, 2.4593]	1.5454 [0.6725, 2.4183]		1.3062 [-0.4486, 3.0610]	3.5237 [1.8861, 5.1613]	
Nausea	21	27	0.5448	21	37	0.7442
	1.3012 [0.5810, 2.0214]	1.0065 [0.3738, 1.6391]		1.3071 [0.2985, 2.3156]	1.5149 [0.7586, 2.2712]	
Vomiting	6	7	0.4162	6	6	0.5608
	0.9233 [-0.0294, 1.8760]	1.4229 [0.5425, 2.3033]		0.7710 [0.4766, 1.0653]	0.8957 [0.6013, 1.1900]	
Diarrhea	18	22	0.7199	18	20	0.1761
	2.9046 [1.4354, 4.3738]	3.2599 [1.9328, 4.5869]		2.9042 [1.4238, 4.3846]	4.3112 [2.9090, 5.7134]	

* Statistically significant difference was observed at the 5% significance level

Table 12. Secondary outcome 3) ANCOVA results (PPS)

Symptom	Placebo vs CP-COV03 300 mg			Placebo vs CP-COV03 450 mg		
	LS Mean [95% CI]		p-value	LS Mean [95% CI]		p-value
	Placebo	300 mg		Placebo	450 mg	
Fever	5	9	0.2307	5	7	0.5604
	3.3615 [0.0059, 6.7170]	0.9103 [-1.5582, 3.3788]		3.8793 [-1.7259, 9.4845]	1.8005 [-2.8183, 6.4193]	
Cough	69	61	0.5510	69	72	0.7660
	4.8783 [3.7568, 5.9999]	4.3835 [3.1906, 5.5764]		4.9011 [3.7211, 6.0810]	5.1504 [3.9953, 6.3054]	
Sore throat	68	66	0.0259*	68	73	0.0040*
	3.1212 [2.3269, 3.9154]	1.8297 [1.0234, 2.6360]		3.0833 [2.3633, 3.8033]	1.6005 [0.9057, 2.2954]	
Headache	59	52	0.3905	59	55	0.3089
	2.3447 [1.5955, 3.0940]	1.8685 [1.0704, 2.6666]		2.3655 [1.6027, 3.1283]	1.7988 [1.0087, 2.5889]	
Muscle ache	64	57		64	64	

Authors' response to Reviewer's comments

Symptom	Placebo vs CP-COV03 300 mg			Placebo vs CP-COV03 450 mg		
	LS Mean [95% CI]		p-value	LS Mean [95% CI]		p-value
	Placebo	300 mg		Placebo	450 mg	
	1.9290 [1.3783, 2.4797]	1.1148 [0.5310, 1.6985]	0.0475*	1.9258 [1.3449, 2.5068]	1.3554 [0.7745, 1.9364]	0.1731
Chill	55	48		55	53	
	0.9659 [0.6524, 1.2795]	0.8411 [0.5054, 1.1768]	0.5918	0.9638 [0.4237, 1.5039]	1.3395 [0.7892, 1.8897]	0.3364
Stuffy or runny nose	54	57		54	57	
	4.3414 [3.1444, 5.5384]	4.1853 [3.0205, 5.3501]	0.8539	4.3569 [3.0935, 5.6202]	4.7672 [3.5378, 5.9966]	0.6469
Fatigue	60	60		60	60	
	2.7467 [1.9309, 3.5625]	1.4199 [0.6041, 2.2358]	0.0251*	2.7492 [1.7730, 3.7253]	3.0258 [2.0497, 4.0020]	0.6928
Difficulty of breathing	19	20		19	22	
	1.6339 [0.6085, 2.6593]	1.6728 [0.6735, 2.6721]	0.9564	1.3550 [-0.4231, 3.1330]	3.2389 [1.5907, 4.8871]	0.1301
Nausea	17	22		17	35	
	0.7914 [0.5422, 1.0407]	0.6839 [0.4659, 0.9019]	0.5219	0.8091 [-0.2821, 1.9002]	1.5070 [0.7536, 2.2604]	0.3007
Vomiting	5	6		5	5	
	1.0059 [-0.2005, 2.2122]	1.5784 [0.4805, 2.6764]	0.4488	0.6991 [0.4099, 0.9883]	0.8009 [0.5117, 1.0901]	0.6097
Diarrhea	16	21		16	20	
	3.4706 [2.0569, 4.8842]	3.1891 [1.9597, 4.4184]	0.7648	3.0623 [1.4397, 4.6850]	4.4251 [2.9820, 5.8683]	0.2213

* Statistically significant difference was observed at the 5% significance level

In the ITT analysis, the ANCOVA analysis indicated that sore throat showed the most significant difference between the groups at a 5% significance level ($p=0.0740$). The LSMEAN [95% confidence interval] for the time (days) taken for a decrease of more than 1 point in sore throat symptoms was 3.18 [2.45, 3.91] days for the placebo group and 2.25 [1.53, 2.97] days for CP-COV03 300 mg dose group. Significant differences were observed at the 5% significance level in the time (days) taken for a decrease of more than 1 point in sore throat symptoms between the placebo and CP-COV03 450 mg dose group ($p=0.0032$), with LSMEANs of 3.16 [2.50, 3.81] days for the placebo group and 1.75 [1.09,

Authors' response to Reviewer's comments

2.41] days for CP-COV03 450 mg dose group, indicating a 1.41 days faster improvement in CP-COV03 450 mg dose group.

For PPS analysis, ANCOVA results showed statistically significant differences at the 5% level for symptoms of sore throat, muscle ache, and fatigue between the placebo and CP-COV03 300 mg dose group. CP-COV03 300 mg dose group showed faster symptom improvement than the placebo group by 1.29 days for sore throat, 0.82 days for muscle pain, and 1.33 days for fatigue. The LSMEAN [95% confidence intervals] for the time (days) taken for a decrease of more than one point in these symptoms were as follows:

Sore throat ($p=0.0259$), placebo group 3.12 [2.33, 3.92] / CP-COV03 300 mg dose group 1.83 [1.02, 2.64]

Muscle ache ($p=0.0475$), placebo group 1.93 [1.38, 2.48] / CP-COV03 300 mg dose group 1.11 [0.53, 1.70]

Fatigue ($p=0.0251$), placebo group 2.75 [1.93, 3.56] / CP-COV03 300 mg dose group 1.42 [0.60, 2.24]

In the placebo and CP-COV03 450 mg dose group, the time taken for the sore throat symptom to decrease by more than 1 point showed a significant difference at the 5% significance level ($p=0.0040$). The LSMEAN [95% CI] for the sore throat symptom improvement was 3.08 [2.36, 3.80] for the placebo group and 1.60 [0.91, 2.30] for CP-COV03 450 mg dose group, indicating that CP-COV03 450 mg dose group experienced a sore throat symptom improvement 1.48 days faster than the Placebo group.

4) Total and average all COVID-19 symptom scores change by Day 14

For the placebo, CP-COV03 300 mg dose, and CP-COV03 450 mg dose group, descriptive statistics for the total and average changes in COVID-19 symptom scores by Day 14 are presented in Tables 13–16. Changes were calculated as the difference from each participant's baseline values, excluding symptoms that were not present.

Total scores and averages were analyzed for each day, using baseline values, age, and severity as covariates in an ANCOVA analysis. The results for the total scores and averages by day are presented in

Tables

17–20.

In the ITT analysis, both the total and average scores for the placebo and CP-COV03 300 mg dose group consistently decreased, with significant differences observed at certain points for total scores. The ANCOVA analysis for total score changes showed a significant difference at Day 4 at the 5% significance level ($p=0.0293$), with the LSMEAN [95% CI] for the Placebo and CP-COV03 300 mg dose group showing 4.01 [3.32, 4.71] and 5.11 [4.42, 5.80], respectively. No significant differences were observed between the placebo and CP-COV03 300 mg dose group across all days for average score changes. The lowest p -value for differences was observed on Day 4 ($p=0.0523$), with the LSMEAN [95% CI] being 1.10 [1.01, 1.19] for the placebo and 1.22 [1.13, 1.30] for CP-COV03 300 mg dose group.

In the placebo and CP-COV03 450 mg dose group, both total scores and averages consistently decreased, and significant differences between the two groups were observed at certain points. The ANCOVA analysis for the change in total scores showed significant differences between the groups at Days 2.5, 3.5, 4, and 4.5 with a significance level of 5%, however, no statistically significant differences

Authors' response to Reviewer's comments

were observed on other days. The LSMEANs [95% confidence intervals] for the days where significant differences were as follows:

Day 2.5 ($p=0.0497$): placebo group 7.32 [6.47, 8.17], CP-COV03 450 mg dose group 8.53 [7.67, 9.39]

Day 3.5 ($p=0.0476$): placebo group 5.10 [4.44, 5.76], CP-COV03 450 mg dose group 6.05 [5.38, 6.71]

Day 4 ($p=0.0182$): placebo group 4.02 [3.48, 4.56], CP-COV03 450 mg dose group 4.95 [4.40, 5.49]

Day 4.5 ($p=0.0150$): placebo group 3.55 [2.99, 4.12], CP-COV03 450 mg dose group 4.55 [3.98, 5.11]

Significant differences for average score changes at the 5% significance level were observed on Days 4, 4.5, and 6.5 between the placebo and CP-COV03 450 mg dose group. Significant differences and their LSMEAN [95% CI] are as follows:

Day 4 ($p=0.0500$), placebo group 1.10 [1.03, 1.17] / CP-COV03 450 mg dose group 1.20 [1.13, 1.28]

Day 4.5 ($p=0.0079$), placebo group 1.04 [0.97, 1.12] / CP-COV03 450 mg dose group 1.19 [1.12, 1.27]

Day 6.5 ($p=0.0408$), placebo group 0.92 [0.82, 1.01] / CP-COV03 450 mg dose group 1.06 [0.96, 1.15]

In the PPS analysis, total and average scores for the placebo and CP-COV03 300 mg dose group also consistently decreased, with significant differences observed at certain points for average scores. No significant differences were found for total score changes at the 5% significance level. The lowest p -value for differences was observed on Day 7 ($p=0.0764$), with the LSMEAN [95% CI] for the placebo and CP-COV03 300 mg dose group showing 2.55 [2.07, 3.03] and 1.91 [1.41, 2.42], respectively. Significant differences for average score changes at the 5% significance level were observed on Days 7, 7.5, 9, 9.5, 10, 10.5, and 11 between the placebo and CP-COV03 300 mg dose group. Significant differences and their LSMEAN [95% CI] are as follows:

Day 7 ($p=0.0212$), placebo group 0.94 [0.84, 1.05] / CP-COV03 300 mg dose group 0.76 [0.65, 0.87]

Day 7.5 ($p=0.0096$), placebo group 0.88 [0.76, 0.99] / CP-COV03 300 mg dose group 0.66 [0.54, 0.78]

Day 9 ($p=0.0199$), placebo group 0.83 [0.71, 0.96] / CP-COV03 300 mg dose group 0.62 [0.50, 0.75]

Day 9.5 ($p=0.0127$), placebo group 0.83 [0.71, 0.95] / CP-COV03 300 mg dose group 0.60 [0.48, 0.73]

Day 10 ($p=0.0073$), placebo group 0.80 [0.69, 0.92] / CP-COV03 300 mg dose group 0.57 [0.44, 0.69]

Day 10.5 ($p=0.0033$), placebo group 0.80 [0.68, 0.92] / CP-COV03 300 mg dose group 0.53 [0.41, 0.66]

Day 11 ($p=0.0471$), placebo group 0.76 [0.64, 0.88] / CP-COV03 300 mg dose group 0.59 [0.46, 0.71]

In the placebo and CP-COV03 450 mg dose group, both total scores and averages were observed to decrease consistently, and significant differences between the two groups were identified at certain points. The ANCOVA analysis for the change in total scores revealed a significant difference between the groups on Day 4.5 ($p=0.0227$), with LSMEANs [95% confidence intervals] of 3.36 [2.74, 3.97] for the placebo group and 4.36 [3.76, 4.96] for CP-COV03 450 mg dose group. The analysis for the change in averages showed significant differences between the placebo and CP-COV03 450 mg dose group on

Authors' response to Reviewer's comments

Days 4, 4.5, 5, and 6.5, with a 5% significance level. The LSMEANs [95% confidence intervals] for the days where significant differences were observed are as follows:

Day 4 ($p=0.0401$), placebo group 1.09 [1.00, 1.17] / CP-COV03 450 mg dose group 1.21 [1.13, 1.29]

Day 4.5 ($p=0.0096$), placebo group 1.01 [0.92, 1.10] / CP-COV03 450 mg dose group 1.17 [1.09, 1.26]

Day 5 ($p=0.0268$), placebo group 0.98 [0.88, 1.08] / CP-COV03 450 mg dose group 1.13 [1.04, 1.23]

Day 6.5 ($p=0.0310$), placebo group 0.88 [0.78, 0.98] / CP-COV03 450 mg dose group 1.04 [0.94, 1.14]

Table 13. Secondary outcome 4) descriptive statistics for the change in total symptom score (ITT)

Category		Placebo N=98		CP-COV03 300 mg N=99		CP-COV03 450 mg N=96	
		Values (Day#)	Change (Day# - baseline)	Values (Day#)	Change (Day# - baseline)	Values (Day#)	Change (Day# - baseline)
Baseline	N	98	NA	99	NA	96	NA
	Mean±SD	11.39±4.93	NA	11.96±5.54	NA	11.58±5.33	NA
	Median [Min, Max]	10.50 [4.00, 25.00]	NA	11.00 [4.00, 26.00]	NA	11.00 [4.00, 24.00]	NA
Day 2	N	98	98	99	99	96	96
	Mean±SD	9.66±4.74	-1.72±5.10	9.54±4.42	-2.42±6.15	9.90±4.88	-1.69±5.96
	Median [Min, Max]	9.00 [1.00, 22.00]	-2.00 [-15.00, 13.00]	9.00 [1.00, 24.00]	-3.00 [-21.00, 17.00]	9.00 [2.00, 34.00]	-2.00 [-13.00, 29.00]
Day 2.5	N	98	98	99	99	96	96
	Mean±SD	7.28±4.07	-4.11±5.20	8.15±4.13	-3.81±6.02	8.57±4.90	-3.01±5.98
	Median [Min, Max]	6.50 [0.00, 20.00]	-4.00 [-16.00, 13.00]	8.00 [1.00, 21.00]	-4.00 [-18.00, 14.00]	7.50 [2.00, 28.00]	-3.00 [-17.00, 23.00]
Day 3	N	98	98	99	99	96	96
	Mean±SD	6.10±3.62	-5.29±5.09	6.76±4.16	-5.20±6.60	6.72±3.80	-4.86±5.50
	Median [Min, Max]	5.00 [0.00, 19.00]	-5.00 [-17.00, 12.00]	6.00 [1.00, 22.00]	-5.00 [-19.00, 14.00]	6.00 [1.00, 19.00]	-5.00 [-19.00, 14.00]
Day 3.5	N	98	98	99	99	96	96
	Mean±SD	5.06±3.59	-6.33±5.17	5.79±4.18	-6.17±6.74	6.08±3.44	-5.50±5.17
	Median	4.00	-6.50	5.00	-6.00	5.00	-5.50

Category		Placebo N=98		CP-COV03 300 mg N=99		CP-COV03 450 mg N=96	
		Values (Day#)	Change (Day# - baseline)	Values (Day#)	Change (Day# - baseline)	Values (Day#)	Change (Day# - baseline)
	[Min, Max]	[0.00, 18.00]	[-20.00, 11.00]	[0.00, 24.00]	[-20.00, 16.00]	[0.00, 16.00]	[-18.00, 9.00]
Day 4	N	98	98	99	99	96	96
	Mean±SD	4.00±2.69	-7.39±5.22	5.12±4.11	-6.84±6.94	4.97±2.99	-6.61±4.96
	Median	3.00	-7.00	4.00	-7.00	4.00	-7.00
	[Min, Max]	[0.00, 11.00]	[-25.00, 4.00]	[0.00, 25.00]	[-20.00, 17.00]	[0.00, 14.00]	[-19.00, 7.00]
Day 4.5	N	98	98	99	99	96	96
	Mean±SD	3.53±2.80	-7.86±5.35	4.26±3.57	-7.70±6.81	4.57±2.94	-7.01±5.45
	Median	3.00	-8.00	4.00	-7.00	4.00	-7.00
	[Min, Max]	[0.00, 13.00]	[-22.00, 4.00]	[0.00, 17.00]	[-23.00, 9.00]	[0.00, 13.00]	[-20.00, 7.00]
Day 5	N	98	98	99	99	96	96
	Mean±SD	3.07±2.61	-8.32±5.25	3.74±3.32	-8.22±6.61	3.82±2.71	-7.76±5.55
	Median	2.00	-8.00	3.00	-8.00	3.00	-8.00
	[Min, Max]	[0.00, 13.00]	[-25.00, 2.00]	[0.00, 16.00]	[-23.00, 10.00]	[0.00, 13.00]	[-20.00, 5.00]
Day 5.5	N	98	98	99	99	96	96
	Mean±SD	2.76±2.68	-8.63±5.26	3.07±3.03	-8.89±6.38	3.51±2.78	-8.07±5.36
	Median	2.00	-8.00	2.00	-9.00	3.00	-8.00
	[Min, Max]	[0.00, 13.00]	[-23.00, 2.00]	[0.00, 16.00]	[-24.00, 9.00]	[0.00, 16.00]	[-21.00, 3.00]
Day 6	N	98	98	99	99	96	96
	Mean±SD	2.36±2.23	-9.03±5.39	2.90±3.56	-9.06±6.80	2.89±2.67	-8.70±5.28
	Median	2.00	-9.00	2.00	-9.00	2.00	-8.50
	[Min, Max]	[0.00, 10.00]	[-25.00, 1.00]	[0.00, 22.00]	[-24.00, 15.00]	[0.00, 14.00]	[-21.00, 3.00]

Category		Placebo N=98		CP-COV03 300 mg N=99		CP-COV03 450 mg N=96	
		Values (Day#)	Change (Day# - baseline)	Values (Day#)	Change (Day# - baseline)	Values (Day#)	Change (Day# - baseline)
Day 6.5	N	98	98	99	99	96	96
	Mean±SD	2.65±2.34	-8.73±5.48	2.98±3.47	-8.98±6.85	3.26±2.71	-8.32±5.21
	Median [Min, Max]	2.00 [0.00, 10.00]	-8.00 [-23.00, 2.00]	2.00 [0.00, 20.00]	-8.00 [-24.00, 13.00]	3.00 [0.00, 13.00]	-8.00 [-21.00, 3.00]
Day 7	N	98	98	99	99	96	96
	Mean±SD	2.63±2.10	-8.76±5.41	2.54±2.76	-9.42±6.43	2.63±2.14	-8.96±5.04
	Median [Min, Max]	2.50 [0.00, 11.00]	-8.00 [-23.00, 0.00]	2.00 [0.00, 16.00]	-9.00 [-24.00, 9.00]	2.00 [0.00, 11.00]	-9.00 [-21.00, 2.00]
Day 7.5	N	98	98	99	99	96	96
	Mean±SD	2.44±2.17	-8.95±5.44	2.31±2.78	-9.65±6.43	2.57±2.18	-9.01±5.18
	Median [Min, Max]	2.00 [0.00, 10.00]	-8.00 [-23.00, 4.00]	2.00 [0.00, 16.00]	-9.00 [-25.00, 9.00]	2.00 [0.00, 13.00]	-8.50 [-20.00, 1.00]
Day 8	N	98	98	99	99	96	96
	Mean±SD	2.12±2.01	-9.27±5.32	2.25±2.63	-9.71±6.40	2.14±1.91	-9.45±5.15
	Median [Min, Max]	2.00 [0.00, 8.00]	-8.50 [-23.00, 2.00]	1.00 [0.00, 15.00]	-9.00 [-25.00, 9.00]	2.00 [0.00, 10.00]	-9.00 [-21.00, -1.00]
Day 8.5	N	98	98	99	99	96	96
	Mean±SD	1.98±2.00	-9.41±5.35	2.07±2.66	-9.89±6.50	2.17±1.87	-9.42±5.06
	Median [Min, Max]	1.00 [0.00, 8.00]	-9.00 [-23.00, 1.00]	1.00 [0.00, 15.00]	-10.00 [-25.00, 10.00]	2.00 [0.00, 9.00]	-9.00 [-20.00, -1.00]
Day 9	N	98	98	99	99	96	96

Category		Placebo N=98		CP-COV03 300 mg N=99		CP-COV03 450 mg N=96	
		Values (Day#)	Change (Day# - baseline)	Values (Day#)	Change (Day# - baseline)	Values (Day#)	Change (Day# - baseline)
	Mean±SD	2.06±2.02	-9.33±5.15	2.05±2.67	-9.91±6.48	1.94±1.95	-9.65±5.16
	Median [Min, Max]	2.00 [0.00, 10.00]	-9.00 [-23.00, 0.00]	1.00 [0.00, 15.00]	-9.00 [-25.00, 10.00]	2.00 [0.00, 11.00]	-9.00 [-20.00, -1.00]
	N	98	98	99	99	96	96
Day 9.5	Mean±SD	1.96±1.96	-9.43±5.14	2.02±2.71	-9.94±6.49	1.98±1.83	-9.60±5.13
	Median [Min, Max]	1.00 [0.00, 10.00]	-9.00 [-23.00, -1.00]	1.00 [0.00, 15.00]	-9.00 [-25.00, 8.00]	2.00 [0.00, 8.00]	-9.00 [-21.00, -1.00]
	N	98	98	99	99	96	96
Day 10	Mean±SD	1.85±1.81	-9.54±4.97	1.80±2.67	-10.16±6.61	1.83±1.86	-9.75±5.13
	Median [Min, Max]	2.00 [0.00, 11.00]	-9.00 [-23.00, -2.00]	1.00 [0.00, 15.00]	-9.00 [-26.00, 10.00]	1.00 [0.00, 9.00]	-9.50 [-21.00, -1.00]
	N	98	98	99	99	96	96
Day 10.5	Mean±SD	1.88±1.87	-9.51±4.99	1.78±2.68	-10.18±6.64	1.82±1.90	-9.76±5.21
	Median [Min, Max]	1.00 [0.00, 10.00]	-9.00 [-23.00, 0.00]	1.00 [0.00, 15.00]	-9.00 [-26.00, 9.00]	1.00 [0.00, 11.00]	-9.00 [-21.00, -1.00]
	N	98	98	99	99	96	96
Day 11	Mean±SD	1.70±1.87	-9.68±5.07	1.85±2.67	-10.11±6.62	1.60±1.78	-9.98±5.17
	Median [Min, Max]	1.00 [0.00, 9.00]	-9.00 [-23.00, -1.00]	1.00 [0.00, 16.00]	-9.00 [-26.00, 9.00]	1.00 [0.00, 11.00]	-10.00 [-21.00, -1.00]
	N	98	98	99	99	96	96
Day 11.5	Mean±SD	1.63±1.76	-9.76±5.04	1.76±2.62	-10.20±6.59	1.64±1.65	-9.95±5.12
	N	98	98	99	99	96	96

Category		Placebo N=98		CP-COV03 300 mg N=99		CP-COV03 450 mg N=96	
		Values (Day#)	Change (Day# - baseline)	Values (Day#)	Change (Day# - baseline)	Values (Day#)	Change (Day# - baseline)
Day 12	Median [Min, Max]	1.00 [0.00, 8.00]	-9.00 [-23.00, -1.00]	1.00 [0.00, 17.00]	-10.00 [-26.00, 10.00]	1.00 [0.00, 8.00]	-10.00 [-21.00, -1.00]
	N	98	98	99	99	96	96
	Mean±SD	1.47±1.52	-9.92±5.21	1.73±2.68	-10.23±6.64	1.55±1.76	-10.03±5.08
Day 12.5	Median [Min, Max]	1.00 [0.00, 7.00]	-9.00 [-25.00, -1.00]	1.00 [0.00, 15.00]	-10.00 [-26.00, 11.00]	1.00 [0.00, 11.00]	-10.00 [-20.00, -1.00]
	N	98	98	99	99	96	96
	Mean±SD	1.54±1.82	-9.85±5.01	1.71±2.48	-10.25±6.49	1.52±1.60	-10.06±5.24
Day 13	Median [Min, Max]	1.00 [0.00, 10.00]	-9.00 [-23.00, -1.00]	1.00 [0.00, 14.00]	-9.00 [-26.00, 7.00]	1.00 [0.00, 8.00]	-9.50 [-22.00, -1.00]
	N	98	98	99	99	96	96
	Mean±SD	1.50±1.84	-9.89±4.95	1.54±2.46	-10.42±6.45	1.48±1.92	-10.10±5.17
Day 13.5	Median [Min, Max]	1.00 [0.00, 11.00]	-9.00 [-23.00, -1.00]	1.00 [0.00, 14.00]	-10.00 [-26.00, 7.00]	1.00 [0.00, 22.00]	-10.00 [-22.00, 7.00]
	N	98	98	99	99	96	96
	Mean±SD	1.39±1.90	-10.00±4.95	1.53±2.46	-10.43±6.37	1.63±2.58	-9.96±5.49
Day 14	Median	1.00	-9.00	1.00	-10.00	1.00[0.00, 12.00]	-10.00
	N	98	98	99	99	96	96
	Mean±SD	1.28±1.88	-10.11±4.86	1.52±2.43	-10.44±6.43	1.42±2.05	-10.17±5.08

Category		Placebo N=98		CP-COV03 300 mg N=99		CP-COV03 450 mg N=96	
		Values (Day#)	Change (Day# - baseline)	Values (Day#)	Change (Day# - baseline)	Values (Day#)	Change (Day# - baseline)
	[Min, Max]	[0.00, 12.00]	[-23.00, -2.00]	[0.00, 14.00]	[-26.00, 7.00]		[-21.00, -1.00]
Day 14.5	N	98	98	99	99	96	96
	Mean±SD	1.40±1.92	-9.99±4.86	1.47±2.48	-10.48±6.46	1.44±1.97	-10.15±5.18
	Median	1.00	-9.00	1.00	-10.00	1.00	-10.00
	[Min, Max]	[0.00, 11.00]	[-23.00, -1.00]	[0.00, 15.00]	[-26.00, 8.00]	[0.00, 12.00]	[-21.00, -1.00]

Table 14. Secondary outcome 4) descriptive statistics for the change in total symptom score (PPS)

Category		Placebo N=77		CP-COV03 300 mg N=70		CP-COV03 450 mg N=80	
		Values (Day#)	Change (Day# - baseline)	Values (Day#)	Change (Day# - baseline)	Values (Day#)	Change (Day# - baseline)
Baseline	N	77	NA	70	NA	80	NA
	Mean±SD	11.75±4.67	NA	13.10±5.73	NA	12.40±5.24	NA
	Median [Min, Max]	11.00 [5.00, 25.00]	NA	13.00 [4.00, 26.00]	NA	12.50 [4.00, 24.00]	NA
Day 2	N	77	77	70	70	80	80
	Mean±SD	9.64±4.41	-2.12±4.93	9.21±3.78	-3.89±5.37	9.68±4.21	-2.73±4.54
	Median [Min, Max]	9.00 [3.00, 22.00]	-2.00 [-15.00, 11.00]	9.00 [2.00, 20.00]	-3.00 [-21.00, 9.00]	9.00 [2.00, 20.00]	-3.00 [-13.00, 7.00]
Day 2.5	N	77	77	70	70	80	80

Category		Placebo N=77		CP-COV03 300 mg N=70		CP-COV03 450 mg N=80	
		Values (Day#)	Change (Day# - baseline)	Values (Day#)	Change (Day# - baseline)	Values (Day#)	Change (Day# - baseline)
	Mean±SD	7.10±3.74	-4.65±5.12	7.61±3.73	-5.49±4.98	8.48±4.64	-3.93±5.18
	Median	7.00	-5.00	8.00	-5.00	7.50	-4.00
	[Min, Max]	[0.00, 19.00]	[-16.00, 8.00]	[1.00, 19.00]	[-18.00, 8.00]	[2.00, 25.00]	[-17.00, 11.00]
Day 3	N	77	77	70	70	80	80
	Mean±SD	5.91±3.42	-5.84±4.87	6.03±3.44	-7.07±5.55	6.41±3.67	-5.99±4.91
	Median	5.00	-6.00	6.00	-7.00	6.00	-5.00
	[Min, Max]	[0.00, 16.00]	[-17.00, 7.00]	[1.00, 17.00]	[-19.00, 7.00]	[1.00, 16.00]	[-19.00, 3.00]
Day 3.5	N	77	77	70	70	80	80
	Mean±SD	4.96±3.50	-6.79±4.96	5.03±3.35	-8.07±5.74	5.99±3.55	-6.41±4.90
	Median	4.00	-7.00	5.00	-7.50	5.00	-6.00
	[Min, Max]	[0.00, 16.00]	[-20.00, 3.00]	[0.00, 14.00]	[-20.00, 10.00]	[0.00, 16.00]	[-18.00, 8.00]
Day 4	N	77	77	70	70	80	80
	Mean±SD	3.87±2.68	-7.88±5.13	4.29±3.28	-8.81±5.99	4.79±3.05	-7.61±4.56
	Median	3.00	-7.00	4.00	-8.00	4.00	-8.00
	[Min, Max]	[0.00, 11.00]	[-25.00, 4.00]	[0.00, 14.00]	[-20.00, 9.00]	[0.00, 14.00]	[-19.00, 4.00]
Day 4.5	N	77	77	70	70	80	80
	Mean±SD	3.29±2.65	-8.47±5.09	3.49±2.71	-9.61±6.13	4.43±2.95	-7.98±5.05
	Median	3.00	-8.00	3.00	-9.00	4.00	-8.00
	[Min, Max]	[0.00, 9.00]	[-22.00, 3.00]	[0.00, 13.00]	[-23.00, 9.00]	[0.00, 13.00]	[-20.00, 7.00]
Day 5	N	77	77	70	70	80	80
	Mean±SD	2.87±2.44	-8.88±5.12	3.09±2.74	-10.01±6.12	3.76±2.76	-8.64±5.29

Category		Placebo N=77		CP-COV03 300 mg N=70		CP-COV03 450 mg N=80	
		Values (Day#)	Change (Day# - baseline)	Values (Day#)	Change (Day# - baseline)	Values (Day#)	Change (Day# - baseline)
Day 5.5	Median [Min, Max]	2.00 [0.00, 10.00]	-9.00 [-25.00, 2.00]	3.00 [0.00, 14.00]	-10.00 [-23.00, 10.00]	3.00 [0.00, 13.00]	-9.00 [-20.00, 5.00]
	N	77	77	70	70	80	80
	Mean±SD	2.55±2.46	-9.21±5.13	2.46±2.42	-10.64±5.92	3.43±2.89	-8.98±5.10
Day 6	Median [Min, Max]	2.00 [0.00, 11.00]	-9.00 [-23.00, 2.00]	2.00 [0.00, 13.00]	-10.00 [-24.00, 9.00]	3.00 [0.00, 16.00]	-9.50 [-21.00, 3.00]
	N	77	77	70	70	80	80
	Mean±SD	2.12±2.05	-9.64±5.13	2.36±2.77	-10.74±6.21	2.89±2.71	-9.51±5.09
Day 6.5	Median [Min, Max]	2.00 [0.00, 8.00]	-9.00 [-25.00, 0.00]	2.00 [0.00, 16.00]	-11.00 [-24.00, 12.00]	2.00 [0.00, 14.00]	-10.00 [-21.00, 3.00]
	N	77	77	70	70	80	80
	Mean±SD	2.47±2.16	-9.29±5.11	2.13±2.52	-10.97±6.23	3.10±2.68	-9.30±4.83
Day 7	Median [Min, Max]	2.00 [0.00, 8.00]	-9.00 [-23.00, 1.00]	1.50 [0.00, 15.00]	-10.50 [-24.00, 11.00]	2.00 [0.00, 13.00]	-9.00 [-21.00, 0.00]
	N	77	77	70	70	80	80
	Mean±SD	2.55±1.96	-9.21±5.16	1.91±2.24	-11.19±6.09	2.61±2.22	-9.79±4.88
Day 7.5	Median [Min, Max]	3.00 [0.00, 8.00]	-8.00 [-23.00, 0.00]	1.00 [0.00, 12.00]	-10.50 [-24.00, 8.00]	2.00 [0.00, 11.00]	-10.00 [-21.00, 0.00]
	N	77	77	70	70	80	80
	Mean±SD	2.31±2.00	-9.44±5.06	1.81±2.39	-11.29±6.18	2.49±2.27	-9.91±4.95
Day 7.5	Median	2.00	-8.00	1.00	-11.00	2.00	-10.00
	N	77	77	70	70	80	80
	Mean±SD	2.31±2.00	-9.44±5.06	1.81±2.39	-11.29±6.18	2.49±2.27	-9.91±4.95

Category		Placebo N=77		CP-COV03 300 mg N=70		CP-COV03 450 mg N=80	
		Values (Day#)	Change (Day# - baseline)	Values (Day#)	Change (Day# - baseline)	Values (Day#)	Change (Day# - baseline)
	[Min, Max]	[0.00, 8.00]	[-23.00, 1.00]	[0.00, 13.00]	[-25.00, 9.00]	[0.00, 13.00]	[-20.00, 0.00]
Day 8	N	77	77	70	70	80	80
	Mean±SD	2.16±1.98	-9.60±5.06	1.74±2.21	-11.36±6.21	2.09±2.03	-10.31±4.95
	Median [Min, Max]	2.00 [0.00, 7.00]	-9.00 [-23.00, 0.00]	1.00 [0.00, 13.00]	-10.00 [-25.00, 9.00]	2.00 [0.00, 10.00]	-10.00 [-21.00, -2.00]
Day 8.5	N	77	77	70	70	80	80
	Mean±SD	2.01±1.98	-9.74±5.14	1.59±2.16	-11.51±6.31	2.15±1.98	-10.25±4.86
	Median [Min, Max]	1.00 [0.00, 7.00]	-9.00 [-23.00, 1.00]	1.00 [0.00, 14.00]	-11.00 [-25.00, 10.00]	2.00 [0.00, 9.00]	-10.00 [-20.00, -1.00]
Day 9	N	77	77	70	70	80	80
	Mean±SD	2.18±2.08	-9.57±4.92	1.56±2.27	-11.54±6.31	1.91±2.06	-10.49±4.99
	Median [Min, Max]	2.00 [0.00, 10.00]	-9.00 [-23.00, 0.00]	1.00 [0.00, 14.00]	-11.00 [-25.00, 10.00]	1.50 [0.00, 11.00]	-10.00 [-20.00, -2.00]
Day 9.5	N	77	77	70	70	80	80
	Mean±SD	2.04±2.02	-9.71±4.92	1.49±2.15	-11.61±6.25	1.99±1.91	-10.41±4.97
	Median [Min, Max]	1.00 [0.00, 10.00]	-9.00 [-23.00, -1.00]	1.00 [0.00, 12.00]	-11.00 [-25.00, 8.00]	2.00 [0.00, 8.00]	-10.00 [-21.00, -1.00]
Day 10	N	77	77	70	70	80	80
	Mean±SD	1.90±1.86	-9.86±4.67	1.33±2.17	-11.77±6.43	1.81±1.91	-10.59±4.96
	Median [Min, Max]	2.00 [0.00, 11.00]	-9.00 [-23.00, -2.00]	1.00 [0.00, 14.00]	-11.00 [-26.00, 10.00]	1.00 [0.00, 9.00]	-10.00 [-21.00, -2.00]

Category		Placebo N=77		CP-COV03 300 mg N=70		CP-COV03 450 mg N=80	
		Values (Day#)	Change (Day# - baseline)	Values (Day#)	Change (Day# - baseline)	Values (Day#)	Change (Day# - baseline)
Day 10.5	N	77	77	70	70	80	80
	Mean±SD	1.94±1.93	-9.82±4.76	1.30±2.23	-11.80±6.46	1.83±1.96	-10.58±5.10
	Median [Min, Max]	2.00 [0.00, 10.00]	-9.00 [-23.00, 0.00]	0.50 [0.00, 13.00]	-11.00 [-26.00, 9.00]	1.00 [0.00, 11.00]	-10.00 [-21.00, -2.00]
Day 11	N	77	77	70	70	80	80
	Mean±SD	1.79±1.93	-9.96±4.79	1.37±2.19	-11.73±6.45	1.63±1.88	-10.78±5.06
	Median [Min, Max]	1.00 [0.00, 9.00]	-9.00 [-23.00, -1.00]	1.00 [0.00, 12.00]	-11.00 [-26.00, 8.00]	1.00 [0.00, 11.00]	-10.00 [-21.00, -2.00]
Day 11.5	N	77	77	70	70	80	80
	Mean±SD	1.66±1.81	-10.09±4.73	1.36±2.15	-11.74±6.46	1.64±1.68	-10.76±4.97
	Median [Min, Max]	1.00 [0.00, 8.00]	-9.00 [-23.00, -1.00]	1.00 [0.00, 12.00]	-11.00 [-26.00, 8.00]	1.00 [0.00, 8.00]	-10.50 [-21.00, -2.00]
Day 12	N	77	77	70	70	80	80
	Mean±SD	1.48±1.52	-10.27±4.90	1.39±2.42	-11.71±6.66	1.54±1.80	-10.86±4.91
	Median [Min, Max]	1.00 [0.00, 7.00]	-9.00 [-25.00, -2.00]	1.00 [0.00, 15.00]	-11.00 [-26.00, 11.00]	1.00 [0.00, 11.00]	-10.50 [-20.00, -2.00]
Day 12.5	N	77	77	70	70	80	80
	Mean±SD	1.60±1.86	-10.16±4.65	1.36±2.16	-11.74±6.48	1.50±1.62	-10.90±5.09
	Median [Min, Max]	1.00 [0.00, 10.00]	-9.00 [-23.00, -2.00]	1.00 [0.00, 11.00]	-11.50 [-26.00, 7.00]	1.00 [0.00, 8.00]	-10.50 [-22.00, -3.00]
Day 13	N	77	77	70	70	80	80

Category		Placebo N=77		CP-COV03 300 mg N=70		CP-COV03 450 mg N=80	
		Values (Day#)	Change (Day# - baseline)	Values (Day#)	Change (Day# - baseline)	Values (Day#)	Change (Day# - baseline)
	Mean±SD	1.55±1.90	-10.21±4.55	1.24±2.00	-11.86±6.43	1.48±2.02	-10.93±5.04
	Median [Min, Max]	1.00 [0.00, 10.00]	-9.00 [-23.00, -2.00]	1.00 [0.00, 11.00]	-11.00 [-26.00, 7.00]	1.00 [0.00, 14.00]	-10.50 [-21.00, -1.00]
	N	77	77	70	70	80	80
Day 13.5	Mean±SD	1.51±1.94	-10.25±4.53	1.23±2.01	-11.87±6.31	1.64±2.74	-10.76±5.45
	Median [Min, Max]	1.00 [0.00, 11.00]	-9.00 [-23.00, -2.00]	1.00 [0.00, 10.00]	-11.50 [-26.00, 6.00]	1.00 [0.00, 22.00]	-10.50 [-22.00, 7.00]
	N	77	77	70	70	80	80
Day 14	Mean±SD	1.40±1.99	-10.35±4.46	1.14±1.89	-11.96±6.34	1.46±2.15	-10.94±4.99
	Median [Min, Max]	1.00 [0.00, 12.00]	-9.00 [-23.00, -2.00]	0.50 [0.00, 9.00]	-11.50 [-26.00, 5.00]	1.00 [0.00, 12.00]	-10.50 [-21.00, -3.00]
	N	77	77	70	70	80	80
Day 14.5	Mean±SD	1.48±1.98	-10.27±4.42	1.14±2.09	-11.96±6.43	1.46±2.01	-10.94±5.08
	Median [Min, Max]	1.00 [0.00, 11.00]	-9.00 [-23.00, -3.00]	0.00 [0.00, 12.00]	-11.50 [-26.00, 8.00]	1.00 [0.00, 12.00]	-10.50 [-21.00, -2.00]
	N	77	77	70	70	80	80

Table 15. Secondary outcome 4) descriptive statistics for the change in average symptom score (ITT)

Category		Placebo N=98		CP-COV03 300 mg N=99		CP-COV03 450 mg N=96	
		Values (Day#)	Change (Day# - baseline)	Values (Day#)	Change (Day# - baseline)	Values (Day#)	Change (Day# - baseline)
Baseline	N	98	NA	99	NA	96	NA
	Mean±SD	1.88±0.38	NA	1.88±0.34	NA	1.90±0.36	NA
	Median [Min, Max]	1.83 [1.29, 3.00]	NA	1.86 [1.29, 3.00]	NA	1.95 [1.29, 3.00]	NA
Day 2	N	98	98	99	99	96	96
	Mean±SD	1.52±0.37	-0.36±0.43	1.47±0.34	-0.41±0.45	1.50±0.37	-0.40±0.43
	Median [Min, Max]	1.50 [1.00, 2.50]	-0.33 [-1.50, 0.51]	1.43 [1.00, 2.38]	-0.42 [-1.67, 0.67]	1.50 [1.00, 2.83]	-0.39 [-1.33, 1.17]
Day 2.5	N	98	98	99	99	96	96
	Mean±SD	1.36±0.39	-0.52±0.46	1.37±0.31	-0.51±0.39	1.42±0.39	-0.48±0.42
	Median [Min, Max]	1.33 [0.00, 2.38]	-0.50 [-1.75, 0.60]	1.33 [1.00, 2.50]	-0.50 [-1.43, 0.50]	1.33[1.00, 2.50]	-0.52 [-1.50, 0.67]
Day 3	N	98	98	99	99	96	96
	Mean±SD	1.30±0.39	-0.58±0.50	1.31±0.33	-0.57±0.43	1.29±0.30	-0.60±0.41
	Median [Min, Max]	1.25 [0.00, 3.00]	-0.57 [-2.29, 0.75]	1.25 [1.00, 2.44]	-0.56 [-1.56, 0.67]	1.25 [1.00, 2.33]	-0.54 [-1.75, 0.19]
Day 3.5	N	98	98	99	99	96	96
	Mean±SD	1.23±0.43	-0.65±0.56	1.23±0.38	-0.65±0.49	1.25±0.29	-0.65±0.45
	Median	1.00	-0.60	1.17	-0.63	1.27	-0.57

Category		Placebo N=98		CP-COV03 300 mg N=99		CP-COV03 450 mg N=96	
		Values (Day#)	Change (Day# - baseline)	Values (Day#)	Change (Day# - baseline)	Values (Day#)	Change (Day# - baseline)
	[Min, Max]	[0.00, 3.00]	[-2.33, 1.29]	[0.00, 2.40]	[-2.00, 0.50]	[0.00, 2.00]	[-2.00, 0.40]
Day 4	N	98	98	99	99	96	96
	Mean±SD	1.10±0.41	-0.78±0.53	1.22±0.45	-0.66±0.53	1.21±0.31	-0.69±0.42
	Median	1.00	-0.71	1.17	-0.63	1.00	-0.65
	[Min, Max]	[0.00, 2.33]	[-2.50, 0.33]	[0.00, 2.60]	[-2.00, 0.60]	[0.00, 2.00]	[-2.00, 0.13]
Day 4.5	N	98	98	99	99	96	96
	Mean±SD	1.04±0.45	-0.84±0.58	1.13±0.46	-0.75±0.54	1.19±0.31	-0.70±0.44
	Median	1.00	-0.74	1.00	-0.75	1.00	-0.67
	[Min, Max]	[0.00, 2.00]	[-2.80, 0.20]	[0.00, 2.60]	[-2.00, 0.60]	[0.00, 2.17]	[-2.00, 0.21]
Day 5	N	98	98	99	99	96	96
	Mean±SD	1.01±0.49	-0.87±0.63	1.01±0.49	-0.87±0.56	1.13±0.34	-0.76±0.48
	Median	1.00	-0.71	1.00	-0.86	1.00	-0.67
	[Min, Max]	[0.00, 2.33]	[-2.80, 0.60]	[0.00, 2.33]	[-2.14, 0.42]	[0.00, 2.00]	[-2.29, 0.22]
Day 5.5	N	98	98	99	99	96	96
	Mean±SD	1.01±0.53	-0.87±0.67	0.98±0.52	-0.90±0.57	1.08±0.38	-0.82±0.52
	Median	1.00	-0.73	1.00	-0.83	1.00	-0.67
	[Min, Max]	[0.00, 2.33]	[-2.80, 0.60]	[0.00, 2.60]	[-2.14, 0.67]	[0.00, 2.00]	[-2.63, 0.10]
Day 6	N	98	98	99	99	96	96
	Mean±SD	0.91±0.55	-0.97±0.69	0.91±0.53	-0.97±0.59	0.94±0.54	-0.96±0.66
	Median	1.00	-0.92	1.00	-0.98	1.00	-0.81
	[Min, Max]	1.00[0.00, 2.00]	[-2.80, 0.60]	[0.00, 2.29]	[-2.14, 0.67]	[0.00, 2.00]	[-3.00, 0.60]

Category		Placebo N=98		CP-COV03 300 mg N=99		CP-COV03 450 mg N=96	
		Values (Day#)	Change (Day# - baseline)	Values (Day#)	Change (Day# - baseline)	Values (Day#)	Change (Day# - baseline)
Day 6.5	N	98	98	99	99	96	96
	Mean±SD	0.91±0.48	-0.97±0.63	0.90±0.55	-0.98±0.62	1.06±0.48	-0.84±0.56
	Median [Min, Max]	1.00[0.00, 2.67]	-0.89 [-2.88, 0.92]	1.00 [0.00, 2.17]	-1.00 [-2.14, 0.25]	1.00 [0.00, 2.50]	-0.70 [-2.63, 0.50]
Day 7	N	98	98	99	99	96	96
	Mean±SD	0.95±0.44	-0.93±0.61	0.88±0.52	-1.00±0.61	0.96±0.47	-0.93±0.56
	Median [Min, Max]	1.00 [0.00, 2.50]	-0.75 [-2.88, 0.75]	1.00 [0.00, 2.00]	-1.00 [-2.14, 0.67]	1.00 [0.00, 2.00]	-0.87 [-3.00, 0.00]
Day 7.5	N	98	98	99	99	96	96
	Mean±SD	0.90±0.47	-0.98±0.63	0.77±0.57	-1.11±0.65	0.97±0.47	-0.93±0.56
	Median [Min, Max]	1.00 [0.00, 2.50]	-0.85 [-2.88, 0.75]	1.00 [0.00, 2.17]	-1.00 [-2.56, 0.50]	1.00 [0.00, 2.00]	-0.85 [-3.00, 0.00]
Day 8	N	98	98	99	99	96	96
	Mean±SD	0.83±0.54	-1.05±0.69	0.79±0.57	-1.09±0.65	0.89±0.50	-1.01±0.56
	Median [Min, Max]	1.00 [0.00, 3.00]	-1.00 [-2.88, 1.50]	1.00 [0.00, 2.60]	-1.00 [-2.56, 0.60]	1.00 [0.00, 2.00]	-1.00 [-3.00, 0.40]
Day 8.5	N	98	98	99	99	96	96
	Mean±SD	0.79±0.52	-1.09±0.69	0.75±0.57	-1.13±0.66	0.94±0.55	-0.95±0.61
	Median [Min, Max]	1.00 [0.00, 2.00]	-1.00 [-2.88, 0.50]	1.00 [0.00, 2.25]	-1.00 [-2.56, 0.50]	1.00 [0.00, 3.00]	-0.89 [-3.00, 0.67]
Day 9	N	98	98	99	99	96	96

Category		Placebo N=98		CP-COV03 300 mg N=99		CP-COV03 450 mg N=96	
		Values (Day#)	Change (Day# - baseline)	Values (Day#)	Change (Day# - baseline)	Values (Day#)	Change (Day# - baseline)
	Mean±SD	0.82±0.55	-1.06±0.69	0.73±0.58	-1.15±0.67	0.84±0.57	-1.06±0.63
	Median	1.00	-1.00	1.00	-1.00	1.00	-1.00
	[Min, Max]	[0.00, 3.00]	[-2.88, 0.71]	[0.00, 2.00]	[-2.56, 0.50]	[0.00, 3.00]	[-3.00, 0.67]
Day 9.5	N	98	98	99	99	96	96
	Mean±SD	0.80±0.52	-1.08±0.67	0.70±0.57	-1.18±0.65	0.89±0.56	-1.01±0.64
	Median	1.00	-1.00	1.00	-1.14	1.00	-1.00
	[Min, Max]	[0.00, 2.00]	[-2.88, 0.50]	[0.00, 2.25]	[-2.56, 0.50]	[0.00, 3.00]	[-3.00, 0.67]
Day 10	N	98	98	99	99	96	96
	Mean±SD	0.79±0.52	-1.09±0.66	0.66±0.58	-1.22±0.65	0.79±0.55	-1.11±0.62
	Median	1.00	-1.00	1.00	-1.25	1.00	-1.00
	[Min, Max]	[0.00, 2.20]	[-2.88, 0.50]	[0.00, 2.25]	[-2.60, 0.50]	[0.00, 2.00]	[-3.00, 0.00]
Day 10.5	N	98	98	99	99	96	96
	Mean±SD	0.79±0.52	-1.09±0.65	0.64±0.59	-1.24±0.68	0.81±0.54	-1.09±0.65
	Median	1.00	-1.00	1.00	-1.25	1.00	-1.00
	[Min, Max]	[0.00, 2.00]	[-2.88, 0.25]	[0.00, 2.17]	[-2.60, 0.50]	[0.00, 2.00]	[-3.00, 0.00]
Day 11	N	98	98	99	99	96	96
	Mean±SD	0.73±0.52	-1.15±0.67	0.68±0.59	-1.20±0.68	0.74±0.54	-1.15±0.64
	Median	1.00	-1.00	1.00	-1.20	1.00	-1.00
	[Min, Max]	[0.00, 2.00]	[-2.88, 0.25]	[0.00, 2.67]	[-2.60, 0.67]	[0.00, 2.00]	[-3.00, 0.00]
Day 11.5	N	98	98	99	99	96	96
	Mean±SD	0.74±0.56	-1.14±0.70	0.68±0.60	-1.20±0.69	0.76±0.58	-1.14±0.68

Category		Placebo N=98		CP-COV03 300 mg N=99		CP-COV03 450 mg N=96	
		Values (Day#)	Change (Day# - baseline)	Values (Day#)	Change (Day# - baseline)	Values (Day#)	Change (Day# - baseline)
Day 12	Median [Min, Max]	1.00 [0.00, 2.00]	-1.00 [-2.88, 0.38]	1.00 [0.00, 2.67]	-1.20 [-2.60, 0.67]	1.00 [0.00, 3.00]	-1.00 [-3.00, 1.00]
	N	98	98	99	99	96	96
	Mean±SD	0.70±0.55	-1.18±0.71	0.65±0.59	-1.23±0.68	0.73±0.54	-1.16±0.63
Day 12.5	Median [Min, Max]	1.00 [0.00, 2.00]	-1.06 [-2.88, 0.38]	1.00 [0.00, 2.67]	-1.22 [-2.60, 0.67]	1.00 [0.00, 2.00]	-1.00 [-3.00, 0.00]
	N	98	98	99	99	96	96
	Mean±SD	0.67±0.55	-1.21±0.67	0.69±0.65	-1.19±0.72	0.76±0.59	-1.13±0.71
Day 13	Median [Min, Max]	1.00 [0.00, 2.00]	-1.17 [-2.88, 0.25]	1.00 [0.00, 2.75]	-1.22 [-2.60, 0.75]	1.00 [0.00, 3.00]	-1.00 [-3.00, 1.00]
	N	98	98	99	99	96	96
	Mean±SD	0.66±0.56	-1.22±0.64	0.62±0.60	-1.26±0.70	0.69±0.59	-1.20±0.67
Day 13.5	Median [Min, Max]	1.00 [0.00, 2.00]	-1.25 [-2.88, 0.25]	1.00 [0.00, 2.75]	-1.25 [-2.60, 0.75]	1.00 [0.00, 2.00]	-1.19 [-3.00, 0.14]
	N	98	98	99	99	96	96
	Mean±SD	0.60±0.60	-1.28±0.72	0.62±0.60	-1.26±0.70	0.71±0.57	-1.19±0.66
Day 14	Median [Min, Max]	1.00 [0.00, 2.20]	-1.33 [-2.88, 0.50]	1.00 [0.00, 2.25]	-1.25 [-2.67, 0.50]	1.00 [0.00, 2.44]	-1.10 [-3.00, 0.57]
	N	98	98	99	99	96	96
	Mean±SD	0.59±0.59	-1.29±0.68	0.62±0.58	-1.26±0.67	0.65±0.57	-1.25±0.65
Day 14	Median	1.00	-1.33	1.00	-1.25	1.00	-1.25
	N	98	98	99	99	96	96
	Mean±SD	0.59±0.59	-1.29±0.68	0.62±0.58	-1.26±0.67	0.65±0.57	-1.25±0.65

Category		Placebo N=98		CP-COV03 300 mg N=99		CP-COV03 450 mg N=96	
		Values (Day#)	Change (Day# - baseline)	Values (Day#)	Change (Day# - baseline)	Values (Day#)	Change (Day# - baseline)
	[Min, Max]	[0.00, 2.33]	[-2.88, 0.52]	[0.00, 2.25]	[-2.60, 0.50]	[0.00, 2.00]	[-3.00, 0.00]
Day 14.5	N	98	98	99	99	96	96
	Mean±SD	0.64±0.62	-1.24±0.70	0.61±0.59	-1.27±0.70	0.69±0.62	-1.21±0.69
	Median	1.00	-1.31	1.00	-1.25	1.00	-1.20
	[Min, Max]	[0.00, 2.33]	[-2.88, 0.52]	[0.00, 2.00]	[-2.67, 0.50]	[0.00, 3.00]	[-3.00, 1.00]

Table 16. Secondary outcome 4) descriptive statistics for the change in average symptom score (PPS)

Category		Placebo N=77		CP-COV03 300 mg N=70		CP-COV03 450 mg N=80	
		Values (Day#)	Change (Day# - baseline)	Values (Day#)	Change (Day# - baseline)	Values (Day#)	Change (Day# - baseline)
Baseline	N	77	NA	70	NA	80	NA
	Mean±SD	1.86±0.40	NA	1.89±0.34	NA	1.90±0.38	NA
	Median [Min, Max]	1.80 [1.29, 3.00]	NA	1.87 [1.33, 2.67]	NA	1.88 [1.29, 3.00]	NA
Day 2	N	77	77	70	70	80	80
	Mean±SD	1.47±0.34	-0.38±0.45	1.45±0.33	-0.44±0.43	1.48±0.35	-0.42±0.40
	Median [Min, Max]	1.50 [1.00, 2.29]	-0.34 [-1.50, 0.51]	1.44 [1.00, 2.25]	-0.47 [-1.67, 0.67]	1.50 [1.00, 2.60]	-0.39 [-1.33, 1.03]
Day 2.5	N	77	77	70	70	80	80

Category		Placebo N=77		CP-COV03 300 mg N=70		CP-COV03 450 mg N=80	
		Values (Day#)	Change (Day# - baseline)	Values (Day#)	Change (Day# - baseline)	Values (Day#)	Change (Day# - baseline)
	Mean±SD	1.32±0.37	-0.54±0.49	1.31±0.28	-0.58±0.36	1.41±0.38	-0.48±0.40
	Median [Min, Max]	1.29 [0.00, 2.33]	-0.50 [-1.75, 0.60]	1.29 [1.00, 2.00]	-0.58 [-1.43, 0.30]	1.33 [1.00, 2.50]	-0.52 [-1.50, 0.52]
Day 3	N	77	77	70	70	80	80
	Mean±SD	1.28±0.40	-0.57±0.51	1.25±0.29	-0.64±0.39	1.29±0.30	-0.61±0.41
	Median [Min, Max]	1.20 [0.00, 3.00]	-0.55 [-2.29, 0.75]	1.17 [1.00, 2.00]	-0.63 [-1.56, 0.67]	1.25 [1.00, 2.33]	-0.53 [-1.75, 0.19]
Day 3.5	N	77	77	70	70	80	80
	Mean±SD	1.22±0.44	-0.64±0.57	1.17±0.38	-0.72±0.47	1.26±0.31	-0.64±0.47
	Median [Min, Max]	1.00 [0.00, 3.00]	-0.57 [-2.33, 1.29]	1.06 [0.00, 2.33]	-0.67 [-2.00, 0.33]	1.33 [0.00, 2.00]	-0.57 [-2.00, 0.40]
Day 4	N	77	77	70	70	80	80
	Mean±SD	1.08±0.41	-0.77±0.55	1.16±0.47	-0.73±0.54	1.21±0.32	-0.68±0.43
	Median [Min, Max]	1.00 [0.00, 2.33]	-0.71 [-2.50, 0.29]	1.00 [0.00, 2.60]	-0.75 [-2.00, 0.60]	1.00 [0.00, 2.00]	-0.63 [-2.00, 0.13]
Day 4.5	N	77	77	70	70	80	80
	Mean±SD	1.00±0.46	-0.85±0.60	1.07±0.50	-0.82±0.57	1.18±0.30	-0.72±0.45
	Median [Min, Max]	1.00 [0.00, 2.00]	-0.75 [-2.80, 0.20]	1.00 [0.00, 2.60]	-0.85 [-2.00, 0.60]	1.00 [0.00, 2.17]	-0.67 [-2.00, 0.17]
Day 5	N	77	77	70	70	80	80
	Mean±SD	0.97±0.50	-0.88±0.67	0.97±0.53	-0.92±0.60	1.14±0.36	-0.76±0.50

Category		Placebo N=77		CP-COV03 300 mg N=70		CP-COV03 450 mg N=80	
		Values (Day#)	Change (Day# - baseline)	Values (Day#)	Change (Day# - baseline)	Values (Day#)	Change (Day# - baseline)
Day 5.5	Median [Min, Max]	1.00 [0.00, 2.33]	-0.75 [-2.80, 0.60]	1.00 [0.00, 2.33]	-0.87 [-2.14, 0.42]	1.00 [0.00, 2.00]	-0.67 [-2.29, 0.22]
	N	77	77	70	70	80	80
	Mean±SD	0.98±0.54	-0.88±0.70	0.92±0.54	-0.98±0.59	1.06±0.40	-0.83±0.54
Day 6	Median [Min, Max]	1.00 [0.00, 2.33]	-0.75 [-2.80, 0.60]	1.00 [0.00, 2.60]	-0.86 [-2.14, 0.67]	1.00 [0.00, 2.00]	-0.67 [-2.63, 0.10]
	N	77	77	70	70	80	80
	Mean±SD	0.86±0.54	-0.99±0.70	0.84±0.56	-1.05±0.60	0.93±0.54	-0.97±0.68
Day 6.5	Median [Min, Max]	1.00 [0.00, 2.00]	-1.00 [-2.80, 0.60]	1.00 [0.00, 2.29]	-1.00 [-2.14, 0.67]	1.00 [0.00, 2.00]	-0.79 [-3.00, 0.60]
	N	77	77	70	70	80	80
	Mean±SD	0.87±0.43	-0.99±0.61	0.78±0.53	-1.11±0.60	1.04±0.49	-0.85±0.59
Day 7	Median [Min, Max]	1.00 [0.00, 1.67]	-0.83 [-2.88, -0.13]	1.00 [0.00, 2.14]	-1.00 [-2.14, 0.14]	1.00 [0.00, 2.50]	-0.70 [-2.63, 0.50]
	N	77	77	70	70	80	80
	Mean±SD	0.94±0.41	-0.92±0.60	0.76±0.52	-1.13±0.58	0.93±0.50	-0.96±0.60
Day 7.5	Median [Min, Max]	1.00 [0.00, 2.00]	-0.75 [-2.88, -0.07]	1.00 [0.00, 2.00]	-1.00 [-2.14, 0.67]	1.00 [0.00, 2.00]	-0.83 [-3.00, 0.00]
	N	77	77	70	70	80	80
	Mean±SD	0.87±0.44	-0.98±0.63	0.66±0.55	-1.23±0.61	0.95±0.49	-0.95±0.58
Day 7.5	Median	1.00	-0.83	1.00	-1.25	1.00	-0.83
	N	77	77	70	70	80	80
	Mean±SD	0.87±0.44	-0.98±0.63	0.66±0.55	-1.23±0.61	0.95±0.49	-0.95±0.58

Category		Placebo N=77		CP-COV03 300 mg N=70		CP-COV03 450 mg N=80	
		Values (Day#)	Change (Day# - baseline)	Values (Day#)	Change (Day# - baseline)	Values (Day#)	Change (Day# - baseline)
	[Min, Max]	[0.00, 1.50]	[-2.88, 0.00]	[0.00, 2.17]	[-2.56, 0.17]	[0.00, 2.00]	[-3.00, 0.00]
Day 8	N	77	77	70	70	80	80
	Mean±SD	0.84±0.53	-1.02±0.71	0.70±0.57	-1.19±0.64	0.85±0.54	-1.04±0.59
	Median	1.00	-0.95	1.00	-1.22	1.00	-1.00
	[Min, Max]	[0.00, 3.00]	[-2.88, 1.50]	[0.00, 2.60]	[-2.56, 0.60]	[0.00, 2.00]	[-3.00, 0.40]
Day 8.5	N	77	77	70	70	80	80
	Mean±SD	0.79±0.52	-1.07±0.71	0.68±0.53	-1.21±0.61	0.92±0.58	-0.97±0.63
	Median	1.00	-1.00	1.00	-1.21	1.00	-0.87
	[Min, Max]	[0.00, 2.00]	[-2.88, 0.50]	[0.00, 2.00]	[-2.56, 0.00]	[0.00, 3.00]	[-3.00, 0.67]
Day 9	N	77	77	70	70	80	80
	Mean±SD	0.84±0.50	-1.02±0.69	0.62±0.57	-1.27±0.64	0.80±0.60	-1.09±0.65
	Median	1.00	-0.83	1.00	-1.29	1.00	-1.00
	[Min, Max]	[0.00, 2.00]	[-2.88, 0.50]	[0.00, 2.00]	[-2.56, 0.00]	[0.00, 3.00]	[-3.00, 0.67]
Day 9.5	N	77	77	70	70	80	80
	Mean±SD	0.83±0.51	-1.03±0.69	0.60±0.54	-1.29±0.60	0.86±0.58	-1.03±0.66
	Median	1.00	-0.82	1.00	-1.33	1.00	-0.95
	[Min, Max]	[0.00, 2.00]	[-2.88, 0.50]	[0.00, 2.00]	[-2.56, 0.00]	[0.00, 3.00]	[-3.00, 0.67]
Day 10	N	77	77	70	70	80	80
	Mean±SD	0.81±0.51	-1.05±0.67	0.57±0.53	-1.33±0.59	0.77±0.59	-1.13±0.66
	Median	1.00	-1.00	1.00	-1.33	1.00	-1.00
	[Min, Max]	[0.00, 2.20]	[-2.88, 0.50]	[0.00, 1.75]	[-2.60, -0.25]	[0.00, 2.00]	[-3.00, 0.00]

Category		Placebo N=77		CP-COV03 300 mg N=70		CP-COV03 450 mg N=80	
		Values (Day#)	Change (Day# - baseline)	Values (Day#)	Change (Day# - baseline)	Values (Day#)	Change (Day# - baseline)
Day 10.5	N	77	77	70	70	80	80
	Mean±SD	0.80±0.52	-1.05±0.67	0.53±0.56	-1.36±0.63	0.80±0.56	-1.09±0.68
	Median [Min, Max]	1.00 [0.00, 2.00]	-1.00 [-2.88, 0.25]	0.50 [0.00, 2.17]	-1.41 [-2.60, 0.17]	1.00 [0.00, 2.00]	-1.00 [-3.00, 0.00]
Day 11	N	77	77	70	70	80	80
	Mean±SD	0.76±0.51	-1.09±0.67	0.58±0.54	-1.31±0.61	0.72±0.57	-1.18±0.68
	Median [Min, Max]	1.00 [0.00, 2.00]	-1.00 [-2.88, 0.25]	1.00 [0.00, 1.71]	-1.33 [-2.60, -0.29]	1.00 [0.00, 2.00]	-1.14 [-3.00, 0.00]
Day 11.5	N	77	77	70	70	80	80
	Mean±SD	0.75±0.56	-1.11±0.70	0.58±0.56	-1.31±0.64	0.75±0.59	-1.15±0.70
	Median [Min, Max]	1.00 [0.00, 2.00]	-1.00 [-2.88, 0.38]	1.00 [0.00, 2.00]	-1.33 [-2.60, 0.25]	1.00 [0.00, 3.00]	-1.07 [-3.00, 1.00]
Day 12	N	77	77	70	70	80	80
	Mean±SD	0.70±0.55	-1.16±0.70	0.54±0.54	-1.35±0.62	0.70±0.56	-1.20±0.67
	Median [Min, Max]	1.00 [0.00, 2.00]	-1.00 [-2.88, 0.38]	1.00 [0.00, 1.88]	-1.37 [-2.60, -0.13]	1.00 [0.00, 2.00]	-1.24 [-3.00, 0.00]
Day 12.5	N	77	77	70	70	80	80
	Mean±SD	0.70±0.56	-1.16±0.68	0.60±0.63	-1.29±0.68	0.74±0.62	-1.15±0.75
	Median [Min, Max]	1.00 [0.00, 2.00]	-1.00 [-2.88, 0.25]	1.00 [0.00, 2.75]	-1.33 [-2.60, 0.75]	1.00 [0.00, 3.00]	-1.07 [-3.00, 1.00]
Day 13	N	77	77	70	70	80	80

Category		Placebo N=77		CP-COV03 300 mg N=70		CP-COV03 450 mg N=80	
		Values (Day#)	Change (Day# - baseline)	Values (Day#)	Change (Day# - baseline)	Values (Day#)	Change (Day# - baseline)
	Mean±SD	0.67±0.57	-1.19±0.66	0.54±0.55	-1.35±0.64	0.66±0.61	-1.23±0.71
	Median [Min, Max]	1.00 [0.00, 2.00]	-1.14 [-2.88, 0.25]	1.00 [0.00, 2.20]	-1.37 [-2.60, 0.20]	1.00 [0.00, 2.00]	-1.33 [-3.00, 0.14]
	N	77	77	70	70	80	80
Day 13.5	Mean±SD	0.66±0.61	-1.20±0.71	0.56±0.58	-1.33±0.68	0.70±0.59	-1.20±0.70
	Median [Min, Max]	1.00 [0.00, 2.20]	-1.20 [-2.88, 0.50]	1.00 [0.00, 2.00]	-1.33 [-2.67, 0.25]	1.00 [0.00, 2.44]	-1.19 [-3.00, 0.57]
	N	77	77	70	70	80	80
Day 14	Mean±SD	0.65±0.61	-1.20±0.67	0.53±0.54	-1.36±0.62	0.65±0.58	-1.25±0.67
	Median [Min, Max]	1.00 [0.00, 2.33]	-1.25 [-2.88, 0.52]	0.50 [0.00, 1.80]	-1.41 [-2.60, 0.00]	1.00 [0.00, 2.00]	-1.33 [-3.00, 0.00]
	N	77	77	70	70	80	80
Day 14.5	Mean±SD	0.67±0.63	-1.19±0.69	0.50±0.54	-1.39±0.65	0.70±0.64	-1.19±0.72
	Median [Min, Max]	1.00 [0.00, 2.33]	-1.25 [-2.88, 0.52]	0.00 [0.00, 1.60]	-1.46 [-2.67, -0.15]	1.00 [0.00, 3.00]	-1.24 [-3.00, 1.00]
	N	77	77	70	70	80	80

Table 17. Secondary outcome 4) ANCOVA results for the change in total symptom score (ITT)

Day #	Placebo vs CP-COV03 300 mg			Placebo vs CP-COV03 450 mg		
	LS Mean [95% CI]		p-value	LS Mean [95% CI]		p-value
	Placebo	300 mg		Placebo	450 mg	
Day 2	9.7887 [8.9324, 10.6451]	9.4112 [8.5592, 10.2632]	0.5390	9.6942 [8.8088, 10.5797]	9.8642 [8.9696, 10.7589]	0.7906
Day 2.5	7.3553 [6.5691, 8.1415]	8.0725 [7.2903, 8.8548]	0.2045	7.3188 [6.4703, 8.1673]	8.5287 [7.6714, 9.3860]	0.0497*
Day 3	6.1623 [5.3992, 6.9255]	6.6979 [5.9386, 7.4571]	0.3286	6.1241 [5.4267, 6.8216]	6.6962 [5.9915, 7.4009]	0.2573
Day 3.5	5.1119 [4.3426, 5.8813]	5.7377 [4.9722, 6.5031]	0.2577	5.0986 [4.4411, 5.7560]	6.0452 [5.3809, 6.7095]	0.0476*
Day 4	4.0134 [3.3176, 4.7092]	5.1079 [4.4156, 5.8002]	0.0293*	4.0196 [3.4796, 4.5597]	4.9487 [4.4030, 5.4944]	0.0182*
Day 4.5	3.5464 [2.9039, 4.1889]	4.2470 [3.6078, 4.8862]	0.1297	3.5546 [2.9941, 4.1152]	4.5484 [3.9820, 5.1148]	0.0150*
Day 5	3.0817 [2.4830, 3.6804]	3.7273 [3.1316, 4.3229]	0.1339	3.0810 [2.5531, 3.6089]	3.8131 [3.2797, 4.3465]	0.0563
Day 5.5	2.7718 [2.1987, 3.3449]	3.0542 [2.4840, 3.6244]	0.4924	2.7716 [2.2365, 3.3067]	3.4936 [2.9529, 4.0343]	0.0632
Day 6	2.3617 [1.7651, 2.9584]	2.8944 [2.3008, 3.4881]	0.2142	2.3698 [1.8814, 2.8582]	2.8725 [2.3790, 3.3660]	0.1557
Day 6.5	2.6534 [2.0606, 3.2462]	2.9795 [2.3897, 3.5693]	0.4435	2.6831 [2.1862, 3.1800]	3.2297 [2.7277, 3.7318]	0.1293
Day 7	2.6318 [2.1405, 3.1232]	2.5362 [2.0473, 3.0250]	0.7861	2.6606 [2.2447, 3.0764]	2.5965 [2.1763, 3.0167]	0.8312
Day 7.5	2.4310 [1.9307, 2.9313]	2.3208 [1.8230, 2.8186]	0.7588	2.4460 [2.0152, 2.8768]	2.5656 [2.1303, 3.0009]	0.7011
Day 8	2.1132 [1.6433, 2.5831]	2.2617 [1.7942, 2.7292]	0.6597	2.1380 [1.7490, 2.5271]	2.1195 [1.7264, 2.5126]	0.9474

Day #	Placebo vs CP-COV03 300 mg			Placebo vs CP-COV03 450 mg		
	LS Mean [95% CI]		p-value	LS Mean [95% CI]		p-value
	Placebo	300 mg		Placebo	450 mg	
Day 8.5	1.9720 [1.5010, 2.4430]	2.0782 [1.6096, 2.5468]	0.7534	2.0011 [1.6184, 2.3838]	2.1447 [1.7580, 2.5313]	0.6040
Day 9	2.0601 [1.5842, 2.5360]	2.0516 [1.5782, 2.5251]	0.9802	2.0856 [1.6952, 2.4760]	1.9126 [1.5181, 2.3071]	0.5402
Day 9.5	1.9654 [1.4924, 2.4383]	2.0141 [1.5435, 2.4846]	0.8859	1.9867 [1.6151, 2.3583]	1.9511 [1.5756, 2.3266]	0.8945
Day 10	1.8460 [1.3878, 2.3042]	1.7989 [1.3431, 2.2548]	0.8862	1.8689 [1.5114, 2.2264]	1.8109 [1.4497, 2.1722]	0.8227
Day 10.5	1.8744 [1.4116, 2.3372]	1.7809 [1.3204, 2.2413]	0.7782	1.9024 [1.5337, 2.2711]	1.7976 [1.4251, 2.1701]	0.6943
Day 11	1.6984 [1.2363, 2.1605]	1.8541 [1.3943, 2.3138]	0.6387	1.7216 [1.3613, 2.0819]	1.5863 [1.2222, 1.9503]	0.6036
Day 11.5	1.6223 [1.1743, 2.0704]	1.7678 [1.3220, 2.2136]	0.6510	1.6504 [1.3152, 1.9856]	1.6173 [1.2786, 1.9560]	0.8914
Day 12	1.4426 [1.0101, 1.8751]	1.7538 [1.3235, 2.1841]	0.3165	1.4783 [1.1513, 1.8054]	1.5429 [1.2124, 1.8734]	0.7848
Day 12.5	1.5330 [1.0965, 1.9694]	1.7148 [1.2806, 2.1490]	0.5614	1.5564 [1.2180, 1.8948]	1.5049 [1.1630, 1.8468]	0.8331
Day 13	1.4946 [1.0577, 1.9314]	1.5407 [1.1061, 1.9753]	0.8829	1.5155 [1.1475, 1.8834]	1.4634 [1.0916, 1.8351]	0.8448
Day 13.5	1.3823 [0.9407, 1.8239]	1.5307 [1.0913, 1.9700]	0.6396	1.4054 [0.9586, 1.8522]	1.6070 [1.1556, 2.0584]	0.5327
Day 14	1.2703 [0.8326, 1.7081]	1.5203 [1.0847, 1.9558]	0.4265	1.2929 [0.9127, 1.6730]	1.3989 [1.0148, 1.7830]	0.6996
Day 14.5	1.3886 [0.9428, 1.8344]	1.4840 [1.0404, 1.9275]	0.7657	1.4131 [1.0348, 1.7913]	1.4221 [1.0399, 1.8043]	0.9737

* Statistically significant difference was observed at the 5% significance level

Table 18. Secondary outcome 4) ANCOVA results for the change in total symptom score (PPS)

Day #	Placebo vs CP-COV03 300 mg			Placebo vs CP-COV03 450 mg		
	LS Mean [95% CI]		p-value	LS Mean [95% CI]		p-value
	Placebo	300 mg		Placebo	450 mg	
Day 2	9.8599 [9.0002, 10.7195]	8.9684 [8.0663, 9.8706]	0.1622	9.7664 [8.9156, 10.6172]	9.5498 [8.7152, 10.3844]	0.7207
Day 2.5	7.2968 [6.5132, 8.0805]	7.4021 [6.5797, 8.2245]	0.8559	7.2195 [6.3419, 8.0971]	8.3637 [7.5028, 9.2246]	0.0686
Day 3	6.0613 [5.3206, 6.8021]	5.8611 [5.0837, 6.6385]	0.7148	6.0091 [5.2651, 6.7530]	6.3163 [5.5865, 7.0460]	0.5621
Day 3.5	5.0786 [4.3276, 5.8297]	4.8992 [4.1111, 5.6874]	0.7467	5.0732 [4.3359, 5.8105]	5.8795 [5.1562, 6.6028]	0.1260
Day 4	3.9337 [3.2657, 4.6017]	4.2158 [3.5147, 4.9168]	0.5682	3.9585 [3.3472, 4.5698]	4.7024 [4.1028, 5.3021]	0.0889
Day 4.5	3.3413 [2.7360, 3.9466]	3.4246 [2.7894, 4.0598]	0.8523	3.3565 [2.7449, 3.9681]	4.3569 [3.7569, 4.9568]	0.0227*
Day 5	2.9126 [2.3297, 3.4954]	3.0390 [2.4273, 3.6507]	0.7693	2.9077 [2.3231, 3.4923]	3.7264 [3.1529, 4.2999]	0.0506
Day 5.5	2.5937 [2.0473, 3.1401]	2.4041 [1.8307, 2.9775]	0.6389	2.6040 [2.0097, 3.1983]	3.3686 [2.7856, 3.9516]	0.0723
Day 6	2.1376 [1.5888, 2.6863]	2.3344 [1.7585, 2.9103]	0.6277	2.1439 [1.6044, 2.6834]	2.8615 [2.3323, 3.3907]	0.0632
Day 6.5	2.4802 [1.9471, 3.0134]	2.1146 [1.5551, 2.6741]	0.3544	2.5251 [1.9913, 3.0588]	3.0446 [2.5211, 3.5682]	0.1729
Day 7	2.5459 [2.0660, 3.0259]	1.9137 [1.4101, 2.4174]	0.0764	2.5887 [2.1247, 3.0528]	2.5709 [2.1156, 3.0261]	0.9569
Day 7.5	2.3141 [1.8140, 2.8141]	1.8117 [1.2869, 2.3364]	0.1756	2.3503 [1.8743, 2.8263]	2.4503 [1.9834, 2.9173]	0.7679
Day 8	2.1501 [1.6709, 2.6294]	1.7492 [1.2462, 2.2521]	0.2589	2.1877 [1.7400, 2.6355]	2.0568 [1.6176, 2.4961]	0.6814

Day #	Placebo vs CP-COV03 300 mg			Placebo vs CP-COV03 450 mg		
	LS Mean [95% CI]		p-value	LS Mean [95% CI]		p-value
	Placebo	300 mg		Placebo	450 mg	
Day 8.5	1.9955 [1.5239, 2.4670]	1.6050 [1.1101, 2.0998]	0.2637	2.0478 [1.6068, 2.4887]	2.1165 [1.6840, 2.5491]	0.8266
Day 9	2.1789 [1.6823, 2.6756]	1.5603 [1.0391, 2.0815]	0.0937	2.2225 [1.7652, 2.6798]	1.8733 [1.4247, 2.3220]	0.2845
Day 9.5	2.0453 [1.5699, 2.5208]	1.4787 [0.9798, 1.9777]	0.1086	2.0776 [1.6441, 2.5110]	1.9503 [1.5251, 2.3756]	0.6803
Day 10	1.8924 [1.4310, 2.3538]	1.3326 [0.8484, 1.8168]	0.1024	1.9403 [1.5285, 2.3522]	1.7699 [1.3659, 2.1740]	0.5615
Day 10.5	1.9257 [1.4507, 2.4006]	1.3103 [0.8119, 1.8088]	0.0814	1.9704 [1.5396, 2.4011]	1.7910 [1.3684, 2.2136]	0.5588
Day 11	1.7823 [1.3116, 2.2529]	1.3823 [0.8884, 1.8763]	0.2515	1.8213 [1.3988, 2.2438]	1.5970 [1.1825, 2.0114]	0.4562
Day 11.5	1.6482 [1.1956, 2.1007]	1.3727 [0.8978, 1.8476]	0.4109	1.6964 [1.3125, 2.0803]	1.6047 [1.2282, 1.9813]	0.7375
Day 12	1.4294 [0.9779, 1.8810]	1.4419 [0.9680, 1.9158]	0.9701	1.5020 [1.1305, 1.8735]	1.5168 [1.1523, 1.8812]	0.9556
Day 12.5	1.5821 [1.1237, 2.0404]	1.3740 [0.8930, 1.8551]	0.5395	1.6344 [1.2495, 2.0193]	1.4644 [1.0868, 1.8419]	0.5352
Day 13	1.5413 [1.0965, 1.9861]	1.2474 [0.7806, 1.7142]	0.3722	1.5936 [1.1668, 2.0203]	1.4287 [1.0101, 1.8473]	0.5876
Day 13.5	1.5125 [1.0623, 1.9627]	1.2220 [0.7495, 1.6944]	0.3833	1.5522 [1.0234, 2.0810]	1.5935 [1.0748, 2.1122]	0.9127
Day 14	1.4100 [0.9664, 1.8536]	1.1347 [0.6692, 1.6003]	0.4020	1.4577 [1.0115, 1.9039]	1.4095 [0.9717, 1.8472]	0.8793
Day 14.5	1.4817 [1.0201, 1.9434]	1.1415 [0.6571, 1.6260]	0.3197	1.5298 [1.0969, 1.9627]	1.4151 [0.9904, 1.8398]	0.7098

* Statistically significant difference was observed at the 5% significance level

Table 19. Secondary outcome 4) ANCOVA results for the change in average symptom score (ITT)

Day #	Placebo vs CP-COV03 300 mg			Placebo vs CP-COV03 450 mg		
	LS Mean [95% CI]		p-value	LS Mean [95% CI]		p-value
	Placebo	300 mg		Placebo	450 mg	
Day 2	1.5181 [1.4499, 1.5863]	1.4657 [1.3978, 1.5335]	0.2844	1.5195 [1.4500, 1.5891]	1.4929 [1.4227, 1.5632]	0.5964
Day 2.5	1.3568 [1.2896, 1.4239]	1.3729 [1.3060, 1.4397]	0.7381	1.3614 [1.2880, 1.4349]	1.4167 [1.3425, 1.4909]	0.2989
Day 3	1.3046 [1.2344, 1.3749]	1.3094 [1.2395, 1.3793]	0.9251	1.3046 [1.2363, 1.3730]	1.2938 [1.2248, 1.3629]	0.8267
Day 3.5	1.2320 [1.1510, 1.3130]	1.2244 [1.1438, 1.3051]	0.8965	1.2335 [1.1595, 1.3076]	1.2454 [1.1705, 1.3202]	0.8248
Day 4	1.0995 [1.0137, 1.1854]	1.2195 [1.1341, 1.3049]	0.0523	1.1008 [1.0285, 1.1732]	1.2039 [1.1308, 1.2770]	0.0500*
Day 4.5	1.0434 [0.9527, 1.1340]	1.1266 [1.0363, 1.2168]	0.2015	1.0441 [0.9667, 1.1215]	1.1941 [1.1159, 1.2723]	0.0079*
Day 5	1.0144 [0.9169, 1.1119]	1.0063 [0.9093, 1.1033]	0.9075	1.0132 [0.9289, 1.0975]	1.1329 [1.0477, 1.2181]	0.0508
Day 5.5	1.0171 [0.9140, 1.1201]	0.9711 [0.8685, 1.0736]	0.5338	1.0147 [0.9238, 1.1057]	1.0789 [0.9870, 1.1708]	0.3294
Day 6	0.9138 [0.8069, 1.0206]	0.9078 [0.8014, 1.0141]	0.9379	0.9088 [0.8002, 1.0175]	0.9434 [0.8336, 1.0532]	0.6599
Day 6.5	0.9149 [0.8124, 1.0174]	0.8962 [0.7942, 0.9982]	0.7992	0.9175 [0.8237, 1.0114]	1.0571 [0.9623, 1.1519]	0.0408*
Day 7	0.9564 [0.8592, 1.0536]	0.8747 [0.7780, 0.9714]	0.2419	0.9588 [0.8690, 1.0487]	0.9588 [0.8681, 1.0496]	1.0000
Day 7.5	0.9051 [0.8010, 1.0092]	0.7731 [0.6695, 0.8767]	0.0783	0.9091 [0.8165, 1.0016]	0.9610 [0.8675, 1.0545]	0.4383
Day 8	0.8329 [0.7220, 0.9438]	0.7867 [0.6764, 0.8971]	0.5614	0.8369 [0.7332, 0.9406]	0.8840 [0.7792, 0.9888]	0.5304

Day #	Placebo vs CP-COV03 300 mg			Placebo vs CP-COV03 450 mg		
	LS Mean [95% CI]		p-value	LS Mean [95% CI]		p-value
	Placebo	300 mg		Placebo	450 mg	
Day 8.5	0.7879 [0.6782, 0.8976]	0.7533 [0.6442, 0.8625]	0.6603	0.7924 [0.6851, 0.8997]	0.9378 [0.8294, 1.0462]	0.0620
Day 9	0.8222 [0.7088, 0.9356]	0.7263 [0.6135, 0.8392]	0.2392	0.8285 [0.7168, 0.9402]	0.8338 [0.7210, 0.9467]	0.9477
Day 9.5	0.8034 [0.6940, 0.9128]	0.7037 [0.5949, 0.8126]	0.2048	0.8082 [0.7000, 0.9164]	0.8835 [0.7742, 0.9928]	0.3360
Day 10	0.7937 [0.6832, 0.9042]	0.6572 [0.5472, 0.7672]	0.0861	0.8009 [0.6939, 0.9079]	0.7816 [0.6735, 0.8897]	0.8031
Day 10.5	0.7856 [0.6743, 0.8969]	0.6427 [0.5319, 0.7534]	0.0745	0.7917 [0.6856, 0.8978]	0.8052 [0.6980, 0.9124]	0.8605
Day 11	0.7291 [0.6177, 0.8405]	0.6836 [0.5728, 0.7945]	0.5694	0.7331 [0.6268, 0.8394]	0.7409 [0.6335, 0.8483]	0.9189
Day 11.5	0.7438 [0.6272, 0.8604]	0.6807 [0.5647, 0.7967]	0.4504	0.7482 [0.6339, 0.8625]	0.7568 [0.6413, 0.8723]	0.9178
Day 12	0.6944 [0.5800, 0.8087]	0.6500 [0.5362, 0.7638]	0.5885	0.6972 [0.5870, 0.8074]	0.7324 [0.6211, 0.8437]	0.6585
Day 12.5	0.6717 [0.5509, 0.7925]	0.6946 [0.5744, 0.8147]	0.7918	0.6773 [0.5625, 0.7921]	0.7588 [0.6428, 0.8748]	0.3269
Day 13	0.6610 [0.5451, 0.7769]	0.6238 [0.5085, 0.7391]	0.6546	0.6641 [0.5497, 0.7784]	0.6891 [0.5736, 0.8046]	0.7621
Day 13.5	0.6008 [0.4812, 0.7204]	0.6223 [0.5033, 0.7413]	0.8015	0.6071 [0.4898, 0.7244]	0.7037 [0.5852, 0.8222]	0.2554
Day 14	0.5934 [0.4760, 0.7108]	0.6194 [0.5026, 0.7362]	0.7577	0.5982 [0.4822, 0.7141]	0.6423 [0.5252, 0.7594]	0.5984
Day 14.5	0.6323 [0.5117, 0.7529]	0.6113 [0.4913, 0.7313]	0.8078	0.6402 [0.5171, 0.7632]	0.6866 [0.5622, 0.8109]	0.6020

* Statistically significant difference was observed at the 5% significance level

Table 20. Secondary outcome 4) ANCOVA results for the change in average symptom score (PPS)

Day #	Placebo vs CP-COV03 300 mg			Placebo vs CP-COV03 450 mg		
	LS Mean [95% CI]		p-value	LS Mean [95% CI]		p-value
	Placebo	300 mg		Placebo	450 mg	
Day 2	1.4805 [1.4071, 1.5538]	1.4435 [1.3665, 1.5204]	0.4932	1.4806 [1.4071, 1.5542]	1.4713 [1.3992, 1.5434]	0.8586
Day 2.5	1.3249 [1.2522, 1.3976]	1.3061 [1.2299, 1.3824]	0.7265	1.3269 [1.2466, 1.4071]	1.4072 [1.3285, 1.4860]	0.1606
Day 3	1.2896 [1.2123, 1.3669]	1.2450 [1.1639, 1.3261]	0.4334	1.2884 [1.2111, 1.3656]	1.2844 [1.2087, 1.3602]	0.9430
Day 3.5	1.2196 [1.1270, 1.3121]	1.1667 [1.0696, 1.2638]	0.4382	1.2192 [1.1346, 1.3038]	1.2546 [1.1716, 1.3377]	0.5566
Day 4	1.0862 [0.9867, 1.1857]	1.1599 [1.0555, 1.2643]	0.3151	1.0860 [1.0032, 1.1687]	1.2076 [1.1265, 1.2888]	0.0401*
Day 4.5	1.0075 [0.8982, 1.1168]	1.0654 [0.9507, 1.1800]	0.4722	1.0082 [0.9199, 1.0965]	1.1727 [1.0862, 1.2593]	0.0096*
Day 5	0.9770 [0.8605, 1.0936]	0.9646 [0.8423, 1.0868]	0.8843	0.9768 [0.8775, 1.0760]	1.1344 [1.0370, 1.2318]	0.0268*
Day 5.5	0.9865 [0.8658, 1.1073]	0.9090 [0.7823, 1.0357]	0.3836	0.9852 [0.8796, 1.0908]	1.0565 [0.9529, 1.1601]	0.3433
Day 6	0.8694 [0.7446, 0.9942]	0.8384 [0.7074, 0.9693]	0.7356	0.8641 [0.7418, 0.9864]	0.9288 [0.8088, 1.0487]	0.4582
Day 6.5	0.8758 [0.7667, 0.9848]	0.7734 [0.6590, 0.8878]	0.2036	0.8787 [0.7760, 0.9813]	1.0374 [0.9367, 1.1381]	0.0310*
Day 7	0.9418 [0.8357, 1.0479]	0.7601 [0.6489, 0.8714]	0.0212*	0.9450 [0.8427, 1.0473]	0.9268 [0.8264, 1.0272]	0.8027
Day 7.5	0.8759 [0.7630, 0.9889]	0.6582 [0.5397, 0.7767]	0.0096*	0.8800 [0.7765, 0.9834]	0.9435 [0.8420, 1.0450]	0.3884
Day 8	0.8368 [0.7117, 0.9619]	0.7045 [0.5733, 0.8357]	0.1522	0.8422 [0.7219, 0.9625]	0.8504 [0.7324, 0.9684]	0.9234

Day #	Placebo vs CP-COV03 300 mg			Placebo vs CP-COV03 450 mg		
	LS Mean [95% CI]		p-value	LS Mean [95% CI]		p-value
	Placebo	300 mg		Placebo	450 mg	
Day 8.5	0.7891 [0.6692, 0.9091]	0.6798 [0.5540, 0.8057]	0.2170	0.7928 [0.6683, 0.9174]	0.9184 [0.7962, 1.0405]	0.1580
Day 9	0.8337 [0.7119, 0.9556]	0.6229 [0.4951, 0.7507]	0.0199*	0.8397 [0.7144, 0.9651]	0.8017 [0.6787, 0.9246]	0.6694
Day 9.5	0.8255 [0.7059, 0.9451]	0.6036 [0.4781, 0.7291]	0.0127*	0.8279 [0.7035, 0.9523]	0.8633 [0.7413, 0.9854]	0.6890
Day 10	0.8047 [0.6867, 0.9227]	0.5686 [0.4448, 0.6924]	0.0073*	0.8111 [0.6871, 0.9351]	0.7674 [0.6458, 0.8891]	0.6211
Day 10.5	0.8013 [0.6795, 0.9230]	0.5341 [0.4064, 0.6618]	0.0033*	0.8055 [0.6829, 0.9282]	0.8013 [0.6810, 0.9217]	0.9618
Day 11	0.7617 [0.6425, 0.8809]	0.5863 [0.4613, 0.7113]	0.0471*	0.7642 [0.6407, 0.8877]	0.7193 [0.5981, 0.8405]	0.6097
Day 11.5	0.7429 [0.6153, 0.8705]	0.5837 [0.4499, 0.7176]	0.0917	0.7461 [0.6157, 0.8765]	0.7461 [0.6182, 0.8741]	0.9999
Day 12	0.6981 [0.5741, 0.8222]	0.5431 [0.4130, 0.6733]	0.0911	0.7010 [0.5742, 0.8278]	0.6989 [0.5745, 0.8233]	0.9820
Day 12.5	0.6958 [0.5611, 0.8304]	0.5999 [0.4586, 0.7412]	0.3342	0.6996 [0.5650, 0.8343]	0.7418 [0.6097, 0.8739]	0.6599
Day 13	0.6697 [0.5422, 0.7972]	0.5409 [0.4072, 0.6747]	0.1716	0.6735 [0.5392, 0.8078]	0.6599 [0.5281, 0.7916]	0.8865
Day 13.5	0.6576 [0.5252, 0.7901]	0.5630 [0.4241, 0.7020]	0.3325	0.6653 [0.5293, 0.8014]	0.6904 [0.5569, 0.8238]	0.7959
Day 14	0.6549 [0.5241, 0.7856]	0.5242 [0.3870, 0.6614]	0.1759	0.6568 [0.5229, 0.7908]	0.6421 [0.5107, 0.7736]	0.8775
Day 14.5	0.6690 [0.5359, 0.8020]	0.4979 [0.3583, 0.6376]	0.0824	0.6752 [0.5321, 0.8183]	0.6974 [0.5570, 0.8378]	0.8275

* Statistically significant difference was observed at the 5% significance level

5) Time (days) taken for the total score of all COVID-19 symptoms to decrease by 25% or more and by 50% or more by Day 14

For the placebo, CP-COV03 300 mg dose, and CP-COV03 450 mg dose group, descriptive statistics for the time taken time (days) taken for the total score of all COVID-19 symptoms to decrease by 25% or more and by 50% or more by Day 14 are presented. The time taken for a decrease of more than 25% is shown in Tables 21–22, and for a decrease of more than 50% in Tables 23–24. Participants whose total symptom score did not decrease by at least 25% or 50% by the evening of Day 14 (= Day 14.5) were censored to have the maximum possible value as 13.00 days (maximum value of 13 days = 14.5 days - (first day of dosing + 0.5 day))

Table 21. Secondary outcome 5) descriptive statistics - time (days) taken for the total symptom score to decrease by 25% or more (ITT)

Total score		Placebo N=98	CP-COV03 300 mg N=99	CP-COV03 450 mg N=96
Time (days) taken for the total score of all COVID-19 symptoms to decrease by 25% or more	N	98	99	96
	Mean	2.06	2.56	2.20
	SD	2.54	3.38	2.56
	Median	1.00	1.00	1.25
	Min	0.50	0.50	0.50
	Max	13.00	13.00	13.00

Table 22. Secondary outcome 5) descriptive statistics - time (days) taken for the total symptom score to decrease by 25% or more (PPS)

Total score		Placebo N=77	CP-COV03 300 mg N=70	CP-COV03 450 mg N=80
Time (days) taken for the total score of all COVID-19 symptoms to decrease by 25% or more	N	77	70	80
	Mean	1.68	1.69	1.83
	SD	1.95	2.57	2.27
	Median	1.00	1.00	1.00
	Min	0.50	0.50	0.50
	Max	10.50	13.00	13.00

Table 23. Secondary outcome 5) descriptive statistics - time (days) taken for the total symptom score to decrease by 50% or more (ITT)

Total score		Placebo N=98	CP-COV03 300 mg N=99	CP-COV03 450 mg N=96
	N	98	99	96

Time (days) taken for the total score of all COVID-19 symptoms to decrease by 50% or more	Mean	3.39	3.93	3.46
	SD	3.56	4.06	3.28
	Median	2.00	2.00	2.50
	Min	0.50	0.50	0.50
	Max	13.00	13.00	13.00

Table 24. Secondary outcome 5) descriptive statistics - time (days) taken for the total symptom score to decrease by 50% or more (PPS)

Total score		Placebo N=98	CP-COV03 300 mg N=99	CP-COV03 450 mg N=96
Time (days) taken for the total score of all COVID-19 symptoms to decrease by 50% or more	N	77	70	80
	Mean	3.06	3.04	3.03
	SD	3.34	3.59	3.14
	Median	1.50	1.50	2.00
	Min	0.50	0.50	0.50
	Max	13.00	13.00	13.00

For the ITT analysis, the ANCOVA results indicated that at a 5% significance level, there was no statistically significant difference between the placebo group, CP-COV03 300 mg dose group, and CP-COV03 450 mg dose group in the time taken for a 25% or more, or 50% or more reduction in total symptom score. Similarly, in the PPS analysis, no statistically significant difference was observed between the placebo group and both, CP-COV03 300 mg dose group, and CP-COV03 450 mg dose group in the time taken for a 25% or more, or 50% or more reduction in total symptom score.

6) Presence and number of new COVID-19 symptoms by Day 14 other than pre-assessed symptoms at baseline

The presence and number of new COVID-19 symptoms by Day 14 other than pre-assessed symptoms at baseline for the placebo, CP-COV03 300 mg dose, and CP-COV03 450 mg dose group are presented in Tables 25–26 and the number of new symptoms for each is presented in Tables 27–28. Symptoms that were present at baseline and symptoms that did not appear throughout the study were not assessed.

In ITT, the number of participants who developed new symptoms was 75 (198 cases) in the placebo group, 80 (238 cases) in the CP-COV03 300 mg dose group, and 88 (246 cases) in the CP-COV03 450 mg dose group.

In PPS, the number of participants who developed new symptoms was 55 (143 cases) in the placebo group, 52 (120 cases) in the CP-COV03 300 mg dose group, and 72 (178 cases) in the CP-COV03 450 mg dose group.

Descriptive statistics for the number of new symptoms by Day 14 per participant across all groups are presented in Tables 29–30, followed by an ANCOVA analysis with age and severity as covariates, the results of which are presented in Tables 31–32.

Table 25. Secondary outcome 6) number of new symptoms by Day (ITT)

New symptoms presence	Placebo N=98 case (%)		CP-COV03 300 mg N=99 case (%)		CP-COV03 450 mg N=96 case (%)	
	Number of newly reported symptoms	Cumulative number of newly reported symptoms	Number of newly reported symptoms	Cumulative number of newly reported symptoms	Number of newly reported symptoms	Cumulative number of newly reported symptoms
Day 2	123(62.1)	123(62.1)	143(60.1)	143(60.1)	153(62.2)	153(62.2)
Day 2.5	24(12.1)	147(74.2)	33(13.9)	176(73.9)	32(13.0)	185(75.2)
Day 3	11(5.6)	158(79.8)	16(6.7)	192(80.7)	17(6.9)	202(82.1)
Day 3.5	6(3.0)	164(82.8)	9(3.8)	201(84.5)	8(3.3)	210(85.4)
Day 4	2(1.0)	166(83.8)	8(3.4)	209(87.8)	1(0.4)	211(85.8)
Day 4.5	4(2.0)	170(85.9)	7(2.9)	216(90.8)	9(3.7)	220(89.4)
Day 5	5(2.5)	175(88.4)	6(2.5)	222(93.3)	6(2.4)	226(91.9)
Day 5.5	1(0.5)	176(88.9)	1(0.4)	223(93.7)	2(0.8)	228(92.7)
Day 6	4(2.0)	180(90.9)	4(1.7)	227(95.4)	4(1.6)	232(94.3)
Day 6.5	3(1.5)	183(92.4)	5(2.1)	232(97.5)	4(1.6)	236(95.9)
Day 7	3(1.5)	186(93.9)	0(0.0)	232(97.5)	2(0.8)	238(96.7)
Day 7.5	2(1.0)	188(94.9)	2(0.8)	234(98.3)	1(0.4)	239(97.2)
Day 8	0(0.0)	188(94.9)	0(0.0)	234(98.3)	0(0.0)	239(97.2)
Day 8.5	0(0.0)	188(94.9)	0(0.0)	234(98.3)	2(0.8)	241(98.0)
Day 9	0(0.0)	188(94.9)	1(0.4)	235(98.7)	0(0.0)	241(98.0)
Day 9.5	0(0.0)	188(94.9)	0(0.0)	235(98.7)	0(0.0)	241(98.0)
Day 10	0(0.0)	188(94.9)	1(0.4)	236(99.2)	0(0.0)	241(98.0)
Day 10.5	2(1.0)	190(96.0)	1(0.4)	237(99.6)	0(0.0)	241(98.0)
Day 11	3(1.5)	193(97.5)	0(0.0)	237(99.6)	1(0.4)	242(98.4)
Day 11.5	2(1.0)	195(98.5)	0(0.0)	237(99.6)	0(0.0)	242(98.4)
Day 12	0(0.0)	195(98.5)	1(0.4)	238(100.0)	0(0.0)	242(98.4)
Day 12.5	1(0.5)	196(99.0)	0(0.0)	238(100.0)	0(0.0)	242(98.4)
Day 13	0(0.0)	196(99.0)	0(0.0)	238(100.0)	1(0.4)	243(98.8)
Day 13.5	0(0.0)	196(99.0)	0(0.0)	238(100.0)	2(0.8)	245(99.6)
Day 14	1(0.5)	197(99.5)	0(0.0)	238(100.0)	1(0.4)	246(100.0)

New symptoms presence	Placebo N=98 case (%)		CP-COV03 300 mg N=99 case (%)		CP-COV03 450 mg N=96 case (%)	
	Number of newly reported symptoms	Cumulative number of newly reported symptoms	Number of newly reported symptoms	Cumulative number of newly reported symptoms	Number of newly reported symptoms	Cumulative number of newly reported symptoms
Day 14.5	1(0.5)	198(100.0)	0(0.0)	238(100.0)	0(0.0)	246(100.0)

Table 26. Secondary outcome 6) number of new symptoms by Day (PPS)

New symptoms presence	Placebo N=77 case (%)		CP-COV03 300 mg N=70 case (%)		CP-COV03 450 mg N=80 case (%)	
	Number of newly reported symptoms	Cumulative number of newly reported symptoms	Number of newly reported symptoms	Cumulative number of newly reported symptoms	Number of newly reported symptoms	Cumulative number of newly reported symptoms
Day 2	90(62.9)	90(62.9)	66(55.0)	66(55.0)	103(57.9)	103(57.9)
Day 2.5	19(13.3)	109(76.2)	21(17.5)	87(72.5)	29(16.3)	132(74.2)
Day 3	8(5.6)	117(81.8)	11(9.2)	98(81.7)	10(5.6)	142(79.8)
Day 3.5	4(2.8)	121(84.6)	4(3.3)	102(85.0)	7(3.9)	149(83.7)
Day 4	2(1.4)	123(86.0)	4(3.3)	106(88.3)	0(0.0)	149(83.7)
Day 4.5	3(2.1)	126(88.1)	5(4.2)	111(92.5)	8(4.5)	157(88.2)
Day 5	2(1.4)	128(89.5)	1(0.8)	112(93.3)	6(3.4)	163(91.6)
Day 5.5	0(0.0)	128(89.5)	1(0.8)	113(94.2)	1(0.6)	164(92.1)
Day 6	4(2.8)	132(92.3)	3(2.5)	116(96.7)	4(2.2)	168(94.4)
Day 6.5	1(0.7)	133(93.0)	1(0.8)	117(97.5)	2(1.1)	170(95.5)
Day 7	3(2.1)	136(95.1)	0(0.0)	117(97.5)	1(0.6)	171(96.1)
Day 7.5	0(0.0)	136(95.1)	1(0.8)	118(98.3)	1(0.6)	172(96.6)
Day 8	0(0.0)	136(95.1)	0(0.0)	118(98.3)	0(0.0)	172(96.6)
Day 8.5	0(0.0)	136(95.1)	0(0.0)	118(98.3)	2(1.1)	174(97.8)
Day 9	0(0.0)	136(95.1)	0(0.0)	118(98.3)	0(0.0)	174(97.8)
Day 9.5	0(0.0)	136(95.1)	0(0.0)	118(98.3)	0(0.0)	174(97.8)
Day 10	0(0.0)	136(95.1)	1(0.8)	119(99.2)	0(0.0)	174(97.8)
Day 10.5	1(0.7)	137(95.8)	0(0.0)	119(99.2)	0(0.0)	174(97.8)
Day 11	3(2.1)	140(97.9)	0(0.0)	119(99.2)	1(0.6)	175(98.3)
Day 11.5	1(0.7)	141(98.6)	0(0.0)	119(99.2)	0(0.0)	175(98.3)

New symptoms presence	Placebo N=77 case (%)		CP-COV03 300 mg N=70 case (%)		CP-COV03 450 mg N=80 case (%)	
	Number of newly reported symptoms	Cumulative number of newly reported symptoms	Number of newly reported symptoms	Cumulative number of newly reported symptoms	Number of newly reported symptoms	Cumulative number of newly reported symptoms
Day 12	0(0.0)	141(98.6)	1(0.8)	120(100.0)	0(0.0)	175(98.3)
Day 12.5	0(0.0)	141(98.6)	0(0.0)	120(100.0)	0(0.0)	175(98.3)
Day 13	0(0.0)	141(98.6)	0(0.0)	120(100.0)	1(0.6)	176(98.9)
Day 13.5	0(0.0)	141(98.6)	0(0.0)	120(100.0)	1(0.6)	177(99.4)
Day 14	1(0.7)	142(99.3)	0(0.0)	120(100.0)	1(0.6)	178(100.0)
Day 14.5	1(0.7)	143(100.0)	0(0.0)	120(100.0)	0(0.0)	178(100.0)

Table 27. Secondary outcome 6) number of new symptoms (ITT)

Newly reported symptoms after baseline	Placebo case (%)	CP-COV03 300 mg case (%)	CP-COV03 450 mg case (%)
Total*	198(100.0)	238(100.0)	246(100.0)
Fever	25(12.6)	28(11.8)	27(11.0)
Cough	7(3.5)	12(5.0)	13(5.3)
Sore throat	7(3.5)	5(2.1)	8(3.3)
Headache	20(10.1)	23(9.7)	25(10.2)
Muscle ache	8(4.0)	5(2.1)	10(4.1)
Chill	12(6.1)	11(4.6)	12(4.9)
Stuffy or runny nose	26(13.1)	28(11.8)	23(9.3)
Fatigue	19(9.6)	17(7.1)	21(8.5)
Difficulty of breathing	18(9.1)	15(6.3)	15(6.1)
Nausea	17(8.6)	32(13.4)	24(9.8)
Vomiting	5(2.5)	9(3.8)	7(2.8)
Diarrhea	34(17.2)	53(22.3)	61(24.8)

* Based on the sum of all days up to Day 14

Table 28. Secondary outcome 6) number of new symptoms (PPS)

Newly reported symptoms after baseline	Placebo case (%)	CP-COV03 300 mg case (%)	CP-COV03 450 mg case (%)
Total*	143(100.0)	120(100.0)	178(100.0)
Fever	20(14.0)	14(11.7)	19(10.7)
Cough	5(3.5)	6(5.0)	7(3.9)
Sore throat	5(3.5)	1(0.8)	4(2.2)
Headache	11(7.7)	13(10.8)	18(10.1)
Muscle ache	6(4.2)	3(2.5)	8(4.5)
Chill	6(4.2)	6(5.0)	10(5.6)
Stuffy or runny nose	19(13.3)	12(10.0)	16(9.0)
Fatigue	14(9.8)	5(4.2)	14(7.9)
Difficulty of breathing	13(9.1)	8(6.7)	10(5.6)
Nausea	14(9.8)	15(12.5)	18(10.1)
Vomiting	5(3.5)	4(3.3)	6(3.4)
Diarrhea	25(17.5)	33(27.5)	48(27.0)

* Based on the sum of all days up to Day 14

Table 29. Secondary outcome 6) descriptive statistics for the number of new symptoms (ITT)

Number of new symptoms		Placebo N=98	CP-COV03 300 mg N=99	CP-COV03 450 mg N=96
Number of new symptoms per participant	N	98	99	96
	Mean	2.02	2.40	2.56
	SD	1.95	2.15	2.05
	Median	1.50	2.00	2.00
	Min	0.00	0.00	0.00
	Max	8.00	9.00	9.00

Table 30. Secondary outcome 6) descriptive statistics for the number of new symptoms (PPS)

Number of new symptoms		Placebo N=77	CP-COV03 300 mg N=70	CP-COV03 450 mg N=80
Number of new symptoms per participant	N	77	70	80
	Mean	1.86	1.71	2.23
	SD	1.95	1.72	1.80
	Median	1.00	1.00	2.00
	Min	0.00	0.00	0.00
	Max	8.00	7.00	9.00

Table 31. Secondary outcome 6) ANCOVA results (ITT)

Category	Placebo vs CP-COV03 300 mg			Placebo vs CP-COV03 450 mg		
	LS Mean [95% CI]		p-value	LS Mean [95% CI]		p-value
	Placebo	300 mg		Placebo	450 mg	
ITT	2.0248 [1.6338, 2.4158]	2.3997 [2.0106, 2.7887]	0.1821	2.0171 [1.6342, 2.4000]	2.5659 [2.1790, 2.9527]	0.0486*

* Statistically significant difference was observed at the 5% significance level

Table 32. Secondary outcome 6) ANCOVA results (PPS)

Category	Placebo vs CP-COV03 300 mg			Placebo vs CP-COV03 450 mg		
	LS Mean [95% CI]		p-value	LS Mean [95% CI]		p-value
	Placebo	300 mg		Placebo	450 mg	
PPS	1.8780 [1.4664, 2.2896]	1.6914 [1.2596, 2.1231]	0.5378	1.8777 [1.4597, 2.2958]	2.2052 [1.7950, 2.6153]	0.2716

In ITT, new symptoms were reported until the Day 14 evening in the placebo group, while no new symptoms appeared after the Day 12 morning in the CP-COV03 300 mg dose group. However, the ANCOVA analysis did not show a statistically significant difference in the number of newly reported symptoms between the placebo group and the CP-COV03 300 mg dose group at the 5% significance level. Between the placebo group and the CP-COV03 300 mg dose group, the LSMEAN [95% confidence interval] of the number of newly reported COVID-19 symptoms after baseline was 2.02 [1.63, 2.42] cases for the placebo and 2.40 [2.01, 2.79] cases for the 300 mg dose group.

New symptoms were reported until the Day 14 evening in the placebo group, while no new symptoms appeared after the Day 14 morning in the CP-COV03 450 mg dose group. The ANCOVA analysis showed a statistically significant difference in the number of newly reported symptoms between the placebo group and the CP-COV03 450 mg dose group at the 5% significance level ($p=0.0486$). Between the placebo group and the CP-COV03 450 mg dose group, the LSMEAN [95% confidence interval] was 2.02 [1.63, 2.40] cases for the placebo and 2.57 [2.18, 2.95] cases for the 450 mg dose group.

The most frequently reported new COVID-19 symptoms after baseline were diarrhea (34 cases), stuffy or runny nose (26 cases), and fever (25 cases) in the placebo group; diarrhea (53 cases), nausea (32 cases), fever (28 cases), and stuffy or runny nose (28 cases) in the CP-COV03 300 mg dose group; and diarrhea (61 cases), fever (27 cases), and headache (25 cases) in the CP-COV03 450 mg dose group.

In PPS, new symptoms were reported until the Day 14 evening in the placebo group, while no new symptoms appeared after the Day 12 morning in the CP-COV03 300 mg dose group. However, the ANCOVA analysis did not show a statistically significant difference in the number of newly reported symptoms between the placebo group and the CP-COV03 300 mg dose group at the 5% significance level. Between the placebo group and the CP-COV03 300 mg dose group, the LSMEAN [95% confidence interval] of the number of newly reported COVID-19 symptoms after baseline was 1.88 [1.47, 2.29] cases for the placebo and 1.69 [1.26, 2.12] cases for the 300 mg dose group.

New symptoms were reported until the Day 14 evening in the placebo group, while no new symptoms appeared after the Day 14 morning in the CP-COV03 450 mg dose group. However, the ANCOVA analysis did not show a statistically significant difference in the number of newly reported symptoms between the placebo group and the CP-COV03 450 mg dose group at the 5% significance level. Between the placebo group and the CP-COV03 450 mg dose group, the LSMEAN [95% confidence interval] was 1.88 [1.46, 2.30] cases for the placebo and 2.21 [1.80, 2.62] cases for the 450 mg dose group.

The most frequently reported new COVID-19 symptoms after baseline were diarrhea (25 cases), fever (20 cases), and stuffy or runny nose (19 cases) in the placebo group; diarrhea (33 cases), nausea (15 cases), and fever (14 cases) in the CP-COV03 300 mg dose group; and diarrhea (48 cases), fever (19 cases), headache (18 cases) and nausea (18 cases) in the CP-COV03 450 mg dose group.

7) Changes in SARS-CoV-2 viral load on Day 2, Day 4, Day 6 and Day 8 compared to Day 0 (baseline, before administration of the investigational product)

Tables 33–34 present the descriptive statistics of the changes in SARS-CoV-2 viral load on Day 2, Day 4, Day 6 and Day 8 compared to Day 0 for the placebo group, CP-COV03 300 mg dose, and CP-COV03 450 mg dose group. ANCOVA analysis of viral load on each day was performed with pre-dose viral load, age and severity as covariates, and the results are presented in Tables 35–36.

Table 33. Secondary outcome 7) descriptive statistics for viral load (ITT)

Category		Placebo N=97*		CP-COV03 300 mg N=99		CP-COV03 450 mg N=96	
		Viral load (copy/ μ L) (Day#)	Changes (copy/ μ L) (Day# - Baseline)	Viral load (copy/ μ L) (Day#)	Changes (copy/ μ L) (Day# - Baseline)	Viral load (copy/ μ L) (Day#)	Changes (copy/ μ L) (Day# - Baseline)
Baseline	N	66	NA	68	NA	64	NA
	Mean \pm SD	255466.78 \pm 380003.16	NA	187929.63 \pm 359589.40	NA	254543.91 \pm 369854.73	NA
	Median [Min, Max]	87751.30 [0.00, 1416730.90]	NA	28551.95 [0.00, 1844137.80]	NA	61727.85 [0.00, 1950068.70]	NA
Day 2	N	66	66	67	67	64	64
	Mean \pm SD	209184.70 \pm 340232.40	-46282.08 \pm 413674.57	127748.23 \pm 215091.35	-62918.92 \pm 399629.87	119432.86 \pm 205559.19	- 135111.05 \pm 407691.30
	Median [Min, Max]	32382.00 [0.00, 1466562.00]	-12041.45 [-1170274.70, 1369531.20]	9863.10 [0.00, 804875.80]	-260.30 [-1834274.70, 720880.40]	30073.25 [0.00, 1016710.20]	-8524.30 [-1906124.90, 852987.00]
Day 4	N	66	66	67	67	64	64
	Mean \pm SD	7082.25 \pm 15756.48	- 248384.53 \pm 379357.51	7988.30 \pm 16908.54	- 182678.85 \pm 362387.47	4060.35 \pm 7902.02	- 250483.57 \pm 370959.21
	Median [Min, Max]	1187.25 [0.00, 99986.50]	-80133.00 [-1411962.40, 38966.60]	547.20[0.00, 88372.80]	-17899.90 [-1844076.00, 21236.30]	807.70[0.00, 51871.50]	-59792.90 [-1945687.10, 18485.90]
Day 6	N	66	66	67	67	64	64
	Mean \pm SD	875.72 \pm 1793.65	- 254591.02 \pm 379687.89	612.55 \pm 1575.38	- 190054.60 \pm 361584.99	488.66 \pm 1209.27	- 254055.25 \pm 370017.17
	Median [Min, Max]	113.45 [0.00, 9744.40]	-86834.50 [-1416723.30, 6447.30]	67.50 [0.00, 9982.80]	-30394.70 [-1844115.00, 4468.70]	38.60[0.00, 6552.10]	-61632.00 [-1950011.00, 3364.10]

Day 8	N	66	66	67	67	64	64
	Mean±SD	31.84±70.83	- 255434.90±380000.69	24.14±57.87	- 190643.01±361569.89	13.46±29.71	- 254530.45±369851.19
	Median [Min, Max]	2.50 [0.00, 340.00]	-87748.80 [-1416688.80, 5.50]	2.50[0.00, 339.20]	-30706.70 [-1844020.90, 0.00]	2.50[0.00, 179.10]	-61709.50 [-1950063.30, 0.00]

* Exclude patients with missing viral load data on Day 0 (major protocol deviation)

Table 34. Secondary outcome 7) descriptive statistics for viral load (PPS)

Category		Placebo N=77		CP-COV03 300 mg N=70		CP-COV03 450 mg N=80	
		Viral load (copy/μL) (Day#)	Changes (copy/μL) (Day# - Baseline)	Viral load (copy/μL) (Day#)	Changes (copy/μL) (Day# - Baseline)	Viral load (copy/μL) (Day#)	Changes (copy/μL) (Day# - Baseline)
Baseline	N	49	NA	48	NA	53	NA
	Mean±SD	254116.23±375927.57	NA	234127.93±415127.39	NA	286037.29±394243.88	NA
	Median [Min, Max]	89891.50 [2.50, 1416730.90]	NA	37667.70 [2.50, 1844137.80]	NA	73453.60[18.70, 1950068.70]	NA
Day 2	N	49	49	48	48	53	53
	Mean±SD	243691.53±377347.33	-10424.70±433403.82	101505.70±191804.50	- 132622.23±427545.74	128115.68±215102.09	- 157921.61±435491.05
	Median [Min, Max]	36153.90 [72.30, 1466562.00]	-12281.10 [-1170274.70, 1369531.20]	10820.90 [2.50, 804875.80]	-4494.40 [-1834274.70, 720880.40]	33397.80 [0.00, 1016710.20]	-4965.10 [-1906124.90, 852987.00]
	N	49	49	48	48	53	53

Category		Placebo N=77		CP-COV03 300 mg N=70		CP-COV03 450 mg N=80	
		Viral load (copy/ μ L) (Day#)	Changes (copy/ μ L) (Day# - Baseline)	Viral load (copy/ μ L) (Day#)	Changes (copy/ μ L) (Day# - Baseline)	Viral load (copy/ μ L) (Day#)	Changes (copy/ μ L) (Day# - Baseline)
Day 4	Mean \pm SD	4642.97 \pm 8647.13	- 249473.26 \pm 374727.68	6845.29 \pm 13517.93	- 227282.64 \pm 416103.12	3927.29 \pm 8299.42	- 282110.00 \pm 395380.78
	Median [Min, Max]	1550.50 [0.00, 49907.10]	-85101.40 [-1411962.40, 8005.50]	463.10 [0.00, 72330.10]	-17973.65 [-1844076.00, 21236.30]	803.30 [0.00, 51871.50]	-63664.70 [-1945687.10, 18485.90]
Day 6	N	49	49	48	48	53	53
	Mean \pm SD	769.42 \pm 1717.89	- 253346.80 \pm 375678.84	449.95 \pm 1071.14	- 233677.98 \pm 415050.06	471.13 \pm 1274.37	- 285566.16 \pm 394404.88
	Median [Min, Max]	111.80 [0.00, 9744.40]	-88076.20 [-1416723.30, 0.00]	59.45 [0.00, 5465.90]	-36883.80 [-1844115.00, 240.80]	23.90 [0.00, 6552.10]	-67834.00 [-1950011.00, 3364.10]
Day 8	N	49	49	48	48	53	53
	Mean \pm SD	35.34 \pm 79.13	- 254080.89 \pm 375921.77	27.04 \pm 65.10	- 234100.89 \pm 415105.17	12.87 \pm 28.18	- 286024.42 \pm 394239.18
	Median [Min, Max]	2.50 [0.00, 340.00]	-89889.00 [-1416688.80, -2.50]	2.50 [0.00, 339.20]	-37606.05 [-1844020.90, -2.50]	2.50 [0.00, 179.10]	-73401.30 [-1950063.30, -16.20]

Authors' point-by-point responses to Reviewer's comments

Table 35. Secondary outcome 7) viral load ANCOVA results (ITT)

Category	Placebo vs CP-COV03 300 mg			Placebo vs CP-COV03 450 mg		
	LS Mean [95% CI]		p-value	LS Mean [95% CI]		p-value
	Placebo	300 mg		Placebo	450 mg	
Day 2	204612.7098 [137739.4192, 271486.0004]	132251.9799 [65882.4268, 198621.5330]	0.1321	214196.3325 [147322.1807, 281070.4843]	114264.6087 [46343.5992, 182185.6182]	0.0410†
Day 4	6952.4609 [2977.2464, 10927.6753]	8116.1490 [4170.8788, 12061.4193]	0.6825	6863.7298 [3793.4780, 9933.9816]	4285.6927 [1167.3788, 7404.0066]	0.2481
Day 6	851.3469 [442.9413, 1259.7526]	636.5627 [231.2335, 1041.8919]	0.4628	857.9280 [480.5215, 1235.3346]	507.0117 [123.6972, 890.3262]	0.2012
Day 8	31.0691 [15.3874, 46.7508]	24.9021 [9.3385, 40.4656]	0.5828	31.8916 [18.4803, 45.3029]	13.4087 [-0.2126, 27.0299]	0.0591

* Unit: copy/ μ L

† Observed difference is statistically significant at the 5% level

Table 36. Secondary outcome 7) viral load ANCOVA results (PPS)

Category	Placebo vs CP-COV03 300 mg			Placebo vs CP-COV03 450 mg		
	LS Mean [95% CI]		p-value	LS Mean [95% CI]		p-value
	Placebo	300 mg		Placebo	450 mg	
Day 2	243605.5148 [160956.3339, 326254.6958]	101593.5057 [18086.7575, 185100.2539]	0.0185†	247027.9904 [162014.0310, 332041.9499]	125031.0334 [43310.3119, 206751.7549]	0.0434†
Day 4	4679.1513 [1437.7266, 7920.5759]	6808.3539 [3533.2962, 10083.4116]	0.3615	4678.3489 [2236.6517, 7120.0462]	3894.5793 [1547.4676, 6241.6910]	0.6482
Day 6	769.0482 [364.1986, 1173.8979]	450.3278 [41.2775, 859.3782]	0.2746	741.8508 [313.9466, 1169.7550]	496.6210 [85.2928, 907.9492]	0.4158
Day 8	35.2943 [14.9439, 55.6448]	27.0871 [6.5254, 47.6487]	0.5747	34.9360 [18.1977, 51.6743]	13.2422 [-2.8477, 29.3321]	0.0676

* Unit: copy/ μ L

† Observed difference is statistically significant at the 5% level

Authors' point-by-point responses to Reviewer's comments

In ITT, the ANCOVA analysis showed that on Day 2 (approximately 16 hours after the first dose), the LSMEAN [95% confidence interval] viral load on Day 2 adjusted on the baseline of the placebo and the CP-COV03 300 mg dose groups were 204612.71 [137739.42, 271486.00] and 132251.98 [65882.43, 198621.53], respectively, indicating that the viral load in the CP-COV03 300 mg dose group was approximately 64.6% of that in the placebo group. However, the p-value for the between-group difference on Day 2 was 0.1321, and there was no statistically significant difference between groups at the 5% significance level on any other day.

On Day 2 (approximately 16 hours after the first dose), the LSMEAN [95% confidence interval] viral load on Day 2 adjusted on the baseline of the placebo and the CP-COV03 450 mg dose groups were 214196.33 [147322.18, 281070.48] and 114264.61 [46343.60, 182185.62], respectively, indicating that the viral load in the CP-COV03 450 mg dose group was approximately 53.3% of that in the placebo group. The p-value for the between-group difference on Day 2 was 0.0410, which is statistically significant at the 5% significance level, while the other days did not show a statistically significant difference between groups at the 5% significance level.

In PPS, the descriptive statistics for change in viral load from Day 2 compared to pre-dosing showed a mean decrease of -10424.70 copy/ μ L for the placebo group and -132622.23 copy/ μ L for the CP-COV03 300 mg dose group, indicating approximately a 12.72-fold decrease. Furthermore, the ANCOVA analysis for the same day (Day 2) also showed statistically significant between-group differences at the 5% significance level (p-value=0.0185). The ANCOVA analysis showed a statistically significant difference between groups on Day 2 (approximately 16 hours after the first dose), with a significance level of 5%. The LSMEAN [95% confidence interval] for viral load on Day 2 was 243605.51 [160956.33, 326254.70] for the placebo group and 101593.51 [18086.76, 185100.25] for the CP-COV03 300 mg dose group, indicating that the viral load in the CP-COV03 300 mg dose group was approximately 41.7% of that in the placebo group.

The descriptive statistics for change in viral load from Day 2 compared to pre-dosing showed a mean decrease of -10424.70 copy/ μ L for the placebo group and -157921.61 copy/ μ L for the CP-COV03 450 mg dose group, indicating approximately a 15.15-fold decrease. Furthermore, the ANCOVA analysis for the same day (Day 2) also showed statistically significant between-group differences at the 5% significance level (p-value=0.0434). The ANCOVA analysis showed a statistically significant difference between groups on Day 2 (approximately 16 hours after the first dose), with a significance level of 5%. The LSMEAN [95% confidence interval] for viral load on Day 2 was 247027.99 [162014.03, 332041.95] for the placebo group and 125031.03 [43310.31, 206751.75] for the CP-COV03 450 mg dose group, indicating that the viral load in the CP-COV03 450 mg dose group was approximately 50.6% of that in the placebo group.

8) The dose and dosing frequency of Acetaminophen, Ibuprofen, and antidiarrheal from Day 1 to Day 28

For the placebo, CP-COV03 300 mg dose, and CP-COV03 450 mg dose group, from Day 1 to Day 28, the dosage and number of administrations for rescue medications including Acetaminophen, Ibuprofen, and antidiarrheals were detailed. Specifically, for Acetaminophen and Ibuprofen (for combination drugs, measure only Acetaminophen or Ibuprofen doses only), the number of participants administered (daily N) and dosages are presented in Tables 37–38. Descriptive statistics for the number of administrations of antidiarrheals are provided in Tables 39–40.

Authors' point-by-point responses to Reviewer's comments

Additionally, an ANCOVA analysis, considering age and severity as covariates, was conducted on the total dosages of Acetaminophen and Ibuprofen administered during the period. Due to insufficient data, the number of administrations for antidiarrheals was not presented through ANCOVA analysis. The ANCOVA analysis results showed that there was no statistical significance between the placebo and the CP-COV03 groups on the total amounts of Acetaminophen and Ibuprofen used until Day 28, at the 5% significance level.

Table 37. Secondary outcome 8) Acetaminophen, Ibuprofen dosages (mg) (ITT)

Acetaminophen, Ibuprofen		Placebo N=98	CP-COV03 300 mg N=99	CP-COV03 450 mg N=96
Day 1	N	19	21	17
	Mean	580.26	647.62	641.18
	SD	134.00	141.84	36.38
	Median	650.00	650.00	650.00
	Min	200.00	350.00	500.00
	Max	650.00	1000.00	650.00
Day 2	N	9	22	8
	Mean	652.78	681.82	650.00
	SD	168.84	102.99	0.00
	Median	650.00	650.00	650.00
	Min	325.00	650.00	650.00
	Max	1000.00	1000.00	650.00
Day 3	N	6	16	6
	Mean	595.83	693.75	650.00
	SD	132.68	119.55	0.00
	Median	650.00	650.00	650.00
	Min	325.00	650.00	650.00
	Max	650.00	1000.00	650.00
Day 4	N	5	3	4
	Mean	725.00	766.67	650.00
	SD	283.95	202.07	0.00
	Median	650.00	650.00	650.00
	Min	325.00	650.00	650.00
	Max	1000.00	1000.00	650.00
Day 5	N	6	2	2

Authors' point-by-point responses to Reviewer's comments

Acetaminophen, Ibuprofen		Placebo N=98	CP-COV03 300 mg N=99	CP-COV03 450 mg N=96
	Mean	654.17	825.00	650.00
	SD	213.55	247.49	0.00
	Median	650.00	825.00	650.00
	Min	325.00	650.00	650.00
	Max	1000.00	1000.00	650.00
Day 6	N	2	0	2
	Mean	662.50	-	650.00
	SD	477.30	-	0.00
	Median	662.50	-	650.00
	Min	325.00	-	650.00
	Max	1000.00	-	650.00
Day 7	N	2	0	1
	Mean	662.50	-	400.00
	SD	477.30	-	.
	Median	662.50	-	400.00
	Min	325.00	-	400.00
	Max	1000.00	-	400.00
Day 8	N	2	1	0
	Mean	662.50	1000.00	-
	SD	477.30	.	-
	Median	662.50	1000.00	-
	Min	325.00	1000.00	-
	Max	1000.00	1000.00	-
Day 9	N	3	0	0
	Mean	641.67	-	-
	SD	339.42	-	-
	Median	600.00	-	-
	Min	325.00	-	-
	Max	1000.00	-	-
Day 10	N	2	1	0
	Mean	662.50	1000.00	-
	SD	477.30	.	-
	Median	662.50	1000.00	-

Authors' point-by-point responses to Reviewer's comments

Acetaminophen, Ibuprofen		Placebo N=98	CP-COV03 300 mg N=99	CP-COV03 450 mg N=96
	Min	325.00	1000.00	-
	Max	1000.00	1000.00	-
Day 11	N	2	0	1
	Mean	662.50	-	300.00
	SD	477.30	-	.
	Median	662.50	-	300.00
	Min	325.00	-	300.00
	Max	1000.00	-	300.00
Day 12	N	1	0	0
	Mean	500.00	-	-
	SD	.	-	-
	Median	500.00	-	-
	Min	500.00	-	-
	Max	500.00	-	-
Day 13	N	1	0	1
	Mean	1000.00	-	300.00
	SD	.	-	.
	Median	1000.00	-	300.00
	Min	1000.00	-	300.00
	Max	1000.00	-	300.00
Day 14	N	0	1	0
	Mean	-	300.00	-
	SD	-	.	-
	Median	-	300.00	-
	Min	-	300.00	-
	Max	-	300.00	-
Day 15	N	0	0	0
	Mean	-	-	-
	SD	-	-	-
	Median	-	-	-
	Min	-	-	-
	Max	-	-	-
Day 16	N	1	0	0

Authors' point-by-point responses to Reviewer's comments

Acetaminophen, Ibuprofen		Placebo N=98	CP-COV03 300 mg N=99	CP-COV03 450 mg N=96
	Mean	500.00	-	-
	SD	.	-	-
	Median	500.00	-	-
	Min	500.00	-	-
	Max	500.00	-	-
Day 17	N	0	0	0
	Mean	-	-	-
	SD	-	-	-
	Median	-	-	-
	Min	-	-	-
	Max	-	-	-
Day 18	N	0	0	0
	Mean	-	-	-
	SD	-	-	-
	Median	-	-	-
	Min	-	-	-
	Max	-	-	-
Day 19	N	0	0	0
	Mean	-	-	-
	SD	-	-	-
	Median	-	-	-
	Min	-	-	-
	Max	-	-	-
Day 20	N	0	0	0
	Mean	-	-	-
	SD	-	-	-
	Median	-	-	-
	Min	-	-	-
	Max	-	-	-
Day 21	N	0	0	0
	Mean	-	-	-
	SD	-	-	-
	Median	-	-	-

Authors' point-by-point responses to Reviewer's comments

Acetaminophen, Ibuprofen		Placebo N=98	CP-COV03 300 mg N=99	CP-COV03 450 mg N=96
	Min	-	-	-
	Max	-	-	-
Day 22	N	0	1	0
	Mean	-	1000.00	-
	SD	-	.	-
	Median	-	1000.00	-
	Min	-	1000.00	-
	Max	-	1000.00	-
Day 23	N	0	0	0
	Mean	-	-	-
	SD	-	-	-
	Median	-	-	-
	Min	-	-	-
	Max	-	-	-
Day 24	N	0	0	0
	Mean	-	-	-
	SD	-	-	-
	Median	-	-	-
	Min	-	-	-
	Max	-	-	-
Day 25	N	0	1	0
	Mean	-	1000.00	-
	SD	-	.	-
	Median	-	1000.00	-
	Min	-	1000.00	-
	Max	-	1000.00	-
Day 26	N	0	1	0
	Mean	-	325.00	-
	SD	-	.	-
	Median	-	325.00	-
	Min	-	325.00	-
	Max	-	325.00	-
Day 27	N	0	0	0

Authors' point-by-point responses to Reviewer's comments

Acetaminophen, Ibuprofen		Placebo N=98	CP-COV03 300 mg N=99	CP-COV03 450 mg N=96
	Mean	-	-	-
	SD	-	-	-
	Median	-	-	-
	Min	-	-	-
	Max	-	-	-
Day 28	N	0	0	0
	Mean	-	-	-
	SD	-	-	-
	Median	-	-	-
	Min	-	-	-
	Max	-	-	-

Table 38. Secondary outcome 8) Acetaminophen, Ibuprofen dosages (mg) (PPS)

Acetaminophen, Ibuprofen		Placebo N=77	CP-COV03 300 mg N=70	CP-COV03 450 mg N=80
Day 1	N	13	16	13
	Mean	573.08	646.88	638.46
	SD	139.37	163.78	41.60
	Median	650.00	650.00	650.00
	Min	200.00	350.00	500.00
	Max	650.00	1000.00	650.00
Day 2	N	3	12	6
	Mean	766.67	650.00	650.00
	SD	202.07	0.00	0.00
	Median	650.00	650.00	650.00
	Min	650.00	650.00	650.00
	Max	1000.00	650.00	650.00
Day 3	N	2	6	4
	Mean	650.00	650.00	650.00
	SD	0.00	0.00	0.00
	Median	650.00	650.00	650.00
	Min	650.00	650.00	650.00
	Max	650.00	650.00	650.00

Authors' point-by-point responses to Reviewer's comments

Acetaminophen, Ibuprofen		Placebo N=77	CP-COV03 300 mg N=70	CP-COV03 450 mg N=80
Day 4	N	0	2	2
	Mean	-	825.00	650.00
	SD	-	247.49	0.00
	Median	-	825.00	650.00
	Min	-	650.00	650.00
	Max	-	1000.00	650.00
Day 5	N	1	1	2
	Mean	650.00	1000.00	650.00
	SD	.	.	0.00
	Median	650.00	1000.00	650.00
	Min	650.00	1000.00	650.00
	Max	650.00	1000.00	650.00
Day 6	N	0	0	2
	Mean	-	-	650.00
	SD	-	-	0.00
	Median	-	-	650.00
	Min	-	-	650.00
	Max	-	-	650.00
Day 7	N	0	0	0
	Mean	-	-	-
	SD	-	-	-
	Median	-	-	-
	Min	-	-	-
	Max	-	-	-
Day 8	N	0	0	0
	Mean	-	-	-
	SD	-	-	-
	Median	-	-	-
	Min	-	-	-
	Max	-	-	-
Day 9	N	0	0	0
	Mean	-	-	-
	SD	-	-	-

Authors' point-by-point responses to Reviewer's comments

Acetaminophen, Ibuprofen		Placebo N=77	CP-COV03 300 mg N=70	CP-COV03 450 mg N=80
	Median	-	-	-
	Min	-	-	-
	Max	-	-	-
Day 10	N	0	0	0
	Mean	-	-	-
	SD	-	-	-
	Median	-	-	-
	Min	-	-	-
	Max	-	-	-
Day 11	N	0	0	1
	Mean	-	-	300.00
	SD	-	-	.
	Median	-	-	300.00
	Min	-	-	300.00
	Max	-	-	300.00
Day 12	N	0	0	0
	Mean	-	-	-
	SD	-	-	-
	Median	-	-	-
	Min	-	-	-
	Max	-	-	-
Day 13	N	0	0	1
	Mean	-	-	300.00
	SD	-	-	.
	Median	-	-	300.00
	Min	-	-	300.00
	Max	-	-	300.00
Day 14	N	0	1	0
	Mean	-	300.00	-
	SD	-	.	-
	Median	-	300.00	-
	Min	-	300.00	-
	Max	-	300.00	-

Authors' point-by-point responses to Reviewer's comments

Acetaminophen, Ibuprofen		Placebo N=77	CP-COV03 300 mg N=70	CP-COV03 450 mg N=80
Day 15	N	0	0	0
	Mean	-	-	-
	SD	-	-	-
	Median	-	-	-
	Min	-	-	-
	Max	-	-	-
Day 16	N	1	0	0
	Mean	500.00	-	-
	SD	.	-	-
	Median	500.00	-	-
	Min	500.00	-	-
	Max	500.00	-	-
Day 17	N	0	0	0
	Mean	-	-	-
	SD	-	-	-
	Median	-	-	-
	Min	-	-	-
	Max	-	-	-
Day 18	N	0	0	0
	Mean	-	-	-
	SD	-	-	-
	Median	-	-	-
	Min	-	-	-
	Max	-	-	-
Day 19	N	0	0	0
	Mean	-	-	-
	SD	-	-	-
	Median	-	-	-
	Min	-	-	-
	Max	-	-	-
Day 20	N	0	0	0
	Mean	-	-	-
	SD	-	-	-

Authors' point-by-point responses to Reviewer's comments

Acetaminophen, Ibuprofen		Placebo N=77	CP-COV03 300 mg N=70	CP-COV03 450 mg N=80
	Median	-	-	-
	Min	-	-	-
	Max	-	-	-
Day 21	N	0	0	0
	Mean	-	-	-
	SD	-	-	-
	Median	-	-	-
	Min	-	-	-
	Max	-	-	-
Day 22	N	0	0	0
	Mean	-	-	-
	SD	-	-	-
	Median	-	-	-
	Min	-	-	-
	Max	-	-	-
Day 23	N	0	0	0
	Mean	-	-	-
	SD	-	-	-
	Median	-	-	-
	Min	-	-	-
	Max	-	-	-
Day 24	N	0	0	0
	Mean	-	-	-
	SD	-	-	-
	Median	-	-	-
	Min	-	-	-
	Max	-	-	-
Day 25	N	0	0	0
	Mean	-	-	-
	SD	-	-	-
	Median	-	-	-
	Min	-	-	-
	Max	-	-	-

Authors' point-by-point responses to Reviewer's comments

Acetaminophen, Ibuprofen		Placebo N=77	CP-COV03 300 mg N=70	CP-COV03 450 mg N=80
Day 26	N	0	0	0
	Mean	-	-	-
	SD	-	-	-
	Median	-	-	-
	Min	-	-	-
	Max	-	-	-
Day 27	N	0	0	0
	Mean	-	-	-
	SD	-	-	-
	Median	-	-	-
	Min	-	-	-
	Max	-	-	-
Day 28	N	0	0	0
	Mean	-	-	-
	SD	-	-	-
	Median	-	-	-
	Min	-	-	-
	Max	-	-	-

Table 39. Secondary outcome 8) the number of administrations of antidiarrheals (ITT)

Antidiarrheals		Placebo N=98	CP-COV03 300 mg N=99	CP-COV03 450 mg N=96
Day 2	N	0	0	1
	Mean	-	-	1.00
	SD	-	-	0.00
	Median	-	-	1.00
	Min	-	-	1.00
	Max	-	-	1.00
Day 3	N	0	1	0
	Mean	-	1.00	-
	SD	-	0.00	-
	Median	-	1.00	-
	Min	-	1.00	-

Authors' point-by-point responses to Reviewer's comments

	Max	-	1.00	-
Day 4	N	0	3	0
	Mean	-	1.00	-
	SD	-	0.00	-
	Median	-	1.00	-
	Min	-	1.00	-
	Max	-	1.00	-
Day 5	N	0	3	1
	Mean	-	1.00	1.00
	SD	-	0.00	0.00
	Median	-	1.00	1.00
	Min	-	1.00	1.00
	Max	-	1.00	1.00

Table 40. Secondary outcome 8) the number of administrations of antidiarrheals (PPS)

Antidiarrheals		Placebo N=77	CP-COV03 300 mg N=70	CP-COV03 450 mg N=80
Day 1-3	N	0	0	0
	Mean	-	-	-
	SD	-	-	-
	Median	-	-	-
	Min	-	-	-
	Max	-	-	-
Day 4	N	0	2	0
	Mean	-	1.00	-
	SD	-	0.00	-
	Median	-	1.00	-
	Min	-	1.00	-
	Max	-	1.00	-
Day 5	N	0	3	0
	Mean	-	1.00	-
	SD	-	0.00	-
	Median	-	1.00	-
	Min	-	1.00	-
	Max	-	1.00	-

Authors' point-by-point responses to Reviewer's comments

Day 6–28	N	0	3	0
	Mean	-	1.00	-
	SD	-	0.00	-
	Median	-	1.00	-
	Min	-	1.00	-
	Max	-	1.00	-

9) Proportion of participants with severe COVID-19 progression from Day 1 to Day 28

Over the study period from Day 1 through Day 28, there were no participants with severe COVID-19 progression.

Table 41. Secondary outcome 9) participants progressed to severe COVID-19 (ITT)

Participants who progressed to severe COVID-19 or not		Placebo N=98 n (%)	CP-COV03 300 mg N=99 n (%)	CP-COV03 450 mg N=96 n (%)
Day 1- Day 28	Total	98(100.0)	99(100.0)	96(100.0)
	Progressed to severity	0(0.0)	0(0.0)	0(0.0)
	Not progressed to severity	98(100.0)	99(100.0)	96(100.0)

10) Pharmacokinetic characteristics of niclosamide and correlation between pharmacokinetic parameters and viral load

Table 42 presents the descriptive statistics of the pharmacokinetic variables. In the CP-COV03 450 mg dose group, participant R03031 was dropped out before dosing, and for participant R03060, pharmacokinetic blood sampling was not conducted due to a consent error related to pharmacokinetic blood collection.

The geometric mean of C_{max} for the CP-COV03 300 mg dose group was 285.25 ng/mL (minimum 139.82, maximum 936.53), while for the CP-COV03 450 mg dose group, it was 389.90 ng/mL (minimum 129.39, maximum 1061.79).

Table 42. Secondary outcome 10) descriptive statistics of pharmacokinetic variables for niclosamide (ITT)

Treatment	Parameters	N	Arithmetic Mean	SD	Geometric Mean	Median	Min	Max
CP-COV03 300 mg (N=20)	C _{max} (ng/mL)	20	317.78	186.42	285.25	246.39	139.82	936.53
	AUC _t (ng·h/mL)	20	11046.21	3451.07	10562.09	9754.47	6479.00	17836.36
	C _{max}	18	460.44	275.45	389.90	391.83	129.39	1061.79

Authors' point-by-point responses to Reviewer's comments

CP-COV03	(ng/mL)							
450 mg	AUCt	18	13969.54	5880.64	12876.29	12912.31	6809.88	24605.15
(N=18)	(ng·h/mL)							

Table 43. Mean and confidence interval for niclosamide AUC and Cmax (ITT)

	Geometric Mean	LSMean Ratio (T/R)		ANOVA-CV	
		Point Estimate	90% CI		
AUCt	CP-COV03 300 mg	10562.09	1.2191	[1.0000,1.4862]	36.12
	CP-COV03 450 mg	12876.29			
Cmax	CP-COV03 300 mg	285.25	1.3669	[1.0277,1.8180]	51.99
	CP-COV03 450 mg	389.90			

- Correlation between niclosamide and viral load

To determine the correlation between niclosamide AUCt and viral load (RdRp gene) measured by qPCR, results using the CORR procedure (correlation analysis) of the SAS® program are presented in Table 44, and scatter plots are shown in Figure 1 (CP-COV03 300 mg dose) and Figure 2 (CP-COV03 450 mg dose). For the correlation analysis, niclosamide AUCt was calculated for AUCday0, AUCDay2, AUCDay4, and AUCDay6, and correlation analysis was conducted with the viral load figures corresponding to each date.

There was a significant negative correlation between niclosamide AUCt and viral load in both the CP-COV03 300 mg dose and 450 mg dose groups. Each correlation coefficient (p-value) was -0.3296 (p=0.0101) for the CP-COV03 300 mg dose group and -0.4818 (p=0.0009) for the CP-COV03 450 mg dose group, showing that the negative correlation was more significant in higher doses. In other words, it was found that as the administered dose increased, the viral load decreased more significantly.

Table 44. Secondary outcome 10) correlation between niclosamide and viral load (ITT)

Niclosamide AUCt vs viral load			Niclosamide AUCt	viral load
CP-COV03 300 mg	Pearson correlation coefficient	Niclosamide AUCt	1.0000	-0.3296
		viral load	-0.3296	1.0000
	Significance probability	Niclosamide AUCt	-	0.0101
		viral load	0.0101	-

Authors' point-by-point responses to Reviewer's comments

Niclosamide AUcT vs viral load		Niclosamide AUcT	viral load	
CP-COV03 450 mg	Pearson correlation coefficient	Niclosamide AUcT	1.0000	
		viral load	-0.4818	
	Significance probability	Niclosamide AUcT		0.0009
		viral load	0.0009	

T1 AUcT- Viral_Load

<Figure 1. Scatter plot of niclosamide AUcT and viral load (qPCR-RdRp gene) for the CP-COV03 300 mg dose group>

Authors' point-by-point responses to Reviewer's comments

<Figure 2. Scatter plot of niclosamide AUCt and viral load (qPCR-RdRp gene) for the CP-COV03 450 mg dose group>

Annexure-II

28-Days full data related to recovery

Time (days) for sustained improvement of all symptoms (ITT, n=293)	Placebo (n=98)	CP-COV03 300 mg (n=96)	CP-COV03 450 mg (n=99)
Median to Improvement [95% CI]	13.50 [11.00, 17.00]	10.50 [9.00, 18.00]	14.50 [12.50, 18.50]
Difference vs. placebo	-	3.00	-1.00
p-value	-	0.9305	0.2949

Time (days) for sustained improvement of all symptoms (PPS, n=227)	Placebo (n=77)	CP-COV03 300 mg (n=70)	CP-COV03 450 mg (n=80)
Median to Improvement [95% CI]	15.00 [11.50, 19.00]	9.25 [7.50, 12.50]	13.25 [11.50, 18.50]
Difference vs. placebo	-	5.75	1.75
p-value	-	0.0275	0.8310

Responses to Reviewer #4's Comments

CRITICAL Comments

1. CRITICAL ----- Line 267 ----- “A possible explanation for this is the presence of magnesium oxide (MgO) in the 450 mg formulation, which may have caused gastrointestinal symptoms, thereby confounding participant-reported symptom evaluations.” ----- As written, this seems to imply that the placebo did not match well at least one of the of the active groups. If true, this would be critical issue.

Author’s Response:

We appreciate the reviewer’s observation regarding the sentence in question. The comment, “this seems to imply that the placebo did not match well at least one of the active groups,” appears to reflect an interpretation that magnesium oxide (MgO) was not included in the low dose group of CP-COV03.

However, as indicated in the immediately following sentence (lines 195–197), MgO was indeed included in the low dose group. The relevant sentence is as follows:

“The daily MgO intake in the high dose was 945 mg, higher than the 630 mg in the low dose and exceeding the recommended daily intake of 800 mg.” (lines 195–197)

Nonetheless, to avoid any possible misunderstanding and to improve clarity in accordance with the reviewer’s suggestion, we have revised the sentence as follows:

A possible explanation for this is the higher amount of magnesium oxide (MgO) in the investigational product administered to the high dose group compared to the low dose group. This higher MgO content may have caused gastrointestinal symptoms, thereby confounding participant-reported symptom evaluations. (line 191-195)

We hope this revision adequately addresses the reviewer’s concern and improves the overall clarity of the manuscript.

2. CRITICAL ----- Line 282 ----- “The higher proportion of certain COVID-19 symptoms at baseline in the 450 mg treatment group may have contributed to the lack of statistically significant improvement in symptoms observed in this group” ----- This reads like a failure of the randomisation system to ensure proper balance or the authors to consider all the pertinent issues ahead of randomisation. Or bad luck.

Author’s Response:

Thank you for your comment. The question you raised is the same as the one previously addressed during the first round of revision by another reviewer. We would like to inform you that we have already incorporated a response to that question in the manuscript as follows:

Comment from the reviewer-3: “Study groups were not well balanced at baseline for all characteristics of note. For example those receiving 450 milligrams had more frequent nausea and diarrhea.....”

Our previous response to reviewer-3:

In designing the study, we stratified by severity (mild/moderate) and age, which are factors that could influence the outcomes of the COVID-19 clinical trial, to ensure balance between study groups. Given that there are 12 target symptoms of COVID-19, it is not feasible to achieve a perfect balance across all symptoms between the study groups. However, the analysis showed that, aside

from nausea, there were no statistically significant differences ($P < 0.05$) in the frequency of symptoms between groups. In the case of nausea, it occurred significantly more frequently in the 450 mg dose group, which may have had a negative impact on symptom improvement in this group.

This point has been already addressed in the discussion section of the present manuscript as follows;

“The higher proportion of certain COVID-19 symptoms at baseline in the 450 mg treatment group may have contributed to the lack of statistically significant improvement in symptoms observed in this group. Factors that could influence clinical outcomes, such as age and disease severity, were stratified to maintain balance between groups, but it was not feasible to stratify for all 12 symptoms to ensure group balance. As a result, while there were no statistically significant differences in the number of patients showing symptoms like fever, cough, sore throat, headache, muscle ache, chill, stuffy or runny nose, fatigue, difficulty of breathing, vomiting or diarrhea at baseline between groups, significantly more patients with nausea were included in the 450 mg group (Chi-squared test, placebo vs 300mg, p-value = 0.0092)”. (lines 209-218)

MAJOR Comments

3. MAJOR ----- Line 274 ----- “In an ad-hoc analysis excluding the influence of MgO, th” --- -- How does this work? Did I miss this in the Methods?

Author’s Response:

In the Methods section of the manuscript (lines 491–493), we have already described the ad-hoc analysis method.

While this was previously described in the methods section, we have now further clarified and expanded the description in the revised manuscript to avoid any possible ambiguity and to support Reviewer-4’s understanding.

Accordingly, the method section was modified as follows (lines 424-428):

This censoring method, together with the Cox proportional hazards regression model, was also employed in an ad-hoc analysis designed to assess representative COVID-19 symptoms—such as fever, headache, and sore throat—while specifically excluding the gastrointestinal symptoms due to the high dose of MgO.

4. MAJOR ----- Throughout ----- n/a ----- The version I received is a mess 9before. Did I miss something?

Author’s Response:

Over the course of several rounds of revision, we were requested to provide a substantial amount of supporting data, which may have caused some confusion. To minimize this, we have consolidated Annexure-1 and Annexure-2 as just “Annexure” and updated the labeling accordingly. We hope this resolves the issue and helps clarify your concerns.

5. MAJOR ----- Throughout ----- e.g. “CP-COV03 300 mg dose, 450 mg dose, and placebo groups” ----- The groups are misnamed. The Placebo group has placebo every time but one of the other groups has placebo for much of their tablets. This should be clearer. Or call it “No CP-COV3” and don’t reference the blinding every moment. Or Higher, Lower and None.

Author’s Response:

We acknowledge the inconsistency in naming the groups and have revised the terminology throughout the manuscript to ensure clarity. The placebo group is now consistently referred to as the “placebo” group, while the treatment groups are labeled based on dosages (low dose and high dose).

CLARITY Comments

6. CLARITY ----- Figures 2 and 3 ----- n/a ----- These could be better presented

Author's Response:

We have revised Figures 2 and 3 to improve clarity, including clearer labeling and adjusted formatting to enhance readability.

Fig. 2. Viral load and pharmacokinetic analyses. (A) The adjusted mean change in viral load from baseline of Severe Acute Respiratory Syndrome Coronavirus 2 (SARS-CoV-2) monitored starting from Day 0. Any data point that is more than 3 times the IQR above the third quartile or below the first quartile is an outlier (n = 77 for placebo, n = 70 for low dose, n = 80 for high dose). Error bars represent standard error (S.E.). (B) Pharmacokinetic profiles of niclosamide from clinical trial (n = 20 for placebo, n = 20 for low dose, n = 18 for high dose). Error bars represent standard error (S.E.).

Fig. 3. Correlation between CP-COV03 pharmacokinetic parameters and the viral load of Severe Acute Respiratory Syndrome Coronavirus 2 (SARS-CoV-2). (A) low dose group; (B) High dose group (n = 15 for low dose, n = 11 for High dose groups).

7. CLARITY ----- Line 129 ----- “the primary outcome of the study was to assess the efficacy of CP-COV03 compared to placebo, measured by the median number of days” ----- No mention of median in the Methods? And is quarter and half days in the summary methods misleading when no one is able to have a half-day or quarter-day?

Author’s Response:

We appreciate this observation. The Methods section has been updated as follows (lines 363-366) The time to sustained improvement (in days) was reported as a median value, with half-day intervals recorded based on symptom assessments conducted every morning and evening.

8. CLARITY ----- Line 368 ----- “patients who did not require hospitalization, but they were hospitalized for close observation” ----- So, they did require hospitalisation?

Author’s Response:

The phrase has been revised for clarity. It now explicitly states that certain patients were admitted for observation but did not require hospitalization based on clinical severity criteria as follows (lines 292-299)

“The enrolled participants were mild to moderate COVID-19 patients who did not require hospitalization, but they were hospitalized based on clinical severity criteria for close observation and sample collection by trained nurses for the pharmacokinetics study and viral load measurements and were treated with the study drugs from Day 1 to Day 6 (Table S5). Also, during the study period, individuals infected with COVID-19 in Korea were required to quarantine at home. To minimize unnecessary outings and movement in accordance with this guideline, the enrolled participants were hospitalized”

9. CLARITY ----- Line 371 ----- n/a ----- Starts talking about D1 without defining it. From disease onset, from symptoms, from hospitalisation, from screening, from randomisation: which?

Author’s Response:

Day 0 is the day of randomization. Day 1 is when the IP is first treated. Screening, Baseline, and Day 1 can be performed on the same day (See supplementary file “KEY MILESTONES”).

10. CLARITY ----- Line 371 ----- “The participants took 9 capsules three times a day from Day 1 evening to Day 6 noon, for a total of 5 days” ----- Did they take or were they allocated to take? It’s the latter, isn’t it?

Author’s Response:

We have revised the wording to clarify as follows (lines 299-301)

“The participants were allocated take 9 capsules three times a day from Day 1 evening to Day 6 noon, for a total of 5 days.”

11. CLARITY ----- Line 433 ----- “symptoms to improve and maintain for more than 48 hours until Day 14” ----- I find this difficult to whether it’s time to the start of that 48 hours or the end of the 48 hours for the measurement. And what if symptoms improve on Day 13: does measurement continue? How does this work for censoring when there’s no 48 hours to follow?

Author’s Response:

As shown in Figure S1, the assessment is based on the completion of a 48-hour sustained symptom improvement period. If this 48-hour window cannot be completed by Day 14, the case is considered censored. For example, if symptom improvement begins on the evening of Day 13 and is maintained through to the evening of Day 14, the full 48-hour period is not met within the study window, and the case is therefore treated as censored.

Figure S1. Example of symptom improvement date evaluation. A) On Day 4, the symptom score decreased from 1 to 0 and was sustained for 48 hours, indicating sustained symptom improvement. The Day 6 was recorded as the symptom improvement date. B) On Day 3, the symptom score decreased from 1 to 0 and was sustained for 48 hours, indicating sustained symptom improvement; however, symptom recurrence was observed on Day 7. The last date the recurred symptom improved was Day 7.5 and was sustained for 48 hours, resulting in Day 9.5 being recorded as the symptom improvement date.

12. CLARITY ----- Line 454 ----- n/a ----- Why didn’t bloods continue to Day 14?

Author’s Response:

Due to ethical considerations, blood sampling was minimized. The purpose of blood collection was to assess the steady-state levels of the drug during the dosing period; therefore, sampling was conducted only during the period of drug administration.

Also, considering the half-life of niclosamide, it is expected that there will be no detectable level of niclosamide in the blood by Day 14. Hence, obtaining a blood sample on that day would be of limited value, as the drug concentration data would not reflect a relevant pharmacokinetic profile.

13. CLARITY ----- Line 472 ----- “The statistical analysis plan (version 2.0) was approved” ----- Approved by who?

Author’s Response:

The statistical analysis plan (version 2.0) was approved by sponsor (Hyundai Bioscience, Co. Ltd, Seoul, Republic of Korea) (lines 405-406)

14. CLARITY ----- Line 478 ----- “, except for those with a protocol violation that could affect the assessment of antiviral activity.” ----- Expand what is meant by protocol violation. Not just eligibility? Could the violation be more common on arm than that others?

Author’s Response:

The specific cases of major protocol violations are already detailed in **Table S2** in the original file as attached below. These violations did not occur more frequently in any particular treatment group, and there were no statistically significant differences in the number of violations between groups.

Table S2. Breakdowns of protocol deviations leading to exclusion from the PPS

	placebo n (%)	300mg TID n (%)	450mg TID n (%)	Total n (%)
Randomized	100 (100.0)	100 (100.0)	100 (100.0)	100 (100.0)
Completed the study	98 (98.0)	98(98.0)	95(95.0)	291 (97.0)
Dropouts	2 (2.0)	2 (2.0)	5 (5.0)	9 (3.0)
Withdrawal of consent	2 (2.0)	1 (1.0)	4(4.0)	7 (2.3)
Assessment by the principal investigator	0 (0.0)	1 (1.0)	1 (1.0)	2 (0.7)
Dropouts before treatment	2 (2.0)	1 (1.0)	4 (4.0)	7 (2.3)
Treatment within symptom onset > 3 days	5 (5.0)	9 (9.0)	7 (7.0)	21 (7.0)
Treatment within symptom onset ≤ Day3	93 (93.0)	90 (90.0)	89 (89.0)	272 (93.7)
PPS	77 (77.0)	70 (70.0)	80 (80.0)	227 (75.7)
PPS exclusion*	23 (23.0)	30 (30.0)	20 (20.0)	73 (24.3)
Major protocol deviations	14 (14.0)	19 (19.0)	10 (10.0)	43 (14.3)
Concomitant drugs affecting COVID-19 symptoms	9 (9.0)	14 (14.0)	5 (5.0)	28 (9.3)
Dropouts	2 (2.0)	2 (2.0)	5 (5.0)	9 (3.0)
PCR missing or <0 in baseline	1 (1.0)	2 (2.0)	2 (2.0)	5 (1.7)

	placebo n (%)	300mg TID n (%)	450mg TID n (%)	Total n (%)
Major protocol deviations*	14 (14.0)	19 (19.0)	10 (10.0)	43 (14.3)
Viral load test omission	3 (3.0)	1 (1.0)	0 (0.0)	4 (1.3)
Randomized after screening test omission	1 (1.0)	0 (0.0)	1 (1.0)	2 (0.7)
Randomization error – PK group allocation	0 (0.0)	0 (0.0)	1 (1.0)	1 (0.3)
Randomization error – Severity allocation	9 (9.0)	9 (9.0)	3 (3.0)	21 (7.0)
Prohibited concomitant medication	0 (0.0)	1 (1.0)	3 (3.0)	4 (1.3)
Re-consent with updated consent form not obtained	3 (3.0)	7 (7.0)	2 (2.0)	12 (4.0)
IP treatment error	1 (1.0)	1 (1.0)	0 (0.0)	2 (0.7)
Consent form error	0 (0.0)	1 (1.0)	0 (0.0)	1 (0.3)

*There is overlap in the number of participants across the sub-categories

15. CLARITY ----- Line 481 ----- “This clinical trial was designed by referencing the clinical trial protocols of Paxlovid® and ensitrelvir utilizing preliminary information such as the drug administration days, efficacy analysis group, and end points” ----- Give trial registration number

Author’s Response:

The information has been provided in the revised manuscript as follows; (lines 346-348)

This clinical trial was designed by referencing the clinical trial protocols of Paxlovid® (NCT04960202) and ensitrelvir (jRCT2031210350)

16. CLARITY ----- Line 483 ----- n/a ----- What’s the time unit here? Days would be imprecise, perhaps, when only going to 14 days whereas hours would be spurious precision and open to possible measurement bias.

Author’s Response:

Thank you for this thoughtful observation. We agree that precision in time measurement is important. In our study, time to symptom improvement was recorded in whole days, as finer resolution (such as in hours) was not feasible due to the practical constraints of patient monitoring and data collection in a real-world clinical setting. While this may introduce some imprecision, we believe the use of whole days strikes a reasonable balance between clinical relevance and feasibility, and helps minimize potential measurement bias associated with subjective symptom reporting.

17. CLARITY ----- Table 2 ----- n/a ----- Can it be a TEAE if there was no treatment? (Placebo) Does these also include problems from Covid-19?

Table 2 – Treatment-Emergent Adverse Events (TEAEs) in the Placebo Group

Author’s Response:

We confirm that TEAEs in the placebo group include adverse events related to COVID-19 symptoms. This has been explicitly mentioned in the table footnote for clarity.

Table 2. Summary of adverse events (Safety analysis set)

	Placebo (n=98)	Low dose (n=99)	High dose (n=96)
Patients with any TEAE, n (%)	26 (26.5)	21 (21.2)	33 (34.4)
Patients with any serious TEAE or death, n	0	0	0
Patients with TEAEs leading to treatment discontinuation, n	0	0	0
Patients with sequelae of TEAE, n	0	0	0
Cases of any TEAE, n	37	32	47
TEAEs occurring in $\geq 2\%$ in either group			
Cardiomegaly, n	5	2	7
Abdominal pain upper, n	0	0	2
Dyspepsia, n	0	2	0
Pneumonia, n	8	9	7
Blood glucose increased, n	5	1	3
White blood cell count decreased, n	2	1	1
Eosinophil count increased, n	0	0	2
Lipase increased, n	1	2	2
Gamma-glutamyl transferase increased, n	2	0	1
Ageusia, n	0	1	2
Dysmenorrhea, n	0	0	2
Urticaria, n	1	1	3
Cases of any treatment-related AE, n	19	12	22
Treatment-related AEs occurring in $\geq 2\%$ in either group			
Blood glucose increased, n	5	1	3
White blood cell count decreased, n	2	1	1
Eosinophil count increased, n	0	0	2
Lipase increased, n	1	2	2
Urticaria, n	1	1	3

AE: adverse event, TEAE: Treatment emergent adverse events

18. CLARITY ----- Throughout ----- n/a ----- What was the role of Rescue medication on outcome measure recording? Was this a protocol violation?

Author's Response:

The role of rescue medication is given as below in the revised manuscript. Also, administering rescue medication was permitted not deemed a protocol violation.

“Rescue medication was administered for ethical reasons to protect clinical trial participants. The number of administrations was recorded, and it was shown that there were no differences in the usage of rescue medication between treatment groups”

(lines 133-135)

NOTE Comments

19. NOTE ----- e.g. Lines 175 & 235 ----- n/a ----- Bits of Results belong in Discussion and the Discussion has quite a lot of repetition of the Results

Author's Response:

We have revised the manuscript to ensure that all results are presented in the Results section, while the Discussion focuses on interpretation and broader implications.

20. NOTE ----- Line 131 ----- “the median number of days required for the improvement of targeted COVID-19 symptoms was 9.0 days (95% CI, 7.00-10.00), 12.25 days (95% CI, 9.50-NR), and 13.0 days (95% CI, 10.50-NR) in the 300 mg dose, 450 mg dose, and placebo groups, respectively.” ----- The lower dose is doing better. Odd?

Author's Response:

We appreciate the reviewer's observation regarding the seemingly paradoxical result, where the 300 mg dose group showed faster symptom improvement compared to the 450 mg group. While this may appear counterintuitive at first glance, it aligns with the pharmacokinetic and pharmacodynamic characteristics of CP-COV03 that we've described in the revised manuscript. Although the 450 mg dose yielded a higher total drug exposure (AUC_t of 12,876.29 ng·h/mL) than the 300 mg dose (AUC_t of 10,562.09 ng·h/mL), both doses achieved steady-state plasma concentrations that were pharmacologically relevant. This is important because niclosamide's antiviral mechanism — primarily through the induction of autophagy — is known to be time-dependent rather than concentration-dependent. In other words, once a threshold concentration sufficient to trigger autophagy is reached, increasing the dose further does not necessarily enhance efficacy.

In this case, the 300 mg dose likely achieved the pharmacodynamic ceiling needed to induce autophagy and exert antiviral effects. The higher 450 mg dose may have offered no additional benefit in terms of antiviral activity and, in some cases, could have introduced a higher pharmacologic burden or systemic stress, which might have slightly delayed clinical improvement. This is reflected in the similar viral load reduction between the two dose groups and the faster resolution of symptoms in the 300 mg group.

These findings support the idea that CP-COV03 efficacy is not strictly dose-proportional and underscore the importance of identifying the optimal therapeutic window rather than assuming higher doses will yield better outcomes. We have added a clearer explanation of this point in the discussion section to better contextualize these findings.

Please refer to the following lines in revised manuscript (lines 150-156)

These findings suggest that CP-COV03's antiviral efficacy is time-dependent (sustained exposure over time; Figure 2B) rather than concentration-dependent³⁴⁻³⁶. This is attributed to the mechanism of action of niclosamide, the active ingredient in CP-COV03, which induces autophagy, resulting in antiviral effects. The plasma concentration necessary to sustain autophagy induction appears to have been achieved with the low dose indicating that it reaches a pharmacodynamic ceiling.

21. (Refer Page 9, lines 207-217) NOTE ----- Line 131 ----- “the median number of days required for the improvement of targeted COVID-19 symptoms was 9.0 days (95% CI, 7.00-10.00), 12.25 days (95% CI, 9.50-NR), and 13.0 days (95% CI, 10.50-NR) in the 300 mg dose, 450 mg dose, and placebo groups, respectively.” ----- What does NR mean and how is this helpful? Can't it be estimated by HR method?

Author's Response:

We have replaced NR with ND (not determined) throughout the revised manuscript, as ND is a more accurate expression.

Confidence intervals for the median survival time are often calculated using non-parametric methods. These methods rely on the availability of survival data beyond the median to estimate variability. The upper confidence interval for the median survival time might not be determined if the survival function (the probability of being alive at a given time) doesn't reach 0.5 (50% survival) within the observed time frame.

22. NOTE ----- Line 82 ----- “Although niclosamide was well tolerated, but ended in failure, since there was no significant difference in the oropharyngeal clearance of SARS-CoV-2 between the placebo and niclosamide groups³¹.” ----- Not a fair definition of “failure” – the trial recruited and gave an answer. Just because it wasn't the answer they wanted, doesn't make the trial a failure.

Author's Response:

We agree with your comment and accordingly we have modified the sentences as follows:

The sentence has been revised to state that the trial “A prior randomized trial in patients with mild to moderate COVID-19 using pristine niclosamide did not significantly shorten symptom duration, likely due to these limitations.” (lines 66–69)

23. NOTE ----- Throughout ----- n/a ----- Almost certainly an issue of the journal system but there's a lot of files here and it's really difficult to know what's in what: the file names don't match the cross-referencing.

Author's Response:

We sincerely appreciate the reviewer's feedback and apologize for any confusion caused. For clarity and ease of reference, we have consolidated Annexure-1 and Annexure-2 into a single document, which is now clearly labeled and included in this revised submission. We hope this addresses the concern and provides a comprehensive view of the study findings.

TRIVIAL Comments

24. TRIVIAL ----- Figure 1 ----- n/a ----- The groups are presented in a different order to in the text. Keep it the same. (Figure 1's order is better)

Author's Response:

The text has been updated to follow the same order (placebo, low dose and high dose) as Figure 1 for consistency.

25. TRIVIAL ----- Intro ----- n/a ----- Feels too long

Author's Response:

The introduction has been slightly **condensed** to improve focus as reviewer commented.

26. TRIVIAL ----- Line 357 ----- n/a ----- The Methods section is difficult to find

Author's Response:

The method section is clearly provided in the manuscript in lines 282-471.

27. TRIVIAL ----- Line 377 ----- "The safety and efficacy assessments were conducted according to the study schedule." ----- As opposed to what?

Author's Response:

This phrase has been revised to clarify that all assessments followed a predefined protocol as follows (line 303):

"The safety and efficacy assessments were conducted following a predefined protocol".

28. TRIVIAL ----- Title ----- n/a ----- No "and" needed

Author's Response:

The unnecessary "and" has been removed for conciseness and as follows:

A randomized, double-blind, placebo-controlled trial of niclosamide nanohybrid for the treatment of patients with mild to moderate COVID-19

29. Reviewer #4 (Remarks on code availability):

They say it will be made available. I haven't looked at the code (not checked if it's actually there) as I had other priorities for the researchers to address.

Code Availability Statement

Author's Response:

Thank you for your comment. We are providing the code as requested (click the link here: <https://github.com/jhchoy1/CPCOV03-CODE>).

Authors' responses to the Reviewers' comments

Reviewer #4 (Remarks to the Author):

1. I appreciated the clarifications from the authors. It was interesting to know that some of the answers had already been in the manuscript, but had got lost in the complex wording and presentation. I have gone through and offer some further comments.

Response to Reviewer

Thank you for your kind feedback and for taking the time to review our manuscript again. We appreciate your note that some of our previous points were already in the manuscript but may have been unclear due to complex wording. Based on your suggestion, we have revised the text to improve clarity and presentation, making sure the main points are easier to follow.

We also reviewed your additional comments carefully and made the necessary changes. We hope these revisions address your concerns and improve the overall quality of the manuscript.

Thank you once again for your helpful suggestions.

2. The imbalance in individual symptoms could be made reassuring by making clear in the Methods that the severity score accounts across all of the symptoms and by putting severity (overall) above individual symptoms in Table 1.

Response to Reviewer

Thank you for this helpful suggestion.

As advised, we have clarified in the **Methods** section that the severity score reflects a cumulative assessment across all symptoms, rather than evaluating each symptom in isolation.

The primary objective of the study was to assess the efficacy of CP-COV03 as compared with the placebo measured by the number of days required for overall improvement in the severity score, which reflects the composite of all targeted COVID-19 symptoms sustained for more than 48 hours until Day 14. (page 15, **lines 365-367**)

Additionally, we have revised **Table 1** to place the **overall severity score** above the individual symptoms, making the structure and emphasis more intuitive and aligned with your recommendation.

We appreciate your insightful input, which has helped improve the clarity and presentation of our data.

Table 1. Baseline characteristics (intention-to-treat population)

	The placebo group (n=98)	The low dose group (n=99)	The high dose group (n=96)
Demographics			
Age (y), mean (SD)	43.79 (12.94)	42.18 (13.50)	41.89 (12.32)
Age groups			
19–29 yrs., n (%)	17 (17.4)	24 (24.2)	20 (20.8)
30–39 yrs., n (%)	21 (21.4)	16 (16.2)	21 (21.9)
40–49 yrs., n (%)	25 (25.5)	25 (25.3)	29 (30.2)
50–59 yrs., n (%)	21 (21.4)	25 (25.3)	16 (16.7)
≥60 yrs., n (%)	14 (14.3)	9 (9.1)	10 (10.4)
Sex			
Male, n (%)	62 (63.3)	70 (70.7)	61 (63.5)
Female, n (%)	36 (36.7)	29 (29.3)	35 (36.5)
Height, cm, mean (SD)	168.81 (8.57)	169.29 (8.39)	169.77 (7.79)
Weight, kg, mean (SD)	70.92 (14.05)	70.04 (13.16)	69.79 (12.99)
Comorbidities			
Hypertension, n (%)	11 (11.0)	7 (7.1)	4 (4.2)
Diabetes mellitus, n (%)	2 (2.0)	5 (5.1)	2 (2.1)
Hyperlipidemia, n (%)	4 (4.1)	11 (11.1)	5 (5.2)
COVID-19 symptoms			
Fever, n (%)	9 (9.2)	12 (12.1)	7 (7.3)
Chill, n (%)	87 (88.8)	82 (82.8)	82 (85.4)
Muscle ache, n (%)	86 (87.8)	88 (88.9)	85 (88.5)
Headache, n (%)	69 (70.4)	70 (70.7)	63 (65.6)
Fatigue, n (%)	80 (81.6)	84 (84.8)	73 (76)
Cough, n (%)	63 (64.3)	66 (66.7)	57 (59.4)
Sore throat, n (%)	66 (67.3)	70 (70.7)	64 (66.7)
Stuffy or runny nose, n (%)	70 (71.4)	75 (75.8)	68 (70.8)
Difficulty of breathing, n (%)	21 (21.4)	23 (23.2)	24 (25)
Nausea	21 (21.4)	27 (27.3)	37 (38.5)
Vomiting	6 (6.1)	7 (7.1)	6 (6.3)
Diarrhea	18 (18.4)	22 (22.2)	20 (20.8)
Mean total symptom score (±SD)	11.39 (±4.93)	11.96 (±5.54)	11.58 (±5.33)
COVID-19 severity			
Mild, n (%)	84 (85.7)	86 (86.9)	86 (89.6)
Moderate, n (%)	14 (14.3)	13 (13.1)	10 (10.4)

COVID-19: coronavirus disease 2019, SD: standard deviation

3. The Results call out Table 47 of the Supplement. There's more than 50 tables. I'm not going to read this in detail: it's impossible for this reviewer. Most tables in the Supplement are not referenced from the paper.

Response to Reviewer

Thank you for pointing this out.

We have already referenced all the Tables given in supplement in the manuscript; however, those appeared in the Annexure ones are not chronologically referenced due to the very fact that all of them are directly correlated with the secondary endpoints (Tables 1-50) and were provided based on the previous revisions by Referees 1-3. We hope this revision makes the supplementary materials more accessible and aligned with the core narrative of the paper. Thank you for highlighting this important usability issue.

4. I did not engage much with the Discussion last time as I had too many questions to be sure I would understand it. At 1,600 words, it's way too long and seems to repeat a lot of the methods and results.

Response to Reviewer

Thank you for your constructive feedback.

We understand your concern regarding the length and redundancy of the discussion section. In response, we were trying to reduce some of the discussion part (100 words, as revised as follows; **Pages 6-8; lines 142-179**). However, many of the discussion parts become lengthened to make readers more understandable as suggested by previous Referees 1-3.

In this study, designed to support emergency use authorization (EUA), the low dose group demonstrated a statistically significant reduction in both viral load and day (reported as a median value with half day intervals) required for overall improvement in the severity score of all targeted COVID-19 symptom to alleviate the 12 targeted COVID-19 symptoms compared to the placebo group.

Although the high dose group had a higher AUCt than the low dose one (12,876.29 vs. 10,562.09 ng·h/mL), both dose groups showed similar pharmacokinetic profiles and achieved steady-state levels, leading to comparable reductions in viral load. This suggests that CP-COV03's efficacy is time-dependent rather than concentration-dependent³⁴⁻³⁶, consistent with niclosamide's mechanism of inducing autophagy. The low dose group appears to have reached the pharmacodynamic ceiling needed for sustained antiviral action.

Analysis of the primary endpoint in the PPS population showed that the low dose group achieved sustained symptom improvement in a median of 9.0 days (95% CI, 7.00–10.00), significantly shorter than the 13.0 days required by the placebo group (95% CI, 10.50–ND; P = 0.0083). In the secondary endpoint, the low dose group showed generally faster improvement in individual symptoms, with sore throat, headache, and

fatigue resolving significantly earlier than in the placebo group (Sore throat: $p=0.0168$, the placebo group 5.23 [95% CI, 4.53-5.93] / the low dose group 3.99 [95% CI, 3.26-4.72]; Headache: $p=0.0285$, the placebo group 5.48 [95% CI, 4.72-6.24] / the low dose group 4.25 [95% CI, 3.46-5.04]; Fatigue: $p=0.0116$, the placebo group 5.62 [95% CI, 4.81-6.42] / the low dose group 4.10 [95% CI, 3.24-4.95]) (Table 6 in Annexure).

The mean number of days required for each of the 12 targeted COVID-19 symptoms to improve and be sustained for more than 48 hours was further analyzed across the ITT, PPS, and mITT groups and is provided in supplementary Table S4. In the ITT population, the median time to resolution of all 12 symptoms was 10.0 days (95% CI, 8.50-12.50) for the low dose group and 12.25 days (95% CI, 10.50-ND) for the placebo group, though not statistically significant. However, in the additional analysis of the mITT-1 population—comprising participants who received the study drug within 3 days of symptom onset—the median time to improvement of targeted symptoms was significantly shorter in the low dose group at 9.0 days (95% CI, 7.50–10.50) compared to 12.5 days (95% CI, 10.50–ND) in the placebo group ($P = 0.024$). Furthermore, in the high-risk subgroup within the mITT-1 population (aged ≥ 60 years, or with obesity, chronic conditions such as diabetes or hypertension, immunocompromised status, or long-term immunosuppressant use), the low dose group showed a median improvement time of 7.5 days (95% CI, 7.00–9.00) versus 12.5 days (95% CI, 8.00–ND) for the placebo group, also reaching statistical significance ($P = 0.017$).

5. Giving “not determined” as an upper bound for median time is unhelpful. This can be estimated through modelling, but it is always better to read up from a pre-specified time on the x-axis (which will always give bounds to the estimate) than across from the y-axis (which can give the same problem the researchers have here).

Response to Reviewer

In time-to-event analysis (e.g., Kaplan–Meier), the occurrence of “Not Determined (ND)” is a statistically expected outcome when the event of interest does not occur in a sufficient proportion of participants within the 14-day observation period.

6. The edits have made it mostly tidier and easier to read but there are still points of inconsistency to make me unsure about my understanding:

In labelling of allocated groups every time

Trial vs study (particularly in “completed the ...”)

Outcome (prognosis) vs outcome measure (how the prognosis was measured) vs endpoint (or event – the key (bad) thing being looked before in the outcome measure)

Response to Reviewer

We thank the reviewer for these detailed and helpful comments. We have carefully reviewed the manuscript to address the specific inconsistencies as noted:

We have standardized the dose group terminology such that the 300 mg or 900 mg dose group is referred to as the **low-dose group**, and the 450 mg or 1350 mg dose group is referred to as the **high-dose group**. In line with the analysis of time to recovery, we have revised "**primary outcome**" to "**primary endpoint**", and similarly, "**secondary outcome**" to "**secondary endpoint**". We have also updated "**primary outcome measure**" to "**primary endpoint measure**" where appropriate.

Additionally, we have revised all mentions of "**clinical study**" to "**clinical trial**" for consistency.

We believe these revisions have substantially improved clarity and consistency, enhancing the overall understanding of the manuscript. We appreciate the reviewer’s guidance in strengthening these aspects.

7. A reminder is needed in the Results and Discussion that the smallest unit of time in the time-to-event analyses as a half-day.

Response to Reviewer

As pointed out, we have now added a clear statement in both the Results and Discussion sections to indicate that the smallest unit of time used in the time-to-event analyses is a half-day. This clarification ensures transparency in the interpretation of the time-related data and supports accurate understanding of the analysis granularity.

In this study, designed to support emergency use authorization (EUA), the low dose group demonstrated a statistically significant reduction in both viral load and day (reported as a median value with half day intervals) required for overall improvement in the severity score of all targeted COVID-19 symptom to alleviate the 12 targeted COVID-19 symptoms compared to the placebo group. (page 6, line 142-145)

The primary objective of the study was to assess the efficacy of CP-COV03 as compared with the placebo measured by the number of days required for targeted COVID-19 symptoms to improve and maintain for more than 48 hours until Day 14. The time to sustained improvement (in days) was reported as a median value, with half-day intervals recorded based on symptom assessments conducted every morning and evening. (page 15, line 352-357)

8. Colours of groups in Figure 3 could usefully match the allocated colour in Figure 2

Response to Reviewer

To enhance visual consistency and improve reader comprehension, we have updated the group colours in Figure 3 to match those used in Figure 2. This alignment allows for easier cross-referencing between figures and a clearer understanding of group comparisons throughout the manuscript.

Fig. 3. Correlation between CP-COV03 pharmacokinetic parameters and the viral load of Severe Acute Respiratory Syndrome Coronavirus 2 (SARS-CoV-2). (A) the low dose group; (B) the high dose group (n = 15 for the low dose group, n = 11 for the high dose group).

9. The use of middle-endian date formats (month first) is anti-scientific but I think this is a problems with the journal rather than the authors. Similarly, putting the Methods after the Results feels like an excuse to finesse what happened: again, a “feature” of the journal, rather than a problem by the authors, I suspect.

Response to Reviewer

As noted, both the date formatting and the placement of the Methods section reflect the journal’s prescribed style rather than authorial decisions. We have adhered to the journal’s formatting and structural requirements throughout the manuscript.

10. The use of “dropout” sounds careless. Talk about early cessation of participation. Similar, “completed the study” is complex, because I think they mean the participant completed the maximum amount of time they needed to contribute which isn’t directly related to the time required by the researchers to complete the study overall.

Response to Reviewer

As commented by reviewer, we have replaced the term “dropout” with more appropriate and precise language such as “early cessation of participation” to avoid any unintended connotations and to reflect participant behavior more accurately.

Additionally, we have revised instances of “completed the study” with “completed the planned study follow-up period” to clarify that it refers specifically to participants who completed the maximum follow-up period required for their individual contribution. These changes have been implemented throughout the manuscript to enhance clarity and accuracy.